# Learning a Neuron by a Shallow ReLU Network: Dynamics and Implicit Bias for Correlated Inputs

**Dmitry Chistikov**[*]
University of Warwick
d.chistikov@warwick.ac.uk

**Matthias Englert**[*]
University of Warwick
m.englert@warwick.ac.uk

**Ranko Lazić**[*]
University of Warwick
r.s.lazic@warwick.ac.uk

## Abstract

We prove that, for the fundamental regression task of learning a single neuron, training a one-hidden layer ReLU network of any width by gradient flow from a small initialisation converges to zero loss and is implicitly biased to minimise the rank of network parameters. By assuming that the training points are correlated with the teacher neuron, we complement previous work that considered orthogonal datasets. Our results are based on a detailed non-asymptotic analysis of the dynamics of each hidden neuron throughout the training. We also show and characterise a surprising distinction in this setting between interpolator networks of minimal rank and those of minimal Euclidean norm. Finally we perform a range of numerical experiments, which corroborate our theoretical findings.

## 1 Introduction

One of the grand challenges for machine learning research is to understand how overparameterised neural networks are able to fit perfectly the training examples and simultaneously to generalise well to unseen data [Zhang, Bengio, Hardt, Recht, and Vinyals, 2021]. The double-descent phenomenon [Belkin, Hsu, Ma, and Mandal, 2019], where increasing the neural network capacity beyond the interpolation threshold can eventually reduce the test loss much further than could be achieved around the underparameterised "sweet spot", is a mystery from the standpoint of classical machine learning theory. This has been observed to happen even for training without explicit regularisers.

**Implicit bias of gradient-based algorithms.** A key hypothesis towards explaining the double-descent phenomenon is that the gradient-based algorithms that are used for training are *implicitly biased* (or *implicitly regularised*) [Neyshabur, Bhojanapalli, McAllester, and Srebro, 2017] to converge to solutions that in addition to fitting the training examples have certain properties which cause them to generalise well. It has attracted much attention in recent years from the research community, which has made substantial progress in uncovering implicit biases of training algorithms in many important settings [Vardi, 2023]. For example, for classification tasks, and for homogeneous networks (which is a wide class that includes ReLU networks provided they contain neither biases at levels deeper than the first nor residual connections), Lyu and Li [2020] and Ji and Telgarsky [2020] established that gradient flow is biased towards maximising the classification margin in parameter space, in the sense that once the training loss gets sufficiently small, the direction of the parameters subsequently converges to a Karush-Kuhn-Tucker point of the margin maximisation problem.

---

[*]Equal contribution.

37th Conference on Neural Information Processing Systems (NeurIPS 2023).

Insights gained in this foundational research direction have not only shed light on overparameterised generalisation, but have been applied to tackle other central problems, such as the susceptibility of networks trained by gradient-based algorithms to adversarial examples [Vardi, Yehudai, and Shamir, 2022] and the possibility of extracting training data from network parameters [Haim, Vardi, Yehudai, Shamir, and Irani, 2022].

**Regression tasks and initialisation scale.**    Showing the implicit bias for regression tasks, where the loss function is commonly mean square, has turned out to be more challenging than for classification tasks, where loss functions typically have exponential tails. A major difference is that, whereas most of the results for classification do not depend on how the network parameters are initialised, the scale of the initialisation has been observed to affect decisively the implicit bias of gradient-based algorithms for regression [Woodworth, Gunasekar, Lee, Moroshko, Savarese, Golan, Soudry, and Srebro, 2020]. When it is large so that the training follows the *lazy regime*, we tend to have fast convergence to a global minimum of the loss, however without an implicit bias towards sparsity and with limited generalisation [Jacot, Ged, Şimşek, Hongler, and Gabriel, 2021]. The focus, albeit at the price of uncertain convergence and lengthier training, has therefore been on the *rich regime* where the initialisation scale is small.

Considerable advances have been achieved for linear networks. For example, Azulay, Moroshko, Nacson, Woodworth, Srebro, Globerson, and Soudry [2021] and Yun, Krishnan, and Mobahi [2021] proved that gradient flow is biased to minimise the Euclidean norm of the predictor for one-hidden layer linear networks with infinitesimally small initialisation, and that the same holds also for deeper linear networks under an additional assumption on their initialisation. A related extensive line of work is on implicit bias of gradient-based algorithms for matrix factorisation and reconstruction, which has been a fruitful test-bed for regression using multi-layer networks. For example, Gunasekar, Woodworth, Bhojanapalli, Neyshabur, and Srebro [2017] proved that, under a commutativity restriction and starting from a small initialisation, gradient flow is biased to minimise the nuclear norm of the solution matrix; they also conjectured that the restriction can be dropped, which after a number of subsequent works was refuted by Li, Luo, and Lyu [2021], leading to a detailed analysis of both underparameterised and overparameterised regimes by Jin, Li, Lyu, Du, and Lee [2023].

For non-linear networks, such as those with the popular ReLU activation, progress has been difficult. Indeed, Vardi and Shamir [2021] showed that precisely characterising the implicit bias via a non-trivial regularisation function is impossible already for single-neuron one-hidden layer ReLU networks, and Timor, Vardi, and Shamir [2023] showed that gradient flow is not biased towards low-rank parameter matrices for multiple-output ReLU networks already with one hidden layer and small training datasets.

**ReLU networks and training dynamics.**    We suggest that, in order to further substantially our knowledge of convergence, implicit bias, and generalisation for regression tasks using non-linear networks, we need to understand more thoroughly the dynamics throughout the gradient-based training. This is because of the observed strong influence that initialisation has on solutions, but is challenging due to the highly non-convex optimisation landscape. To this end, evidence and intuition were provided by Maennel, Bousquet, and Gelly [2018], Li et al. [2021], and Jacot et al. [2021], who conjectured that, from sufficiently small initialisations, after an initial phase where the neurons get aligned to a number of directions that depend only on the dataset, training causes the parameters to pass close to a sequence of saddle points, during which their rank increases gradually but stays low.

The first comprehensive analysis in this vein was accomplished by Boursier, Pillaud-Vivien, and Flammarion [2022], who focused on orthogonal datasets (which are therefore of cardinality less than or equal to the input dimension), and established that, for one-hidden layer ReLU networks, gradient flow from an infinitesimal initialisation converges to zero loss and is implicitly biased to minimise the Euclidean norm of the network parameters. They also showed that, per sign class of the training labels (positive or negative), minimising the Euclidean norm of the interpolator networks coincides with minimising their rank.

**Our contributions.**    We tackle the main challenge posed by Boursier et al. [2022], namely handling datasets that are not orthogonal. A major obstacle to doing so is that, whereas the analysis of the training dynamics in the orthogonal case made extensive use of an almost complete separation between a turning phase and a growth phase for all hidden neurons, non-orthogonal datasets cause

considerably more complex dynamics in which hidden neurons follow training trajectories that simultaneously evolve their directions and norms [Boursier et al., 2022, Appendix A].

To analyse this involved dynamics in a reasonably clean setting, we consider the training of one-hidden layer ReLU networks by gradient flow from a small balanced initialisation on datasets that are labelled by a teacher ReLU neuron with which all the training points are correlated. More precisely, we assume that the angles between the training points and the teacher neuron are less than $\pi/4$, which implies that all angles between training points are less than $\pi/2$. The latter restriction has featured per label class in many works in the literature (such as by Phuong and Lampert [2021] and Wang and Pilanci [2022]), and the former is satisfied for example if the training points can be obtained by summing the teacher neuron $\boldsymbol{v}^*$ with arbitrary vectors of length less than $\|\boldsymbol{v}^*\|/\sqrt{2}$. All our other assumptions are very mild, either satisfied with probability exponentially close to 1 by any standard random initialisation, or excluding corner cases of Lebesgue measure zero.

Our contributions can be summarised as follows.

- We provide a detailed **non-asymptotic analysis** of the dynamics of each hidden neuron throughout the training, and show that it applies whenever the initialisation scale $\lambda$ is below a **precise bound** which is polynomial in the network width $m$ and exponential in the training dataset cardinality $n$. Moreover, our analysis applies for any input dimension $d > 1$, for any $n \geq d$ (otherwise exact learning of the teacher neuron may not be possible), for any $m$, and without assuming any specific random distribution for the initialisation. In particular, we demonstrate that the role of the overparameterisation in this setting is to ensure that initially at least one hidden neuron with a positive last-layer weight has in its active half-space at least one training point.

- We show that, during a first phase of the training, all active hidden neurons with a positive last-layer weight **get aligned** to a single direction which is positively correlated with all training points, whereas all active hidden neurons with a negative last-layer weight get turned away from all training points so that they deactivate. In contrast to the orthogonal dataset case where the sets of training points that are in the active half-spaces of the neurons are essentially constant during the training, in our correlated setting this first phase in general consists, for each neuron, of a different **sequence of stages** during which the cardinality of the set of training points in its active half-space gradually increases or decreases, respectively.

- We show that, during the rest of the training, the bundle of aligned hidden neurons with their last-layer weights, formed by the end of the first phase, grows and turns as it travels from near the origin to near the teacher neuron, and **does not separate**. To establish the latter property, which is the most involved part of this work, we identify a set in predictor space that depends only on $\lambda$ and the training dataset, and prove: first, that the trajectory of the bundle **stays inside the set**; and second, that this implies that the directional gradients of the individual neurons are such that the angles between them are non-increasing.

- We prove that, after the training departs from the initial saddle, which takes time logarithmic in $\lambda$ and linear in $d$, the gradient satisfies a Polyak-Łojasiewicz inequality and consequently the loss **converges to zero exponentially fast**.

- We prove that, although for any fixed $\lambda$ the angles in the bundle of active hidden neurons do not in general converge to zero as the training time tends to infinity, if we let $\lambda$ tend to zero then the networks to which the training converges have a limit: a network of rank 1, in which all non-zero hidden neurons are positive scalings of the teacher neuron and have positive last-layer weights. This establishes that gradient flow from an infinitesimal initialisation is **implicitly biased** to select interpolator networks of **minimal rank**. Note also that the limit network is identical in predictor space to the teacher neuron.

- We show that, surprisingly, among all networks with zero loss, there may exist some whose Euclidean norm is smaller than that of any network of rank 1. Moreover, we prove that this is the case if and only if a certain condition on angles determined by the training dataset is satisfied. This result might be seen as **refuting the conjecture** of Boursier et al. [2022, section 3.2] that the implicit bias to minimise Euclidean parameter norm holds beyond the orthogonal setting, and adding some weight to the hypothesis of Razin and Cohen [2020]. The counterexample networks in our proof have rank 2 and make essential use of the ReLU non-linearity.

- We perform numerical experiments that indicate that the training dynamics and the implicit bias we theoretically established occur in practical settings in which some of our assumptions are relaxed. In particular, gradient flow is replaced by gradient descent with a realistic learning rate,

the initialisation scales are small but not nearly as small as in the theory, and the angles between the teacher neuron and the training points are distributed around $\pi/4$.

We further discuss related work, prove all theoretical results, and provide additional material on our experiments, in the appendix.

## 2    Preliminaries

**Notation.**    We write: $[n]$ for the set $\{1, \ldots, n\}$, $\|\boldsymbol{v}\|$ for the Euclidean length of a vector $\boldsymbol{v}$, $\overline{\boldsymbol{v}} := \boldsymbol{v}/\|\boldsymbol{v}\|$ for the normalised vector, $\measuredangle(\boldsymbol{v}, \boldsymbol{v}') := \arccos(\overline{\boldsymbol{v}}^\top \overline{\boldsymbol{v}}')$ for the angle between $\boldsymbol{v}$ and $\boldsymbol{v}'$, and $\mathrm{cone}\{\boldsymbol{v}_1, \ldots, \boldsymbol{v}_n\} := \{\sum_{i=1}^n \beta_i \boldsymbol{v}_i \mid \beta_1, \ldots, \beta_n \geq 0\}$ for the cone generated by vectors $\boldsymbol{v}_1, \ldots, \boldsymbol{v}_n$.

**One-hidden layer ReLU network.**    For an input $\boldsymbol{x} \in \mathbb{R}^d$, the output of the network is

$$h_{\boldsymbol{\theta}}(\boldsymbol{x}) := \sum_{j=1}^m a_j\, \sigma(\boldsymbol{w}_j^\top \boldsymbol{x}) \,,$$

where $m$ is the width, the parameters $\boldsymbol{\theta} = (\boldsymbol{a}, \boldsymbol{W}) \in \mathbb{R}^m \times \mathbb{R}^{m \times d}$ consist of last-layer weights $\boldsymbol{a} = [a_1, \ldots, a_m]$ and hidden-layer weights $\boldsymbol{W}^\top = [\boldsymbol{w}_1, \ldots, \boldsymbol{w}_m]$, and $\sigma(u) := \max\{u, 0\}$ is the ReLU function.

**Balanced initialisation.**    For all $j \in [m]$ let

$$\boldsymbol{w}_j^0 := \lambda\, \boldsymbol{z}_j \qquad\qquad\qquad a_j^0 := s_j \|\boldsymbol{w}_j^0\|$$

where $\lambda > 0$ is the initialisation scale, $\boldsymbol{z}_j \in \mathbb{R}^d \setminus \{\boldsymbol{0}\}$, and $s_j \in \{\pm 1\}$.

A precise upper bound on $\lambda$ will be stated in Assumption 2.

We regard the initial unscaled hidden-layer weights $\boldsymbol{z}_j$ and last-layer signs $s_j$ as given, without assuming any specific random distributions for them. For example, we might have that each $\boldsymbol{z}_j$ consists of $d$ independent centred Gaussians with variance $\frac{1}{dm}$ and each $s_j$ is uniform over $\{\pm 1\}$.

We consider only initialisations for which the layers are balanced, i.e. $|a_j^0| = \|\boldsymbol{w}_j^0\|$ for all $j \in [m]$. Since more generally each difference $(a_j^t)^2 - \|\boldsymbol{w}_j^t\|^2$ is constant throughout training [Du, Hu, and Lee, 2018, Theorem 2.1] and we focus on small initialisation scales that tend to zero, this restriction (which is also present in Boursier et al. [2022]) is minor but simplifies our analysis.

**Neuron-labelled correlated inputs.**    The teacher neuron $\boldsymbol{v}^* \in \mathbb{R}^d$ and the training dataset $\{(\boldsymbol{x}_i, y_i)\}_{i=1}^n \subseteq (\mathbb{R}^d \setminus \{\boldsymbol{0}\}) \times \mathbb{R}$ are such that for all $i$ we have

$$y_i = \sigma(\boldsymbol{v}^{*\top} \boldsymbol{x}_i) \qquad\qquad \measuredangle(\boldsymbol{v}^*, \boldsymbol{x}_i) < \pi/4 \,.$$

In particular, since the angles between $\boldsymbol{v}^*$ and the training points $\boldsymbol{x}_i$ are acute, each label $y_i$ is positive.

To apply our results to a network with biases in the hidden layer and to a teacher neuron with a bias, one can work in dimension $d+1$ and extend the training points to $\begin{bmatrix} \boldsymbol{x}_i \\ 1 \end{bmatrix}$.

**Mean square loss gradient flow.**    For the regression task of learning the teacher neuron by the one-hidden layer ReLU network, we use the standard mean square empirical loss

$$L(\boldsymbol{\theta}) := \frac{1}{2n} \sum_{i=1}^n (y_i - h_{\boldsymbol{\theta}}(\boldsymbol{x}_i))^2 \,.$$

Our theoretical analysis concentrates on training by gradient flow, which from an initialisation as above evolves the network parameters by descending along the gradient of the loss by infinitesimal steps in continuous time [Li, Tai, and E, 2019]. Formally, we consider any parameter trajectory $\boldsymbol{\theta}^t \colon [0, \infty) \to \mathbb{R}^m \times \mathbb{R}^{m \times d}$ that is absolutely continuous on every compact subinterval, and that satisfies the differential inclusion

$$\mathrm{d}\boldsymbol{\theta}^t / \mathrm{d}t \in -\partial L(\boldsymbol{\theta}^t) \quad \text{for almost all } t \in [0, \infty) \,,$$

where $\partial L$ denotes the Clarke [1975] subdifferential of the loss function (which is locally Lipschitz).

We work with the Clarke subdifferential, which is a generalisation of the gradient, because the ReLU activation is not differentiable at 0, which causes non-differentiability of the loss function [Bolte, Daniilidis, Ley, and Mazet, 2010]. Although it follows from our results that, in our setting, the derivative of the ReLU can be fixed as $\sigma'(0) := 0$ like in the orthogonal case [Boursier et al., 2022, Appendix D], and the gradient flow trajectories are uniquely defined, that is not a priori clear; hence we work with the unrestricted Clarke subdifferential of the ReLU. We also remark that, in other settings, $\sigma'(0)$ cannot be fixed in this way due to gradient flow subtrajectories that correspond to gradient descent zig-zagging along a ReLU boundary (cf. e.g. Maennel et al. [2018, section 9.4]).

**Basic observations.** We establish the formulas for the derivatives of the last-layer weights and the hidden neurons; and that throughout the training, the signs of the last-layer weights do not change, and their absolute values track the norms of the corresponding hidden neurons. The latter property holds for all times $t$ by continuity and enables us to focus the analysis on the hidden neurons.

**Proposition 1.** *For all $j \in [m]$ and almost all $t \in [0, \infty)$ we have:*

(i) $\mathrm{d}a_j^t/\mathrm{d}t = {\boldsymbol{w}_j^t}^\top \boldsymbol{g}_j^t$ *and* $\mathrm{d}\boldsymbol{w}_j^t/\mathrm{d}t = a_j^t\, \boldsymbol{g}_j^t$, *where* $\boldsymbol{g}_j^t \in \frac{1}{n}\sum_{i=1}^n (y_i - h_{\boldsymbol{\theta}^t}(\boldsymbol{x}_i))\, \partial\sigma({\boldsymbol{w}_j^t}^\top \boldsymbol{x}_i)\, \boldsymbol{x}_i$;

(ii) $a_j^t = s_j \|\boldsymbol{w}_j^t\| \neq 0$.

The definition in part (i) of the vectors $\boldsymbol{g}_j^t$ that govern the dynamics is a membership because the subdifferential of the ReLU at 0 is the set of all values between 0 and 1, i.e. $\partial\sigma(0) = [0, 1]$.

## 3    Assumptions

To state our assumptions precisely, we introduce some additional notation. Let

$$I_+(\boldsymbol{v}) := \{i \in [n] \,|\, \boldsymbol{v}^\top \boldsymbol{x}_i > 0\} \quad I_0(\boldsymbol{v}) := \{i \in [n] \,|\, \boldsymbol{v}^\top \boldsymbol{x}_i = 0\} \quad I_-(\boldsymbol{v}) := \{i \in [n] \,|\, \boldsymbol{v}^\top \boldsymbol{x}_i < 0\}$$

denote the sets of indices of training points that are, respectively, either inside or on the boundary or outside of the non-negative half-space of a vector $\boldsymbol{v}$. Then let

$$J_+ := \{j \in [m] \;|\; I_+(\boldsymbol{z}_j) \neq \emptyset \wedge s_j = +1\} \quad J_- := \{j \in [m] \;|\; I_+(\boldsymbol{z}_j) \neq \emptyset \wedge s_j = -1\}$$

be the sets of indices of hidden neurons that are initially active on at least one training point and whose last-layer signs are, respectively, positive or negative. Also let

$$\boldsymbol{X} := [\boldsymbol{x}_1, \ldots, \boldsymbol{x}_n] \qquad\qquad \boldsymbol{\gamma}_I := \frac{1}{n}\sum_{i \in I} y_i \boldsymbol{x}_i$$

denote the matrix whose columns are all the training points, and the sum of all training points whose indices are in a set $I$, weighted by the corresponding labels and divided by $n$.

Moreover we define, for each $j \in J_+ \cup J_-$, a continuous trajectory $\boldsymbol{\alpha}_j^t$ in $\mathbb{R}^d$ by

$$\boldsymbol{\alpha}_j^0 := \boldsymbol{z}_j \qquad\qquad \mathrm{d}\boldsymbol{\alpha}_j^t/\mathrm{d}t := s_j \|\boldsymbol{\alpha}_j^t\|\, \boldsymbol{\gamma}_{I_+(\boldsymbol{\alpha}_j^t)} \quad \text{for all } t \in (0, \infty)\,.$$

Thus, starting from the unscaled initialisation $\boldsymbol{z}_j$ of the corresponding hidden neuron, $\boldsymbol{\alpha}_j^t$ follows a dynamics obtained from that of $\boldsymbol{w}_j^t$ in Proposition 1 (i) and (ii) by replacing the vector $\boldsymbol{g}_j^t$ by $\boldsymbol{\gamma}_{I_+(\boldsymbol{\alpha}_j^t)}$, which amounts to removing from $\boldsymbol{g}_j^t$ the network output terms and the activation boundary summands. These trajectories will be useful as *yardsticks* in our analysis of the first phase of the training.

**Assumption 1.**    (i) $d > 1$, $\mathrm{span}\{\boldsymbol{x}_1, \ldots, \boldsymbol{x}_n\} = \mathbb{R}^d$, *and* $\|\boldsymbol{v}^*\| = 1$.

(ii) $J_+ \neq \emptyset$, $I_0(\boldsymbol{z}_j) = \emptyset$ *for all* $j \in [m]$, *and* $\angle(\boldsymbol{z}_j, \boldsymbol{\gamma}_{[n]}) > 0$ *for all* $j \in J_-$.

(iii) $\overline{\boldsymbol{x}}_1, \ldots, \overline{\boldsymbol{x}}_n$ *are distinct, the eigenvalues of* $\frac{1}{n}\boldsymbol{X}\boldsymbol{X}^\top$ *are distinct, and* $\boldsymbol{v}^*$ *does not belong to a span of fewer than $d$ eigenvectors of* $\frac{1}{n}\boldsymbol{X}\boldsymbol{X}^\top$.

(iv) $|I_0(\boldsymbol{\alpha}_j^t)| \leq 1$ *for all* $j \in J_+ \cup J_-$ *and all* $t \in [0, \infty)$.

(v) *For all $j \in [m]$ and all $0 \leq T < T'$, if for all $t \in (T, T')$ we have $I_+(\boldsymbol{w}_j^t) = I_0(\boldsymbol{w}_j^{T'}) \neq \emptyset$ and $I_0(\boldsymbol{w}_j^t) = I_+(\boldsymbol{w}_j^{T'}) = \emptyset$, then for all $t \geq T'$ we have $\boldsymbol{w}_j^t = \boldsymbol{w}_j^{T'}$.*

This assumption is very mild. Part (i) excludes the trivial univariate case without biases (for univariate inputs with biases one needs $d = 2$), ensures that exact learning is possible, and fixes the length of the teacher neuron to streamline the presentation. Part (ii) assumes that, initially: at least one hidden neuron with a positive last-layer weight has in its active half-space at least one training point, no training point is at a ReLU boundary, and no hidden neuron with a negative last-layer weight is perfectly aligned with the $\boldsymbol{\gamma}_{[n]}$ vector; this holds with probability at least $1 - (3/4)^m$ for any continuous symmetric distribution of the unscaled hidden-neuron initialisations, e.g. $\boldsymbol{z}_j \overset{\text{i.i.d.}}{\sim} \mathcal{N}(\boldsymbol{0}, \frac{1}{dm}\boldsymbol{I}_d)$, and the uniform distribution of the last-layer signs $s_j \overset{\text{i.i.d.}}{\sim} \mathcal{U}\{\pm 1\}$. Parts (iii) and (iv) exclude corner cases of Lebesgue measure zero; observe that $\frac{1}{n}\boldsymbol{X}\boldsymbol{X}^\top$ is positive-definite, and that (iv) rules out a yardstick trajectory encountering two or more training points in its half-space boundary at exactly the same time. Part (v) excludes some unrealistic gradient flows that might otherwise be possible due to the use of the subdifferential: it specifies that, whenever a neuron deactivates (i.e. all training points exit its positive half-space), then it stays deactivated for the remainder of the training.

Before our next assumption, we define several further quantities. Let $\eta_1 > \cdots > \eta_d > 0$ denote the eigenvalues of $\frac{1}{n}\boldsymbol{X}\boldsymbol{X}^\top$, and let $\boldsymbol{u}_1, \ldots, \boldsymbol{u}_d$ denote the corresponding unit-length eigenvectors such that $\boldsymbol{v}^* = \sum_{k=1}^d \nu_k^* \boldsymbol{u}_k$ for some $\nu_1^*, \ldots, \nu_d^* > 0$. Also, for each $j \in J_+ \cup J_-$, let $n_j := |I_{-s_j}(\boldsymbol{z}_j)|$ be the number of training points that should enter into or exit from the non-negative half-space along the trajectory $\boldsymbol{\alpha}_j^t$ depending on whether the sign $s_j$ is positive or negative (respectively), and let

$$\varphi_j^t := \measuredangle(\boldsymbol{\alpha}_j^t, \boldsymbol{\gamma}_{I_+(\boldsymbol{\alpha}_j^t)}) \quad \text{for all } t \in [0, \infty) \text{ such that } I_+(\boldsymbol{\alpha}_j^t) \neq \emptyset$$

be the evolving angle between $\boldsymbol{\alpha}_j^t$ and the vector governing its dynamics (if any). Then the existence of the times at which the entries or the exits occur is confirmed in the following.

**Proposition 2.** *For all $j \in J_+ \cup J_-$ there exist a unique enumeration $i_j^1, \ldots, i_j^{n_j}$ of $I_{-s_j}(\boldsymbol{z}_j)$ and unique $0 = \tau_j^0 < \tau_j^1 < \cdots < \tau_j^{n_j}$ such that for all $\ell \in [n_j]$:*

*(i)* $I_{s_j}(\boldsymbol{\alpha}_j^t) = I_{s_j}(\boldsymbol{z}_j) \cup \{i_j^1, \ldots, i_j^{\ell-1}\}$ *for all* $t \in (\tau_j^{\ell-1}, \tau_j^\ell)$;

*(ii)* $I_0(\boldsymbol{\alpha}_j^t) = \emptyset$ *for all* $t \in (\tau_j^{\ell-1}, \tau_j^\ell)$, *and* $I_0\left(\boldsymbol{\alpha}_j^{\tau_j^\ell}\right) = \{i_j^\ell\}$.

Finally we define two measurements of the unscaled initialisation and the training dataset, which are positive thanks to Assumption 1, and which will simplify the presentation of our results.

$$\delta := \min \left\{ \begin{array}{c} \min_{i \in [n]} \|\boldsymbol{x}_i\|, \ \min_{i,i' \in [n]} \overline{\boldsymbol{x}}_i^\top \overline{\boldsymbol{x}}_{i'}, \ \min_{k \in [d-1]}(\sqrt{\eta_k} - \sqrt{\eta_{k+1}})(d-1), \ \sqrt{\eta_d}, \\[2mm] \min_{k \in [d]} \nu_k^* \sqrt{d}, \ \min_{j \in [m]} \|\boldsymbol{z}_j\|, \ \min_{j \in J_+} \cos \varphi_j^0, \ \min_{j \in J_-} \sin \varphi_j^0, \\[2mm] \min \left\{ |\overline{\boldsymbol{\alpha}}_j^t{}^\top \overline{\boldsymbol{x}}_i| \ \middle| \ \begin{array}{c} j \in J_+ \cup J_- \wedge \ell \in [n_j] \\ \wedge\, t \in [\tau_j^{\ell-1}, \tau_j^\ell] \wedge i \in [n] \\ \wedge\, i \neq i_j^\ell \wedge (\ell \neq 1 \Rightarrow i \neq i_j^{\ell-1}) \end{array} \right\}, \ \min_{j \in J_-} \overline{\boldsymbol{\alpha}}_j^0{}^\top \overline{\boldsymbol{x}}_{i_j^1}, \\[2mm] \min\{\tau_j^\ell - \tau_j^{\ell-1} \mid j \in J_+ \cup J_- \wedge \ell \in [n_j]\} \end{array} \right\}$$

$$\Delta := \max\{\max_{i \in [n]} \|\boldsymbol{x}_i\|, \ \max_{j \in [m]} \|\boldsymbol{z}_j\|, \ 1\} .$$

**Assumption 2.** $0 < \varepsilon \leq \frac{1}{4}$ and $\lambda \leq \left(m\, n^{9n\Delta^2/\delta^3}\right)^{-3/\varepsilon}$.

The quantity $\varepsilon$ introduced here has no effect on the network training, but is a parameter of our analysis, so that varying it within the assumed range tightens some of the resulting bounds while loosening others. The assumed bound on the initialisation scale $\lambda$ is polynomial in the network width $m$ and exponential in the dataset cardinality $n$. The latter is also the case in Boursier et al. [2022], where the bound was stated informally and without its dependence on parameters other than $m$ and $n$.

## 4  First phase: alignment or deactivation

We show that, for each initially active hidden neuron, if its last-layer sign is positive then it turns to include in its active half-space all training points that were initially outside, whereas if its last-layer

sign is negative then it turns to remove from its active half-space all training points that were initially inside. Moreover, those training points cross the activation boundary in the same order as they cross the half-space boundary of the corresponding yardstick trajectory $\boldsymbol{\alpha}_j^t$, and at approximately the same times (cf. Proposition 2).

**Lemma 3.** *For all $j \in J_+ \cup J_-$ there exist unique $0 = t_j^0 < t_j^1 < \ldots < t_j^{n_j}$ such that for all $\ell \in [n_j]$:*

(i) $I_{s_j}(\boldsymbol{w}_j^t) = I_{s_j}(\boldsymbol{z}_j) \cup \{i_j^1, \ldots, i_j^{\ell-1}\}$ *for all $t \in (t_j^{\ell-1}, t_j^{\ell})$;*

(ii) $I_0(\boldsymbol{w}_j^t) = \emptyset$ *for all $t \in (t_j^{\ell-1}, t_j^{\ell})$, and $I_0\left(\boldsymbol{w}_j^{t_j^\ell}\right) = \{i_j^\ell\}$;*

(iii) $|\tau_j^\ell - t_j^\ell| \leq \lambda^{1-\left(1+\frac{3\ell-1}{3n_j}\right)\varepsilon}$.

The preceding lemma is proved by establishing, for this first phase of the training, non-asymptotic upper bounds on the Euclidean norms of the hidden neurons and hence on the absolute values of the network outputs, and inductively over the stage index $\ell$, on the distances between the unit-sphere normalisations of $\boldsymbol{\alpha}_j^t$ and $\boldsymbol{w}_j^t$. Based on that analysis, we then obtain that each negative-sign hidden neuron does not grow from its initial length and deactivates by time $T_0 := \max_{j \in J_+ \cup J_-} \tau_j^{n_j} + 1$.

**Lemma 4.** *For all $j \in J_-$ we have:*
$$\|\boldsymbol{w}_j^{T_0}\| \leq \lambda\|\boldsymbol{z}_j\| \qquad\qquad \boldsymbol{w}_j^t = \boldsymbol{w}_j^{T_0} \quad \text{for all } t \geq T_0 \ .$$

We also obtain that, up to a later time $T_1 := \varepsilon \ln(1/\lambda)/\|\boldsymbol{\gamma}_{[n]}\|$, each positive-sign hidden neuron: grows but keeps its length below $2\|\boldsymbol{z}_j\|\lambda^{1-\varepsilon}$, continues to align to the vector $\boldsymbol{\gamma}_{[n]}$ up to a cosine of at least $1 - \lambda^\varepsilon$, and maintains bounded by $\lambda^{1-3\varepsilon}$ the difference between the logarithm of its length divided by the initialisation scale and the logarithm of the corresponding yardstick vector length.

**Lemma 5.** *For all $j \in J_+$ we have:*
$$\|\boldsymbol{w}_j^{T_1}\| < 2\|\boldsymbol{z}_j\|\lambda^{1-\varepsilon} \qquad \overline{\boldsymbol{w}}_j^{T_1\top}\overline{\boldsymbol{\gamma}}_{[n]} \geq 1 - \lambda^\varepsilon \qquad |\ln\|\boldsymbol{\alpha}_j^{T_1}\| - \ln\|\boldsymbol{w}_j^{T_1}/\lambda\|| \leq \lambda^{1-3\varepsilon} \ .$$

## 5 Second phase: growth and convergence

We next analyse the gradient flow subsequent to the deactivation of the negative-sign hidden neurons by time $T_0$ and the alignment of the positive-sign ones up to time $T_1$, and establish that the loss converges to zero at a rate which is exponential and does not depend on the initialisation scale $\lambda$.

**Theorem 6.** *Under Assumptions 1 and 2, there exists a time $T_2 < \ln(1/\lambda)(4+\varepsilon)d\Delta^2/\delta^6$ such that for all $t \geq 0$ we have $L(\boldsymbol{\theta}^{T_2+t}) < 0.5\,\Delta^2\,\mathrm{e}^{-t\cdot 0.4\,\delta^4/\Delta^2}$.*

In particular, for $\varepsilon = 1/4$ and $\lambda = \left((m\,n^n)^{9\Delta^2/\delta^3}\right)^{-3/\varepsilon}$ (cf. Assumption 2), the first bound in Theorem 6 becomes $T_2 < (\ln m + n\ln n)\,d \cdot 17 \cdot 27\,\Delta^4/\delta^9$.

The proof of Theorem 6 is in large part geometric, with a key role played by a set $\mathcal{S} := \mathcal{S}_1 \cup \cdots \cup \mathcal{S}_d$ in predictor space, whose constituent subsets are defined as

$$\mathcal{S}_\ell := \left\{ \boldsymbol{v} = \sum_{k=1}^d \nu_k \boldsymbol{u}_k \ \middle| \ \bigwedge_{1 \leq k < \ell} \Omega_k \ \wedge \ \Phi_\ell \ \wedge \bigwedge_{\ell \leq k < k' \leq d} (\Psi_{k,k'}^\downarrow \wedge \Psi_{k,k'}^\uparrow) \ \wedge \ \Xi \right\},$$

where the individual constraints are as follows (here $\eta_0 := \infty$ so that e.g. $\frac{\eta_1}{2\eta_0} = 0$):

$$\Omega_k: \ 1 < \frac{\nu_k}{\nu_k^*} \qquad \Phi_\ell: \ \frac{\eta_\ell}{2\eta_{\ell-1}} < \frac{\nu_\ell}{\nu_\ell^*} \leq 1 \qquad \Psi_{k,k'}^\downarrow: \ \frac{\eta_{k'}}{2\eta_k}\frac{\nu_k}{\nu_k^*} < \frac{\nu_{k'}}{\nu_{k'}^*}$$

$$\Xi: \ \overline{\boldsymbol{v}}^\top\overline{\boldsymbol{XX}^\top(\boldsymbol{v}^*-\boldsymbol{v})} > \lambda^{\varepsilon/3} \qquad\qquad \Psi_{k,k'}^\uparrow: \ \frac{\nu_{k'}}{\nu_{k'}^*} < 1 - \left(1 - \frac{\nu_k}{\nu_k^*}\right)^{\frac{1}{2}+\frac{\eta_{k'}}{2\eta_k}} \ .$$

Thus $\mathcal{S}$ is connected, open, and constrained by $\Xi$ to be within the ellipsoid $\boldsymbol{v}^\top\boldsymbol{XX}^\top(\boldsymbol{v}^*-\boldsymbol{v}) = 0$ which is centred at $\frac{\boldsymbol{v}^*}{2}$, with the remaining constraints slicing off further regions by straight or curved boundary surfaces.

In the most complex component of this work, we show that, for all $t \geq T_1$, the trajectory of the sum $\boldsymbol{v}^t := \sum_{j \in J_+} a_j^t \boldsymbol{w}_j^t$ of the active hidden neurons weighted by the last layer stays inside $\mathcal{S}$, and the cosines of the angles between the neurons remain above $1 - 4\lambda^\varepsilon$. This involves proving that each face of the boundary of $\mathcal{S}$ is repelling for the training dynamics when approached from the inside; we remark that, although that is in general false for the entire boundary of the constraint $\Xi$, it is in particular true for its remainder after the slicing off by the other constraints. We also show that all points in $\mathcal{S}$ are positively correlated with all training points, which together with the preceding facts implies that, during this second phase of the training, the network behaves approximately like a linear one-hidden layer one-neuron network. Then, as the cornerstone of the rest of the proof, we show that, for all $t \geq T_2$, the gradient of the loss satisfies a Polyak-Łojasiewicz inequality $\|\nabla L(\boldsymbol{\theta}^t)\|^2 > \frac{2\eta_d \|\boldsymbol{\gamma}_{[n]}\|}{5\eta_1} L(\boldsymbol{\theta}^t)$. Here $T_2 := \inf\{t \geq T_1 \mid \nu_1^t / \nu_1^* \geq 1/2\}$ is a time by which the network has departed from the initial saddle, more precisely when the first coordinate $\nu_1^t$ of the bundle vector $\boldsymbol{v}^t$ with respect to the basis consisting of the eigenvectors of the matrix $\frac{1}{n} \boldsymbol{X} \boldsymbol{X}^\top$ crosses the half-way threshold to the first coordinate $\nu_1^*$ of the teacher neuron.

The interior of the ellipsoid in the constraint $\Xi$ actually consists of all vectors that have an acute angle with the derivative of the training dynamics in predictor space, and the "padding" of $\lambda^{\varepsilon/3}$ is present because the derivative of the bundle vector $\boldsymbol{v}^t$ is "noisy" due to the latter being made up of the approximately aligned neurons. The remaining constraints delimit the subsets $\mathcal{S}_1, \ldots, \mathcal{S}_d$ of the set $\mathcal{S}$, through which the bundle vector $\boldsymbol{v}_t$ passes in that order, with each unique "handover" from $\mathcal{S}_\ell$ to $\mathcal{S}_{\ell+1}$ happening exactly when the corresponding coordinate $\nu_\ell^t$ exceeds its target $\nu_\ell^*$. The non-linearity of the constraints $\Psi_{k,k'}^\uparrow$ is needed to ensure the repelling for the training dynamics.

## 6  Implicit bias of gradient flow

Let us denote the set of all balanced networks by
$$\Theta := \{(\boldsymbol{a}, \boldsymbol{W}) \in \mathbb{R}^m \times \mathbb{R}^{m \times d} \mid \forall j \in [m] \colon |a_j| = \|\boldsymbol{w}_j\|\}$$
and the subset in which all non-zero hidden neurons are positive scalings of $\boldsymbol{v}^*$, have positive last-layer weights, and have lengths whose squares sum up to $\|\boldsymbol{v}^*\| = 1$, by
$$\Theta_{\boldsymbol{v}^*} := \{(\boldsymbol{a}, \boldsymbol{W}) \in \Theta \mid \textstyle\sum_{j=1}^m \|\boldsymbol{w}_j\|^2 = 1 \ \wedge \ \forall j \in [m] \colon \boldsymbol{w}_j \neq \boldsymbol{0} \Rightarrow (\overline{\boldsymbol{w}}_j = \boldsymbol{v}^* \wedge a_j > 0)\} \, .$$

Our main result establishes that, as the initialisation scale $\lambda$ tends to zero, the networks with zero loss to which the gradient flow converges tend to a network in $\Theta_{\boldsymbol{v}^*}$. The explicit subscripts indicate the dependence on $\lambda$ of the parameter vectors. The proof builds on the preceding results and involves a careful control of accumulations of approximation errors over lengthy time intervals.

**Theorem 7.** *Under Assumptions 1 and 2, $L\left(\lim_{t \to \infty} \boldsymbol{\theta}_\lambda^t\right) = 0$ and $\lim_{\lambda \to 0^+} \lim_{t \to \infty} \boldsymbol{\theta}_\lambda^t \in \Theta_{\boldsymbol{v}^*}$.*

## 7  Interpolators with minimum norm

To compare the set $\Theta_{\boldsymbol{v}^*}$ of balanced rank-1 interpolator networks with the set of all minimum-norm interpolator networks, in this section we focus on training datasets of cardinality $d$, we assume the network width is greater than 1 (otherwise the rank is necessarily 1), and we exclude the threshold case of Lebesgue measure zero where $\mathcal{M} = 0$. The latter measurement of the training dataset is defined below in terms of angles between the teacher neuron and vectors in any two cones generated by different generators of the dual of the cone of all training points.

Let $[\boldsymbol{\chi}_1, \ldots, \boldsymbol{\chi}_d]^\top := \boldsymbol{X}^{-1}$ and
$$\mathcal{M} := \max \left\{ \cos \angle(\boldsymbol{p}, \boldsymbol{q}) - \sin \angle(\boldsymbol{p}, \boldsymbol{v}^*) \; \middle| \; \begin{array}{c} \emptyset \subsetneq K \subsetneq [d] \\ \wedge \ \boldsymbol{0} \neq \boldsymbol{p} \in \operatorname{cone}\{\boldsymbol{\chi}_k \mid k \in K\} \\ \wedge \ \boldsymbol{0} \neq \boldsymbol{q} \in \operatorname{cone}\{\boldsymbol{\chi}_k \mid k \notin K\} \end{array} \right\} \, .$$

**Assumption 3.** $n = d$, $m > 1$, and $\mathcal{M} \neq 0$.

We obtain that, surprisingly, $\Theta_{\boldsymbol{v}^*}$ equals the set of all interpolators with minimum Euclidean norm if $\mathcal{M} < 0$, but otherwise they are disjoint.

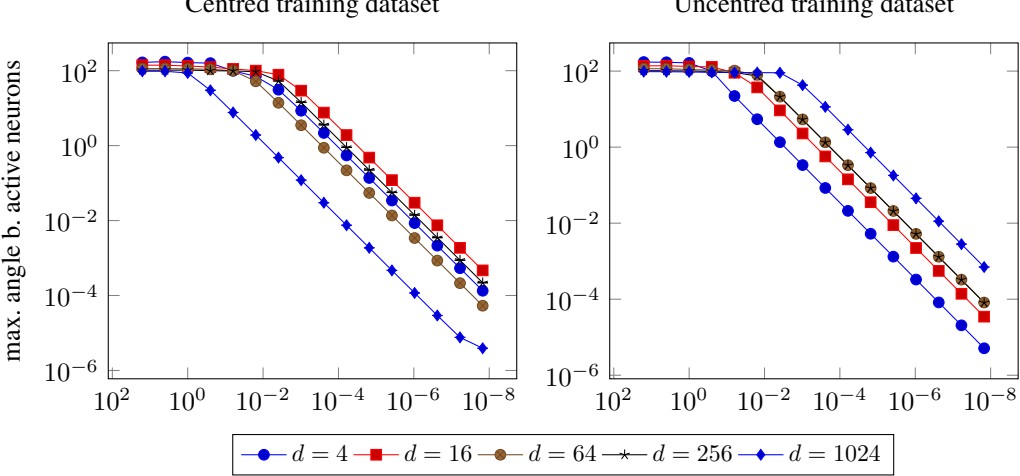

Figure 1: Dependence of the maximum angle between active hidden neurons on the initialisation scale $\lambda$, for two generation schemes of the training dataset and a range of input dimensions, at the end of the training. Both axes are logarithmic, and each point plotted shows the median over five trials.

**Theorem 8.** *Under Assumptions 1 and 3:*

*(i) if $\mathcal{M} < 0$ then $\Theta_{\boldsymbol{v}^*}$ is the set of all global minimisers of $\|\boldsymbol{\theta}\|^2$ subject to $L(\boldsymbol{\theta}) = 0$;*

*(ii) if $\mathcal{M} > 0$ then no point in $\Theta_{\boldsymbol{v}^*}$ is a global minimiser of $\|\boldsymbol{\theta}\|^2$ subject to $L(\boldsymbol{\theta}) = 0$.*

For each of the two cases, we provide a family of example datasets in the appendix. We remark that a sufficient condition for $\mathcal{M} < 0$ to hold is that the inner product of any two distinct rows $\boldsymbol{\chi}_k$ of the inverse of the dataset matrix $\boldsymbol{X}$ is non-positive, i.e. that the inverse of the Gram matrix of the dataset (in our setting this Gram matrix is positive) is a Z-matrix (cf. e.g. Fiedler and Pták [1962]). Also, if the training points were orthogonal then all the $\cos \angle(\boldsymbol{p}, \boldsymbol{q})$ terms in the definition of $\mathcal{M}$ would be zero and consequently we would have $\mathcal{M} < 0$; this is consistent with the result that, per sign class of the training labels in the orthogonal setting, minimising the Euclidean norm of interpolators coincides with minimising their rank [Boursier et al., 2022, Appendix C].

## 8   Experiments

We consider two schemes for generating the training dataset, where $\mathbb{S}^{d-1}$ is the unit sphere in $\mathbb{R}^d$.

**Centred:** We sample $\boldsymbol{\mu}$ from $\mathcal{U}(\mathbb{S}^{d-1})$, then sample $\boldsymbol{x}_1, \ldots, \boldsymbol{x}_d$ from $\mathcal{N}(\boldsymbol{\mu}, \frac{\rho}{d}\boldsymbol{I}_d)$ where $\rho = 1$, and finally set $\boldsymbol{v}^* = \boldsymbol{\mu}$. This distribution has the property that, in high dimensions, the angles between the teacher neuron $\boldsymbol{v}^*$ and the training points $\boldsymbol{x}_i$ concentrate around $\pi/4$. We exclude rare cases where some of these angles exceed $\pi/2$.

**Uncentred:** This is the same, except that we use $\rho = \sqrt{2} - 1$, sample one extra point $\boldsymbol{x}_0$, and finally set $\boldsymbol{v}^* = \overline{\boldsymbol{x}}_0$. Here the angles between $\boldsymbol{v}^*$ and $\boldsymbol{x}_i$ also concentrate around $\pi/4$ in high dimensions, but the expected distance between $\boldsymbol{v}^*$ and $\boldsymbol{\mu}$ is $\sqrt{\rho}$.

For each of the two dataset schemes, we train a one-hidden layer ReLU network of width $m = 200$ by gradient descent with learning rate $0.01$, from a balanced initialisation such that $\boldsymbol{z}_j \overset{\text{i.i.d.}}{\sim} \mathcal{N}(\boldsymbol{0}, \frac{1}{dm}\boldsymbol{I}_d)$ and $s_j \overset{\text{i.i.d.}}{\sim} \mathcal{U}\{\pm 1\}$, and for a range of initialisation scales $\lambda$ and input dimensions $d$.[2]

We present in Figure 1 some results from considering initialisation scales $\lambda = 4^2, 4^1, \ldots, 4^{-12}, 4^{-13}$ and input dimensions $d = 4, 16, 64, 256, 1024$, where we train until the number of iterations

---

[2]We are making code to run the experiments available at https://github.com/englert-m/shallow_ReLU_dynamics.

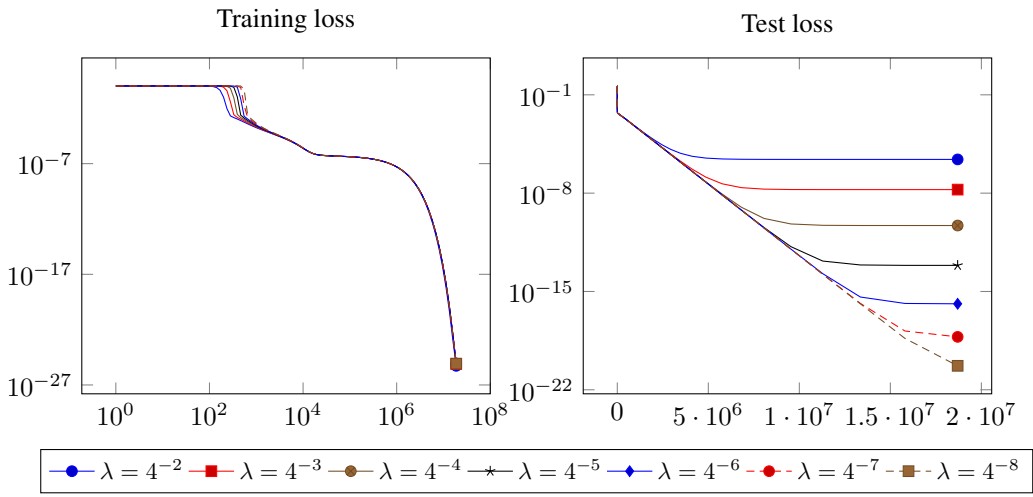

Figure 2: Evolution of the training loss, and of an outside distribution test loss, during training for an example centred training dataset in dimension 16 and width 200. The horizontal axes, logarithmic for the training loss and linear for the test loss, show iterations. The vertical axes are logarithmic.

reaches $2 \cdot 10^7$ or the loss drops below $10^{-9}$. The plots are in line with Theorem 7, showing how the maximum angle between active hidden neurons at the end of the training decreases with $\lambda$.

Figure 2 on the left illustrates the exponentially fast convergence of the training loss (cf. Theorem 6), and on the right how the implicit bias can result in good generalisation. The test loss is computed over an input distribution which is different from that of the training points, namely we sample 64 test inputs from $\mathcal{N}(\mathbf{0}, \boldsymbol{I}_d)$. These plots are for initialisation scales $\lambda = 4^{-2}, 4^{-3}, \ldots, 4^{-7}, 4^{-8}$.

## 9 Conclusion

We provided a detailed analysis of the dynamics of training a shallow ReLU network by gradient flow from a small initialisation for learning a single neuron which is correlated with the training points, establishing convergence to zero loss and implicit bias to rank minimisation in parameter space. We believe that in particular the geometric insights we obtained in order to deal with the complexities of the multi-stage alignment of hidden neurons followed by the simultaneous evolution of their norms and directions, will be useful to the community in the ongoing quest to understand implicit bias of gradient-based algorithms for regression tasks using non-linear networks.

A major direction for future work is to bridge the gap between, on one hand, our assumption that the angles between the teacher neuron and the training points are less than $\pi/4$, and the other, the assumption of Boursier et al. [2022] that the training points are orthogonal, while keeping a fine granularity of description. We expect this to be difficult because it seems to require handling bundles of approximately aligned neurons which may have changing sets of training points in their active half-spaces and which may separate during the training. However, it should be straightforward to extend our results to orthogonally separable datasets and two teacher ReLU neurons, where each of the latter has an arbitrary sign, labels one of the two classes of training points, and has angles less than $\pi/4$ with them; the gradient flow would then pass close to a second saddle point, where the labels of one of the classes have been nearly fitted but the hidden neurons that will fit the labels of the other class are still small. We report on related numerical experiments in the appendix.

We also obtained a condition on the dataset that determines whether rank minimisation and Euclidean norm minimisation for interpolator networks coincide or are distinct. Although this dichotomy remains true if the $\pi/4$ correlation bound is relaxed to $\pi/2$, the implicit bias of gradient flow in that extended setting is an open question. Other directions for future work include considering multi-neuron teacher networks, student networks with more than one hidden layer, further non-linear activation functions, and gradient descent instead of gradient flow; also refining the bounds on the initialisation scale and the convergence time.

## Acknowledgments and Disclosure of Funding

We acknowledge the Centre for Discrete Mathematics and Its Applications at the University of Warwick for partial support, and the Scientific Computing Research Technology Platform at the University of Warwick for providing the compute cluster on which the experiments presented in this paper were run.

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
