# Contents

## A   Related work

Here we further discuss a selection of related work.

**Early training phase.** Our analysis of the first phase of the training builds on those of Maennel et al. [2018], who considered unrestricted datasets but focused on asymptotic results; and of Boursier et al. [2022], who restricted the datasets to orthogonal but provided detailed non-asymptotic bounds. In particular, our normalised yardstick trajectories $\overline{\boldsymbol{\alpha}}_j^t$ are analogous to $\vec{U}_i(t)$ in Maennel et al. [2018, section 9.6] and $\widetilde{w}_j^t$ in Boursier et al. [2022, Appendix B.6]. Our contribution in this part, in relation to these two works, is to extend the fine-grained description for the orthogonal case to the more involved correlated case, which exhibits a sequence of intermediate stages, obtaining detailed non-asymptotic bounds including for the initialisation scale $\lambda$; the latter are essential for our analysis of the subsequent second phase of the training and our proof of the implicit bias when $\lambda$ tends to zero.

Non-asymptotic bounds for early training of one-hidden layer networks were shown by Lyu, Li, Wang, and Arora [2021] with the Leaky-ReLU non-linearity, logistic loss, and linearly separable data which are either symmetric or have a principal direction and uniformly labelled support vectors; and by Min, Vidal, and Mallada [2023] with the ReLU non-linearity, exponential loss, and orthogonally separable data. Wang and Ma [2022] studied early training by gradient descent of one-hidden layer ReLU networks, focusing on a non-balanced random initialisation and on obtaining a lower bound for the Euclidean norm of the gradient. Another related work is by Xu and Du [2023], where the hidden layer is trained, every last-layer weight is fixed to $1$, and the loss is over a Gaussian data population; consequently some aspects of the training dynamics are simpler, and already the first phase aligns the neurons to the teacher.

**Learning a single neuron.** A number of previous works studied learning a single neuron by gradient-based algorithms, in settings including realisable without bias [Yehudai and Shamir, 2020], agnostic and noisy [Frei, Cao, and Gu, 2020], realisable with bias [Vardi, Yehudai, and Shamir, 2021], multi layer [Lee, Sim, and Ye, 2022], and overparameterised [Xu and Du, 2023]. In particular, Vardi et al. [2021] proved exponentially fast convergence of gradient descent to the global minimum by geometric and algebraic arguments, for two complementary sets of assumptions on the data distribution and the student initialisation; Xu and Du [2023] determined that, when the student network has width at least $2$ and only its hidden layer is trained, the speed of convergence drops to cubic; and Lee et al. [2022] studied how neuron depth and initialisation scale affect the speed of

convergence. In contrast to the settings in those works, our student network has arbitrary width and both its layers are trained.

**Convergence for one-hidden layer ReLU networks and mean square loss.** Further related convergence results were obtained by Jentzen and Riekert [2023], who proved that, for one-hidden layer ReLU networks trained by gradient flow with respect to a mean square population loss, if the data is one-dimensional, the target function is affine, the initial loss is sufficiently small, and the training trajectory is bounded, then it converges to zero loss; and that if moreover the network is width-one then the boundedness assumption is not needed.

Also with univariate data, Stewart, Bach, Berthet, and Vert [2023] compared the features learnt by one-hidden layer ReLU networks for the square loss and the cross-entropy loss, postulating that sparseness in the regression case may cause optimisation difficulties, and reporting synthetic experiments that support the claim.

**Properties transferred from parameter space to function space.** The implicit bias in parameter space that we established, namely to interpolator networks of rank 1, has a clear implication in function space, namely the resulting function is identical to that defined by the teacher neuron. However, we also compared that set of interpolator networks with the one obtained by minimising the Euclidean norm. The question of what functions the latter networks define is in general non-trivial; for one-hidden layer ReLU networks, it was studied in the univariate case by Savarese, Evron, Soudry, and Srebro [2019] and Ergen and Pilanci [2021], and in the multivariate case by Ongie, Willett, Soudry, and Srebro [2020]. More recently, Boursier and Flammarion [2023] investigated further the univariate case, elucidating the consequences of whether the norm takes into account the bias terms.

**Regression using diagonal linear networks.** Even, Pesme, Gunasekar, and Flammarion [2023] and Pesme and Flammarion [2023] considered implicit bias for regression tasks, focusing on diagonal networks with linear activation. The former proved convergence and compared implicit biases of gradient descent and stochastic gradient descent with large learning rates. The latter studied gradient flow from a vanishing initialisation and provided a full description of training trajectories, showing that they jump from saddle to saddle until reaching the minimum $\ell_1$-norm solution.

**Classification using Leaky-ReLU networks.** Building on the implicit bias to margin maximisation in parameter space for homogenous networks [Lyu and Li, 2020, Ji and Telgarsky, 2020], convergence to a linear classifier was shown by Lyu et al. [2021], Sarussi, Brutzkus, and Globerson [2021], and Frei, Vardi, Bartlett, Srebro, and Hu [2023c] for one-hidden layer networks with the Leaky-ReLU activation and several sets of assumptions that include linear separability of the data. Frei, Vardi, Bartlett, and Srebro [2023a] subsequently established that in two kinds of distributional settings benign overfitting occurs, namely the predictors interpolate noisy training data and simultaneously generalise well to unseen test data.

**Implicit bias and adversarial examples.** In addition to Vardi et al. [2021], likewise in the context of one-hidden layer ReLU networks and exponentially-tailed loss functions, consequences for adversarial examples of the implicit bias to margin maximisation in parameter space were investigated by Englert and Lazić [2022], who showed that for orthogonally separable training datasets it may prevent adversarial reprogrammability; and by Frei, Vardi, Bartlett, and Srebro [2023b], who showed that for clustered data it leads to non-robust solutions even though robust networks that fit the data exist. In contrast to those works, which focus on gradient flow, Melamed, Yehudai, and Vardi [2023] considered possibly stochastic gradient descent and established that, when data belongs to a low-dimensional linear subspace, the training produces non-robust solutions, but decreasing the initialisation scale or adding a Euclidean norm regulariser increases robustness to orthogonal adversarial perturbations.

# B   Proof for the preliminaries

First we note that the gradient flow ensures that the loss monotonically decreases, at the rate equal to the square of the Euclidean norm of the gradient.

**Proposition 9** (by Davis, Drusvyatskiy, Kakade, and Lee [2020, Lemma 5.2]). *For almost all* $t \in [0, \infty)$ *we have* $\mathrm{d}L(\boldsymbol{\theta}^t)/\mathrm{d}t = -\|\mathrm{d}\boldsymbol{\theta}^t/\mathrm{d}t\|^2$.

Then we show the following proposition from section 2.

**Proposition 1.** *For all* $j \in [m]$ *and almost all* $t \in [0, \infty)$ *we have:*

(i) $\mathrm{d}a_j^t/\mathrm{d}t = \boldsymbol{w}_j^{t\top} \boldsymbol{g}_j^t$ *and* $\mathrm{d}\boldsymbol{w}_j^t/\mathrm{d}t = a_j^t \boldsymbol{g}_j^t$, *where* $\boldsymbol{g}_j^t \in \frac{1}{n}\sum_{i=1}^n (y_i - h_{\boldsymbol{\theta}^t}(\boldsymbol{x}_i)) \, \partial\sigma(\boldsymbol{w}_j^{t\top} \boldsymbol{x}_i) \, \boldsymbol{x}_i$;

(ii) $a_j^t = s_j \|\boldsymbol{w}_j^t\| \neq 0$.

*Proof.* Part (i) follows by straightforward calculations. For part (ii), by (i), for all $j \in [m]$ and almost all $t \in [0, \infty)$ we have

$$\mathrm{d}(a_j^t)^2/\mathrm{d}t = 2a_j^t \boldsymbol{w}_j^{t\top} \boldsymbol{g}_j^t = \mathrm{d}\|\boldsymbol{w}_j^t\|^2/\mathrm{d}t \ ,$$

and so $(a_j^t)^2 - \|\boldsymbol{w}_j^t\|^2$ is constant and therefore zero for all $t$ by the initialisation and continuity. It remains to show that $a_j^t \neq 0$ for all $t$. Observe that, by Proposition 9, for almost all $t$, provided $a_j^t \neq 0$ we have

$$\mathrm{d}\ln(a_j^t)^2/\mathrm{d}t \geq -2\|\boldsymbol{g}_j^t\| \geq -2\sqrt{2L(\boldsymbol{\theta}^t)} \max_{i=1}^n \|\boldsymbol{x}_i\| \geq -2\sqrt{2L(\boldsymbol{\theta}^0)} \max_{i=1}^n \|\boldsymbol{x}_i\| \ .$$

Hence, by continuity, for all $t$ we have

$$(a_j^t)^2 \geq (a_j^0)^2 \exp\left(-2t\sqrt{2L(\boldsymbol{\theta}^0)} \max_{i=1}^n \|\boldsymbol{x}_i\|\right) \ . \qquad \square$$

Also from Proposition 1 we obtain the formulas for the derivatives of the spherical coordinates of the hidden neurons, i.e. of their logarithmic Euclidean norms and their unit normalisations.

**Corollary 10.** *For all* $j \in [m]$ *and almost all* $t \in [0, \infty)$ *we have*

$$\mathrm{d}\ln\|\boldsymbol{w}_j^t\|/\mathrm{d}t = s_j \overline{\boldsymbol{w}}_j^{t\top} \boldsymbol{g}_j^t \qquad\qquad \mathrm{d}\overline{\boldsymbol{w}}_j^t/\mathrm{d}t = s_j(\boldsymbol{g}_j^t - \overline{\boldsymbol{w}}_j^t \overline{\boldsymbol{w}}_j^{t\top} \boldsymbol{g}_j^t) \ .$$

## C   Proofs for the assumptions

In this section we prove Proposition 2 from section 3, which is subsumed by Proposition 14 below, and then show as Corollary 15 that the cases excluded by Assumption 1 (iv) have Lebesgue measure zero, as claimed in section 3.

We first establish several elementary properties of the yardstick trajectories $\boldsymbol{\alpha}_j^t$.

**Proposition 11.** *For all* $j \in J_+ \cup J_-$, *the following statements hold.*

(i) $\boldsymbol{\gamma}_{I_+(\boldsymbol{\alpha}_j^t)} = \mathbf{0}$ *if and only if* $I_+(\boldsymbol{\alpha}_j^t) = \emptyset$.

(ii) $(\boldsymbol{\alpha}_j^t)^\top \boldsymbol{\gamma}_{I_+(\boldsymbol{\alpha}_j^t)} \geq 0$. *Moreover,* $(\boldsymbol{\alpha}_j^t)^\top \boldsymbol{\gamma}_{I_+(\boldsymbol{\alpha}_j^t)} = 0$ *if and only if* $(\boldsymbol{\alpha}_j^t)^\top \boldsymbol{x}_i \leq 0$ *for all* $i \in [n]$.

(iii) *If* $(\boldsymbol{\alpha}_j^t)^\top \boldsymbol{\gamma}_{I_+(\boldsymbol{\alpha}_j^t)} = \|\boldsymbol{\alpha}_j^t\|\|\boldsymbol{\gamma}_{I_+(\boldsymbol{\alpha}_j^t)}\|$ *for some* $t$, *then* $I_+(\boldsymbol{\alpha}_j^t) = \emptyset$ *or* $I_+(\boldsymbol{\alpha}_j^t) = [n]$.

(iv) $\boldsymbol{\alpha}_j^t \neq \mathbf{0}$ *for all* $t \in [0, \infty)$.

*Proof.* For part (i), only the left-to-right implication requires a proof. Let $\boldsymbol{\gamma}_{I_+(\boldsymbol{\alpha}_j^t)} = \frac{1}{n}\sum_i y_i \boldsymbol{x}_i$, where the summation is over $i \in I_+(\boldsymbol{\alpha}_j^t)$; then $\|\boldsymbol{\gamma}_{I_+(\boldsymbol{\alpha}_j^t)}\|^2 = \frac{1}{n^2}\sum_{i,k} y_i y_k \boldsymbol{x}_i^\top \boldsymbol{x}_k$. Since all training points are positively correlated, and all coefficients $y_i$ that are present in the sum are positive, $\|\boldsymbol{\gamma}_{I_+(\boldsymbol{\alpha}_j^t)}\|^2 > 0$ unless the sum contains no terms; that is, unless $I_+(\boldsymbol{\alpha}_j^t) = \emptyset$.

For part (ii), observe that $\boldsymbol{\gamma}_{I_+(\boldsymbol{\alpha}_j^t)}$ is a positive linear combination of $\boldsymbol{x}_k$ for $k \in I_+(\boldsymbol{\alpha}_j^t)$, and every such $\boldsymbol{x}_k$ forms an acute angle with $\boldsymbol{\alpha}_j^t$; thus, $(\boldsymbol{\alpha}_j^t)^\top \boldsymbol{\gamma}_{I_+(\boldsymbol{\alpha}_j^t)} > 0$ unless $\boldsymbol{\gamma}_{I_+(\boldsymbol{\alpha}_j^t)} = \mathbf{0}$.

For part (iii), we again use the definition of $\boldsymbol{\gamma}_{I_+(\boldsymbol{\alpha}_j^t)}$ and the fact that $\boldsymbol{x}_i^\top \boldsymbol{x}_k > 0$ for all $i, k \in [n]$. If $I_+(\boldsymbol{\alpha}_j^t) \neq \emptyset$, then $\boldsymbol{\alpha}_j^t \neq \boldsymbol{0}$ and $\boldsymbol{x}_i^\top \boldsymbol{\alpha}_j^t = \boldsymbol{x}_i^\top \boldsymbol{\gamma}_{I_+(\boldsymbol{\alpha}_j^t)} \cdot \|\boldsymbol{\alpha}_j^t\|/\|\boldsymbol{\gamma}_{I_+(\boldsymbol{\alpha}_j^t)}\| > 0$ for all $i$, and thus $I_+(\boldsymbol{\alpha}_j^t) = [n]$.

For part (iv), note that

$$\frac{\mathrm{d}\|\boldsymbol{\alpha}_j^t\|^2}{\mathrm{d}t} = 2(\boldsymbol{\alpha}_j^t)^\top \cdot s_j \|\boldsymbol{\alpha}_j^t\| \boldsymbol{\gamma}_{I_+(\boldsymbol{\alpha}_j^t)} = \|\boldsymbol{\alpha}_j^t\|^2 \|\boldsymbol{\gamma}_{I_+(\boldsymbol{\alpha}_j^t)}\| \cdot 2 s_j \cos \measuredangle(\boldsymbol{\alpha}_j^t, \boldsymbol{\gamma}_{I_+(\boldsymbol{\alpha}_j^t)}) \,,$$

where the final product is 0 if $\boldsymbol{\alpha}_j^t = \boldsymbol{0}$. It follows that $\mathrm{d}\|\boldsymbol{\alpha}_j^t\|^2/\mathrm{d}t \geq -2\|\boldsymbol{\alpha}_j^t\|^2 \|\boldsymbol{\gamma}_{[n]}\|$ for all $t \in [0, \infty)$. By Grönwall's inequality, $\|\boldsymbol{\alpha}_j^t\|^2 \geq \|\boldsymbol{\alpha}_j^0\|^2 \, \mathrm{e}^{-2\|\boldsymbol{\gamma}_{[n]}\| t} > 0$ for all $t$, and it remains to recall that $\boldsymbol{\alpha}_j^0 = \boldsymbol{z}_j$ was initially chosen to be non-zero. $\qquad\square$

The next proposition establishes continuity and monotonicity properties, assuming that the trajectory $\boldsymbol{\alpha}_j^t$ crosses between regions of continuity of the right-hand side of the ODE finitely many times.

**Proposition 12.** *Let* $j \in J_+ \cup J_-$. *Let* $T > 0$ *be such that* $I_0(\boldsymbol{\alpha}_j^t)$ *is only non-empty for finitely many* $t \in [0, T]$. *Then the following statements hold.*

(i) *The map* $t \mapsto (\boldsymbol{\alpha}_j^t)^\top \boldsymbol{\gamma}_{I_+(\boldsymbol{\alpha}_j^t)}$ *from* $[0, T]$ *to* $\mathbb{R}$ *is continuous.*

(ii) *If* $\boldsymbol{\gamma}_{I_+(\boldsymbol{\alpha}_j^t)} = \boldsymbol{0}$ *for some* $t \in [0, T]$, *then* $t = T$.

(iii) $\boldsymbol{\gamma}_{I_+(\boldsymbol{\alpha}_j^t)} = \begin{cases} \lim_{\xi \to t^-} \boldsymbol{\gamma}_{I_+(\boldsymbol{\alpha}_j^\xi)} & \text{if } s_j = +1 \text{ and } t \in (0, T], \\ \lim_{\xi \to t^+} \boldsymbol{\gamma}_{I_+(\boldsymbol{\alpha}_j^\xi)} & \text{if } s_j = -1 \text{ and } t \in [0, T). \end{cases}$

(iv) *For each* $i \in [n]$, $s_j(\boldsymbol{\alpha}_j^t)^\top \boldsymbol{x}_i$ *is a strictly increasing function of* $t$ *on* $[0, T]$. *Furthermore, for all* $i \in [n]$, *whenever* $0 \leq t_1 < t_2 \leq T$:

- *if* $s_j = +1$ *and* $i \in I_0(\boldsymbol{\alpha}_j^{t_1}) \cup I_+(\boldsymbol{\alpha}_j^{t_1})$, *then* $i \in I_+(\boldsymbol{\alpha}_j^{t_2})$;
- *if* $s_j = -1$ *and* $i \notin I_+(\boldsymbol{\alpha}_j^{t_1})$, *then* $i \notin I_0(\boldsymbol{\alpha}_j^{t_2}) \cup I_+(\boldsymbol{\alpha}_j^{t_2})$.

(v) *Let* $t_0 \in [0, T)$ *be either* 0 *or a point of discontinuity of* $\boldsymbol{\gamma}_{I_+(\boldsymbol{\alpha}_j^t)}$. *If* $s_j = -1$, *then* $\lim_{t \to t_0^+} \cos \varphi_j^t \neq 1$.

We remark that the assumption of Proposition 12 that the set $\{t \geq 0 \mid I_0(\boldsymbol{\alpha}_j^t) \neq \emptyset\}$ has a finite intersection with $[0, T]$ will be justified in Proposition 14, in the following sense: we will show that the assumption holds for every segment $[0, T]$ such that $\boldsymbol{\gamma}_{I_+(\boldsymbol{\alpha}_j^t)} \neq \boldsymbol{0}$ for all $t \in [0, T)$.

*Proof.* For part (i), since the map $t \mapsto \boldsymbol{\alpha}_j^t$ from $[0, \infty)$ to $\mathbb{R}^d$ is continuous by definition, it suffices to consider points of discontinuity of $\boldsymbol{\gamma}_{I_+(\boldsymbol{\alpha}_j^t)}$. The set $I_0(\boldsymbol{\alpha}_j^{t_0})$ is necessarily non-empty at every such point $t_0 \in [0, T]$. Let us fix such a $t_0$; then one-sided limits of $\boldsymbol{\gamma}_{I_+(\boldsymbol{\alpha}_j^t)}$ as $t \to t_0^+$ and $t \to t_0^-$ exist (excepting $t \to 0^-$ and $t \to T^+$, which we do not consider). Here we used the assumption from the statement of the proposition: in a small enough neighbourhood of $t_0$ there are no other points $t$ for which $I_0(\boldsymbol{\alpha}_j^t)$ is non-empty; this assumption could have been avoided if necessary. The value of $(\boldsymbol{\alpha}_j^{t_0})^\top \boldsymbol{\gamma}_{I_+(\boldsymbol{\alpha}_j^{t_0})}$ may only differ from (either of) the one-sided limits of $(\boldsymbol{\alpha}_j^t)^\top \boldsymbol{\gamma}_{I_+(\boldsymbol{\alpha}_j^t)}$ as $t \to t_0^\pm$ by the summands $(\boldsymbol{\alpha}_j^{t_0})^\top \cdot \frac{1}{n} y_i \boldsymbol{x}_i$ with $i \in I_0(\boldsymbol{\alpha}_j^{t_0})$. But every such summand is equal to 0 anyway by the definition of $I_0$.

Before establishing part (ii), we first prove a weaker version of part (iv), namely non-strict monotonicity: for each $i \in [n]$, $s_j(\boldsymbol{\alpha}_j^t)^\top \boldsymbol{x}_i$ is a non-decreasing function of $t$ on $[0, T]$. To this end, we consider the derivatives

$$\frac{\mathrm{d}(\boldsymbol{\alpha}_j^t)^\top \boldsymbol{x}_i}{\mathrm{d}t} = s_j \|\boldsymbol{\alpha}_j^t\| \cdot (\boldsymbol{\gamma}_{I_+(\boldsymbol{\alpha}_j^t)})^\top \boldsymbol{x}_i$$

inside each interval $(t_1, t_2)$ on which $\boldsymbol{\gamma}_{I_+(\boldsymbol{\alpha}_j^t)}$ is continuous. Observe that $(\boldsymbol{\gamma}_{I_+(\boldsymbol{\alpha}_j^t)})^\top \boldsymbol{x}_i \geq 0$ for each $i \in [n]$, since training points are pairwise positively correlated. Therefore, $s_j(\boldsymbol{\alpha}_j^t)^\top \boldsymbol{x}_i$ is

non-decreasing on $(t_1, t_2)$. Since $\boldsymbol{\alpha}_j^t$ is continuous, and there are only finitely many points at which $\boldsymbol{\gamma}_{I_+(\boldsymbol{\alpha}_j^t)}$ is discontinuous, $s_j(\boldsymbol{\alpha}_j^t)^\top \boldsymbol{x}_i$ is also non-decreasing on $[0, T]$.

We now establish part (ii). If $\boldsymbol{\gamma}_{I_+(\boldsymbol{\alpha}_j^t)} = \boldsymbol{0}$ for some $t \in [0, T]$, then

$$t_0 := \inf\{t \in [0, T] \mid I_+(\boldsymbol{\alpha}_j^t) = \emptyset\}$$

is well-defined by Proposition 11, part (i). Observe that if $I_+(\boldsymbol{\alpha}_j^{t_0})$ is non-empty, then $I_+(\boldsymbol{\alpha}_j^t)$ is non-empty for all $t$ in a small neighbourhood of $t_0$ by the continuity of $\boldsymbol{\alpha}_j^t$. Therefore, $I_+(\boldsymbol{\alpha}_j^{t_0}) = \emptyset$ by our choice of $t_0$. At the same time, notice that $0 < t_0$ because we assume $j \in J_+ \cup J_-$. Furthermore, for all $t' < t_0$ there is some $i \in [n]$ with $(\boldsymbol{\alpha}_j^{t'})^\top \boldsymbol{x}_i > 0$. By compactness and by the (non-strict) monotonicity property proved above, there exists a single $i \in [n]$ such that $(\boldsymbol{\alpha}_j^{t'})^\top \boldsymbol{x}_i > 0$ for all $t' < t_0$. Once again by continuity, we have $(\boldsymbol{\alpha}_j^{t_0})^\top \boldsymbol{x}_i \geq 0$. Since $I_+(\boldsymbol{\alpha}_j^{t_0}) = \emptyset$, we conclude that $i \in I_0(\boldsymbol{\alpha}_j^{t_0})$.

We have shown that, assuming $\boldsymbol{\gamma}_{I_+(\boldsymbol{\alpha}_j^t)} = \boldsymbol{0}$ for some $t \in [0, T]$, the existence of $t_0 \in (0, t]$ such that $I_+(\boldsymbol{\alpha}_j^{t_0}) = \emptyset$ and $I_0(\boldsymbol{\alpha}_j^{t_0}) \neq \emptyset$. It follows that $\boldsymbol{\alpha}_j^t = \boldsymbol{\alpha}_j^{t_0}$ and $I_0(\boldsymbol{\alpha}_j^t) = I_0(\boldsymbol{\alpha}_j^{t_0})$ for all $t \geq t_0$. Under the assumptions of the proposition, this means $t_0 = t = T$. This completes the proof of part (ii).

We now proceed to part (iv), proving *strict* monotonicity of each $s_j(\boldsymbol{\alpha}_j^t)^\top \boldsymbol{x}_i$. By part (ii), for every interval $(t_1, t_2)$ on which $\boldsymbol{\gamma}_{I_+(\boldsymbol{\alpha}_j^t)}$ is continuous, we have in fact $\boldsymbol{\gamma}_{I_+(\boldsymbol{\alpha}_j^t)} \neq \boldsymbol{0}$. Therefore, $(\boldsymbol{\gamma}_{I_+(\boldsymbol{\alpha}_j^t)})^\top \boldsymbol{x}_i > 0$, because training points are pairwise positively correlated, and each $s_j(\boldsymbol{\alpha}_j^t)^\top \boldsymbol{x}_i$ strictly increases on $(t_1, t_2)$. Since $\boldsymbol{\alpha}_j^t$ is continuous, and there are only finitely many points at which $\boldsymbol{\gamma}_{I_+(\boldsymbol{\alpha}_j^t)}$ is discontinuous, $s_j(\boldsymbol{\alpha}_j^t)^\top \boldsymbol{x}_i$ is also strictly increasing on $[0, T]$. The two remaining implications in the statement of part (iv) follow.

Part (iii) is a consequence of part (iv).

To establish part (v), firstly observe that, by part (iii) and by the continuity of $\boldsymbol{\alpha}_j^t$, the function $\cos \varphi_j^t$ is in fact right-continuous at $t_0$: the one-sided limit in question exists and is equal to $\cos \varphi_j^{t_0}$. By Proposition 11, part (iii), if $\cos \varphi_j^{t_0} = 1$, then $I_+(\boldsymbol{\alpha}_j^{t_0}) = [n]$. We now consider two cases. If $t_0 = 0$, then $\boldsymbol{\alpha}_j^{t_0} = \boldsymbol{z}_j$, but this is ruled out by Assumption 1, part (ii). Otherwise $t_0$ is a point of discontinuity of $\boldsymbol{\gamma}_{I_+(\boldsymbol{\alpha}_j^t)}$. Since $\boldsymbol{\alpha}_j^t$ is a continuous function of $t$, the set $I_0(\boldsymbol{\alpha}_j^{t_0})$ is necessarily non-empty, but this is also a contradiction because this set must be disjoint from $I_+(\boldsymbol{\alpha}_j^{t_0})$. $\qquad\square$

The following proposition strengthens the previously proved statement (Proposition 11, part (iv)) that $\boldsymbol{\alpha}_j^t \neq \boldsymbol{0}$, bounding $\|\boldsymbol{\alpha}_j^t\|$ from below. In the sequel, we will require analytic expressions for two related quantities:

$$\frac{\mathrm{d}\|\boldsymbol{\alpha}_j^t\|}{\mathrm{d}t} = \frac{\mathrm{d}\sqrt{\|\boldsymbol{\alpha}_j^t\|^2}}{\mathrm{d}t} = \frac{1}{2\|\boldsymbol{\alpha}_j^t\|} \cdot \frac{\mathrm{d}\|\boldsymbol{\alpha}_j^t\|^2}{\mathrm{d}t} = \frac{2(\boldsymbol{\alpha}_j^t)^\top \cdot s_j\|\boldsymbol{\alpha}_j^t\|\boldsymbol{\gamma}_{I_+(\boldsymbol{\alpha}_j^t)}}{2\|\boldsymbol{\alpha}_j^t\|} = s_j(\boldsymbol{\alpha}_j^t)^\top \boldsymbol{\gamma}_{I_+(\boldsymbol{\alpha}_j^t)}\,,$$

$$\frac{\mathrm{d}}{\mathrm{d}t}\left(\frac{\boldsymbol{\alpha}_j^t}{\|\boldsymbol{\alpha}_j^t\|}\right) = \frac{\frac{\mathrm{d}\boldsymbol{\alpha}_j^t}{\mathrm{d}t} \cdot \|\boldsymbol{\alpha}_j^t\| - \frac{\mathrm{d}\|\boldsymbol{\alpha}_j^t\|}{\mathrm{d}t} \cdot \boldsymbol{\alpha}_j^t}{\|\boldsymbol{\alpha}_j^t\|^2} = \frac{s_j\|\boldsymbol{\alpha}_j^t\|^2 \boldsymbol{\gamma}_{I_+(\boldsymbol{\alpha}_j^t)} - s_j \cdot (\boldsymbol{\alpha}_j^t)^\top \boldsymbol{\gamma}_{I_+(\boldsymbol{\alpha}_j^t)} \cdot \boldsymbol{\alpha}_j^t}{\|\boldsymbol{\alpha}_j^t\|^2}\,.$$

**Proposition 13.** *Let $0 \leq t_1 < t_2$ be such that $I_0(\boldsymbol{\alpha}_j^t) = \emptyset$ for all $t \in (t_1, t_2)$ and $I_0(\boldsymbol{\alpha}_j^t) \neq \emptyset$ for at most finitely many $t \in [0, t_1]$.*

(i) *If $t_1$ is either $0$ or a point of discontinuity of $\boldsymbol{\gamma}_{I_+(\boldsymbol{\alpha}_j^t)}$, then there is a constant $\mu > 0$, only dependent on $t_1$ but not on $t$ or $t_2$, such that $\|\boldsymbol{\alpha}_j^t\| \geq \mu$ for all $t \in (t_1, t_2)$.*

(ii) *Suppose $\boldsymbol{\gamma}_{I_+(\boldsymbol{\alpha}_j^t)} = \boldsymbol{\gamma}$ for all $t \in (t_1, t_2)$, and denote $\varphi_j^{t_1^+} := \angle(\boldsymbol{\alpha}_j^{t_1}, \boldsymbol{\gamma}) = \arccos\big((\overline{\boldsymbol{\alpha}}_j^{t_1})^\top \overline{\boldsymbol{\gamma}}\big)$. Then for all $t \in (t_1, t_2)$ we have*

$$\|\boldsymbol{\alpha}_j^t\| = \tfrac{1}{2} \cdot (1 + s_j \cos \varphi_j^{t_1^+}) \cdot \|\boldsymbol{\alpha}_j^{t_1}\| \cdot \mathrm{e}^{\|\boldsymbol{\gamma}\|(t-t_1)} +$$

$$\tfrac{1}{2} \cdot (1 - s_j \cos \varphi_j^{t_1^+}) \cdot \|\boldsymbol{\alpha}_j^{t_1}\| \cdot \mathrm{e}^{-\|\boldsymbol{\gamma}\|(t-t_1)}\,.$$

*Proof.* We establish part (ii) first. The functions $\|\boldsymbol{\alpha}_j^t\|$ and $(\boldsymbol{\alpha}_j^t)^\top \boldsymbol{\gamma}$ satisfy, for all $t \in (t_1, t_2)$, the following system of ordinary differential equations:

$$\frac{\mathrm{d}\|\boldsymbol{\alpha}_j^t\|}{\mathrm{d}t} = s_j \cdot (\boldsymbol{\alpha}_j^t)^\top \boldsymbol{\gamma}\,, \qquad \frac{\mathrm{d}\big((\boldsymbol{\alpha}_j^t)^\top \boldsymbol{\gamma}\big)}{\mathrm{d}t} = s_j \|\boldsymbol{\gamma}\|^2 \|\boldsymbol{\alpha}_j^t\|\,.$$

By the standard theory of linear ODEs, the solution can be sought in the form

$$\|\boldsymbol{\alpha}_j^t\| = c_1 \,\mathrm{e}^{\|\boldsymbol{\gamma}\|t} + c_2 \,\mathrm{e}^{-\|\boldsymbol{\gamma}\|t}\,,$$
$$(\boldsymbol{\alpha}_j^t)^\top \boldsymbol{\gamma} = (c_1 \,\mathrm{e}^{\|\boldsymbol{\gamma}\|t} - c_2 \,\mathrm{e}^{-\|\boldsymbol{\gamma}\|t}) \cdot s_j \|\boldsymbol{\gamma}\|\,.$$

The constants $c_1$ and $c_2$ are chosen based on the initial conditions as $t \to t_1^+$, i.e., they should satisfy the following system of linear equations:

$$\begin{bmatrix} \mathrm{e}^{\|\boldsymbol{\gamma}\|t_1} & \mathrm{e}^{-\|\boldsymbol{\gamma}\|t_1} \\ \mathrm{e}^{\|\boldsymbol{\gamma}\|t_1} & -\mathrm{e}^{-\|\boldsymbol{\gamma}\|t_1} \end{bmatrix} \begin{bmatrix} c_1 \\ c_2 \end{bmatrix} = \begin{bmatrix} \|\boldsymbol{\alpha}_j^{t_1}\| \\ s_j(\boldsymbol{\alpha}_j^{t_1})^\top \overline{\boldsymbol{\gamma}} \end{bmatrix}\,.$$

Here we rely on the continuity of $t \mapsto \boldsymbol{\alpha}_j^t$. We obtain

$$\|\boldsymbol{\alpha}_j^t\| = \tfrac{1}{2} \cdot \Big(\|\boldsymbol{\alpha}_j^{t_1}\| + s_j(\boldsymbol{\alpha}_j^{t_1})^\top \overline{\boldsymbol{\gamma}}\Big) \cdot \mathrm{e}^{\|\boldsymbol{\gamma}\|(t-t_1)} +$$
$$\tfrac{1}{2} \cdot \Big(\|\boldsymbol{\alpha}_j^{t_1}\| - s_j(\boldsymbol{\alpha}_j^{t_1})^\top \overline{\boldsymbol{\gamma}}\Big) \cdot \mathrm{e}^{-\|\boldsymbol{\gamma}\|(t-t_1)}\,,$$

which can then be rewritten in the required form.

We now establish part (i). By the continuity of dot products $(\boldsymbol{\alpha}_j^t)^\top \boldsymbol{x}_i$, there exists a vector $\boldsymbol{\gamma} \in \mathbb{R}^d$ such that $\boldsymbol{\gamma}_{I_+(\boldsymbol{\alpha}_j^t)} = \boldsymbol{\gamma}$ for all $t \in (t_1, t_2)$. In the degenerate case, $\boldsymbol{\gamma} = \boldsymbol{0}$, we have $\boldsymbol{\alpha}_j^t = \boldsymbol{\alpha}_j^{t_1}$ for all $t \in (t_1, t_2)$. Hence, we can choose $\mu := \|\boldsymbol{\alpha}_j^{t_1}\|$, which is positive by Proposition 11, part (iv). We will therefore assume $\boldsymbol{\gamma} \neq \boldsymbol{0}$. The idea is to rely on Proposition 13, part (ii), noting that $\cos\varphi_j^{t_1^+} = \lim_{t \to t_1^+}(\overline{\boldsymbol{\alpha}}_j^t)^\top \overline{\boldsymbol{\gamma}} = (\overline{\boldsymbol{\alpha}}_j^{t_1})^\top \overline{\boldsymbol{\gamma}} \geq 0$, by Proposition 11, part (ii), and by continuity of $\boldsymbol{\alpha}_j^t$. So, if $s_j = +1$, then clearly $\|\boldsymbol{\alpha}_j^t\| \geq \tfrac{1}{2}\|\boldsymbol{\alpha}_j^{t_1}\| =: \mu$. If $s_j = -1$, then, again dropping the second term in the closed-form expression for $\|\boldsymbol{\alpha}_j^t\|$, we obtain $\|\boldsymbol{\alpha}_j^t\| \geq \tfrac{1}{2}\|\boldsymbol{\alpha}_j^{t_1}\| \cdot (1 - \lim_{t \to t_1^+} \cos\varphi_j^t) \cdot 1$. By Proposition 12, part (v), $\lim_{t \to t_1^+} \cos\varphi_j^t < 1$, which completes the proof. $\square$

**Proposition 14.** *For all* $j \in J_+ \cup J_-$ *there exist a unique enumeration* $i_j^1, \ldots, i_j^{n_j}$ *of* $I_{-s_j}(\boldsymbol{z}_j)$ *and unique* $\tau_j^1, \ldots, \tau_j^{n_j} \in [0, \infty)$ *such that for all* $\ell \in [n_j]$ *the following hold, where* $\tau_j^0 := 0$, $\varphi_j^{(\ell-1)^+} := \lim_{t \to (\tau_j^{\ell-1})^+} \varphi_j^t$, $\varphi_j^{\ell^-} := \lim_{t \to (\tau_j^\ell)^-} \varphi_j^t$, *and*

$$I_j^\ell := \begin{cases} I_+(\boldsymbol{z}_j) \cup \{i_j^1, \ldots, i_j^{\ell-1}\} & \text{if } s_j = 1, \\ I_+(\boldsymbol{z}_j) \setminus \{i_j^1, \ldots, i_j^{\ell-1}\} & \text{if } s_j = -1 \text{:}\end{cases}$$

(i) $i_j^\ell = \arg\min\left\{-s_j\left(\overline{\boldsymbol{\alpha}}_j^{\tau_j^{\ell-1}}\right)^\top \overline{\boldsymbol{x}}_i \Big/ \overline{\boldsymbol{\gamma}}_{I_j^\ell}^\top \overline{\boldsymbol{x}}_i \ \Big| \ i \in I_{-s_j}(\boldsymbol{z}_j) \setminus \{i_j^1, \ldots, i_j^{\ell-1}\}\right\}$;

(ii) $\sin\left(\varphi_j^{(\ell-1)^+} - \varphi_j^{\ell^-}\right) \Big/ \sin\varphi_j^{\ell^-} = -\left(\overline{\boldsymbol{\alpha}}_j^{\tau_j^{\ell-1}}\right)^\top \overline{\boldsymbol{x}}_{i_j^\ell} \Big/ \overline{\boldsymbol{\gamma}}_{I_j^\ell}^\top \overline{\boldsymbol{x}}_{i_j^\ell}$;

(iii) $\tau_j^{\ell-1} < \tau_j^\ell$;

(iv) $I_+(\boldsymbol{\alpha}_j^t) = I_j^\ell$ for all $t \in (\tau_j^{\ell-1}, \tau_j^\ell)$;

(v) $I_0(\boldsymbol{\alpha}_j^t) = \emptyset$ for all $t \in (\tau_j^{\ell-1}, \tau_j^\ell)$, and $I_0\left(\boldsymbol{\alpha}_j^{\tau_j^\ell}\right) = \{i_j^\ell\}$;

(vi) $\cos\varphi_j^t = \tanh\left(\mathrm{artanh}\cos\varphi_j^{(\ell-1)^+} + s_j \left\|\boldsymbol{\gamma}_{I_j^\ell}\right\|(t - \tau_j^{\ell-1})\right)$ for all $t \in (\tau_j^{\ell-1}, \tau_j^\ell)$;

(vii) if $\ell < n_j$ then $\left\|\boldsymbol{\gamma}_{I_j^\ell}\right\|\cos\varphi_j^{\ell^-} = \left\|\boldsymbol{\gamma}_{I_j^{\ell+1}}\right\|\cos\varphi_j^{\ell^+}$;

*(viii)* if $\ell = n_j$ and $s_j = 1$ then $\left\| \boldsymbol{\gamma}_{I_j^\ell} \right\| \cos \varphi_j^{\ell^-} = \|\boldsymbol{\gamma}_{[n]}\| \cos \lim_{t \to (\tau_j^\ell)^+} \varphi_j^t$;

*(ix)* if $\ell = n_j$ and $s_j = -1$ then $\varphi_j^{\ell^-} = \pi/2$;

*(x)* $\overline{\boldsymbol{\alpha}}_j^t = \left( \sin(\varphi_j^t) \, \overline{\boldsymbol{\alpha}}_j^{\tau_j^{\ell-1}} + \sin\left( \varphi_j^{(\ell-1)^+} - \varphi_j^t \right) \overline{\boldsymbol{\gamma}}_{I_j^\ell} \right) \Big/ \sin \varphi_j^{(\ell-1)^+}$ for all $t \in (\tau_j^{\ell-1}, \tau_j^\ell)$;

*(xi)* $s_j \, \mathrm{d} \, \overline{\boldsymbol{\alpha}}_j^{t\top} \, \overline{\boldsymbol{x}}_i / \mathrm{d}t \geq \boldsymbol{\gamma}_{I_j^\ell}^\top \overline{\boldsymbol{x}}_i$ for all $i \notin I_j^\ell$ and all $t \in (\tau_j^{\ell-1}, \tau_j^\ell)$;

*(xii)* if $s_j = -1$, then $\mathrm{d} \, \overline{\boldsymbol{\alpha}}_j^{t\top} \, \overline{\boldsymbol{x}}_{i_j^\ell} / \mathrm{d}t < 0$ and $\mathrm{d}^2 \, \overline{\boldsymbol{\alpha}}_j^{t\top} \, \overline{\boldsymbol{x}}_{i_j^\ell} / \mathrm{d}t^2 < 0$ for all $t \in (\tau_j^{\ell-1}, \tau_j^\ell)$.

*Proof.* Throughout, we let $j \in J_+ \cup J_-$ stay fixed but arbitrary.

*Parts (i), (iii), (iv), and (v).* We establish these parts by a common inductive argument. The induction is on $\ell$, ranging from 1 to $n_j$. We do not separate the base case. We first notice that $\mathrm{d}((\boldsymbol{\alpha}_j^t)^\top \overline{\boldsymbol{x}}_i)/\mathrm{d}t = s_j \|\boldsymbol{\alpha}_j^t\| \cdot (\boldsymbol{\gamma}_{I_+(\boldsymbol{\alpha}_j^t)})^\top \overline{\boldsymbol{x}}_i$. In particular, for any fixed $i \in [n]$ we can write

$$s_j (\boldsymbol{\alpha}_j^\xi)^\top \overline{\boldsymbol{x}}_i = s_j \left( \boldsymbol{\alpha}_j^{\tau_j^{\ell-1}} \right)^\top \overline{\boldsymbol{x}}_i + \int_{\tau_j^{\ell-1}}^\xi \|\boldsymbol{\alpha}_j^t\| \cdot (\boldsymbol{\gamma}_{I_+(\boldsymbol{\alpha}_j^t)})^\top \overline{\boldsymbol{x}}_i \, \mathrm{d}t \; . \tag{1}$$

This equality holds for any $\xi > \tau_j^{\ell-1}$ as long as the integral on the right-hand side is well-defined; we first need to justify the existence of an appropriate $\xi > \tau_j^{\ell-1}$. This is not automatic because the function $\boldsymbol{\gamma}_{I_+(\boldsymbol{\alpha}_j^t)}$ is not assumed continuous. We consider two cases.

If $\ell = 1$, then $I_0 \left( \boldsymbol{\alpha}_j^{\tau_j^{\ell-1}} \right) = \emptyset$ by Assumption 1, part (ii), and thus $\left( \boldsymbol{\alpha}_j^{\tau_j^{\ell-1}} \right)^\top \overline{\boldsymbol{x}}_i$ are all non-zero; by continuity of $\boldsymbol{\alpha}_j^t$, this holds in a sufficiently small right-neighbourhood of $\tau_j^{\ell-1}$. Thus, if the set $\{t > \tau_j^{\ell-1} \mid I_0(\boldsymbol{\alpha}_j^t) \neq \emptyset\}$ is non-empty, its infimum is strictly greater than $\tau_j^{\ell-1}$, and we can pick this infimum as $\xi$. In fact, we will show below that the set cannot be empty, but for now let us say that it is safe to pick any $\xi > \tau_j^{\ell-1}$ in this hypothetical situation.

Now suppose $\ell > 1$, then by the inductive hypothesis $I_0 \left( \boldsymbol{\alpha}_j^{\tau_j^{\ell-1}} \right) = \{i_j^{\ell-1}\}$. For all $i \neq i_j^{\ell-1}$, we have $\left( \boldsymbol{\alpha}_j^{\tau_j^{\ell-1}} \right)^\top \overline{\boldsymbol{x}}_i \neq 0$ and thus each $(\boldsymbol{\alpha}_j^t)^\top \overline{\boldsymbol{x}}_i$ maintains the sign in some right-neighbourhood of $\tau_j^{\ell-1}$. For $i = i_j^{\ell-1}$, rewrite Equation 1 as

$$s_j (\boldsymbol{\alpha}_j^\xi)^\top \overline{\boldsymbol{x}}_{i_j^{\ell-1}} = \int_{\tau_j^{\ell-1}}^\xi \|\boldsymbol{\alpha}_j^t\| \cdot (\boldsymbol{\gamma}_{I_+(\boldsymbol{\alpha}_j^t)})^\top \overline{\boldsymbol{x}}_{i_j^{\ell-1}} \, \mathrm{d}t \; , \tag{2}$$

where the integrand is non-negative. Therefore, the function $s_j (\boldsymbol{\alpha}_j^\xi)^\top \overline{\boldsymbol{x}}_{i_j^{\ell-1}}$ is non-negative for all $\xi > \tau_j^{\ell-1}$ and moreover is non-decreasing. Consider the set $I_j^\ell$ defined in the statement of the proposition; we have $\emptyset \neq I_j^\ell = I_+(\boldsymbol{\alpha}_j^t)$ and $(\boldsymbol{\gamma}_{I_+(\boldsymbol{\alpha}_j^t)})^\top \overline{\boldsymbol{x}}_{i_j^{\ell-1}} > 0$ for all $t$ greater than $\tau_j^{\ell-1}$ in a small neighbourhood of $\tau_j^{\ell-1}$. (Note that $\boldsymbol{\gamma}_{I_+(\boldsymbol{\alpha}_j^t)} \neq \mathbf{0}$ by Proposition 11, part (i).) So we can replace $\boldsymbol{\gamma}_{I_+(\boldsymbol{\alpha}_j^t)}$ with $\boldsymbol{\gamma}_{I_j^\ell}$ in Equation 2 if $\xi$ is close enough to $\tau_j^{\ell-1}$; we have now shown the existence of a $\xi > \tau_j^{\ell-1}$ such that $I_+(\boldsymbol{\alpha}_j^t) = \boldsymbol{\gamma}_{I_j^\ell}$ for all $t \in (\tau_j^{\ell-1}, \xi)$. (In fact, we can again choose $\xi := \inf\{t > \tau_j^{\ell-1} \mid I_0(\boldsymbol{\alpha}_j^t) \neq \emptyset\}$.)

Having found an appropriate $\xi$, let us observe that, by the inductive hypothesis (part (v)) and by the choice of $\xi$, the set $I_0(\boldsymbol{\alpha}_j^t)$ is only non-empty for finitely many time points $t \in [0, \xi]$. Let us consider

$$I' := \left\{ i \in [n] \;\middle|\; s_j \left( \boldsymbol{\alpha}_j^{\tau_j^{\ell-1}} \right)^\top \overline{\boldsymbol{x}}_i < 0 \right\} = I_{-s_j}(\boldsymbol{z}_j) \setminus \{i_j^1, \ldots, i_j^{\ell-1}\} \; .$$

Recalling that all training points are positively correlated, we see that $(\boldsymbol{\gamma}_{I_j^\ell})^\top \overline{\boldsymbol{x}}_i > 0$. By Proposition 13, part (i), the integrand in Equation 1 is lower-bounded by $\mu \cdot (\boldsymbol{\gamma}_{I_j^\ell})^\top \overline{\boldsymbol{x}}_i$ for all $t$. Therefore, for each $i \in I'$ the expression on the right-hand side of Equation 1 tends to $+\infty$ if we let, formally, $\xi \to +\infty$. Since for $\xi = \tau_j^{\ell-1}$ each of the right-hand sides is negative if $i \in I'$, there exists some $\xi > \tau_j^{\ell-1}$ and an $i \in I'$ for which the left-hand side, $(\boldsymbol{\alpha}_j^\xi)^\top \overline{\boldsymbol{x}}_i$, becomes 0. Rewriting Equation 1 as $\int_{\tau_j^{\ell-1}}^{\xi} \|\boldsymbol{\alpha}_j^t\| \, \mathrm{d}t = r_i$, where $r_i := -s_j \left(\boldsymbol{\alpha}_j^{\tau_j^{\ell-1}}\right)^\top \overline{\boldsymbol{x}}_i \Big/ (\boldsymbol{\gamma}_{I_j^\ell})^\top \overline{\boldsymbol{x}}_i > 0$, we observe that the integral on the left-hand side does not depend on $i$. Thus, the smallest $\xi > \tau_j^{\ell-1}$ for which the integral is equal to $r_i$ for some $i \in I'$ is the earliest time point after $\tau_j^{\ell-1}$ at which the set $I_0(\boldsymbol{\alpha}_j^\xi)$ becomes non-empty. This value of $\xi$ is then, by definition, $\tau_j^\ell$. Since for $\xi = \tau_j^{\ell-1}$ the integral is zero and since $r_i > 0$ for all $i \in I'$, we also have $I_0(\boldsymbol{\alpha}_j^\xi) = \{i_j^\ell\}$ where $i = i_j^\ell$ is the index of the smallest $r_i$ among $i \in I'$; this $i$ is unique by Assumption 1, part (iv).

To complete the proof of the inductive step for part (i), it remains to note that rescaling each $r_i$ by a factor of $\left\|\boldsymbol{\gamma}_{I_j^\ell}\right\| \Big/ \left\|\boldsymbol{\alpha}_j^{\tau_j^{\ell-1}}\right\|$ does not change the $\arg\min$.

Notice that, for the current value of $\ell$ we have also justified the inequality $\tau_j^{\ell-1} < \tau_j^\ell$ of part (iii), as well as equalities $I_+(\boldsymbol{\alpha}_j^t) = I_j^\ell$ and $I_0(\boldsymbol{\alpha}_j^t) = \emptyset$ for all $t \in (\tau_j^{\ell-1}, \tau_j^\ell)$, and $I_0\left(\boldsymbol{\alpha}_j^{\tau_j^\ell}\right) = \{i_j^\ell\}$, which together comprise parts (iv) and (v). This completes the inductive argument, proving parts (i), (iii), (iv), and (v).

*Intermediate summary.* We have already established uniqueness of the enumeration $i_j^1, \dots, i_j^{n_j}$ and time points $\tau_j^1, \dots, \tau_j^{n_j}$: part (iii) requires that the latter be sorted in the ascending order, and our argument for the choice of $\tau_j^\ell$ makes it clear that there is always only one possibility, if we want to require (as parts (iv) and (v) do) that $I_+(\boldsymbol{\alpha}_j^t)$ remain constant in between $\tau_j^{\ell-1}$ and $\tau_j^\ell$, and $I_0\left(\boldsymbol{\alpha}_j^{\tau_j^\ell}\right)$ be non-empty. In addition, we now know that the assumption of Proposition 12 holds for every time segment $[0, T]$ such that $\boldsymbol{\gamma}_{I_+(\boldsymbol{\alpha}_j^t)} \neq \boldsymbol{0}$ for all $t \in [0, T)$; and in particular up to $T = \tau_j^{n_j}$.

*Parts (vii), (viii), and (ix).* For part (vii), suppose $\ell < n_j$. By Proposition 12, part (iii), both one-sided limits of $\cos \varphi_j^t$ as $t \to (\tau_j^\ell)^\pm$ exist. By part (iv) of the current proposition,

$$\lim_{t \to (\tau_j^\ell)^-} \cos \varphi_j^t = \frac{\left(\boldsymbol{\alpha}_j^{\tau_j^\ell}\right)^\top \boldsymbol{\gamma}_{I_j^\ell}}{\left\|\boldsymbol{\alpha}_j^{\tau_j^\ell}\right\| \left\|\boldsymbol{\gamma}_{I_j^\ell}\right\|} \qquad \text{and} \qquad \lim_{t \to (\tau_j^\ell)^+} \cos \varphi_j^t = \frac{\left(\boldsymbol{\alpha}_j^{\tau_j^\ell}\right)^\top \boldsymbol{\gamma}_{I_j^{\ell+1}}}{\left\|\boldsymbol{\alpha}_j^{\tau_j^\ell}\right\| \left\|\boldsymbol{\gamma}_{I_j^{\ell+1}}\right\|},$$

and we notice that the numerators are equal by Proposition 12, part (i). Multiplying each limit by the norm of the corresponding $\boldsymbol{\gamma}$, we obtain the desired equation.

Part (viii) follows from the same calculations in the case $\ell = n_j$, where instead of $I_j^{\ell+1}$ we use $I_j^{n_j} \cup \{i_j^{n_j}\} = [n]$.

For part (ix) we observe that $I_j^{n_j} = \{i_j^{n_j}\}$, so we have $I_+\left(\boldsymbol{\alpha}_j^{\tau_j^{n_j}}\right) = \emptyset$ and $I_0\left(\boldsymbol{\alpha}_j^{\tau_j^{n_j}}\right) = \{i_j^{n_j}\}$, so indeed $\cos \varphi_j^t \to 0$ as $t \to (\tau_j^{n_j})^-$.

*Part (vi).* We rely on the facts that $I_0(\boldsymbol{\alpha}_j^t) = \emptyset$ and that $\boldsymbol{\gamma}_{I_+(\boldsymbol{\alpha}_j^t)} \equiv \boldsymbol{\gamma}_{I_j^\ell}$ for all $t \in (\tau_j^{\ell-1}, \tau_j^\ell)$, proved in parts (iv) and (v). Notice that

$$\frac{\mathrm{d}\left((\boldsymbol{\alpha}_j^t)^\top \boldsymbol{\gamma}_{I_j^\ell}\right)}{\mathrm{d}t} = s_j \|\boldsymbol{\alpha}_j^t\| \left\|\boldsymbol{\gamma}_{I_j^\ell}\right\|^2$$

and, using a previously obtained formula for $\mathrm{d}\|\boldsymbol{\alpha}_j^t\|/\mathrm{d}t$ (just before Proposition 13),

$$
\begin{aligned}
\frac{\mathrm{d}\cos\varphi_j^t}{\mathrm{d}t} &= \frac{\mathrm{d}}{\mathrm{d}t}\left(\frac{(\boldsymbol{\alpha}_j^t)^\top \boldsymbol{\gamma}_{I_j^\ell}}{\|\boldsymbol{\alpha}_j^t\|\cdot\|\boldsymbol{\gamma}_{I_j^\ell}\|}\right) \\
&= \frac{\frac{\mathrm{d}}{\mathrm{d}t}\big((\boldsymbol{\alpha}_j^t)^\top \boldsymbol{\gamma}_{I_j^\ell}\big)\cdot\|\boldsymbol{\alpha}_j^t\|\cdot\|\boldsymbol{\gamma}_{I_j^\ell}\| - \|\boldsymbol{\gamma}_{I_j^\ell}\|\cdot\frac{\mathrm{d}\|\boldsymbol{\alpha}_j^t\|}{\mathrm{d}t}\cdot(\boldsymbol{\alpha}_j^t)^\top \boldsymbol{\gamma}_{I_j^\ell}}{\|\boldsymbol{\alpha}_j^t\|^2\|\boldsymbol{\gamma}_{I_j^\ell}\|^2} \\
&= \frac{s_j\|\boldsymbol{\alpha}_j^t\|\|\boldsymbol{\gamma}_{I_j^\ell}\|^2\|\boldsymbol{\alpha}_j^t\|\|\boldsymbol{\gamma}_{I_j^\ell}\| - \|\boldsymbol{\gamma}_{I_j^\ell}\|(\boldsymbol{\alpha}_j^t)^\top \boldsymbol{\gamma}_{I_j^\ell}\cdot s_j(\boldsymbol{\alpha}_j^t)^\top \boldsymbol{\gamma}_{I_j^\ell}}{\|\boldsymbol{\alpha}_j^t\|^2\|\boldsymbol{\gamma}_{I_j^\ell}\|^2} \\
&= s_j\|\boldsymbol{\gamma}_{I_j^\ell}\|(1-\cos^2\varphi_j^t)\,.
\end{aligned}
$$

Separating variables, we obtain

$$
\frac{\mathrm{d}\cos\varphi_j^t}{1-\cos^2\varphi_j^t} = s_j\|\boldsymbol{\gamma}_{I_j^\ell}\|\mathrm{d}t\,,
$$

and so $\operatorname{artanh}\cos\varphi_j^t = s_j\|\boldsymbol{\gamma}_{I_j^\ell}\| t + C$ for $t \in (\tau_j^{\ell-1}, \tau_j^\ell)$, where the constant $C$ is determined from the initial condition $\lim_{t\to(\tau_j^{\ell-1})^+}\operatorname{artanh}\cos\varphi_j^t = s_j\|\boldsymbol{\gamma}_{I_j^\ell}\|\tau_j^{\ell-1} + C$. The left-hand side is well-defined, since $\cos\varphi_j^{(\ell-1)^+} \notin \{-1,1\}$. Indeed, $\cos\varphi_j^t \ge 0$ for all $t$ by Proposition 11, part (ii), so the limit cannot be negative; it thus suffices to rule out the value 1. The case $s_j = -1$ is already handled in Proposition 12, part (v). For $s_j = +1$, we observe that, if $\ell > 1$, then $\cos\varphi_j^{(\ell-1)^-} > \cos\varphi_j^{(\ell-1)^+}$ by part (vii) of the current proposition, since $\|\boldsymbol{\gamma}_{I_j^{\ell-1}}\|^2 < \|\boldsymbol{\gamma}_{I_j^{\ell-1}\cup\{i_j^{\ell-1}\}}\|^2 = \|\boldsymbol{\gamma}_{I_j^\ell}\|^2$ thanks to the positive correlation between training points; hence $\cos\varphi_j^{(\ell-1)^+} < 1$. Finally, $\ell = 1$ implies $\cos\varphi_j^{(\ell-1)^+} = \lim_{t\to 0^+}\cos\varphi_j^t = \cos\angle(\boldsymbol{z}_j, \boldsymbol{\gamma}_{I_+(\boldsymbol{\alpha}_j^0)}) = 1$, and in this case $I_+(\boldsymbol{\alpha}_j^0) = [n]$ by Proposition 11, part (iii). Hence, $I_{-s_j}(\boldsymbol{z}_j) = \emptyset$ and $n_j = 0$, a contradiction. In conclusion, we have thus argued that $\cos\varphi_j^{(\ell-1)^+} \notin \{-1,1\}$ in all cases, so $\operatorname{artanh}\cos\varphi_j^t = s_j\|\boldsymbol{\gamma}_{I_j^\ell}\|(t - \tau_j^{\ell-1}) + \operatorname{artanh}\cos\varphi_j^{(\ell-1)^+}$ and it remains to take the hyperbolic tangent on both sides of this equation to prove part (vi) for $t \in (\tau_j^{\ell-1}, \tau_j^\ell)$.

*Part (x).* We rely on the result of part (vi). Recall the analytic expression for the derivative $\mathrm{d}\overline{\boldsymbol{\alpha}}_j^t/\mathrm{d}t$, obtained just before Proposition 13. Notice that, for all $t \in (\tau_j^{\ell-1}, \tau_j^\ell)$, the derivative $\mathrm{d}\overline{\boldsymbol{\alpha}}_j^t/\mathrm{d}t$ belongs to the linear subspace spanned by vectors $\overline{\boldsymbol{\alpha}}_j^t$ and $\overline{\boldsymbol{\gamma}}_{I_j^\ell}$. It follows that $\overline{\boldsymbol{\alpha}}_j^t$ can be expressed as a linear combination of two fixed vectors, $\overline{\boldsymbol{\alpha}}_j^{\tau_j^{\ell-1}}$ and $\overline{\boldsymbol{\gamma}}_{I_j^\ell}$. It is thus sufficient to check that the vector

$$
\boldsymbol{f} := \left(\sin(\varphi_j^t)\,\overline{\boldsymbol{\alpha}}_j^{\tau_j^{\ell-1}} + \sin\left(\varphi_j^{(\ell-1)^+} - \varphi_j^t\right)\overline{\boldsymbol{\gamma}}_{I_j^\ell}\right)\Big/\sin\varphi_j^{(\ell-1)^+}
$$

has norm 1 and forms an angle of $\varphi_j^t$ with $\boldsymbol{\gamma}_{I_j^\ell}$. We have

$$
\begin{aligned}
\|\boldsymbol{f}\|^2 = {}&\left(\frac{\sin\varphi_j^t}{\sin\varphi_j^{(\ell-1)^+}}\right)^2 + \left(\frac{\sin\left(\varphi_j^{(\ell-1)^+} - \varphi_j^t\right)}{\sin\varphi_j^{(\ell-1)^+}}\right)^2 \\
&+ 2\cdot\frac{\sin\varphi_j^t\,\sin\left(\varphi_j^{(\ell-1)^+} - \varphi_j^t\right)}{\sin^2\varphi_j^{(\ell-1)^+}}\cdot\cos\varphi_j^{(\ell-1)^+}\,.
\end{aligned}
$$

Denote $a = \varphi_j^t$ and $b = \varphi_j^{(\ell-1)^+} - \varphi_j^t$, then

$$\|\boldsymbol{f}\|^2 = \frac{\sin^2 a + \sin^2 b + 2\sin a \sin b \cos(a+b)}{\sin^2(a+b)}$$

$$= \frac{\sin^2 a + \sin^2 b + 2\sin a \sin b \left(\cos a \cos b - \sin a \sin b\right)}{(\sin a \cos b + \cos a \sin b)^2}$$

$$= \frac{\sin^2 a + \sin^2 b + 2\sin a \sin b \left(\cos a \cos b - \sin a \sin b\right)}{\sin^2 a \cos^2 b + 2\sin a \cos b \cos a \sin b + \cos^2 a \sin^2 b}$$

$$= \frac{\sin^2 a + \sin^2 b + 2\sin a \sin b \cos a \cos b - 2\sin^2 a \sin^2 b}{\sin^2 a \left(1 - \sin^2 b\right) + 2\sin a \cos b \cos a \sin b + \left(1 - \sin^2 a\right)\sin^2 b}$$

$$= 1 .$$

To verify the second claim, observe that

$$\boldsymbol{f}^\top \overline{\boldsymbol{\gamma}}_{I_j^\ell} = \frac{\sin \varphi_j^t}{\sin \varphi_j^{(\ell-1)^+}} \cdot \left(\overline{\boldsymbol{\alpha}}_j^{\tau_j^{\ell-1}}\right)^\top \overline{\boldsymbol{\gamma}}_{I_j^\ell} + \frac{\sin\left(\varphi_j^{(\ell-1)^+} - \varphi_j^t\right)}{\sin \varphi_j^{(\ell-1)^+}} \cdot \left\|\overline{\boldsymbol{\gamma}}_{I_j^\ell}\right\|^2$$

$$= \frac{\sin \varphi_j^t \, \cos \varphi_j^{(\ell-1)^+} + \sin \varphi_j^{(\ell-1)^+} \cos \varphi_j^t - \cos \varphi_j^{(\ell-1)^+} \sin \varphi_j^t}{\sin \varphi_j^{(\ell-1)^+}}$$

$$= \cos \varphi_j^t .$$

We must still check still that the vector $\boldsymbol{f}$ is on the correct side of $\overline{\boldsymbol{\gamma}}_{I_j^\ell}$: indeed, there are two arcs on the unit circle that connect the endpoint of vector $\overline{\boldsymbol{\gamma}}_{I_j^\ell}$ with a point at arc length $\varphi_j^t$ away from it. However, this check is easy: for $t = \tau_j^{\ell-1}$, only one of these arcs connects $\overline{\boldsymbol{\gamma}}_{I_j^\ell}$ to $\overline{\boldsymbol{\alpha}}_j^{\tau_j^{\ell-1}}$, and we can see that $\boldsymbol{f} \to \overline{\boldsymbol{\alpha}}_j^{\tau_j^{\ell-1}}$ as $\varphi_j^t \to \varphi_j^{(\ell-1)^+}$.

*Part (ii).* Let $t \to (\tau_j^\ell)^-$ in the equation of part (x), and take the dot product of each side with $\overline{\boldsymbol{x}}_{i_j^\ell}$. Observe that $\left(\overline{\boldsymbol{\alpha}}_j^{\tau_j^\ell}\right)^\top \overline{\boldsymbol{x}}_{i_j^\ell} = 0$, because $I_0\left(\boldsymbol{\alpha}_j^{\tau_j^\ell}\right) = \{i_j^\ell\}$ by part (v). We obtain

$$0 = \frac{\sin \varphi_j^{\ell^-} \cdot \left(\overline{\boldsymbol{\alpha}}_j^{\tau_j^{\ell-1}}\right)^\top \overline{\boldsymbol{x}}_{i_j^\ell} + \sin\left(\varphi_j^{(\ell-1)^+} - \varphi_j^{\ell^-}\right) \cdot \left(\overline{\boldsymbol{\gamma}}_{I_j^\ell}\right)^\top \overline{\boldsymbol{x}}_{i_j^\ell}}{\sin \varphi_j^{(\ell-1)^+}}$$

and the required equation follows. It remains to note that $\sin \varphi_j^{\ell^-} \neq 0$ because otherwise either $\varphi_j^{(\ell-1)^+} - \varphi_j^{\ell^-} \in \{-\pi, 0, \pi\}$ or $\left(\overline{\boldsymbol{\gamma}}_{I_j^\ell}\right)^\top \overline{\boldsymbol{x}}_{i_j^\ell} = 0$. The former is impossible because we know already from part (vi) that $\cos \varphi_j^t \in (0,1)$ when $t \in (\tau_j^{\ell-1}, \tau_j^\ell)$, and $\varphi_j^t \geq 0$ by definition, so $\varphi_j^{(\ell-1)^+} = \varphi_j^{\ell^-}$ but this would still contradict part (vi). The latter is impossible because $I_j^\ell \neq \emptyset$ and, by Proposition 11, part (i), the dot product must be positive due to positive correlation between training points.

*Part (xi).* Consider any interval $(t_1, t_2)$ such that $I_0(\boldsymbol{\alpha}_j^t) = \emptyset$ for all $t \in (t_1, t_2)$, and let $\boldsymbol{\gamma} := \boldsymbol{\gamma}_{I_+(\boldsymbol{\alpha}_j^t)}$; the choice of $t$ in the interval is immaterial by the continuity of the map $t \mapsto \boldsymbol{\alpha}_j^t$ and of the dot product function with a fixed vector $\boldsymbol{x}_i$. We have $\boldsymbol{\gamma} = \boldsymbol{\gamma}_{I_j^\ell}$ when $t \in (\tau_j^{\ell-1}, \tau_j^\ell)$ by part (iv). Recall our calculations for the derivative of $\|\boldsymbol{\alpha}_j^t\|$ and of $\overline{\boldsymbol{\alpha}}_j^t$ (before Proposition 13). We have $\mathrm{d}\overline{\boldsymbol{\alpha}}_j^t/\mathrm{d}t = s_j\, \boldsymbol{p}$, where $\boldsymbol{p} := \boldsymbol{\gamma} - \overline{\boldsymbol{\alpha}}_j^t \cdot (\overline{\boldsymbol{\alpha}}_j^t)^\top \boldsymbol{\gamma}$ is the vector obtained by subtracting from $\boldsymbol{\gamma}$ its orthogonal projection onto the line with direction $\boldsymbol{\alpha}_j^t$. Then

$$s_j \frac{\mathrm{d}(\overline{\boldsymbol{\alpha}}_j^t)^\top \boldsymbol{x}_i}{\mathrm{d}t} = \boldsymbol{\gamma}^\top \boldsymbol{x}_i - (\overline{\boldsymbol{\alpha}}_j^t)^\top \boldsymbol{x}_i \cdot (\overline{\boldsymbol{\alpha}}_j^t)^\top \boldsymbol{\gamma} .$$

For $t \in (t_1, t_2)$, we have $(\overline{\boldsymbol{\alpha}}_j^t)^\top \boldsymbol{x}_i \leq 0$ because $i \notin I_+(\boldsymbol{\alpha}_j^t)$. Recall that $(\overline{\boldsymbol{\alpha}}_j^t)^\top \boldsymbol{\gamma} \geq 0$ by Proposition 11, part (ii). We have shown that $-(\overline{\boldsymbol{\alpha}}_j^t)^\top \boldsymbol{x}_i \cdot (\overline{\boldsymbol{\alpha}}_j^t)^\top \boldsymbol{\gamma} \geq 0$, completing the proof of part (xi).

*Part (xii).* We continue the calculation from part (xi) assuming that $s_j = -1$ and $i = i_j^\ell$. For the function $g(t) \coloneqq (\overline{\boldsymbol{\alpha}}_j^t)^\top \overline{\boldsymbol{x}}_{i_j^\ell}$, we have

$$\frac{\mathrm{d}g}{\mathrm{d}t} = -\boldsymbol{\gamma}^\top \overline{\boldsymbol{x}}_{i_j^\ell} + (\overline{\boldsymbol{\alpha}}_j^t)^\top \overline{\boldsymbol{x}}_{i_j^\ell} \cdot (\overline{\boldsymbol{\alpha}}_j^t)^\top \boldsymbol{\gamma} \, ,$$

$$\frac{1}{\|\boldsymbol{\gamma}\|} \frac{\mathrm{d}^2 g}{\mathrm{d}t^2} = \frac{\mathrm{d}}{\mathrm{d}t} \left( (\overline{\boldsymbol{\alpha}}_j^t)^\top \overline{\boldsymbol{\gamma}} \right) \cdot (\overline{\boldsymbol{\alpha}}_j^t)^\top \overline{\boldsymbol{x}}_{i_j^\ell} + (\overline{\boldsymbol{\alpha}}_j^t)^\top \overline{\boldsymbol{\gamma}} \cdot \frac{\mathrm{d}}{\mathrm{d}t} \left( (\overline{\boldsymbol{\alpha}}_j^t)^\top \overline{\boldsymbol{x}}_{i_j^\ell} \right)$$

$$= \frac{\mathrm{d} \cos \varphi_j^t}{\mathrm{d}t} \cdot g + \cos \varphi_j^t \cdot \frac{\mathrm{d}g}{\mathrm{d}t} \, .$$

We will show the following two properties:

- $\mathrm{d}g/\mathrm{d}t < 0$ as $t \to (\tau_j^{\ell-1})^+$;

- if $\mathrm{d}g/\mathrm{d}t = 0$ for some $t$, then $\mathrm{d}^2 g/\mathrm{d}t^2 < 0$ for the same $t$.

Together, these properties imply that $\mathrm{d}g/\mathrm{d}t < 0$ throughout the interval $(\tau_j^{\ell-1}, \tau_j^\ell)$. Indeed, assume otherwise for the sake of contradiction, then $\mathrm{d}g/\mathrm{d}t = 0$ at some point $t_0 \in (\tau_j^{\ell-1}, \tau_j^\ell)$. By the second property, $g$ must have a local maximum at $t_0$, and in particular $\mathrm{d}g/\mathrm{d}t > 0$ for all $t < t_0$ close enough to $t_0$. By the first property, the minimum of $\mathrm{d}g/\mathrm{d}t$ on $(\tau_j^{\ell-1}, t_0)$ exists and is attained at an interior point of the interval. But this contradicts the second property.

Let us now justify the properties. For the first property, notice that

$$\left. \frac{\mathrm{d}g}{\mathrm{d}t} \right|_{t \to (\tau_j^{\ell-1})^+} = -\|\boldsymbol{\gamma}\| \cdot \left( \overline{\boldsymbol{\gamma}}^\top \overline{\boldsymbol{x}}_{i_j^\ell} - \left( \overline{\boldsymbol{\alpha}}_j^{\tau_j^{\ell-1}} \right)^\top \overline{\boldsymbol{x}}_{i_j^\ell} \cdot \left( \overline{\boldsymbol{\alpha}}_j^{\tau_j^{\ell-1}} \right)^\top \overline{\boldsymbol{\gamma}} \right)$$

$$= -\|\boldsymbol{\gamma}\| \cdot (\overline{\boldsymbol{\gamma}}^\top \overline{\boldsymbol{x}}_{i_j^\ell}) \cdot \left( 1 - \frac{\left( \overline{\boldsymbol{\alpha}}_j^{\tau_j^{\ell-1}} \right)^\top \overline{\boldsymbol{x}}_{i_j^\ell}}{\overline{\boldsymbol{\gamma}}^\top \overline{\boldsymbol{x}}_{i_j^\ell}} \cdot \left( \overline{\boldsymbol{\alpha}}_j^{\tau_j^{\ell-1}} \right)^\top \overline{\boldsymbol{\gamma}} \right) \, .$$

Here, $\overline{\boldsymbol{\gamma}}^\top \overline{\boldsymbol{x}}_{i_j^\ell} > 0$ since $i_j^\ell \in I_j^\ell$. The value of the ratio $\left( \overline{\boldsymbol{\alpha}}_j^{\tau_j^{\ell-1}} \right)^\top \overline{\boldsymbol{x}}_{i_j^\ell} \Big/ \overline{\boldsymbol{\gamma}}^\top \overline{\boldsymbol{x}}_{i_j^\ell}$ appears in the statement of part (i), and in particular replacing the index $i_j^\ell$ with any other $i \in I_j^\ell$ would result in a higher (positive) value. Therefore, if we assume for the sake of contradiction that the right-hand side in the last equation is non-negative, then it will remain non-negative if $i_j^\ell$ is replaced with every other $i \in I_j^\ell$. In other words, if $g(t) = (\overline{\boldsymbol{\alpha}}_j^t)^\top \overline{\boldsymbol{x}}_{i_j^\ell}$ is non-decreasing in a right-neighbourhood of $\tau_j^{\ell-1}$, so is every dot product $(\overline{\boldsymbol{\alpha}}_j^t)^\top \overline{\boldsymbol{x}}_i$ with $i \in I_j^\ell$. But then their linear combination with positive coefficients $y_i \|\boldsymbol{x}_i\|/n$ is also non-decreasing. This, however, is not possible because this linear combination is $(\overline{\boldsymbol{\alpha}}_j^t)^\top \boldsymbol{\gamma}_{I_+(\boldsymbol{\alpha}_j^t)}$, and we already saw in the proof of part (xi) that $\mathrm{d}\overline{\boldsymbol{\alpha}}_j^t/\mathrm{d}t = s_j \, \boldsymbol{p}$, where $\boldsymbol{p}$ is an orthogonal projection of $\boldsymbol{\gamma}_{I_+(\boldsymbol{\alpha}_j^t)}$ onto a proper subspace. By standard properties of projections we must have $\boldsymbol{p}^\top \boldsymbol{\gamma}_{I_+(\boldsymbol{\alpha}_j^t)} > 0$ and, since $s_j = -1$, $\mathrm{d}\big((\overline{\boldsymbol{\alpha}}_j^t)^\top \boldsymbol{\gamma}_{I_+(\boldsymbol{\alpha}_j^t)}\big)/\mathrm{d}t < 0$, which is a contradiction. (The case $\boldsymbol{p}^\top \boldsymbol{\gamma}_{I_+(\boldsymbol{\alpha}_j^t)} = 0$ is impossible by Proposition 12, part (v), as we would then have $\cos \varphi_j^{(\ell-1)^+} = 1$.) This concludes the proof of the first property.

The second property follows directly from the equation for $\mathrm{d}^2 g/\mathrm{d}t^2$, because $\cos \varphi_j^t$ decreases by part (vi) and because $g > 0$.

For the sign of second derivative in general, it remains to consider the second term. The first factor is positive by Proposition 11, part (ii); and we just proved above that $\mathrm{d}g/\mathrm{d}t < 0$. (Note that $\boldsymbol{\gamma} \neq \boldsymbol{0}$ by Proposition 11, part (i), because $i_j^\ell \in I_+(\boldsymbol{\alpha}_j^t)$.) This completes the proof of part (xii). $\qquad \square$

**Corollary 15.** *For all $j \in J_+ \cup J_-$, the set of all $\boldsymbol{z}_j \in \mathbb{R}^d$ such that $|I_0(\boldsymbol{\alpha}_j^t)| > 1$ for some $t \in [0, \infty)$ has Lebesgue measure zero.*

*Proof.* A single yardstick trajectory at any time $t$ follows a direction $\boldsymbol{\gamma}_S$ for some $S \subseteq [n]$. The set $S$ changes at most $n$ times, namely at the crossing of $\bigcup_{i \in [n]} H_i$, where $H_i$ is the set of vectors orthogonal to the training point $\boldsymbol{x}_i$. (The proof of this fact does not rely on Assumption 1, part (iv). It is a consequence of Proposition 12, part (iv). We note that the assumption of Proposition 12 is shown to be valid in the proof of Proposition 14, under "Intermediate summary" on page 21.)

The union $U$ of all $H_i \cap H_k$, $i < k$, is a union of finitely many subspaces of dimension $d-2$ (because no two training points are collinear by Assumption 1, part (iii)). Consider all the vectors $\boldsymbol{u}$ such that the yardstick trajectory starting at $\boldsymbol{u}$ passes through $U$. We claim that this is a set of zero measure. Indeed:

- Every convex polyhedron $P$ of dimension $d - 2$, for example $H_i \cap H_k$, can be reached by a straight-line trajectory (without change of direction) from a convex polyhedron $P'$ of dimension at most $d - 1$, i.e., of co-dimension at least 1.

- The previous change of direction occurs at the intersection of the polyhedron $P'$ and the union of all $H_i$. This intersection is a finite union of convex polyhedra of co-dimension at least 2, because, for all nonempty subsets $S \subseteq [n]$, the vector $\boldsymbol{\gamma}_S$ cannot belong to any $H_i$, thanks to the 45-degree condition. To each of these polyhedra, the previous bullet point applies.

- No more than $n$ changes of direction may take place along a single trajectory.

Thus, all vectors from which a point in $U$ can be reached along a yardstick trajectory belong to a finite union of affine subspaces of co-dimension 1. Thus, they form a measure zero set. $\qquad\square$

# D  Proofs for the first phase

Here we prove Lemma 3, Lemma 4, and Lemma 5, as well as a number of related results. The former are subsumed by Lemma 19, Lemma 21, and Lemma 23 below.

Recall the definitions of $\delta$ and $\Delta$ in section 3:

$$
\delta := \min \left\{
\begin{array}{c}
\min_{i \in [n]} \|\boldsymbol{x}_i\|, \ \min_{i,i' \in [n]} \overline{\boldsymbol{x}}_i^\top \overline{\boldsymbol{x}}_{i'}, \ \min_{k \in [d-1]}(\sqrt{\eta_k} - \sqrt{\eta_{k+1}})(d-1), \ \sqrt{\eta_d}, \\[4pt]
\min_{k \in [d]} \nu_k^* \sqrt{d}, \ \min_{j \in [m]} \|\boldsymbol{z}_j\|, \ \min_{j \in J_+} \cos \varphi_j^0, \ \min_{j \in J_-} \sin \varphi_j^0, \\[4pt]
\min \left\{ |\overline{\boldsymbol{\alpha}}_j^{t\top} \overline{\boldsymbol{x}}_i| \ \middle| \ \begin{array}{l} j \in J_+ \cup J_- \wedge \ell \in [n_j] \\ \wedge\, t \in [\tau_j^{\ell-1}, \tau_j^\ell] \wedge i \in [n] \\ \wedge\, i \neq i_j^\ell \wedge (\ell \neq 1 \Rightarrow i \neq i_j^{\ell-1}) \end{array} \right\}, \ \min_{j \in J_-} \overline{\boldsymbol{\alpha}}_j^{0\top} \overline{\boldsymbol{x}}_{i_j^1}, \\[4pt]
\min\{\tau_j^\ell - \tau_j^{\ell-1} \mid j \in J_+ \cup J_- \wedge \ell \in [n_j]\}
\end{array}
\right\}
$$

$\Delta := \max\{\max_{i \in [n]} \|\boldsymbol{x}_i\|, \ \max_{j \in [m]} \|\boldsymbol{z}_j\|, \ 1\}$.

Thus $\delta$ is the minimum of: the length of any training point, the cosine of the angle between any two training points, the difference between the square roots of any consecutive eigenvalues adjusted by the dimension, the square root of the smallest eigenvalue, the smallest eigenvector coordinate of the teacher neuron adjusted by the square root of the dimension, the length of any unscaled hidden-neuron initialisation, the cosine or sine of the angle between it (if active) and the corresponding vector $\boldsymbol{\gamma}_I$ depending on whether the last-layer sign is positive or negative (respectively), the absolute cosine of any angle between a trajectory point $\boldsymbol{\alpha}_j^t$ and a training point which is neither the previous nor the next to cross the half-space boundary, the cosine of the angle between any initial negative-sign active hidden neuron and the first data point to cross the boundary, and the time between any two consecutive crossings; and $\Delta \geq 1$ is the maximum length of any traning point or unscaled hidden-neuron initialisation.

First we observe that, immediately from the definitions in section 3 of the vectors $\boldsymbol{\gamma}_I$, the matrix $\boldsymbol{X}$, the eigenvalues $\eta_k$, the eigenvectors $\boldsymbol{u}_k$, and the coordinates $\nu_k^*$ of the teacher neuron with respect to the basis consisting of the eigenvectors, we have the following two alternative expressions for the vector $\boldsymbol{\gamma}_{[n]}$.

**Proposition 16.** $\boldsymbol{\gamma}_{[n]} = \frac{1}{n} \boldsymbol{X} \boldsymbol{X}^\top \boldsymbol{v}^* = \sum_{k=1}^d \eta_k \nu_k^* \boldsymbol{u}_k.$

Then we establish upper bounds on: the largest eigenvalue of the matrix $\frac{1}{n}\boldsymbol{X}\boldsymbol{X}^\top$, the ratio of any two consecutive eigenvalues in their decreasing ordering, the Euclidean lengths of the vectors $\boldsymbol{\gamma}_I$, the cosines of the angles $\varphi_j^t$ that measure alignment of the yardstick trajectories $\boldsymbol{\alpha}_j^t$ (both defined in section 3) mapped backwards through the hyperbolic tangent sigmoid, and the finish time of the last intermediate alignment stage of a yardstick trajectory; and lower bounds on: the Euclidean lengths of the vectors $\boldsymbol{\gamma}_I$, and the cosines of the angles between a yardstick trajectory and the training point that is the next to enter or exit its active half-space.

**Proposition 17.** *(i)* $\eta_1 \leq \Delta^2$.

*(ii)* $\frac{\eta_{k+1}}{\eta_k} \leq \left(1 - \frac{\delta}{(d-1)\Delta}\right)^2$ *for all $k \in [d-1]$.*

*(iii)* $\frac{\delta^{5/2}|I|}{\sqrt{2n}} \leq \|\boldsymbol{\gamma}_I\| \leq \frac{\Delta^2|I|}{n}$ *for all $I \subseteq [n]$, and $\delta^2 \leq \|\boldsymbol{\gamma}_{[n]}\|$.*

*(iv)* $\max_{j \in J_+}^{\ell \in [n_j]} \operatorname{artanh} \cos \varphi_j^{\ell^-} < \ln\left(\frac{2}{\delta}\right)$.

*(v)* $\max_{j \in J_-}^{\ell \in [n_j]} \operatorname{artanh} \cos \varphi_j^{(\ell-1)^+} < \ln\left(\frac{2}{\delta}\right)$.

*(vi)* $\max_{j \in J_+ \cup J_-} \tau_j^{n_j} < \frac{4n \ln n}{\delta^3}$.

*(vii)* $\left|\overline{\boldsymbol{\alpha}}_j^{t\top} \overline{\boldsymbol{x}}_{i_j^\ell}\right| \geq \frac{2\delta^4}{3n}(\tau_j^\ell - t)$ *for all $j \in J_+ \cup J_-$, $\ell \in [n_j]$, and $t \in [\tau_j^{\ell-1}, \tau_j^\ell]$.*

*Proof.* For part (i), we have

$$\eta_1 = \left\|\frac{1}{n}\boldsymbol{X}\boldsymbol{X}^\top \boldsymbol{u}_1\right\| = \frac{1}{n}\left\|\sum_{i \in [n]} \boldsymbol{x}_i \boldsymbol{x}_i^\top \boldsymbol{u}_1\right\| \leq \frac{1}{n}\sum_{i \in [n]} \|\boldsymbol{x}_i \boldsymbol{x}_i^\top \boldsymbol{u}_1\| \leq \max_{i \in [n]} \|\boldsymbol{x}_i\|^2 = \Delta^2 .$$

For part (ii), supposing $k \in [d-1]$, by part (i) we have

$$\frac{\sqrt{\eta_{k+1}}}{\sqrt{\eta_k}} = 1 - \frac{\sqrt{\eta_k} - \sqrt{\eta_{k+1}}}{\sqrt{\eta_k}} \leq 1 - \frac{\sqrt{\eta_k} - \sqrt{\eta_{k+1}}}{\sqrt{\eta_1}} \leq 1 - \frac{\delta}{(d-1)\Delta} .$$

For part (iii), supposing $I \subseteq [n]$, recalling that $\measuredangle(\boldsymbol{v}^*, \boldsymbol{x}_i) < \pi/4$ for all $i \in [n]$ we have

$$\|\boldsymbol{\gamma}_I\| = \frac{1}{n}\left\|\sum_{i \in I} y_i \boldsymbol{x}_i\right\| = \frac{1}{n}\sqrt{\sum_{i,i' \in I} y_i y_{i'} \boldsymbol{x}_i^\top \boldsymbol{x}_{i'}} \geq \frac{|I|}{n}\min_{i,i' \in I}\sqrt{y_i y_{i'} \boldsymbol{x}_i^\top \boldsymbol{x}_{i'}}$$

$$= \frac{|I|}{n}\min_{i,i' \in I}\sqrt{\boldsymbol{v}^{*\top}\boldsymbol{x}_i \cdot \boldsymbol{v}^{*\top}\boldsymbol{x}_{i'} \cdot \boldsymbol{x}_i^\top \boldsymbol{x}_{i'}} \geq \frac{|I|}{n}\sqrt{\left(\frac{\delta}{\sqrt{2}}\right)^2 \delta^3} = \frac{\delta^{5/2}|I|}{\sqrt{2n}} ,$$

and we have

$$|\boldsymbol{\gamma}_I\| \leq \frac{1}{n}\sum_{i \in I} y_i \|\boldsymbol{x}_i\| \leq \frac{|I|}{n}\max_{i \in I}\boldsymbol{v}^{*\top}\boldsymbol{x}_i \cdot \|\boldsymbol{x}_i\| \leq \frac{|I|}{n}\max_{i \in I}\|\boldsymbol{x}_i\|^2 \leq \frac{\Delta^2|I|}{n} .$$

Also, recalling Proposition 16 we have $\|\boldsymbol{\gamma}_{[n]}\| = \|\frac{1}{n}\boldsymbol{X}\boldsymbol{X}^\top \boldsymbol{v}^*\| \geq \eta_d \geq \delta^2$.

For part (iv), supposing $j \in J_+$ and $\ell \in [n_j]$, and observing that $\operatorname{artanh} q = \frac{1}{2}\ln\left(\frac{1+q}{1-q}\right) < \frac{1}{2}\ln\left(\frac{2}{1-q}\right)$ for all $|q| < 1$, by Proposition 14 (iv) and (v) we have

$$\operatorname{artanh} \cos \varphi_j^{\ell^-} = \operatorname{artanh} \lim_{t \to (\tau_j^\ell)^-} \cos \measuredangle\left(\boldsymbol{\alpha}_j^t, \boldsymbol{\gamma}_{I_j^\ell}\right)$$

$$< \operatorname{artanh} \sin \measuredangle\left(\boldsymbol{x}_{i_j^\ell}, \boldsymbol{\gamma}_{I_j^\ell}\right) = \operatorname{artanh}\sqrt{1 - \cos^2 \measuredangle\left(\boldsymbol{x}_{i_j^\ell}, \boldsymbol{\gamma}_{I_j^\ell}\right)} \leq \operatorname{artanh}\sqrt{1 - \delta^2}$$

$$< \operatorname{artanh}\left(1 - \frac{\delta^2}{2}\right) < \frac{1}{2}\ln\left(\frac{4}{\delta^2}\right) = \ln\left(\frac{2}{\delta}\right) .$$

Part (v) follows analogously, once we recall that, for all $j \in J_-$, by Assumption 1 (ii) we have $\cos \varphi_j^{0^+} = \cos \varphi_j^0 = \sqrt{1 - \sin^2 \varphi_j^0} \le \sqrt{1 - \delta^2}$.

For part (vi), supposing $j \in J_+ \cup J_-$ we have

$$
\begin{aligned}
\tau_j^{n_j} &= \sum_{\ell \in [n_j]} \tau_j^\ell - \tau_j^{\ell-1} && \text{since } \tau_j^0 := 0 \text{ in Proposition 14} \\
&\le \sum_{\ell \in [n_j]} \frac{\ln\left(\frac{2}{\delta}\right)}{\left\| \boldsymbol{\gamma}_{I_j^\ell} \right\|} && \text{by parts (iv) and (v), and Proposition 14 (vi)} \\
&\le \frac{\sqrt{2}n}{\delta^{5/2}} \ln\left(\frac{2}{\delta}\right) \sum_{\ell \in [n_j]} \frac{1}{|I_j^\ell|} && \text{by part (iii)} \\
&\le \frac{\sqrt{2}n}{\delta^{5/2}} \ln\left(\frac{2}{\delta}\right) \sum_{i \in [n]} \frac{1}{i} && \text{by the definition of } I_j^\ell \text{ in Proposition 14} \\
&< \frac{\sqrt{2}n(1 + \ln n)}{\delta^{5/2}} \ln\left(\frac{2}{\delta}\right) && \text{by properties of the harmonic series} \\
&< \frac{7n \ln n}{2\delta^{5/2}} \ln\left(\frac{2}{\delta}\right) && \text{since } n \ge 2 \text{ by Assumption 1 (i)} \\
&< \frac{4n \ln n}{\delta^3} && \text{since } \ln\left(\frac{2}{\delta}\right) < \frac{8}{7\sqrt{\delta}}.
\end{aligned}
$$

For part (vii), if $s_j = 1$ then by Proposition 14 (xi) and by part (iii) we have

$$
\inf_{t \in (\tau_j^{\ell-1}, \tau_j^\ell)} \frac{\mathrm{d}\,\overline{\boldsymbol{\alpha}}_j^{t\,\top} \boldsymbol{x}_{i_j^\ell}}{\mathrm{d}t} \ge \boldsymbol{\gamma}_{I_j^\ell}^\top \overline{\boldsymbol{x}}_{i_j^\ell} \ge \frac{\delta^{7/2}}{\sqrt{2}n} \ .
$$

If $s_j = -1$ then by Proposition 14 (xii) we have that $\overline{\boldsymbol{\alpha}}_j^{t\,\top} \overline{\boldsymbol{x}}_{i_j^\ell}$ is concave on $[\tau_j^{\ell-1}, \tau_j^\ell]$, so by part (v), by Proposition 14 (vi), by part (iii), and since $\ln\left(\frac{2}{\delta}\right) < \frac{3}{2\sqrt{2\delta}}$, for all $t \in [\tau_j^{\ell-1}, \tau_j^\ell)$ we have

$$
\frac{\overline{\boldsymbol{\alpha}}_j^{t\,\top} \overline{\boldsymbol{x}}_{i_j^\ell}}{\tau_j^\ell - t} \ge \frac{\overline{\boldsymbol{\alpha}}_j^{\tau_j^{\ell-1}\,\top} \overline{\boldsymbol{x}}_{i_j^\ell}}{\tau_j^\ell - \tau_j^{\ell-1}} \ge \frac{\delta}{\frac{\sqrt{2}n}{\delta^{5/2}} \ln\left(\frac{2}{\delta}\right)} = \frac{\delta^{7/2}}{\sqrt{2}n} \frac{1}{\ln\left(\frac{2}{\delta}\right)} > \frac{2\delta^4}{3n} \ . \qquad \square
$$

Recall from section 4 that $T_0 = \max_{j \in J_+ \cup J_-} \tau_j^{n_j} + 1$ and $T_1 = \varepsilon \ln(1/\lambda)/\|\boldsymbol{\gamma}_{[n]}\|$.

**Proposition 18.** $T_0 < T_1/3$.

*Proof.* We have

$$
\begin{aligned}
T_1/3 &\ge 9n \ln n \frac{\Delta^2}{\delta^3} \Big/ \|\boldsymbol{\gamma}_{[n]}\| && \text{by Assumption 2} \\
&\ge \frac{9n \ln n}{\delta^3} && \text{by Proposition 17 (iii)} \\
&> \frac{4n \ln n}{\delta^3} + 1 && \text{since } n \ge 2 \text{ by Assumption 1 (i)} \\
&> T_0 && \text{by Proposition 17 (vi).} \qquad \square
\end{aligned}
$$

The following lemma states that, throughout the first phase of the training, the lengths of the positive-sign initially active hidden neurons are non-decreasing, and the lengths of the negative-sign initially active hidden neurons are non-increasing; and it provides a time-sensitive upper bound for the former. As for the remaining hidden neurons, i.e. those whose indices are not in the sets $J_+$ and $J_-$, they are by definition inactive at initialisation, and by Assumption 1 (ii) have no training points in their activation boundaries, so they do not change throughout the training.

**Lemma 19.** *(i)* $\|\boldsymbol{w}_j^t\| < 2\|\boldsymbol{z}_j\|\lambda^{1-\varepsilon\,t/T_1}$ *for all* $j \in J_+$ *and all* $t \in [0, T_1]$.

*(ii)* $s_j\,\mathrm{d}\|\boldsymbol{w}_j^t\|/\mathrm{d}t \geq 0$ *for all* $j \in J_+ \cup J_-$ *and almost all* $t \in [0, T_1]$.

*Proof.* First we establish the following, which implies (i).

*Claim* 20. $\|\boldsymbol{w}_j^t\| < 2\|\boldsymbol{z}_j\|\lambda^{1-\varepsilon\,t/T_1}$ for all $j \in [m]$ and all $t \in [0, T_1]$.

*Proof of claim.* Assume for a contradiction that this fails, and let $t \in [0, T_1]$ be the smallest such that $\|\boldsymbol{w}_j^t\| \geq 2\|\boldsymbol{z}_j\|\lambda^{1-\varepsilon\,t/T_1}$ for some $j \in [m]$. Then we have

$$
\begin{aligned}
\|\boldsymbol{w}_j^t\| &\leq \lambda\|\boldsymbol{z}_j\|\mathrm{e}^{t\,\max_{t' \in [0,t]}\left\|\boldsymbol{g}_j^{t'}\right\|} && \text{by Proposition 1 and Grönwall's inequality} \\
&\leq \lambda\|\boldsymbol{z}_j\|\mathrm{e}^{t\left(\|\boldsymbol{\gamma}_{[n]}\| + \max_{t' \in [0,t]}^{i \in [n]}\left|h_{\boldsymbol{\theta}^{t'}}(\boldsymbol{x}_i)\right|\|\boldsymbol{x}_i\|\right)} && \text{by the definition of } \boldsymbol{g}_j^{t'} \text{ in Proposition 1 (i)} \\
&\leq \lambda\|\boldsymbol{z}_j\|\mathrm{e}^{t\left(\|\boldsymbol{\gamma}_{[n]}\| + m\,\max_{t' \in [0,t]}^{j' \in [m], i \in [n]}\left\|\boldsymbol{w}_{j'}^{t'}\right\|^2\|\boldsymbol{x}_i\|^2\right)} && \text{by the definition of } h_{\boldsymbol{\theta}^{t'}} \text{ in section 2} \\
&\leq \lambda\|\boldsymbol{z}_j\|\mathrm{e}^{t(\|\boldsymbol{\gamma}_{[n]}\| + \lambda^{2-2\varepsilon}4m\Delta^4)} && \text{since } \left\|\boldsymbol{w}_{j'}^{t'}\right\| \leq 2\left\|\boldsymbol{z}_{j'}\right\|\lambda^{1-\varepsilon\,t'/T_1} \\
&\leq \lambda\|\boldsymbol{z}_j\|\mathrm{e}^{t\|\boldsymbol{\gamma}_{[n]}\|}\mathrm{e}^{T_1\lambda^{2-2\varepsilon}4m\Delta^4} && \text{since } t \leq T_1 \\
&= \|\boldsymbol{z}_j\|\lambda^{1-\varepsilon\,t/T_1}\lambda^{-\varepsilon\lambda^{2-2\varepsilon}4m\Delta^4/\|\boldsymbol{\gamma}_{[n]}\|} && \text{since } \mathrm{e}^{T_1\|\boldsymbol{\gamma}_{[n]}\|} = \lambda^{-\varepsilon} \\
&\leq \|\boldsymbol{z}_j\|\lambda^{1-\varepsilon\,t/T_1}\lambda^{-\varepsilon\lambda^{2-2\varepsilon}4m\Delta^4/\delta^2} && \text{by Proposition 17 (iii)} \\
&= \|\boldsymbol{z}_j\|\lambda^{1-\varepsilon\,t/T_1}\mathrm{e}^{\ln(\lambda^{-\varepsilon})\lambda^{2-2\varepsilon}4m\Delta^4/\delta^2} && \text{since exp and ln are inverses} \\
&< \|\boldsymbol{z}_j\|\lambda^{1-\varepsilon\,t/T_1}\mathrm{e}^{\lambda^{2-3\varepsilon}4m\Delta^4/\delta^2} && \text{since } \lambda^{-\varepsilon} > \ln(\lambda^{-\varepsilon}) \\
&< \|\boldsymbol{z}_j\|\lambda^{1-\varepsilon\,t/T_1}\mathrm{e}^{\lambda^{2-4\varepsilon}} && \text{since } \lambda^{-\varepsilon} \geq m^3\,n^{9\cdot3n\Delta^2/\delta^3} > 4m\Delta^4/\delta^2 \\
&< 2\|\boldsymbol{z}_j\|\lambda^{1-\varepsilon\,t/T_1} && \text{since } \lambda^{2-4\varepsilon} \leq \lambda \leq 2^{-9\cdot2\cdot3\cdot4} < \ln 2. \qquad \square
\end{aligned}
$$

To prove (ii), observing that by Proposition 1 for all $j \in [m]$ and almost all $t \in [0, \infty)$ we have

$$
s_j\,\mathrm{d}\|\boldsymbol{w}_j^t\|/\mathrm{d}t = \boldsymbol{w}_j^{t\top}\boldsymbol{g}_j^t \in \frac{1}{n}\sum_{i=1}^{n}(y_i - h_{\boldsymbol{\theta}^t}(\boldsymbol{x}_i))\,\partial\sigma(\boldsymbol{w}_j^{t\top}\boldsymbol{x}_i)\,\boldsymbol{w}_j^{t\top}\boldsymbol{x}_i \subseteq \sum_{i=1}^{n}(y_i - h_{\boldsymbol{\theta}^t}(\boldsymbol{x}_i))\,[0, \infty)\,,
$$

it suffices to show that for all $t \in [0, T_1]$ and all $i \in [n]$ we have $|h_{\boldsymbol{\theta}^t}(\boldsymbol{x}_i)| \leq y_i$. Indeed

$$
\begin{aligned}
|h_{\boldsymbol{\theta}^t}(\boldsymbol{x}_i)| &\leq m\,\max_{j=1}^{m}\|\boldsymbol{w}_j^t\|^2\|\boldsymbol{x}_i\| && \text{by the definition of } h_{\boldsymbol{\theta}^{t'}} \text{ in section 2} \\
&< 4m\Delta^3\lambda^{2-2\varepsilon} && \text{by Claim 20} \\
&< \frac{\delta}{\sqrt{2}} && \text{since } \lambda^{2\varepsilon-2} \geq m^{3\cdot6}\,n^{9\cdot3\cdot6n\Delta^2/\delta^3} > 4\sqrt{2}m\Delta^3/\delta \\
&< y_i && \text{since } \angle(\boldsymbol{v}^*, \boldsymbol{x}_i) < \pi/4. \qquad \square
\end{aligned}
$$

The next lemma provides a detailed description of the intermediate alignment stages for each hidden neuron. It states that the training points enter or exit the active half-space of each positive-sign or negative-sign (respectively) hidden neuron in the same order as they do for the positive half-space of the corresponding yardstick trajectory, and it provides non-asymptotic bounds for each difference: between a dynamics-governing vector $\boldsymbol{g}_j^t$ (defined in Proposition 1 (i)) and the corresponding intermediate yardstick target $\boldsymbol{\gamma}_{I_j^\ell}$ (the sets $I_j^\ell$ of indices of training points that are in the active half-space of hidden neuron $j$ during stage $\ell$ are defined in Proposition 14), between the unit-sphere normalisations of a hidden neuron and the corresponding yardstick vector, and between the corresponding boundary crossing times for a hidden neuron and its yardstick vector. In particular, it shows that each negative-sign initially active hidden neuron $j$ deactivates at time $t_j^{n_j}$, and hence before time $T_0$. The proof of the lemma is inductive over the stage index $\ell$, and involves carefully controlling the differences between the trajectories on the unit sphere of the hidden neurons and their yardstick vectors; this is non-trivial because, in contrast to the latter which have separate individual dynamics, the dynamics of the former are joint since each governing vector $\boldsymbol{g}_j^t$ depends on the outputs of the whole network.

**Lemma 21.** *For all $j \in J_+ \cup J_-$ there exist unique $t_j^1, \ldots, t_j^{n_j} \in [0, \infty)$ such that for all $\ell \in [n_j]$ the following hold, where $t_j^0 := 0$:*

(i) $I_+(\boldsymbol{w}_j^t) = I_j^\ell$ *for all* $t \in (t_j^{\ell-1}, t_j^\ell)$;

(ii) $I_0(\boldsymbol{w}_j^t) = \emptyset$ *for all* $t \in (t_j^{\ell-1}, t_j^\ell)$, *and* $I_0\left(\boldsymbol{w}_j^{t_j^\ell}\right) = \{i_j^\ell\}$;

(iii) $\|\boldsymbol{\gamma}_{I_j^\ell} - \boldsymbol{g}_j^t\| \leq \lambda^{2-\varepsilon}$ *for all* $t \in (t_j^{\ell-1}, t_j^\ell)$;

(iv) $\|\overline{\boldsymbol{\alpha}}_j^t - \overline{\boldsymbol{w}}_j^t\| \leq \lambda^{1 - \left(1 + \frac{3\ell}{3n_j}\right)\varepsilon}$ *for all* $t \in (t_j^{\ell-1}, \max\{t_j^\ell, \tau_j^\ell\}]$;

(v) $|\tau_j^\ell - t_j^\ell| \leq \lambda^{1 - \left(1 + \frac{3\ell-1}{3n_j}\right)\varepsilon}$.

*Proof.* For all $j \in J_+ \cup J_-$ and all $\ell \in [n_j]$ let

$$\mathsf{t}_j^0 := 0 \qquad\qquad \mathsf{t}_j^{-\ell} := \tau_j^\ell - \lambda^{1 - \left(1 + \frac{3\ell-1}{3n_j}\right)\varepsilon} \qquad\qquad \mathsf{t}_j^\ell := \tau_j^\ell + \lambda^{1 - \left(1 + \frac{3\ell-1}{3n_j}\right)\varepsilon} \ .$$

For all $j \in J_+ \cup J_-$ and all $\ell \in [n_j]$, define a continuous $\mathsf{w}_j^t$ by

$$\mathsf{w}_j^{\mathsf{t}_j^{\ell-1}} := \boldsymbol{\alpha}_j^{\mathsf{t}_j^{\ell-1}} \qquad\qquad d\mathsf{w}_j^t/dt := s_j \|\mathsf{w}_j^t\| \boldsymbol{\gamma}_{I_j^\ell} \quad \text{for all } t \in (\mathsf{t}_j^{\ell-1}, \mathsf{t}_j^\ell] \ .$$

Thus $\mathsf{w}_j^t$ equals $\boldsymbol{\alpha}_j^t$ on $[\mathsf{t}_j^{\ell-1}, \tau_j^\ell]$, and $\mathsf{w}_j^t$ continues with the same dynamics on $(\tau_j^\ell, \mathsf{t}_j^\ell]$.

We first observe that, until the largest of the times $\mathsf{t}_j^\ell$, the network outputs at all the training points remain small. Specifically, for all $t \in [0, \max_{j \in J_+ \cup J_-} \mathsf{t}_j^{n_j}]$ we have

$$\frac{1}{n} \sum_{i=1}^n |h_{\boldsymbol{\theta}^t}(\boldsymbol{x}_i)| \|\boldsymbol{x}_i\| \leq m \left(\max_{j=1}^m \|\boldsymbol{w}_j^t\|^2\right)\left(\max_{i=1}^n \|\boldsymbol{x}_i\|^2\right) \quad \text{by the definition of } h_{\boldsymbol{\theta}^t} \text{ in section 2}$$

$$< 4m\Delta^4 \lambda^{2 - 2\varepsilon/3} \qquad\qquad\qquad\qquad \text{by Proposition 18 and Lemma 19}$$

$$< \lambda^{2-\varepsilon} \qquad\qquad\qquad\qquad\qquad\qquad \text{since } \lambda^{-\varepsilon/3} \geq m\, n^{9n\Delta^2/\delta^3} > 4m\Delta^4.$$

That shows that part (iii) of the lemma is implied by parts (i), (ii), and (v).

Let $j \in J_+ \cup J_-$ be fixed for the remainder of the proof.

We proceed to show the lemma by induction on $\ell \in [n_j]$, where we make use of the following inductive hypothesis:

$$\text{at } t = \mathsf{t}_j^{\ell-1} \text{ we have } I_+(\boldsymbol{w}_j^t) = I_j^\ell, \ I_0(\boldsymbol{w}_j^t) = \emptyset, \text{ and } \|\boldsymbol{\alpha}_j^t - \overline{\boldsymbol{w}}_j^t\| \leq \lambda^{1 - \left(1 + \frac{3\ell-3}{3n_j}\right)\varepsilon}.$$

That holds for $\ell = 1$ by Assumption 1 (ii) and since $\overline{\boldsymbol{\alpha}}_j^0 = \overline{\boldsymbol{z}_j} = \overline{\lambda \boldsymbol{z}_j} = \overline{\boldsymbol{w}}_j^0$.

Consider $\ell \in [n_j]$. By the inductive hypothesis, at $t = \mathsf{t}_j^{\ell-1}$ we have $1 - \overline{\boldsymbol{w}}_j^{t\top} \overline{\boldsymbol{w}}_j^t \leq \frac{1}{2}\lambda^{2 - 2\left(1 + \frac{3\ell-3}{3n_j}\right)\varepsilon}$.

Let $\mathsf{T} := \min\{t > \mathsf{t}_j^{\ell-1} \mid I_0(\boldsymbol{w}_j^t) \neq \emptyset\}$.

If $s_j = 1$, then for all $t \in (\mathsf{t}_j^{\ell-1}, \min\{\mathsf{t}_j^\ell, \mathsf{T}\})$ we have

$$\mathrm{d}(1 - \overline{\mathsf{w}}_j^{t\top} \overline{\boldsymbol{w}}_j^t)/\mathrm{d}t$$

$$= -\overline{\mathsf{w}}_j^{t\top} \boldsymbol{g}_j^t - \overline{\boldsymbol{w}}_j^{t\top} \boldsymbol{\gamma}_{I_j^\ell} + \overline{\mathsf{w}}_j^{t\top} \overline{\boldsymbol{w}}_j^t (\overline{\mathsf{w}}_j^{t\top} \boldsymbol{\gamma}_{I_j^\ell} + \overline{\boldsymbol{w}}_j^{t\top} \boldsymbol{g}_j^t) \qquad \text{by Corollary 10 and since } I_0(\boldsymbol{w}_j^t) = \emptyset$$

$$< 2\lambda^{2-\varepsilon} - (1 - \overline{\mathsf{w}}_j^{t\top} \overline{\boldsymbol{w}}_j^t)(\overline{\mathsf{w}}_j^t + \overline{\boldsymbol{w}}_j^t)^\top \boldsymbol{\gamma}_{I_j^\ell} \qquad \text{since } \|\boldsymbol{\gamma}_{I_j^\ell} - \boldsymbol{g}_j^t\| < \lambda^{2-\varepsilon}$$

$$\leq 2\lambda^{2-\varepsilon} - (1 - \overline{\mathsf{w}}_j^{t\top} \overline{\boldsymbol{w}}_j^t)\,\overline{\mathsf{w}}_j^{t\top} \boldsymbol{\gamma}_{I_j^\ell} \qquad \text{since } I_+(\boldsymbol{w}_j^t) = I_j^\ell$$

$$\leq 2\lambda^{2-\varepsilon} - (1 - \overline{\mathsf{w}}_j^{t\top} \overline{\boldsymbol{w}}_j^t)\,\big\|\boldsymbol{\gamma}_{I_j^1}\big\| \cos\varphi_j^0 \qquad \text{by Proposition 14 (vi) and (vii)}$$

$$\leq 2\lambda^{2-\varepsilon} - (1 - \overline{\mathsf{w}}_j^{t\top} \overline{\boldsymbol{w}}_j^t)\,\frac{\delta^{7/2}}{\sqrt{2}n} \qquad \text{by Proposition 17 (iii)}$$

$$\leq 4\lambda^{\left(1+2\frac{3\ell-3}{3n_j}\right)\varepsilon}\left(\tfrac{1}{2}\lambda^{2-2\left(1+\frac{3\ell-3}{3n_j}\right)\varepsilon} - (1 - \overline{\mathsf{w}}_j^{t\top} \overline{\boldsymbol{w}}_j^t)\right) \qquad \text{since } \lambda^\varepsilon \leq n^{-27n/\delta^3} < \left(\frac{\delta^3}{n}\right)^{27} < \frac{\delta^{7/2}}{4\sqrt{2}n},$$

so for all $t \in (\mathsf{t}_j^{\ell-1}, \min\{\mathsf{t}_j^\ell, \mathsf{T}\}]$ we have $1 - \overline{\mathsf{w}}_j^{t\top} \overline{\boldsymbol{w}}_j^t \leq \tfrac{1}{2}\lambda^{2-2\left(1+\frac{3\ell-3}{3n_j}\right)\varepsilon}$ and thus $\|\overline{\mathsf{w}}_j^t - \overline{\boldsymbol{w}}_j^t\| \leq \lambda^{1-\left(1+\frac{3\ell-3}{3n_j}\right)\varepsilon}$.

If $s_j = -1$ then for all $t \in (\mathsf{t}_j^{\ell-1}, \min\{\mathsf{t}_j^\ell, \mathsf{T}\})$ we have

$$\mathrm{d}\left(\tfrac{1}{2}\lambda^{2-2\left(1+\frac{3\ell-3}{3n_j}\right)\varepsilon} + (1 - \overline{\mathsf{w}}_j^{t\top} \overline{\boldsymbol{w}}_j^t)\right)/\mathrm{d}t$$

$$= \overline{\mathsf{w}}_j^{t\top} \boldsymbol{g}_j^t + \overline{\boldsymbol{w}}_j^{t\top} \boldsymbol{\gamma}_{I_j^\ell} - \overline{\mathsf{w}}_j^{t\top} \overline{\boldsymbol{w}}_j^t (\overline{\mathsf{w}}_j^{t\top} \boldsymbol{\gamma}_{I_j^\ell} + \overline{\boldsymbol{w}}_j^{t\top} \boldsymbol{g}_j^t) \qquad \text{by Corollary 10 and since } I_0(\boldsymbol{w}_j^t) = \emptyset$$

$$< 2\lambda^{2-\varepsilon} + (1 - \overline{\mathsf{w}}_j^{t\top} \overline{\boldsymbol{w}}_j^t)(\overline{\mathsf{w}}_j^t + \overline{\boldsymbol{w}}_j^t)^\top \boldsymbol{\gamma}_{I_j^\ell} \qquad \text{since } \|\boldsymbol{\gamma}_{I_j^\ell} - \boldsymbol{g}_j^t\| < \lambda^{2-\varepsilon}$$

$$\leq 2\big\|\boldsymbol{\gamma}_{I_j^\ell}\big\|\left(\tfrac{1}{2}\lambda^{2-2\left(1+\frac{3\ell-3}{3n_j}\right)\varepsilon} + (1 - \overline{\mathsf{w}}_j^{t\top} \overline{\boldsymbol{w}}_j^t)\right) \qquad \text{since } \lambda^\varepsilon \leq n^{-27n/\delta^3} < \frac{\delta^{5/2}}{2\sqrt{2}n},$$

so for all $t \in (\mathsf{t}_j^{\ell-1}, \min\{\mathsf{t}_j^\ell, \mathsf{T}\}]$ we have

$$1 - \overline{\mathsf{w}}_j^{t\top} \overline{\boldsymbol{w}}_j^t$$

$$\leq \lambda^{2-2\left(1+\frac{3\ell-3}{3n_j}\right)\varepsilon}\left(\exp\left(2\big\|\boldsymbol{\gamma}_{I_j^\ell}\big\|(t - \mathsf{t}_j^{\ell-1})\right) - \tfrac{1}{2}\right) \qquad \text{by Grönwall's inequality}$$

$$< \lambda^{2-2\left(1+\frac{3\ell-3}{3n_j}\right)\varepsilon}\left(\exp\left(2\big\|\boldsymbol{\gamma}_{I_j^\ell}\big\|(\tau_j^\ell - \tau_j^{\ell-1} + \lambda^{1-2\varepsilon})\right) - \tfrac{1}{2}\right) \qquad \text{by the definitions of } \mathsf{t}_j^0 \text{ and } \mathsf{t}_j^\ell$$

$$< \lambda^{2-2\left(1+\frac{3\ell-3}{3n_j}\right)\varepsilon}\left(\exp\left(3\big\|\boldsymbol{\gamma}_{I_j^\ell}\big\|(\tau_j^\ell - \tau_j^{\ell-1})\right) - \tfrac{1}{2}\right) \qquad \text{since } \lambda^{1-2\varepsilon} \leq n^{-9\cdot6n/\delta^3} < \frac{\delta}{2}$$

$$< \lambda^{2-2\left(1+\frac{3\ell-3}{3n_j}\right)\varepsilon}\left(\frac{8}{\delta^3} - \tfrac{1}{2}\right) \qquad \begin{array}{l}\text{by Proposition 14 (vi)} \\ \text{and Proposition 17 (v)}\end{array}$$

$$< \tfrac{1}{2}\lambda^{2-2\left(1+\frac{3\ell-2}{3n_j}\right)\varepsilon} \qquad \text{since } \lambda^{-\frac{2\varepsilon}{3n}} \geq n^{9\cdot2/\delta^3} > \frac{16}{\delta^3}$$

and thus $\|\overline{\mathsf{w}}_j^t - \overline{\boldsymbol{w}}_j^t\| \leq \lambda^{1-\left(1+\frac{3\ell-2}{3n_j}\right)\varepsilon}$.

For all $i \in [n]$ such that $i \neq i_j^\ell$ and if $\ell \neq 1$ then $i \neq i_j^{\ell-1}$, for all $t \in [\tau_j^\ell, \mathsf{t}_j^\ell]$ we have

$$|\overline{\mathsf{w}}_j^{t\top} \boldsymbol{x}_i| \geq \delta - \left(\max_{t'\in[\tau_j^\ell,t]}\big\|\mathrm{d}\overline{\mathsf{w}}_j^{t'}/\mathrm{d}t'\big\|\right)(t - \tau_j^\ell) \qquad \text{since } |\overline{\mathsf{w}}_j^{t\top} \boldsymbol{x}_i| \geq \delta \text{ for } t = \tau_j^\ell$$

$$\geq \delta - \big\|\boldsymbol{\gamma}_{I_j^\ell}\big\|(t - \tau_j^\ell) \qquad \text{by properties of projection}$$

$$\geq \delta - \Delta^2(t - \tau_j^\ell) \qquad \text{by Proposition 17 (iii)}$$

$$> \delta - \Delta^2\lambda^{1-2\varepsilon} \qquad \text{by the definition of } \mathsf{t}_j^\ell$$

$$> \delta/2 \qquad \text{since } \lambda^{1-2\varepsilon} \leq n^{-9\cdot6n\Delta^2/\delta^3} < \frac{\delta}{2\Delta^2},$$

so for all $t \in (\mathsf{t}_j^{\ell-1}, \min\{\mathsf{t}_j^\ell, \mathsf{T}\}]$ we have $\left|\overline{\boldsymbol{w}}_j^{t\top} \boldsymbol{x}_i\right| > \delta/2 - \lambda^{1-\left(1+\frac{3\ell-2}{3n_j}\right)\varepsilon} > \delta/4$.

If $\ell \neq 1$ then for all $t \in (\mathsf{t}_j^{\ell-1}, \min\{\mathsf{t}_j^\ell, \mathsf{T}\})$ we have

$$
\begin{aligned}
&\mathrm{d}\, s_j\, \overline{\boldsymbol{w}}_j^{t\top} \boldsymbol{x}_{i_j^{\ell-1}} \Big/ \mathrm{d}t \\
&= \boldsymbol{g}_j^{t\top} \boldsymbol{x}_{i_j^{\ell-1}} - \overline{\boldsymbol{w}}_j^{t\top} \boldsymbol{g}_j^t\, \overline{\boldsymbol{w}}_j^{t\top} \boldsymbol{x}_{i_j^{\ell-1}} && \text{by Corollary 10 and since } I_0(\boldsymbol{w}_j^t) = \emptyset \\
&> \boldsymbol{\gamma}_{I_j^\ell}^\top \boldsymbol{x}_{i_j^{\ell-1}} - \left\|\boldsymbol{\gamma}_{I_j^\ell}\right\| \left|\overline{\boldsymbol{w}}_j^{t\top} \boldsymbol{x}_{i_j^{\ell-1}}\right| - 2\lambda^{2-\varepsilon} && \text{since } \left\|\boldsymbol{\gamma}_{I_j^\ell} - \boldsymbol{g}_j^t\right\| < \lambda^{2-\varepsilon} \\
&> \frac{\delta^3}{\sqrt{2n}} - \left\|\boldsymbol{\gamma}_{I_j^\ell}\right\| \left|\overline{\boldsymbol{w}}_j^{t\top} \boldsymbol{x}_{i_j^{\ell-1}}\right| - 2\lambda^{2-\varepsilon} && \text{recalling } \forall i \in [n]\colon \measuredangle(\boldsymbol{v}^*, \boldsymbol{x}_i) < \pi/4 \\
&\geq \frac{\delta^3}{\sqrt{2n}} - \Delta^2 \left|\overline{\boldsymbol{w}}_j^{t\top} \boldsymbol{x}_{i_j^{\ell-1}}\right| - 2\lambda^{2-\varepsilon} && \text{by Proposition 17 (iii)} \\
&> \frac{\delta^3}{2\sqrt{2n}} - \Delta^2 \left|\overline{\boldsymbol{w}}_j^{t\top} \boldsymbol{x}_{i_j^{\ell-1}}\right| && \text{since } \lambda^{2-\varepsilon} \leq n^{-9\cdot 3\cdot 7n/\delta^3} < \frac{\delta^3}{4\sqrt{2n}},
\end{aligned}
$$

so $i_j^{\ell-1} \notin I_0(\boldsymbol{w}_j^t)$ at $t = \min\{\mathsf{t}_j^\ell, \mathsf{T}\}$ since otherwise the continuous curve $\overline{\boldsymbol{w}}_j^{t\top} \boldsymbol{x}_{i_j^{\ell-1}}$ would have different signs to the right of $\mathsf{t}_j^{\ell-1}$ and to the left of $\min\{\mathsf{t}_j^\ell, \mathsf{T}\}$ without crossing zero in between.

Assume for a contradiction that $\mathsf{T} < \mathsf{t}_j^{-\ell}$. Then $I_0(\boldsymbol{w}_j^\mathsf{T}) = \{i_j^\ell\}$. But also

$$
\begin{aligned}
\left|\overline{\mathsf{w}}_j^{\mathsf{T}\top} \boldsymbol{x}_{i_j^\ell}\right| &\geq \left|\overline{\mathsf{w}}_j^{\mathsf{T}\top} \boldsymbol{x}_{i_j^\ell}\right| - \lambda^{1-\left(1+\frac{3\ell-2}{3n_j}\right)\varepsilon} && \text{since } \left\|\overline{\mathsf{w}}_j^\mathsf{T} - \overline{\mathsf{w}}_j^\mathsf{T}\right\| \leq \lambda^{1-\left(1+\frac{3\ell-2}{3n_j}\right)\varepsilon} \\
&= \left|\overline{\boldsymbol{\alpha}}_j^{\mathsf{T}\top} \boldsymbol{x}_{i_j^\ell}\right| - \lambda^{1-\left(1+\frac{3\ell-2}{3n_j}\right)\varepsilon} && \text{since } \mathsf{w}_j^\mathsf{T} = \boldsymbol{\alpha}_j^\mathsf{T} \\
&\geq \frac{2\delta^4}{3n}(\tau_j^\ell - \mathsf{T}) - \lambda^{1-\left(1+\frac{3\ell-2}{3n_j}\right)\varepsilon} && \text{by Proposition 17 (vii)} \\
&> \frac{2\delta^4}{3n}\lambda^{1-\left(1+\frac{3\ell-1}{3n_j}\right)\varepsilon} - \lambda^{1-\left(1+\frac{3\ell-2}{3n_j}\right)\varepsilon} && \text{by the definition of } \mathsf{t}_j^{-\ell} \\
&= \lambda^{1-\left(1+\frac{3\ell-1}{3n_j}\right)\varepsilon}\left(\frac{2\delta^4}{3n} - \lambda^{\frac{\varepsilon}{3n_j}}\right) && \text{calculation} \\
&> 0 && \text{since } \lambda^{\frac{\varepsilon}{3n}} \leq n^{-9/\delta^3} < \frac{\delta^{3\cdot 7}}{n^2}.
\end{aligned}
$$

For all $t \in [\mathsf{t}_j^{-\ell}, \mathsf{t}_j^\ell]$ we have

$$
\begin{aligned}
&\left|\mathrm{d}\,\overline{\mathsf{w}}_j^{t\top} \boldsymbol{x}_{i_j^\ell} \Big/ \mathrm{d}t\right| \\
&= \left|\boldsymbol{\gamma}_{I_j^\ell}^\top \boldsymbol{x}_{i_j^\ell} - \overline{\mathsf{w}}_j^{t\top} \boldsymbol{\gamma}_{I_j^\ell}\, \overline{\mathsf{w}}_j^{t\top} \boldsymbol{x}_{i_j^\ell}\right| && \text{by the definition of } \mathsf{w}_j^t \\
&\geq \left|\boldsymbol{\gamma}_{I_j^\ell}^\top \boldsymbol{x}_{i_j^\ell}\right| - \left|\overline{\mathsf{w}}_j^{t\top} \boldsymbol{\gamma}_{I_j^\ell}\, \overline{\mathsf{w}}_j^{t\top} \boldsymbol{x}_{i_j^\ell}\right| && \text{by properties of absolute value} \\
&> \frac{\delta^3}{\sqrt{2n}} - \left|\overline{\mathsf{w}}_j^{t\top} \boldsymbol{\gamma}_{I_j^\ell}\, \overline{\mathsf{w}}_j^{t\top} \boldsymbol{x}_{i_j^\ell}\right| && \text{recalling } \forall i \in [n]\colon \measuredangle(\boldsymbol{v}^*, \boldsymbol{x}_i) < \pi/4 \\
&\geq \frac{\delta^3}{\sqrt{2n}} - \Delta^2 \left|\overline{\mathsf{w}}_j^{t\top} \boldsymbol{x}_{i_j^\ell}\right| && \text{by Proposition 17 (iii)} \\
&\geq \frac{\delta^3}{\sqrt{2n}} - \Delta^2 \left(\max_{t' \in [\mathsf{t}_j^{-\ell}, \mathsf{t}_j^\ell]} \left\|\mathrm{d}\overline{\mathsf{w}}_j^{t'}/\mathrm{d}t'\right\|\right) |t - \tau_j^\ell| && \text{since } \overline{\mathsf{w}}_j^{t\top} \boldsymbol{x}_{i_j^\ell} = 0 \text{ at } t = \tau_j^\ell \\
&\geq \frac{\delta^3}{\sqrt{2n}} - \Delta^4 |t - \tau_j^\ell| && \text{by properties of projection} \\
&> \frac{\delta^3}{\sqrt{2n}} - \Delta^4 \lambda^{1-2\varepsilon} && \text{by the definitions of } \mathsf{t}_j^{-\ell} \text{ and } \mathsf{t}_j^\ell \\
&> \frac{\delta^3}{2\sqrt{2n}} && \text{since } \lambda^{1-2\varepsilon} \leq n^{-9\cdot 6n\Delta^2/\delta^3} < \frac{\delta^3}{2\sqrt{2n}\Delta^4}
\end{aligned}
$$

$$> \lambda^{\frac{\varepsilon}{3n}} \qquad\qquad\qquad \text{since } \lambda^{\frac{\varepsilon}{3n}} \le n^{-9/\delta^3},$$

so at $t = \mathsf{t}_j^{-\ell}$ and at $t = \mathsf{t}_j^{\ell}$ it holds that $\overline{\mathsf{w}}_j^{t\top} \boldsymbol{x}_{i_j^\ell}$ has different signs and absolute values greater than $\lambda^{\frac{\varepsilon}{3n}} \lambda^{1-\left(1+\frac{3\ell-1}{3n_j}\right)\varepsilon} \ge \lambda^{1-\left(1+\frac{3\ell-2}{3n_j}\right)\varepsilon}$.

Assume for a contradiction that $\mathsf{T} > \mathsf{t}_j^{\ell}$. Then at $t = \mathsf{t}_j^{-\ell}$ and at $t = \mathsf{t}_j^{\ell}$ it holds that $\|\overline{\mathsf{w}}_j^t - \overline{\boldsymbol{w}}_j^t\| \le \lambda^{1-\left(1+\frac{3\ell-2}{3n_j}\right)\varepsilon}$, so $\boldsymbol{w}_j^{t\top} \boldsymbol{x}_{i_j^\ell}$ has different signs.

Therefore $\mathsf{T} \in [\mathsf{t}_j^{-\ell}, \mathsf{t}_j^{\ell}]$ and $I_0(\boldsymbol{w}_j^{\mathsf{T}}) = \{i_j^\ell\}$. Let $t_j^\ell := \mathsf{T}$.

To complete the proof, it suffices to show that, for all $t \in (t_j^\ell, \mathsf{t}_j^{\ell}]$, we have $\|\overline{\boldsymbol{\alpha}}_j^t - \overline{\boldsymbol{w}}_j^t\| \le \lambda^{1-\left(1+\frac{3\ell}{3n_j}\right)\varepsilon}$, and if $\ell \ne n_j$ then $I_+(\boldsymbol{w}_j^t) = I_j^{\ell+1}$ and $I_0(\boldsymbol{w}_j^t) = \emptyset$.

Since at $t = \min\{\tau_j^\ell, t_j^\ell\}$ we have $\|\overline{\boldsymbol{\alpha}}_j^t - \overline{\boldsymbol{w}}_j^t\| = \|\overline{\boldsymbol{w}}_j^t - \overline{\boldsymbol{w}}_j^t\| \le \lambda^{1-\left(1+\frac{3\ell-2}{3n_j}\right)\varepsilon}$, and for almost all $t \in (\min\{\tau_j^\ell, t_j^\ell\}, \mathsf{t}_j^{\ell})$ we have

$$
\begin{aligned}
|\mathrm{d}\|\overline{\boldsymbol{\alpha}}_j^t - \overline{\boldsymbol{w}}_j^t\|/\mathrm{d}t| &\le \|\mathrm{d}(\overline{\boldsymbol{\alpha}}_j^t - \overline{\boldsymbol{w}}_j^t)/\mathrm{d}t\| && \text{by properties of projection} \\
&\le \|\mathrm{d}\overline{\boldsymbol{\alpha}}_j^t/\mathrm{d}t\| + \|\mathrm{d}\overline{\boldsymbol{w}}_j^t/\mathrm{d}t\| && \text{by the triangle inequality} \\
&\le \left(\left\|\boldsymbol{\gamma}_{I_+(\boldsymbol{\alpha}_j^t)}\right\| + \|\boldsymbol{g}_j^t\|\right) && \text{by properties of projection} \\
&\le \left(\left\|\boldsymbol{\gamma}_{I_+(\boldsymbol{\alpha}_j^t)}\right\| + 2\left\|\boldsymbol{\gamma}_{I_+(\boldsymbol{w}_j^t)\cup I_0(\boldsymbol{w}_j^t)}\right\|\right) && \text{since } \forall i \in [n]\colon |h_{\boldsymbol{\theta}^t}(\boldsymbol{x}_i)| \le y_i \\
& && \text{by the proof of Lemma 19 (ii)} \\
&\le 3\Delta^2 && \text{by Proposition 17 (iii),}
\end{aligned}
$$

it follows that for all $t \in [\min\{\tau_j^\ell, t_j^\ell\}, \mathsf{t}_j^{\ell}]$ we have

$$
\begin{aligned}
\|\overline{\boldsymbol{\alpha}}_j^t - \overline{\boldsymbol{w}}_j^t\| &\le \lambda^{1-\left(1+\frac{3\ell-2}{3n_j}\right)\varepsilon} + 6\Delta^2 \lambda^{1-\left(1+\frac{3\ell-1}{3n_j}\right)\varepsilon} && \text{by the definitions of } \mathsf{t}_j^{-\ell} \text{ and } \mathsf{t}_j^{\ell} \\
&< 7\Delta^2 \lambda^{1-\left(1+\frac{3\ell-1}{3n_j}\right)\varepsilon} && \text{since } \lambda^{1-\left(1+\frac{3\ell-2}{3n_j}\right)\varepsilon} < \lambda^{1-\left(1+\frac{3\ell-1}{3n_j}\right)\varepsilon} \\
&< \lambda^{1-\left(1+\frac{3\ell}{3n_j}\right)\varepsilon} && \text{since } \lambda^{-\frac{\varepsilon}{3n}} \ge n^{9\Delta^2} > n^3\Delta^{2\cdot6}.
\end{aligned}
$$

If $\ell \ne n_j$ then for all $i \ne i_j^\ell$ and all $t \in [\mathsf{t}_j^{-\ell}, \mathsf{t}_j^{\ell}]$ we have

$$|\boldsymbol{\alpha}_j^{t\top} \boldsymbol{x}_i| \ge \delta - \Delta^2|t - \tau_j^\ell| > \delta - \Delta^2\lambda^{1-2\varepsilon} > \delta/2 ,$$

so for all $t \in (t_j^\ell, \mathsf{t}_j^{\ell}]$ we have $|\overline{\boldsymbol{w}}_j^{t\top} \boldsymbol{x}_i| > \delta/2 - \lambda^{1-\left(1+\frac{3\ell}{3n_j}\right)\varepsilon} > \delta/4$. Also, for almost all $t \in (t_j^\ell, \mathsf{t}_j^{\ell})$ the following holds, where $I := I_j^\ell \cap I_j^{\ell+1}$:

$$
\begin{aligned}
\mathrm{d}\, s_j \overline{\boldsymbol{w}}_j^{t\top} \boldsymbol{x}_{i_j^\ell} \Big/ \mathrm{d}t & \\
= \boldsymbol{g}_j^{t\top} \overline{\boldsymbol{x}}_{i_j^\ell} - \overline{\boldsymbol{w}}_j^{t\top} \boldsymbol{g}_j^t \, \overline{\boldsymbol{w}}_j^{t\top} \overline{\boldsymbol{x}}_{i_j^\ell} & \text{by Corollary 10} \\
> \mathsf{g}_j^{t\top} \overline{\boldsymbol{x}}_{i_j^\ell} - \overline{\boldsymbol{w}}_j^{t\top} \boldsymbol{g}_j^t \left|\overline{\boldsymbol{w}}_j^{t\top} \overline{\boldsymbol{x}}_{i_j^\ell}\right| - 2\lambda^{2-\varepsilon} & \text{since } \tfrac{1}{n}\sum_{i=1}^n |h_{\boldsymbol{\theta}^t}(\boldsymbol{x}_i)|\|\boldsymbol{x}_i\| < \lambda^{2-\varepsilon}, \\
& \text{where } \exists\varsigma_j^t \in [0,1]\colon \boldsymbol{g}_j^t = \boldsymbol{\gamma}_I + \tfrac{1}{n}y_{i_j^\ell}\varsigma_j^t \boldsymbol{x}_{i_j^\ell} \\
\ge \boldsymbol{\gamma}_I^\top \overline{\boldsymbol{x}}_{i_j^\ell} - \overline{\boldsymbol{w}}_j^{t\top} \boldsymbol{\gamma}_I \left|\overline{\boldsymbol{w}}_j^{t\top} \overline{\boldsymbol{x}}_{i_j^\ell}\right| - 2\lambda^{2-\varepsilon} & \text{since } \boldsymbol{x}_{i_j^\ell}^\top \overline{\boldsymbol{x}}_{i_j^\ell} \ge \overline{\boldsymbol{w}}_j^{t\top} \boldsymbol{x}_{i_j^\ell} \left|\overline{\boldsymbol{w}}_j^{t\top} \overline{\boldsymbol{x}}_{i_j^\ell}\right| \\
\ge \boldsymbol{\gamma}_I^\top \overline{\boldsymbol{x}}_{i_j^\ell} - \|\boldsymbol{\gamma}_I\| \left|\overline{\boldsymbol{w}}_j^{t\top} \overline{\boldsymbol{x}}_{i_j^\ell}\right| - 2\lambda^{2-\varepsilon} & \text{since } \overline{\boldsymbol{w}}_j^{t\top} \boldsymbol{\gamma}_I \le \|\boldsymbol{\gamma}_I\| \\
> \frac{\delta^3}{\sqrt{2}n} - \|\boldsymbol{\gamma}_I\| \left|\overline{\boldsymbol{w}}_j^{t\top} \overline{\boldsymbol{x}}_{i_j^\ell}\right| - 2\lambda^{2-\varepsilon} & \text{recalling } \forall i \in [n]\colon \angle(\boldsymbol{v}^*, \boldsymbol{x}_i) < \pi/4 \\
\ge \frac{\delta^3}{\sqrt{2}n} - \Delta^2 \left|\overline{\boldsymbol{w}}_j^{t\top} \overline{\boldsymbol{x}}_{i_j^\ell}\right| - 2\lambda^{2-\varepsilon} & \text{by Proposition 17 (iii)} \\
> \frac{\delta^3}{2\sqrt{2}n} - \Delta^2 \left|\overline{\boldsymbol{w}}_j^{t\top} \overline{\boldsymbol{x}}_{i_j^\ell}\right| & \text{since } \lambda^{2-\varepsilon} \le n^{-9\cdot3\cdot7n/\delta^3} < \frac{\delta^3}{4\sqrt{2}n}.
\end{aligned}
$$

Hence $i_j^\ell \notin I_0\big(\boldsymbol{w}_j^{t'}\big)$ for all $t' \in (t_j^\ell, \mathsf{t}_j^\ell]$ since otherwise the continuous curve $\overline{\boldsymbol{w}}_j^{t\,\top} \boldsymbol{x}_{i_j^\ell}$ would have different signs to the right of $t_j^\ell$ and to the left of the smallest such $t'$ without crossing zero in between. $\qquad\square$

For each positive-sign hidden neuron, after the completion of all of the intermediate alignment stages (if any), its yardstick vector proceeds to align to the vector $\boldsymbol{\gamma}_{[n]}$. For the cosine of the angle between them, whose starting point is the angle $\varphi_j^{n_j^+}$ defined below, from the proof of Proposition 14 (vi) we obtain the following expression, which will be useful in the proof of the next lemma. We also remark that the angle $\varphi_j^{n_j^+}$ already featured in Proposition 14 (viii).

**Proposition 22.** *For all $j \in J_+$ and all $t > \tau_j^{n_j}$ we have $I_+(\boldsymbol{\alpha}_j^t) = [n]$ and*

$$\cos \varphi_j^t = \tanh\left(\operatorname{artanh} \cos \varphi_j^{n_j^+} + \|\boldsymbol{\gamma}_{[n]}\|(t - \tau_j^{n_j})\right),$$

*where $\varphi_j^{n_j^+} := \lim_{t \to \left(\tau_j^{n_j}\right)^+} \varphi_j^t$.*

The final lemma in this section establishes non-asymptotic bounds for the final alignment stage in the first phase of the training, which we consider to end at time $T_1$. During it, each positive-sign hidden neuron: continues to grow but keeps its length below $2\|\boldsymbol{z}_j\|\lambda^{1-\varepsilon}$, aligns to the vector $\boldsymbol{\gamma}_{[n]}$ up to a cosine of at least $1 - \lambda^\varepsilon$, and maintains bounded by $\lambda^{1-3\varepsilon}$ the difference between the logarithm of its length divided by the initialisation scale and the logarithm of the corresponding yardstick vector length. Establishing the latter bound, which is stated in part (iv) and will be instrumental in the proof of the implicit bias (cf. Lemma 34), involves putting together the bounds in Lemma 21 (iii), (iv), and (v), and Lemma 23 (i) and (ii) on the dynamics-governing vectors, the unit-sphere normalisations, and the boundary crossing times, over the lengthy time period up to $T_1$ which depends on the initialisation scale $\lambda$.

**Lemma 23.** *For all $j \in J_+$ we have:*

(i) $\|\boldsymbol{\gamma}_{[n]} - \boldsymbol{g}_j^t\| \leq \lambda^{2-3\varepsilon}$ *for all $t \in (t_j^{n_j}, T_1]$;*

(ii) $\|\overline{\boldsymbol{\alpha}}_j^t - \overline{\boldsymbol{w}}_j^t\| \leq \lambda^{1-2\varepsilon}$ *for all $t \in (t_j^{n_j}, T_1]$;*

(iii) $\overline{\boldsymbol{w}}_j^{T_1\,\top} \overline{\boldsymbol{\gamma}}_{[n]} \geq 1 - \lambda^\varepsilon$;

(iv) $\left|\ln \|\boldsymbol{\alpha}_j^{T_1}\| - \ln \|\boldsymbol{w}_j^{T_1}/\lambda\|\right| \leq \lambda^{1-3\varepsilon}$.

*Proof.* For all $t \in [0, T_1]$ we have

$$\frac{1}{n}\sum_{i=1}^n |h_{\boldsymbol{\theta}^t}(\boldsymbol{x}_i)|\|\boldsymbol{x}_i\| \leq m\left(\max_{j=1}^m \|\boldsymbol{w}_j^t\|^2\right)\left(\max_{i=1}^n \|\boldsymbol{x}_i\|^2\right) \quad \text{by the definition of } h_{\boldsymbol{\theta}^t} \text{ in section 2}$$

$$< 4m\Delta^4 \lambda^{2-2\varepsilon} \qquad\qquad\qquad\qquad\qquad \text{by Lemma 19}$$

$$< \lambda^{2-3\varepsilon} \qquad\qquad\qquad\qquad\qquad\qquad \text{since } \lambda^{-\varepsilon} \geq m^3 \, n^{9\cdot3n\Delta^2/\delta^3} > 4m\Delta^4.$$

Suppose $j \in J_+$.

Assume for a contradiction that $I_0(\boldsymbol{w}_j^t) \neq \emptyset$ for some $t \in (t_j^{n_j}, T_1]$, and let $\mathsf{T} > t_j^{n_j}$ be the smallest such that $\boldsymbol{w}_j^{\mathsf{T}\,\top} \boldsymbol{x}_i = 0$ for some $i \in [n]$. Then for all $t \in (t_j^{n_j}, \mathsf{T})$ we have

$$\mathrm{d}\,\overline{\boldsymbol{w}}_j^{t\,\top} \boldsymbol{x}_i/\mathrm{d}t = \boldsymbol{g}_j^{t\,\top} \boldsymbol{x}_i - \overline{\boldsymbol{w}}_j^{t\,\top} \boldsymbol{g}_j^t \, \overline{\boldsymbol{w}}_j^{t\,\top} \boldsymbol{x}_i \qquad\qquad \text{by Corollary 10 and since } I_0(\boldsymbol{w}_j^t) = \emptyset$$

$$> \boldsymbol{\gamma}_{[n]}^\top \boldsymbol{x}_i - \|\boldsymbol{\gamma}_{[n]}\| \, \overline{\boldsymbol{w}}_j^{t\,\top} \boldsymbol{x}_i - 2\lambda^{2-3\varepsilon} \qquad \text{since } \|\boldsymbol{\gamma}_{[n]} - \boldsymbol{g}_j^t\| < \lambda^{2-3\varepsilon}$$

$$\geq \delta^3 - \Delta^2 \, \overline{\boldsymbol{w}}_j^{t\,\top} \boldsymbol{x}_i - 2\lambda^{2-3\varepsilon} \qquad\qquad \text{by Proposition 17 (iii)}$$

$$> \frac{\delta^3}{2} - \Delta^2 \, \overline{\boldsymbol{w}}_j^{t\,\top} \boldsymbol{x}_i \qquad\qquad\qquad \text{since } \lambda^{2-3\varepsilon} \leq n^{-9\cdot3\cdot5n/\delta^3} < \frac{\delta^3}{4},$$

so the continuous curve $\overline{\boldsymbol{w}}_j^{t\top} \boldsymbol{x}_i$ is positive to the right of $t_j^{n_j}$, negative to the left of $\mathsf{T}$, and does not cross zero in between.

Hence $I_0(\boldsymbol{w}_j^t) = \emptyset$ for all $t \in (t_j^{n_j}, T_1]$, which together with the inequality $\frac{1}{n}\sum_{i=1}^n |h_{\boldsymbol{\theta}^t}(\boldsymbol{x}_i)| \|\boldsymbol{x}_i\| < \lambda^{2-3\varepsilon}$ establishes part (i).

By Lemma 21 (iv), for all $t \in [t_j^{n_j}, \max\{t_j^{n_j}, \tau_j^{n_j}\}]$ we have $1 - \overline{\boldsymbol{\alpha}}_j^{t\top} \overline{\boldsymbol{w}}_j^t \le \frac{1}{2}\lambda^{2-4\varepsilon}$.

Moreover, for all $t \in (\max\{t_j^{n_j}, \tau_j^{n_j}\}, T_1)$ we have

$$
\begin{aligned}
&\mathrm{d}(1 - \overline{\boldsymbol{\alpha}}_j^{t\top} \overline{\boldsymbol{w}}_j^t)/\mathrm{d}t \\
&= -\overline{\boldsymbol{\alpha}}_j^{t\top} \boldsymbol{g}_j^t - \overline{\boldsymbol{w}}_j^{t\top} \boldsymbol{\gamma}_{[n]} + \overline{\boldsymbol{\alpha}}_j^{t\top} \overline{\boldsymbol{w}}_j^t (\overline{\boldsymbol{\alpha}}_j^{t\top} \boldsymbol{\gamma}_{[n]} + \overline{\boldsymbol{w}}_j^{t\top} \boldsymbol{g}_j^t) && \text{by Corollary 10 and Proposition 22} \\
&< 2\lambda^{2-3\varepsilon} - (1 - \overline{\boldsymbol{\alpha}}_j^{t\top} \overline{\boldsymbol{w}}_j^t)(\overline{\boldsymbol{\alpha}}_j^t + \overline{\boldsymbol{w}}_j^t)^\top \boldsymbol{\gamma}_{[n]} && \text{since } \|\boldsymbol{\gamma}_{[n]} - \boldsymbol{g}_j^t\| < \lambda^{2-3\varepsilon} \\
&\le 2\lambda^{2-3\varepsilon} - (1 - \overline{\boldsymbol{\alpha}}_j^{t\top} \overline{\boldsymbol{w}}_j^t)\frac{\delta^{7/2}}{\sqrt{2n}} && \text{by Proposition 14 (vi), (vii), and (viii),} \\
&&& \text{and Proposition 17 (iii)} \\
&\le 4\lambda^\varepsilon \left(\frac{1}{2}\lambda^{2-4\varepsilon} - (1 - \overline{\boldsymbol{\alpha}}_j^{t\top} \overline{\boldsymbol{w}}_j^t)\right) && \text{since } \lambda^{-\varepsilon} \ge n^{9\cdot 3n/\delta^3} > \frac{4\sqrt{2n}}{\delta^{7/2}}.
\end{aligned}
$$

Hence for all $t \in (t_j^{n_j}, T_1]$ we have $1 - \overline{\boldsymbol{\alpha}}_j^{t\top} \overline{\boldsymbol{w}}_j^t \le \frac{1}{2}\lambda^{2-4\varepsilon}$ and thus $\|\overline{\boldsymbol{\alpha}}_j^t - \overline{\boldsymbol{w}}_j^t\| \le \lambda^{1-2\varepsilon}$, establishing part (ii).

By Proposition 18 and Proposition 22 we have

$$
\overline{\boldsymbol{\alpha}}_j^{T_1\top} \overline{\boldsymbol{\gamma}}_{[n]} > \tanh\left(\tfrac{2}{3}\|\boldsymbol{\gamma}_{[n]}\| T_1\right) = \tanh\left(\tfrac{2\varepsilon}{3}\ln\left(\tfrac{1}{\lambda}\right)\right) > 1 - 2\lambda^{\frac{4\varepsilon}{3}},
$$

so $\overline{\boldsymbol{w}}_j^{T_1\top} \overline{\boldsymbol{\gamma}}_{[n]} > 1 - 2\lambda^{\frac{4\varepsilon}{3}} - \lambda^{1-2\varepsilon} = 1 - \lambda^\varepsilon \left(2\lambda^{\frac{\varepsilon}{3}} + \lambda^{1-3\varepsilon}\right) > 1 - \lambda^\varepsilon$, establishing part (iii).

Recalling Lemma 21, for all $t \in [0, T_1]$ we have

$$
\left\|\boldsymbol{\gamma}_{I_+(\boldsymbol{\alpha}_j^t)} - \boldsymbol{g}_j^t\right\| \le \begin{cases} \frac{\Delta^2}{n} + \lambda^{2-3\varepsilon} < \frac{2\Delta^2}{n} & \text{if } \exists \ell \in [n_j]: |t - \tau_j^\ell| \le \lambda^{1-2\varepsilon}, \\ \lambda^{2-3\varepsilon} & \text{otherwise} \end{cases}
$$

$$
\|\overline{\boldsymbol{\alpha}}_j^t - \overline{\boldsymbol{w}}_j^t\| \le \lambda^{1-2\varepsilon},
$$

so for almost all $t \in [0, T_1]$ we have

$$
\begin{aligned}
&\left|\frac{\mathrm{d}\ln\|\boldsymbol{\alpha}_j^t\|}{\mathrm{d}t} - \frac{\mathrm{d}\ln\|\boldsymbol{w}_j^t/\lambda\|}{\mathrm{d}t}\right| \\
&= \left|\overline{\boldsymbol{\alpha}}_j^{t\top} \boldsymbol{\gamma}_{I_+(\boldsymbol{\alpha}_j^t)} - \overline{\boldsymbol{w}}_j^{t\top} \boldsymbol{g}_j^t\right| && \text{by Corollary 10} \\
&\le \left|(\overline{\boldsymbol{\alpha}}_j^t - \overline{\boldsymbol{w}}_j^t)^\top \boldsymbol{\gamma}_{I_+(\boldsymbol{\alpha}_j^t)}\right| + \left|\overline{\boldsymbol{w}}_j^{t\top}\left(\boldsymbol{\gamma}_{I_+(\boldsymbol{\alpha}_j^t)} - \boldsymbol{g}_j^t\right)\right| && \text{by properties of absolute value} \\
&\le \lambda^{1-2\varepsilon}\Delta^2 + \begin{cases} \frac{2\Delta^2}{n} & \text{if } \exists \ell \in [n_j]: |t - \tau_j^\ell| \le \lambda^{1-2\varepsilon}, \\ \lambda^{2-3\varepsilon} & \text{otherwise} \end{cases} && \text{by Proposition 17 (iii)} \\
&< \begin{cases} \frac{3\Delta^2}{n} & \text{if } \exists \ell \in [n_j]: |t - \tau_j^\ell| \le \lambda^{1-2\varepsilon}, \\ \lambda^{1-\frac{5\varepsilon}{2}} & \text{otherwise} \end{cases} && \text{since } \lambda^{-\frac{\varepsilon}{2}} \ge n^{\frac{9\cdot 3}{2}n\Delta^2} > 2\Delta^2,
\end{aligned}
$$

and therefore

$$
\begin{aligned}
&|\ln\|\boldsymbol{\alpha}_j^{T_1}\| - \ln\|\boldsymbol{w}_j^{T_1}/\lambda\|| \\
&\le 2n\lambda^{1-2\varepsilon}\frac{3\Delta^2}{n} + (T_1 - 2n\lambda^{1-2\varepsilon})\lambda^{1-\frac{5\varepsilon}{2}} && \text{by the previous inequality} \\
&< 6\Delta^2\lambda^{1-2\varepsilon} + T_1\lambda^{1-\frac{5\varepsilon}{2}} && \text{omitting the negative term} \\
&< \frac{1}{2}\lambda^{1-3\varepsilon} + \frac{1}{\delta^2}\lambda^{1-\frac{5\varepsilon}{2}}\ln(1/\lambda^\varepsilon) && \text{since } \lambda^{-\varepsilon} > 12\Delta^2 \text{ and by Proposition 17 (iii)} \\
&< \frac{1}{2}\lambda^{1-3\varepsilon} + \frac{3}{\delta^2}\lambda^{1-\frac{8\varepsilon}{3}} && \text{since } 3\lambda^{-\frac{\varepsilon}{6}} = 3\sqrt[6]{1/\lambda^\varepsilon} > \ln(1/\lambda^\varepsilon) \\
&< \lambda^{1-3\varepsilon} && \text{since } \lambda^{-\frac{\varepsilon}{3}} \ge n^{9n/\delta^3} > \frac{6}{\delta^2},
\end{aligned}
$$

establishing part (iv). □

# E  Proofs for the second phase

Here we prove a number of results which culminate in Lemma 32 below, whose parts (ii) and (v) establish Theorem 6.

We begin by observing that the eigenvector of the largest eigenvalue of the matrix $\frac{1}{n}\boldsymbol{X}\boldsymbol{X}^\top$ is in the interior of the cone spanned by the training points.

**Proposition 24.** $\boldsymbol{u}_1 \in \mathrm{int}(\mathrm{cone}\{\boldsymbol{x}_1,\ldots,\boldsymbol{x}_n\})$.

*Proof.* If $\boldsymbol{v} \in \mathrm{cone}\{\boldsymbol{x}_1,\ldots,\boldsymbol{x}_n\} \setminus \{\boldsymbol{0}\}$, i.e. $\boldsymbol{v} = \sum_{i=1}^n \beta_i \boldsymbol{x}_i$ for some $\beta_1,\ldots,\beta_n \geq 0$ that are not all zero, then

$$\frac{1}{n}\boldsymbol{X}\boldsymbol{X}^\top \boldsymbol{v} = \frac{1}{n}\sum_{i'=1}^n \left(\sum_{i=1}^n \beta_i \boldsymbol{x}_{i'}^\top \boldsymbol{x}_i\right)\boldsymbol{x}_{i'} \in \mathrm{int}(\mathrm{cone}\{\boldsymbol{x}_1,\ldots,\boldsymbol{x}_n\}) .$$

Thus $\frac{1}{n}\boldsymbol{X}\boldsymbol{X}^\top$ maps $\mathrm{cone}\{\boldsymbol{x}_1,\ldots,\boldsymbol{x}_n\} \setminus \{\boldsymbol{0}\}$ into $\mathrm{int}(\mathrm{cone}\{\boldsymbol{x}_1,\ldots,\boldsymbol{x}_n\})$.

Let $\boldsymbol{v}_0 := \boldsymbol{v}^*$, and $\boldsymbol{v}_{\ell+1} := \frac{1}{n}\boldsymbol{X}\boldsymbol{X}^\top \boldsymbol{v}_\ell$ for all $\ell \in \mathbb{N}$.

Recalling Proposition 16, we have $\boldsymbol{v}_1 = \gamma_{[n]} \in \mathrm{int}(\mathrm{cone}\{\boldsymbol{x}_1,\ldots,\boldsymbol{x}_n\}) \subseteq \mathrm{cone}\{\boldsymbol{x}_1,\ldots,\boldsymbol{x}_n\} \setminus \{\boldsymbol{0}\}$, and so $\boldsymbol{v}_\ell \in \mathrm{int}(\mathrm{cone}\{\boldsymbol{x}_1,\ldots,\boldsymbol{x}_n\})$ for all $\ell \geq 1$.

Since $\boldsymbol{v}_\ell = \sum_{k=1}^d \eta_k^\ell \nu_k \boldsymbol{u}_k$ and $\eta_1$ is strictly the largest eigenvalue, we have that $\angle(\boldsymbol{u}_1, \boldsymbol{v}_\ell) \to 0$ as $\ell \to \infty$. Therefore $\boldsymbol{u}_1 \in \mathrm{cone}\{\boldsymbol{x}_1,\ldots,\boldsymbol{x}_n\} \setminus \{\boldsymbol{0}\}$, but since $\frac{1}{n}\boldsymbol{X}\boldsymbol{X}^\top \boldsymbol{u}_1 = \eta_1 \boldsymbol{u}_1$, in fact $\boldsymbol{u}_1 \in \mathrm{int}(\mathrm{cone}\{\boldsymbol{x}_1,\ldots,\boldsymbol{x}_n\})$. $\qquad\square$

Our next observation is that the key set $\mathcal{S}$ defined in section 5 is strictly contained in the ball with centre $\boldsymbol{v}^*/2$ and radius $\|\boldsymbol{v}^*\|/2 = 1/2$, i.e. that passes through the origin and the teacher neuron, and is centred half-way between them.

**Proposition 25.** *For all $\boldsymbol{v} \in \mathcal{S}$ we have $\boldsymbol{v}^\top(\boldsymbol{v}^* - \boldsymbol{v}) > 0$.*

*Proof.* Suppose $\boldsymbol{v} = \sum_{k=1}^d \nu_k \boldsymbol{u}_k \in \mathcal{S}_\ell$ for some $\ell \in [d]$. Then

$$\boldsymbol{v}^\top(\boldsymbol{v}^* - \boldsymbol{v}) = \sum_{k=1}^d \nu_k(\nu_k^* - \nu_k) > \sum_{k=1}^d \frac{\eta_k}{\eta_\ell}\nu_k(\nu_k^* - \nu_k) = \frac{1}{\eta_\ell}\boldsymbol{v}^\top \frac{1}{n}\boldsymbol{X}\boldsymbol{X}^\top(\boldsymbol{v}^* - \boldsymbol{v}) > 0 . \quad\square$$

The angles between a vector $\boldsymbol{v}$ in $\mathcal{S}$ and the vector obtained by applying the operator $\frac{1}{n}\boldsymbol{X}\boldsymbol{X}^\top$ to the vector $\boldsymbol{v}^* - \boldsymbol{v}$ will be important in what follows. We now show that, if $\boldsymbol{v}$ is in the subset $\mathcal{S}_1$ (also defined in section 5), then the cosine of that angle has a positive lower bound that does not depend on the initialisation scale $\lambda$.

**Proposition 26.** *For all $\boldsymbol{v} = \sum_{k=1}^d \nu_k \boldsymbol{u}_k \in \mathcal{S}_1$ we have*

$$\overline{\boldsymbol{v}}^\top \,\overline{\boldsymbol{X}\boldsymbol{X}^\top(\boldsymbol{v}^* - \boldsymbol{v})} > \frac{1}{2}\left(\frac{\eta_d \nu_d^*}{\|\gamma_{[n]}\|}\right)^2 .$$

*Proof.* Observe that

$$\overline{\boldsymbol{v}}^\top \,\overline{\boldsymbol{X}\boldsymbol{X}^\top(\boldsymbol{v}^* - \boldsymbol{v})} = \overline{\frac{\nu_1^*}{\nu_1}\boldsymbol{v}}^\top \frac{1}{n}\boldsymbol{X}\boldsymbol{X}^\top \frac{\nu_d^*}{\nu_d^* - \nu_d}(\boldsymbol{v}^* - \boldsymbol{v})$$

$$> \frac{\eta_1 \sum_{k=1}^d \eta_k \nu_k^{*2}\frac{\nu_k}{\nu_k^*}\left(1 - \frac{\nu_k}{\nu_k^*}\right)}{\frac{\nu_1}{\nu_1^*}\left(1 - \frac{\nu_d}{\nu_d^*}\right)\|\gamma_{[n]}\|^2}$$

$$> \eta_1 \eta_d \frac{\frac{\nu_d}{\nu_d^*}}{\frac{\nu_1}{\nu_1^*}}\frac{\nu_d^{*2}}{\|\gamma_{[n]}\|^2}$$

$$> \frac{1}{2}\left(\frac{\eta_d \nu_d^*}{\|\gamma_{[n]}\|}\right)^2 . \qquad\square$$

Our final preparatory result is a positive lower bound, however depending on $\lambda$, on the cosine of every angle between a vector $\boldsymbol{v}$ in $\mathcal{S}$ and a training point. The proof relies on the correlation property of our datasets, i.e. that the angles between the training points and the teacher neuron are less than $\pi/4$.

**Proposition 27.** $\overline{\boldsymbol{v}}^{\top}\overline{\boldsymbol{x}}_i > \sqrt{8}\lambda^{\varepsilon/2}$ for all $\boldsymbol{v} \in \mathcal{S}$ and all $i \in [n]$.

*Proof.* Suppose $\boldsymbol{v} = \sum_{k=1}^{d} \nu_k \boldsymbol{u}_k \in \mathcal{S}$.

It suffices to establish that $\cos\angle(\boldsymbol{v}^*, \boldsymbol{v}) \geq \frac{1}{\sqrt{2}} + 2\lambda^{\varepsilon/2}$, because it implies that for all $i \in [n]$ we have

$$\cos\angle(\boldsymbol{v}, \boldsymbol{x}_i) \geq \cos\angle(\boldsymbol{v}^*, \boldsymbol{v})\cos\angle(\boldsymbol{v}^*, \boldsymbol{x}_i) - \sin\angle(\boldsymbol{v}^*, \boldsymbol{v})\sin\angle(\boldsymbol{v}^*, \boldsymbol{x}_i)$$

$$> \frac{1}{\sqrt{2}}\left(\frac{1}{\sqrt{2}} + 2\lambda^{\varepsilon/2}\right) - \frac{1}{\sqrt{2}}\sqrt{1 - \left(\frac{1}{\sqrt{2}} + 2\lambda^{\varepsilon/2}\right)^2}$$

$$= \frac{1}{2} + \sqrt{2}\lambda^{\varepsilon/2} - \sqrt{\frac{1}{2} - \left(\frac{1}{2} + \sqrt{2}\lambda^{\varepsilon/2}\right)^2}$$

$$> \frac{1}{2} + \sqrt{2}\lambda^{\varepsilon/2} - \sqrt{\frac{1}{4} - \sqrt{2}\lambda^{\varepsilon/2}}$$

$$> \frac{1}{2} + \sqrt{2}\lambda^{\varepsilon/2} - \left(\frac{1}{2} - \sqrt{2}\lambda^{\varepsilon/2}\right)$$

$$= \sqrt{8}\lambda^{\varepsilon/2}\,.$$

By Proposition 24, we have $\boldsymbol{u}_1^{\top}\boldsymbol{v}^* > 1/\sqrt{2}$.

If $\boldsymbol{v} \in \mathcal{S}_1$ then

$$\left\|\boldsymbol{v}^* - \frac{\nu_1^*}{\nu_1}\boldsymbol{v}\right\|^2 = \sum_{k=2}^{d}\left(1 - \frac{\nu_1^*}{\nu_1}\frac{\nu_k}{\nu_k^*}\right)^2 \nu_k^{*2} < \sum_{k=2}^{d}\left(1 - \frac{\eta_k}{2\eta_1}\right)^2 \nu_k^{*2} \leq \left(1 - \frac{\eta_d}{2\eta_1}\right)^2 \|\boldsymbol{v}^* - \nu_1^*\boldsymbol{u}_1\|^2\,,$$

so we have

$$\cos\angle(\boldsymbol{v}^*, \boldsymbol{v}) > \sqrt{1 - \frac{1}{2}\left(1 - \frac{\eta_d}{2\eta_1}\right)^2} = \sqrt{\frac{1}{2} + \frac{\eta_d}{2\eta_1} - \frac{\eta_d^2}{8\eta_1^2}} > \sqrt{\frac{1}{2} + \frac{3\eta_d}{8\eta_1}}$$

$$> \frac{1}{\sqrt{2}} + (2 - \sqrt{2})\frac{3\eta_d}{8\eta_1} \geq \frac{1}{\sqrt{2}} + (2 - \sqrt{2})\frac{3\delta^2}{8\Delta^2} > \frac{1}{\sqrt{2}} + 2\lambda^{\varepsilon/2}\,.$$

Otherwise $\boldsymbol{v} \in \mathcal{S}_\ell$ for some $\ell \neq 1$. Then $(\boldsymbol{v} - \nu_1^*\boldsymbol{u}_1)^{\top}(\boldsymbol{v}^* - \boldsymbol{v}) > \boldsymbol{v}^{\top}(\boldsymbol{v}^* - \boldsymbol{v}) > 0$ by Proposition 25. Also $(\boldsymbol{v} - \nu_1^*\boldsymbol{u}_1)^{\top}(\boldsymbol{v}^* - \nu_1^*\boldsymbol{u}_1) = \sum_{k=2}^{d}\nu_k\nu_k^* > \sum_{k=2}^{d}\frac{\eta_k}{2\eta_1}\nu_k^{*2} \geq \frac{\eta_d}{2\eta_1}\|\boldsymbol{v}^* - \nu_1^*\boldsymbol{u}_1\|^2$. Hence

$$\|\boldsymbol{v}^* - \boldsymbol{v}\|^2 \leq \|\boldsymbol{v}^* - \nu_1^*\boldsymbol{u}_1\|^2 - \|\boldsymbol{v} - \nu_1^*\boldsymbol{u}_1\|^2 < \left(1 - \left(\frac{\eta_d}{2\eta_1}\right)^2\right)\|\boldsymbol{v}^* - \nu_1^*\boldsymbol{u}_1\|^2\,,$$

so we have

$$\cos\angle(\boldsymbol{v}^*, \boldsymbol{v}) > \sqrt{\frac{1}{2} + \frac{1}{2}\left(\frac{\eta_d}{2\eta_1}\right)^2} > \frac{1}{\sqrt{2}} + \frac{2 - \sqrt{2}}{2}\left(\frac{\eta_d}{2\eta_1}\right)^2$$

$$\geq \frac{1}{\sqrt{2}} + \frac{2 - \sqrt{2}}{2}\left(\frac{\delta^2}{2\Delta^2}\right)^2 > \frac{1}{\sqrt{2}} + 2\lambda^{\varepsilon/2}\,. \qquad \square$$

For all $t \geq T_1$, recall from section 5 that $\boldsymbol{v}^t = \sum_{j \in J_+} a_j^t \boldsymbol{w}_j^t$, and let

$$\boldsymbol{g}^t := \frac{1}{n}\boldsymbol{X}\boldsymbol{X}^{\top}(\boldsymbol{v}^* - \boldsymbol{v}^t) \qquad\qquad \boldsymbol{f}^t := \|\boldsymbol{v}^t\|(\boldsymbol{g}^t + \overline{\boldsymbol{v}}^t\overline{\boldsymbol{v}}^{t\top}\boldsymbol{g}^t)\,.$$

The next lemma is at the heart of our analysis of the training dynamics. It establishes several key facts that hold at all times $t$ from the start $T_1$ of the second phase, and which form the statement of the lemma as follows.

*Parts (i) and (ii).* The cosines of all angles between hidden neurons that form the aligned bundle remain above $1 - 4\lambda^\varepsilon$, and the bundle vector $\boldsymbol{v}^t$ which was defined as the sum of the constituent hidden neurons multiplied by their last-layer weights stays in the set $\mathcal{S}$. These two properties support each other, e.g. we show that the containment in $\mathcal{S}$ implies that the gradients of the individual hidden neurons are such that the bundle keeps together rather than breaks apart.

*Parts (iii) and (iv).* The network acts linearly on the training points, namely its outputs for the training points equal their inner products with the bundle vector. Moreover, the vectors $\boldsymbol{g}_j^t$ (defined in Proposition 1 (i)) that govern the dynamics are all equal to the vector $\boldsymbol{g}^t$ which is obtained by applying the operator $\frac{1}{n}\boldsymbol{X}\boldsymbol{X}^\top$ to the vector $\boldsymbol{v}^* - \boldsymbol{v}^t$.

*Parts (v) and (vi).* The derivative of the bundle vector $\boldsymbol{v}^t$ with respect to the time $t$ exists, i.e. the issue of the non-differentiability of the ReLU activation at $0$ does not arise in this respect. However, the two-layer dynamics is such that this derivative is in general only approximated by the vector $\boldsymbol{f}^t$ defined above, and we bound that error by a ball centred at $\boldsymbol{f}^t$ whose radius depends on the initialisation scale $\lambda$.

*Parts (vii), (viii) and (ix).* We show that the squared norm of $\boldsymbol{v}^t$ grows exponentially fast as it moves away from the saddle at the origin, obtaining a lower bound on the speed of its increase that does not depend on $\lambda$ as long as $\boldsymbol{v}^t$ is in the subset $\mathcal{S}_1$, and a lower bound that depends on $\lambda$ subsequently. Also we show an upper bound on the speed of decrease of the squared norm of $\boldsymbol{v}^* - \boldsymbol{v}$, i.e. the square of the distance between the bundle vector and the teacher neuron.

Perhaps the most involved segment of the proof proceeds by showing that each face of the boundary of the set $\mathcal{S}$ is repelling towards the interior of $\mathcal{S}$ with respect to the dynamics of the bundle vector $\boldsymbol{v}^t$, whose derivative is approximately $\boldsymbol{f}^t$. A major complication is that this is in general not true for the entire boundary of the "padded ellipsoid" constraint $\Xi$, but holds for its remainder after the slicing off by the other constraints that define $\mathcal{S}$.

**Lemma 28.** *For all $t \geq T_1$ we have:*

- *(i)* $1 - \overline{\boldsymbol{w}}_j^{t\top} \overline{\boldsymbol{w}}_{j'}^t < 4\lambda^\varepsilon$ *for all $j, j' \in J_+$;*
- *(ii)* $\boldsymbol{v}^t \in \mathcal{S}$;
- *(iii)* $h_{\boldsymbol{\theta}^t}(\boldsymbol{x}_i) = \boldsymbol{v}^{t\top}\boldsymbol{x}_i$ *for all $i \in [n]$;*
- *(iv)* $\boldsymbol{g}_j^t = \boldsymbol{g}^t$ *for all $j \in J_+$;*
- *(v)* $\boldsymbol{v}^t$ *is differentiable at $t$;*
- *(vi)* $\|\mathrm{d}\boldsymbol{v}^t/\mathrm{d}t - \boldsymbol{f}^t\| \leq 3\lambda^{\varepsilon/2}\|\boldsymbol{f}^t\|$;
- *(vii)* $\mathrm{d}\|\boldsymbol{v}^t\|^2/\mathrm{d}t \geq (\eta_d \nu_d^*/\|\boldsymbol{\gamma}_{[n]}\|)^2 \|\boldsymbol{v}^t\|^2 \|\boldsymbol{g}^t\|$ *if $\boldsymbol{v}^t \in \mathcal{S}_1$;*
- *(viii)* $\mathrm{d}\|\boldsymbol{v}^t\|^2/\mathrm{d}t \geq 3\lambda^{\varepsilon/3}\|\boldsymbol{v}^t\|^2\|\boldsymbol{g}^t\|$;
- *(ix)* $\mathrm{d}\|\boldsymbol{v}^* - \boldsymbol{v}^t\|^2/\mathrm{d}t \geq -5\eta_1 \|\boldsymbol{v}^t\|\|\boldsymbol{v}^* - \boldsymbol{v}^t\|^2$.

*Proof.* First we establish the following.

*Claim 29.* For all $t \geq T_1$, assertions (i)–(ii) imply assertions (iii)–(ix).

*Proof of claim.* Suppose $t \geq T_1$, and (i) and (ii) are true.

By Lemma 21 and Proposition 27, we have (iii), (iv), and (v).

For (vi), we have

$$\left\|\frac{\mathrm{d}\boldsymbol{v}^t}{\mathrm{d}t} - \boldsymbol{f}^t\right\| = \left\|\sum_{j\in J_+}\frac{\mathrm{d}}{\mathrm{d}t}(\|\boldsymbol{w}_j^t\|^2\,\overline{\boldsymbol{w}}_j^t) - \|\boldsymbol{v}^t\|(\boldsymbol{g}^t + \overline{\boldsymbol{v}}^t\,\overline{\boldsymbol{v}}^{t\top}\boldsymbol{g}^t)\right\|$$

$$= \left\|\sum_{j\in J_+}\|\boldsymbol{w}_j^t\|^2(\boldsymbol{g}^t + \overline{\boldsymbol{w}}_j^t\,\overline{\boldsymbol{w}}_j^{t\top}\boldsymbol{g}^t) - \|\boldsymbol{v}^t\|(\boldsymbol{g}^t + \overline{\boldsymbol{v}}^t\,\overline{\boldsymbol{v}}^{t\top}\boldsymbol{g}^t)\right\|$$

$$= \left\|\sum_{j\in J_+}\left(\|\boldsymbol{w}_j^t\|^2\,\boldsymbol{g}^t + \overline{\boldsymbol{w}}_j^t\|\boldsymbol{w}_j^t\|^2\,\overline{\boldsymbol{w}}_j^{t\top}\boldsymbol{g}^t - \overline{\boldsymbol{v}}^{t\top}\overline{\boldsymbol{w}}_j^t\|\boldsymbol{w}_j^t\|^2\,\boldsymbol{g}^t - \overline{\boldsymbol{v}}^t\|\boldsymbol{w}_j^t\|^2\,\overline{\boldsymbol{w}}_j^{t\top}\boldsymbol{g}^t\right)\right\|$$

$$= \left\|\sum_{j\in J_+}(1 - \overline{\boldsymbol{v}}^{t\top}\overline{\boldsymbol{w}}_j^t)\|\boldsymbol{w}_j^t\|^2\,\boldsymbol{g}^t + \sum_{j\in J_+}(\overline{\boldsymbol{w}}_j^t - \overline{\boldsymbol{v}}^t)\|\boldsymbol{w}_j^t\|^2\,\overline{\boldsymbol{w}}_j^{t\top}\boldsymbol{g}^t\right\|$$

$$< 4\lambda^\varepsilon \sum_{j\in J_+}\|\boldsymbol{w}_j^t\|^2\|\boldsymbol{g}^t\| + \sqrt{8}\lambda^{\varepsilon/2}\sum_{j\in J_+}\|\boldsymbol{w}_j^t\|^2\,\overline{\boldsymbol{w}}_j^{t\top}\boldsymbol{g}^t$$

$$< \frac{4\lambda^\varepsilon}{1 - 4\lambda^\varepsilon}\|\boldsymbol{v}^t\|\|\boldsymbol{g}^t\| + \sqrt{8}\lambda^{\varepsilon/2}\,\boldsymbol{v}^{t\top}\boldsymbol{g}^t$$

$$< (5\lambda^\varepsilon + \sqrt{8}\lambda^{\varepsilon/2})\|\boldsymbol{f}^t\|$$

$$< 3\lambda^{\varepsilon/2}\|\boldsymbol{f}^t\|\ .$$

Now

$$\mathrm{d}\|\boldsymbol{v}^t\|^2/\mathrm{d}t \geq 2(\boldsymbol{v}^{t\top}\boldsymbol{f}^t - 3\lambda^{\varepsilon/2}\|\boldsymbol{v}^t\|\|\boldsymbol{f}^t\|)$$

$$= 2\|\boldsymbol{v}^t\|(2\boldsymbol{v}^{t\top}\boldsymbol{g}^t - 3\lambda^{\varepsilon/2}\|\boldsymbol{f}^t\|)\ ,$$

so for (vii), if $\boldsymbol{v}^t \in \mathcal{S}_1$ then by Proposition 26 we have

$$2\|\boldsymbol{v}^t\|(2\boldsymbol{v}^{t\top}\boldsymbol{g}^t - 3\lambda^{\varepsilon/2}\|\boldsymbol{f}^t\|) > (2(\eta_d\nu_d^*/\|\boldsymbol{\gamma}_{[n]}\|)^2 - 12\lambda^{\varepsilon/2})\|\boldsymbol{v}^t\|^2\|\boldsymbol{g}^t\|$$

$$> (\eta_d\nu_d^*/\|\boldsymbol{\gamma}_{[n]}\|)^2\|\boldsymbol{v}^t\|^2\|\boldsymbol{g}^t\|$$

since

$$\lambda^{\varepsilon/2} \leq n^{-\frac{9\cdot 3n\Delta^2}{2\delta^3}} < \left(\frac{4\delta^3}{9\cdot 3n\Delta^2}\right)^2 < \frac{\delta^6}{12d\Delta^4} \leq (\eta_d\nu_d^*/\|\boldsymbol{\gamma}_{[n]}\|)^2/12\ ,$$

and for (viii), in general we have

$$2\|\boldsymbol{v}^t\|(2\boldsymbol{v}^{t\top}\boldsymbol{g}^t - 3\lambda^{\varepsilon/2}\|\boldsymbol{f}^t\|) > (4\lambda^{\varepsilon/3} - 12\lambda^{\varepsilon/2})\|\boldsymbol{v}^t\|^2\|\boldsymbol{g}^t\|$$

$$> 3\lambda^{\varepsilon/3}\|\boldsymbol{v}^t\|^2\|\boldsymbol{g}^t\|\ .$$

For (ix), we have

$$\mathrm{d}\|\boldsymbol{v}^* - \boldsymbol{v}^t\|^2/\mathrm{d}t \geq -2((\boldsymbol{v}^* - \boldsymbol{v}^t)^\top\boldsymbol{f}^t + 3\lambda^{\varepsilon/2}\|\boldsymbol{v}^* - \boldsymbol{v}^t\|\|\boldsymbol{f}^t\|)$$

$$\geq -4(1 + 3\lambda^{\varepsilon/2})\|\boldsymbol{v}^t\|\|\boldsymbol{g}^t\|\|\boldsymbol{v}^* - \boldsymbol{v}^t\|$$

$$> -5\|\boldsymbol{v}^t\|\|\boldsymbol{g}^t\|\|\boldsymbol{v}^* - \boldsymbol{v}^t\|$$

$$> -5\eta_1\|\boldsymbol{v}^t\|\|\boldsymbol{v}^* - \boldsymbol{v}^t\|$$

since $\|\boldsymbol{g}^t\| = \|\frac{1}{n}\boldsymbol{X}\boldsymbol{X}^\top(\boldsymbol{v}^* - \boldsymbol{v}^t)\| \leq \eta_1\|\boldsymbol{v}^* - \boldsymbol{v}^t\| < \eta_1\|\boldsymbol{v}^*\| = \eta_1$ by Proposition 25. $\qquad\square$

Second we show the following.

*Claim* 30. Assertions (i) and (ii) are true for $t = T_1$.

*Proof of claim.* By Lemma 23 (iii), for all $j, j' \in J_+$ we have

$$
\begin{aligned}
1 - \overline{\boldsymbol{w}}_j^{T_1\top} \overline{\boldsymbol{w}}_{j'}^{T_1} &= \|\overline{\boldsymbol{w}}_j^{T_1} - \overline{\boldsymbol{w}}_{j'}^{T_1}\|^2/2 \\
&< \|\overline{\boldsymbol{w}}_j^{T_1} - \overline{\boldsymbol{\gamma}}_{[n]}\|^2 + \|\overline{\boldsymbol{w}}_{j'}^{T_1} - \overline{\boldsymbol{\gamma}}_{[n]}\|^2 \\
&\leq 4\lambda^\varepsilon
\end{aligned}
$$

where the first inequality is strict unless $\overline{\boldsymbol{w}}_j^{T_1} = \overline{\boldsymbol{\gamma}}_{[n]} = \overline{\boldsymbol{w}}_{j'}^{T_1}$, but in that case $1 - \overline{\boldsymbol{w}}_j^{T_1\top} \overline{\boldsymbol{w}}_{j'}^{T_1} = 0$.

Writing $\boldsymbol{v}^{T_1} = \sum_{k=1}^d \nu_k \boldsymbol{u}_k$, and recalling Proposition 16, Lemma 19 (i), and Lemma 23 (i) and (iii), we obtain that $\boldsymbol{v}^{T_1} \in \mathcal{S}_1$ because:

$\Phi_1$: we have

$$
\frac{\nu_1}{\|\boldsymbol{v}^{T_1}\|} \geq \frac{\eta_1 \nu_1^*}{\|\boldsymbol{\gamma}_{[n]}\|} - \sqrt{2}\lambda^{\varepsilon/2} \geq \frac{4\delta^3}{\sqrt{d}\Delta^2} - \sqrt{2}\lambda^{\varepsilon/2} > 0
$$

and

$$
\frac{\nu_1}{\nu_1^*} \leq \frac{4m\sqrt{d}\Delta^2}{\delta}\lambda^{2-2\varepsilon} < \frac{1}{2} \ ;
$$

$\Psi_{k,k'}^\downarrow$ **for all** $1 \leq k < k' \leq d$: we have

$$
\begin{aligned}
\frac{\nu_k}{\|\boldsymbol{v}^{T_1}\|\eta_k \nu_k^*} &\leq \frac{1}{\|\boldsymbol{\gamma}_{[n]}\|} + \frac{\sqrt{2}\lambda^{\varepsilon/2}}{\eta_k \nu_k^*} \\
&\leq \frac{1}{\|\boldsymbol{\gamma}_{[n]}\|} + \frac{\sqrt{2d}\lambda^{\varepsilon/2}}{\delta^3} \\
&< \frac{2}{\|\boldsymbol{\gamma}_{[n]}\|} - \frac{2\sqrt{2d}\lambda^{\varepsilon/2}}{\delta^3} \\
&\leq \frac{2}{\|\boldsymbol{\gamma}_{[n]}\|} - \frac{2\sqrt{2}\lambda^{\varepsilon/2}}{\eta_{k'}\nu_{k'}^*} \\
&\leq \frac{2\nu_{k'}}{\|\boldsymbol{v}^{T_1}\|\eta_{k'}\nu_{k'}^*} \ ;
\end{aligned}
$$

$\Psi_{k,k'}^\uparrow$ **for all** $1 \leq k < k' \leq d$: by Bernoulli's inequality we have

$$
\begin{aligned}
\left(1 - \frac{\nu_k}{\nu_k^*}\right)^{\frac{\eta_k + \eta_{k'}}{2\eta_k}} &< 1 - \frac{\eta_k + \eta_{k'}}{2\eta_k}\frac{\nu_k}{\nu_k^*} \\
&\leq 1 - \|\boldsymbol{v}^{T_1}\|\frac{\eta_k + \eta_{k'}}{2}\left(\frac{1}{\|\boldsymbol{\gamma}_{[n]}\|} - \frac{\sqrt{2}\lambda^{\varepsilon/2}}{\eta_k \nu_k^*}\right) \\
&= 1 - \|\boldsymbol{v}^{T_1}\|\left(\frac{\eta_{k'}}{\|\boldsymbol{\gamma}_{[n]}\|} + \frac{\eta_k - \eta_{k'}}{2\|\boldsymbol{\gamma}_{[n]}\|} - \frac{\eta_k + \eta_{k'}}{2}\frac{\sqrt{2}\lambda^{\varepsilon/2}}{\eta_k \nu_k^*}\right) \\
&\leq 1 - \|\boldsymbol{v}^{T_1}\|\left(\frac{\eta_{k'}}{\|\boldsymbol{\gamma}_{[n]}\|} + \frac{\delta^2}{d\Delta^2} - \Delta^2\frac{\sqrt{2d}\lambda^{\varepsilon/2}}{\delta^3}\right) \\
&< 1 - \|\boldsymbol{v}^{T_1}\|\left(\frac{\eta_{k'}}{\|\boldsymbol{\gamma}_{[n]}\|} + \Delta^2\frac{\sqrt{2d}\lambda^{\varepsilon/2}}{\delta^3}\right) \\
&\leq 1 - \|\boldsymbol{v}^{T_1}\|\eta_{k'}\left(\frac{1}{\|\boldsymbol{\gamma}_{[n]}\|} + \frac{\sqrt{2}\lambda^{\varepsilon/2}}{\eta_{k'}\nu_{k'}^*}\right) \\
&\leq 1 - \frac{\nu_{k'}}{\nu_{k'}^*} \ ;
\end{aligned}
$$

$\Xi$: we have

$$\begin{aligned}
\overline{\boldsymbol{v}}^{T_1\top} \boldsymbol{g}^{T_1} &\geq \overline{\boldsymbol{v}}^{T_1\top} \boldsymbol{\gamma}_{[n]} - \lambda^{2-3\varepsilon} \\
&\geq (1 - \lambda^\varepsilon)\|\boldsymbol{\gamma}_{[n]}\| - \lambda^{2-3\varepsilon} \\
&> 3\lambda^{\varepsilon/3}\|\boldsymbol{\gamma}_{[n]}\| - \lambda^{2-3\varepsilon} \\
&> 2\lambda^{\varepsilon/3}\|\boldsymbol{\gamma}_{[n]}\| \\
&> \lambda^{\varepsilon/3}(\|\boldsymbol{\gamma}_{[n]}\| + \lambda^{2-3\varepsilon}) \\
&\geq \lambda^{\varepsilon/3}\|\boldsymbol{g}^{T_1}\| \,. \qquad\qquad\qquad\qquad\qquad \square
\end{aligned}$$

Assume for a contradiction that there exists $t > T_1$ such that either (i) or (ii) is false, and let $t$ be the smallest such.

For all $j, j' \in J_+$ we have

$$\begin{aligned}
\mathrm{d}(1 - \overline{\boldsymbol{w}}_j^{t\top}\overline{\boldsymbol{w}}_{j'}^t)/\mathrm{d}t &= -\overline{\boldsymbol{w}}_{j'}^{t\top}(\boldsymbol{g}^t - \overline{\boldsymbol{w}}_j^t\overline{\boldsymbol{w}}_j^{t\top}\boldsymbol{g}^t) - \overline{\boldsymbol{w}}_j^{t\top}(\boldsymbol{g}^t - \overline{\boldsymbol{w}}_{j'}^t\overline{\boldsymbol{w}}_{j'}^{t\top}\boldsymbol{g}^t) \\
&= -(1 - \overline{\boldsymbol{w}}_j^{t\top}\overline{\boldsymbol{w}}_{j'}^t)(\overline{\boldsymbol{w}}_j^t + \overline{\boldsymbol{w}}_{j'}^t)^\top \boldsymbol{g}^t \\
&\leq -2(\lambda^{\varepsilon/3} - \sqrt{8}\lambda^{\varepsilon/2})\|\boldsymbol{g}^t\|(1 - \overline{\boldsymbol{w}}_j^{t\top}\overline{\boldsymbol{w}}_{j'}^t) \\
&\leq -\lambda^{\varepsilon/3}\|\boldsymbol{g}^t\|(1 - \overline{\boldsymbol{w}}_j^{t\top}\overline{\boldsymbol{w}}_{j'}^t) \,.
\end{aligned}$$

Therefore (ii) is false. Hence $\boldsymbol{v}^t$ is in at least one possibly curved face of the boundary of $\mathcal{S}$, and is distinct from $\boldsymbol{0}$ and $\boldsymbol{v}^*$. We consider those faces in the following cases, where we omit the superscripts $t$, assume $\boldsymbol{v} = \sum_{k=1}^d \nu_k \boldsymbol{u}_k \in \mathrm{cl}(\mathcal{S}_\ell)$ for some $\ell \in [d]$, write e.g. $\widehat{\Omega}_k$ for the constraint obtained by replacing the unique strict inequality in $\Omega_k$ by equality, and denote by $\boldsymbol{p}$ a normal vector to the respective face that is at $\boldsymbol{v}$ and on the side of the interior of $\mathcal{S}$. To get a contradiction, it suffices to show in each of the cases that $\overline{\boldsymbol{p}}^\top \boldsymbol{f} > 3\lambda^{\varepsilon/2}$, because by Claim 29 and continuity we have that (vi) is true at $t$, and so $\overline{\boldsymbol{p}}^\top (\mathrm{d}\boldsymbol{v}/\mathrm{d}t) \geq \overline{\boldsymbol{p}}^\top \boldsymbol{f} - 3\lambda^{\varepsilon/2}\|\boldsymbol{f}\| > 0$.

*Case $\widehat{\Omega}_k$ for some $1 \leq k < \ell$.* Picking $\boldsymbol{p} := \boldsymbol{u}_k$, we have

$$\overline{\boldsymbol{p}}^\top \boldsymbol{f} = \nu_k^* \overline{\boldsymbol{v}}^\top \boldsymbol{g}/\|\boldsymbol{f}\| > \frac{\nu_k^*}{2}\overline{\boldsymbol{v}}^\top \overline{\boldsymbol{g}} \geq \frac{\delta}{2\sqrt{d}}\lambda^{\varepsilon/3} > 3\lambda^{\varepsilon/2}$$

since $\lambda^{-\varepsilon/6} \geq n^{\frac{9}{2}n/\delta^3} \geq \frac{9}{\sqrt{2}}\mathrm{e}(\ln 2)\sqrt{d}/\delta^3 > \frac{6\sqrt{d}}{\delta}$.

*Case $\widehat{\Phi}_\ell$.* Necessarily $\ell \neq 1$. Picking $\boldsymbol{p} := \boldsymbol{u}_\ell$, we have

$$\overline{\boldsymbol{p}}^\top \boldsymbol{f} \geq \left(\eta_\ell\left(1 - \frac{\eta_\ell}{2\eta_{\ell-1}}\right)\nu_\ell^* + \frac{1}{\|\boldsymbol{v}\|}\frac{\eta_\ell}{2\eta_{\ell-1}}\nu_\ell^* \overline{\boldsymbol{v}}^\top \boldsymbol{g}\right)\Big/(2\|\boldsymbol{g}\|)$$

$$> \frac{\eta_\ell}{2\eta_1}\left(1 - \frac{\eta_\ell}{2\eta_{\ell-1}}\right)\nu_\ell^* > \frac{\delta^3}{4\sqrt{d}\Delta^2} > 3\lambda^{\varepsilon/2} \,.$$

*Case $\widehat{\Psi}_{k,k'}^\downarrow$ for some $\ell \leq k < k' \leq d$.* Picking $\boldsymbol{p} := -\eta_{k'}\nu_{k'}^*\boldsymbol{u}_k + 2\eta_k\nu_k^*\boldsymbol{u}_{k'}$, we have

$$\begin{aligned}
\overline{\boldsymbol{p}}^\top \boldsymbol{f} &= \eta_k \eta_{k'}\nu_k^*\nu_{k'}^*\left(-\left(1 - \frac{\nu_k}{\nu_k^*}\right) + 2\left(1 - \frac{\nu_{k'}}{\nu_{k'}^*}\right)\right)\|\boldsymbol{v}\|/(\|\boldsymbol{p}\|\|\boldsymbol{f}\|) \\
&= \eta_k \eta_{k'}\nu_k^*\nu_{k'}^*\left(1 + \left(1 - \frac{\eta_{k'}}{\eta_k}\right)\frac{\nu_k}{\nu_k^*}\right)\|\boldsymbol{v}\|/(\|\boldsymbol{p}\|\|\boldsymbol{f}\|) \\
&> \eta_k \eta_{k'}\nu_k^*\nu_{k'}^*/(2\|\boldsymbol{p}\|\|\boldsymbol{g}\|) \\
&> \frac{\eta_k \eta_{k'}\nu_k^*\nu_{k'}^*}{2\eta_1\sqrt{(2\eta_k\nu_k^*)^2 + (\eta_{k'}\nu_{k'}^*)^2}} \\
&= \left(\left(\frac{2\eta_1}{\eta_k}\frac{1}{\nu_k^*}\right)^2 + \left(\frac{2\eta_1}{\eta_{k'}}\frac{2}{\nu_{k'}^*}\right)^2\right)^{-1/2}
\end{aligned}$$

$$\geq \frac{\delta^3}{5\sqrt{2d}\Delta^2}$$
$$> 3\lambda^{\varepsilon/2} \ .$$

*Case* $\widehat{\Psi}^{\uparrow}_{k,k'}$ *for some* $\ell \leq k < k' \leq d$. Picking

$$\boldsymbol{p} := \frac{\eta_k + \eta_{k'}}{2} \nu_{k'}^* \boldsymbol{u}_k - \eta_k \left(1 - \frac{\nu_k}{\nu_k^*}\right)^{\frac{1}{2} - \frac{\eta_{k'}}{2\eta_k}} \nu_k^* \boldsymbol{u}_{k'} \ ,$$

we have

$$\boldsymbol{p}^\top \boldsymbol{g} = \eta_k \eta_{k'} \nu_k^* \nu_{k'}^* \left( \left(\frac{1}{2} + \frac{\eta_k}{2\eta_{k'}}\right)\left(1 - \frac{\nu_k}{\nu_k^*}\right) - \left(1 - \frac{\nu_k}{\nu_k^*}\right)^{\frac{1}{2} - \frac{\eta_{k'}}{2\eta_k}}\left(1 - \frac{\nu_{k'}}{\nu_{k'}^*}\right) \right)$$

$$= \eta_k \eta_{k'} \nu_k^* \nu_{k'}^* \left( \left(\frac{1}{2} + \frac{\eta_k}{2\eta_{k'}}\right)\left(1 - \frac{\nu_k}{\nu_k^*}\right) - \left(1 - \frac{\nu_k}{\nu_k^*}\right) \right)$$

$$= \frac{\eta_k^2 - \eta_k \eta_{k'}}{2}\left(1 - \frac{\nu_k}{\nu_k^*}\right)\nu_k^* \nu_{k'}^*$$

and

$$\boldsymbol{p}^\top \boldsymbol{v} = \frac{\eta_k + \eta_{k'}}{2}\nu_k \nu_{k'}^* - \eta_k \left(1 - \frac{\nu_k}{\nu_k^*}\right)^{\frac{1}{2} - \frac{\eta_{k'}}{2\eta_k}}\nu_{k'}\nu_k^*$$

$$= \eta_k \nu_k^* \nu_{k'}^* \left( \left(\frac{1}{2} + \frac{\eta_{k'}}{2\eta_k}\right)\frac{\nu_k}{\nu_k^*} - \left(1 - \frac{\nu_k}{\nu_k^*}\right)^{\frac{1}{2} - \frac{\eta_{k'}}{2\eta_k}}\left(1 - \left(1 - \frac{\nu_k}{\nu_k^*}\right)^{\frac{1}{2} + \frac{\eta_{k'}}{2\eta_k}}\right) \right)$$

$$= \eta_k \nu_k^* \nu_{k'}^* \left( \left(1 - \left(\frac{1}{2} - \frac{\eta_{k'}}{2\eta_k}\right)\frac{\nu_k}{\nu_k^*}\right) - \left(1 - \frac{\nu_k}{\nu_k^*}\right)^{\frac{1}{2} - \frac{\eta_{k'}}{2\eta_k}} \right)$$

$$> \frac{\eta_k^2 - \eta_{k'}^2}{8\eta_k}\left(\frac{\nu_k}{\nu_k^*}\right)^2 \nu_k^* \nu_{k'}^* \ .$$

Hence if $\nu_k/\nu_k^* \leq 1/2$ then

$$\overline{\boldsymbol{p}}^\top \overline{\boldsymbol{f}} > \frac{\eta_k^2 - \eta_k \eta_{k'}}{4}\nu_k^* \nu_{k'}^* / (2\|\boldsymbol{p}\|\|\boldsymbol{g}\|)$$

$$> \frac{(\eta_k^2 - \eta_k \eta_{k'})\nu_k^* \nu_{k'}^*}{\eta_1 \sqrt{(\eta_k + \eta_{k'})^2 \nu_{k'}^{*,2} + 4\eta_k^2 \nu_k^{*2}}}$$

$$\geq \frac{\delta^6}{\sqrt{2}d^2 \Delta^4}$$

$$> 3\lambda^{\varepsilon/2} \ ,$$

else

$$\overline{\boldsymbol{p}}^\top \overline{\boldsymbol{f}} > \frac{\eta_k^2 - \eta_{k'}^2}{32\eta_k}\nu_k^* \nu_{k'}^* \ \overline{\boldsymbol{v}}^\top \boldsymbol{g} / (\|\boldsymbol{p}\|\|\boldsymbol{f}\|)$$

$$> \frac{(\eta_k^2 - \eta_{k'}^2)\nu_k^* \nu_{k'}^*}{32\eta_k \sqrt{(\eta_k + \eta_{k'})^2 \nu_{k'}^{*,2} + 4\eta_k^2 \nu_k^{*2}}} \ \overline{\boldsymbol{v}}^\top \overline{\boldsymbol{g}}$$

$$\geq \frac{\delta^6}{16\sqrt{2}d^2 \Delta^4}\lambda^{\varepsilon/3}$$

$$> 3\lambda^{\varepsilon/2}$$

since $\lambda^{-\varepsilon/6} \geq n^{\frac{9}{2}n\Delta^2/\delta^3} \geq \left(2^{\frac{9}{4}n\Delta^2/\delta^3}\right)^2 > \left(\frac{11n\Delta^2}{\delta^3}\right)^2 > \frac{48\sqrt{2}d^2\Delta^4}{\delta^6}$.

*Case* $\widehat{\Xi}$. Here $\overline{\boldsymbol{v}}^\top \overline{\boldsymbol{g}} = \lambda^{\varepsilon/3}$.

Hence, by Proposition 26, since $\lambda^{\varepsilon/3} < \frac{1}{2}\frac{\delta^6}{d\Delta^4} \le \frac{1}{2}\left(\frac{\eta_d \nu_d^*}{\|\boldsymbol{\gamma}_{[n]}\|}\right)^2$, necessarily $\ell \ne 1$.

Picking

$$\boldsymbol{p} := \sum_{k=1}^{d} \eta_k \nu_k^* \left(1 - \frac{2\nu_k}{\nu_k^*} - \lambda^{\varepsilon/3}\left(\frac{\nu_k}{\nu_k^*}\frac{\|\boldsymbol{g}\|}{\eta_k \|\boldsymbol{v}\|} - \left(1 - \frac{\nu_k}{\nu_k^*}\right)\frac{\eta_k \|\boldsymbol{v}\|}{\|\boldsymbol{g}\|}\right)\right)\boldsymbol{u}_k \ ,$$

we have

$$\boldsymbol{p}^\top \boldsymbol{g} = \sum_{k=1}^{d} \eta_k^2 \nu_k^{*2}\left(1 - \frac{\nu_k}{\nu_k^*}\right)\left(1 - \frac{2\nu_k}{\nu_k^*} - \lambda^{\varepsilon/3}\left(\frac{\nu_k}{\nu_k^*}\frac{\|\boldsymbol{g}\|}{\eta_k \|\boldsymbol{v}\|} - \left(1 - \frac{\nu_k}{\nu_k^*}\right)\frac{\eta_k \|\boldsymbol{v}\|}{\|\boldsymbol{g}\|}\right)\right)$$

$$= \sum_{k=1}^{d} \eta_k^2 \nu_k^{*2}\left(1 - \frac{\nu_k}{\nu_k^*}\right)\left(1 - \frac{2\nu_k}{\nu_k^*}\right) + \lambda^{\varepsilon/3}\sum_{k=1}^{d}\eta_k^3 \nu_k^{*2}\left(1 - \frac{\nu_k}{\nu_k^*}\right)^2\frac{\|\boldsymbol{v}\|}{\|\boldsymbol{g}\|} - \lambda^{2\varepsilon/3}\|\boldsymbol{g}\|^2$$

$$= \|\boldsymbol{g}\|^2 + \sum_{k=1}^{\ell-1}\eta_k^2 \nu_k^{*2}\frac{\nu_k}{\nu_k^*}\left(\frac{\nu_k}{\nu_k^*} - 1\right) - \sum_{k=\ell}^{d}\eta_k^2 \nu_k^{*2}\frac{\nu_k}{\nu_k^*}\left(1 - \frac{\nu_k}{\nu_k^*}\right)$$

$$\quad + \lambda^{\varepsilon/3}\sum_{k=1}^{d}\eta_k^3 \nu_k^{*2}\left(1 - \frac{\nu_k}{\nu_k^*}\right)^2\frac{\|\boldsymbol{v}\|}{\|\boldsymbol{g}\|} - \lambda^{2\varepsilon/3}\|\boldsymbol{g}\|^2$$

$$\ge \|\boldsymbol{g}\|^2 + \eta_{\ell-1}\sum_{k=1}^{\ell-1}\eta_k \nu_k^{*2}\frac{\nu_k}{\nu_k^*}\left(\frac{\nu_k}{\nu_k^*} - 1\right) - \eta_\ell \sum_{k=\ell}^{d}\eta_k \nu_k^{*2}\frac{\nu_k}{\nu_k^*}\left(1 - \frac{\nu_k}{\nu_k^*}\right)$$

$$\quad + \lambda^{\varepsilon/3}\eta_d \|\boldsymbol{v}\|\|\boldsymbol{g}\| - \lambda^{2\varepsilon/3}\|\boldsymbol{g}\|^2$$

$$= \|\boldsymbol{g}\|^2 + \frac{\eta_{\ell-1} - \eta_\ell}{2}\left(\sum_{k=1}^{\ell-1}\eta_k \nu_k^{*2}\frac{\nu_k}{\nu_k^*}\left(\frac{\nu_k}{\nu_k^*} - 1\right) + \sum_{k=\ell}^{d}\eta_k \nu_k^{*2}\frac{\nu_k}{\nu_k^*}\left(1 - \frac{\nu_k}{\nu_k^*}\right)\right)$$

$$\quad - \lambda^{\varepsilon/3}\left(\frac{\eta_{\ell-1} + \eta_\ell}{2} - \eta_d\right)\|\boldsymbol{v}\|\|\boldsymbol{g}\| - \lambda^{2\varepsilon/3}\|\boldsymbol{g}\|^2$$

$$\ge \left(\frac{1}{8}\left(1 - \frac{\eta_\ell}{\eta_{\ell-1}}\right)\eta_d\left(\min_{k=1}^{\ell-1}\nu_k^*\right) + \|\boldsymbol{g}\| - \lambda^{\varepsilon/3}\left(\frac{\eta_{\ell-1} + \eta_\ell}{2} - \eta_d\right)\|\boldsymbol{v}\| - \lambda^{2\varepsilon/3}\|\boldsymbol{g}\|\right)\|\boldsymbol{g}\|$$

and

$$\boldsymbol{p}^\top \boldsymbol{v} = \sum_{k=1}^{d}\eta_k \nu_k^{*2}\frac{\nu_k}{\nu_k^*}\left(1 - \frac{2\nu_k}{\nu_k^*} - \lambda^{\varepsilon/3}\left(\frac{\nu_k}{\nu_k^*}\frac{\|\boldsymbol{g}\|}{\eta_k \|\boldsymbol{v}\|} - \left(1 - \frac{\nu_k}{\nu_k^*}\right)\frac{\eta_k \|\boldsymbol{v}\|}{\|\boldsymbol{g}\|}\right)\right)$$

$$> -\eta_1\|\boldsymbol{v}\|^2 + \lambda^{\varepsilon/3}\frac{\|\boldsymbol{v}\|}{\|\boldsymbol{g}\|}\sum_{k=1}^{d}\eta_k^2 \nu_k^{*2}\frac{\nu_k}{\nu_k^*}\left(1 - \frac{\nu_k}{\nu_k^*}\right)$$

$$\ge -\eta_1\|\boldsymbol{v}\|^2 - \frac{\lambda^{\varepsilon/3}}{n}\frac{\|\boldsymbol{v}\|}{\|\boldsymbol{g}\|}\left(\eta_1 \sum_{k=1}^{\ell-1}\eta_k \nu_k^{*2}\frac{\nu_k}{\nu_k^*}\left(\frac{\nu_k}{\nu_k^*} - 1\right) - \eta_d\sum_{k=\ell}^{d}\eta_k \nu_k^{*2}\frac{\nu_k}{\nu_k^*}\left(1 - \frac{\nu_k}{\nu_k^*}\right)\right)$$

$$= -\eta_1\|\boldsymbol{v}\|^2(1 - \lambda^{2\varepsilon/3}) - \lambda^{\varepsilon/3}\frac{\|\boldsymbol{v}\|}{\|\boldsymbol{g}\|}(\eta_1 - \eta_d)\sum_{k=\ell}^{d}\eta_k \nu_k^{*2}\frac{\nu_k}{\nu_k^*}\left(1 - \frac{\nu_k}{\nu_k^*}\right)$$

$$\ge -\eta_1\left(1 + \lambda^{\varepsilon/3}\left(1 - \frac{\eta_d}{\eta_1}\right) - \lambda^{2\varepsilon/3}\right)\|\boldsymbol{v}\|^2 \ ,$$

also

$$\|\boldsymbol{p}\| < 2\sqrt{\sum_{k=1}^{d}\left((\eta_1 \nu_k)^2 + (\eta_k(\nu_k^* - \nu_k))^2 + \left(\lambda^{\varepsilon/3}\nu_k\frac{\|\boldsymbol{g}\|}{\|\boldsymbol{v}\|}\right)^2 + \left(\lambda^{\varepsilon/3}\eta_1 \eta_k(\nu_k^* - \nu_k)\frac{\|\boldsymbol{v}\|}{\|\boldsymbol{g}\|}\right)^2\right)}$$

$$= 2\sqrt{1 + \lambda^{2\varepsilon/3}}\sqrt{\eta_1^2\|\boldsymbol{v}\|^2 + \|\boldsymbol{g}\|^2}$$

$$< 2(1 + \lambda^{\varepsilon/3})(\eta_1 \|\boldsymbol{v}\| + \|\boldsymbol{g}\|)$$
$$< 4(1 + \lambda^{\varepsilon/3})\eta_1$$

and

$$\|\boldsymbol{f}\| = (1 + \lambda^{\varepsilon/3})\|\boldsymbol{v}\|\|\boldsymbol{g}\| .$$

Therefore

$$\overline{\boldsymbol{p}}^\top \overline{\boldsymbol{f}} > \left[ \frac{1}{8}\left(1 - \frac{\eta_\ell}{\eta_{\ell-1}}\right)\eta_d \left(\min_{k=1}^{\ell-1} \nu_k^*\right) \right.$$
$$- \lambda^{\varepsilon/3}\left(\frac{\eta_{\ell-1} + \eta_\ell}{2} + \eta_1 - \eta_d\right)\|\boldsymbol{v}\|$$
$$\left. - \lambda^{2\varepsilon/3}((\eta_1 - \eta_d)\|\boldsymbol{v}\| + \|\boldsymbol{g}\|) \right]$$
$$\Big/ 4(1 + \lambda^{\varepsilon/3})^2 \eta_1 \qquad \text{calculation}$$

$$> \frac{1}{40}\left(1 - \frac{\eta_\ell}{\eta_{\ell-1}}\right)\frac{\eta_d}{\eta_1}\left(\min_{k=1}^{\ell-1} \nu_k^*\right)$$
$$- \frac{\lambda^{\varepsilon/3}}{5}\left(\frac{\eta_{\ell-1} + \eta_\ell}{2\eta_1} + 1 - \frac{\eta_d}{\eta_1}\right)\|\boldsymbol{v}\|$$
$$- \frac{\lambda^{2\varepsilon/3}}{5}\left(\left(1 - \frac{\eta_d}{\eta_1}\right)\|\boldsymbol{v}\| + \frac{\|\boldsymbol{g}\|}{\eta_1}\right) \qquad \text{since } \lambda^{\varepsilon/3} \le n^{-9n} \le 2^{-9 \cdot 2} < \sqrt{5/4} - 1$$

$$> \frac{\delta^4}{40 d\sqrt{d}\Delta^3} - \frac{2}{5}\lambda^{\varepsilon/3} - \frac{2}{5}\lambda^{2\varepsilon/3} \qquad \begin{array}{l}\text{by the definitions of } \delta \text{ and } \Delta, \\ \text{Proposition 17 (ii), and Proposition 25}\end{array}$$

$$> \frac{1}{40}\left(\frac{\delta^4}{d\sqrt{d}\Delta^3} - 17\lambda^{\varepsilon/3}\right) \qquad \text{since } \lambda^{\varepsilon/3} \le n^{-9n} \le 2^{-9 \cdot 2} < 1/16$$

$$> \frac{\delta^4}{80 d\sqrt{d}\Delta^3} \qquad \begin{array}{l}\text{since } \lambda^{-\varepsilon/3} \ge n^{9n\Delta^2/\delta^3} \ge \left(2^{6n\Delta^2/\delta^3}\right)^{3/2} \\ \qquad \ge (2^{11}n\Delta^2/\delta^3)^{3/2} > \frac{17 \cdot 80 d\sqrt{d}\Delta^3}{\delta^4}\end{array}$$

$$> 3\lambda^{\varepsilon/2} \qquad \begin{array}{l}\text{since } \lambda^{-\varepsilon/2} \ge n^{\frac{9 \cdot 3}{2}n\Delta^2/\delta^3} \ge \left(2^{9n\Delta^2/\delta^3}\right)^{3/2} \\ \qquad \ge (2^{17}n\Delta^2/\delta^3)^{3/2} > \frac{3 \cdot 80 d\sqrt{d}\Delta^3}{\delta^4} ,\end{array}$$

completing the proof. $\qquad\square$

*Example* 31. To illustrate some aspects of the training dynamics, let us consider a single run of gradient descent with learning rate $0.01$, for a network of width $m = 25$ initialised using $\boldsymbol{z}_j \overset{\text{i.i.d.}}{\sim} \mathcal{N}(\boldsymbol{0}, \frac{1}{dm}\boldsymbol{I}_d)$ and $s_j \overset{\text{i.i.d.}}{\sim} \mathcal{U}\{\pm 1\}$ with scale $\lambda = 4^{-7}$, and on a synthetic uncentred training dataset in dimension $d = 16$ as described in section 8.

Figure 3 shows the coordinates of the vector $\boldsymbol{v}^* - \boldsymbol{v}^t$ in the eigenvectors basis crossing zero one by one exactly in the order of their indices, i.e. in the decreasing order of the corresponding eigenvalues of the matrix $\frac{1}{n}\boldsymbol{X}\boldsymbol{X}^\top$. Thus the bundle vector $\boldsymbol{v}^t$ travels through the subsets $\mathcal{S}_1, \mathcal{S}_2, \dots$ of the set $\mathcal{S}$ exactly in their order, and in line with what we established in the proof of Lemma 28, passing through each $\mathcal{S}_k$ at most once. $\qquad\square$

Let us write $\boldsymbol{v}^t = \sum_{k=1}^d \nu_k^t \boldsymbol{u}_k$.

Recall from section 5 that $T_2 = \inf\{t \ge T_1 \mid \nu_1^t/\nu_1^* \ge 1/2\}$.

Building on the preceding results, the final lemma in this section establishes that the loss converges to zero exponentially fast. To show it, we partition the second phase of the training into two, namely before and after the time $T_2$ which is when the coordinate of the bundle vector $\boldsymbol{v}^t$ with respect to the largest-eigenvalue eigenvector of the matrix $\frac{1}{n}\boldsymbol{X}\boldsymbol{X}^\top$ crosses the half-way threshold to the corresponding coordinate of the teacher neuron $\boldsymbol{v}^*$. The period before $T_2$ (and after the start $T_1$ of

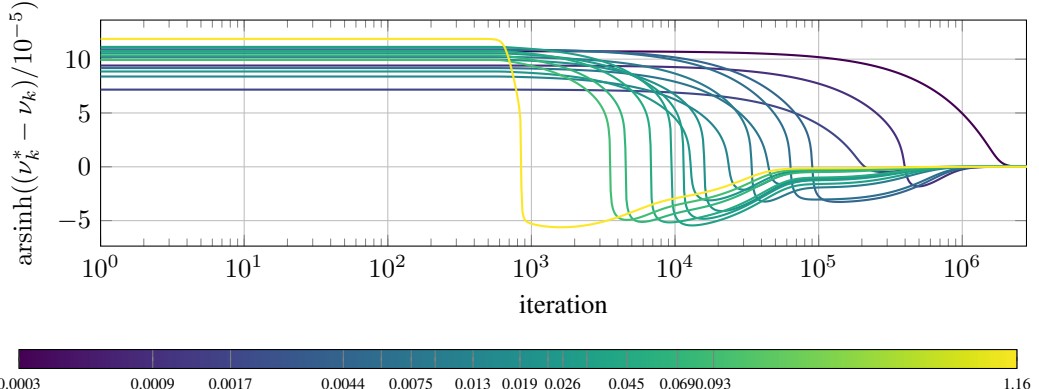

Figure 3: The coordinates in the eigenvectors basis of the difference between the teacher neuron and the weighted sum of the hidden neurons crossing zero in the decreasing eigenvalue order. The horizontal axis is logarithmic. The vertical axis shows the values mapped using the inverse of the hyperbolic sine in order to be able to visualise numbers at different scales on both sides around zero. The colours are picked from a colourmap based on the corresponding eigenvalue.

the second phase) consists of an exponentially fast departure of $v^t$ from near the saddle at the origin, whereas the period after $T_2$ obeys a Polyak-Łojasiewicz inequality which implies exponentially fast convergence.

**Lemma 32.** *(i) $\|v^t\| < \frac{1}{2}$ for all $t \in [T_1, T_2]$.*

 *(ii) $T_2 - T_1 < \ln\left(\frac{1}{\lambda}\right)\frac{(4+\varepsilon/2)d\Delta^2}{\delta^6}$ and $T_2 < \ln\left(\frac{1}{\lambda}\right)\frac{(4+\varepsilon)d\Delta^2}{\delta^6}$.*

 *(iii) $\|v^t\| > \frac{\|\gamma_{[n]}\|}{4\eta_1}$ for all $t \geq T_2$.*

 *(iv) $\|\nabla L(\theta^t)\|^2 > \frac{2\eta_d\|\gamma_{[n]}\|}{5\eta_1}L(\theta^t)$ for all $t \geq T_2$.*

 *(v) $L(\theta^t) < \frac{\Delta^2}{2}\exp\left(-(t - T_2)\frac{2\delta^4}{5\Delta^2}\right)$ for all $t \geq T_2$.*

*Proof.* By Lemma 19 (ii), we have $\|v^{T_1}\| > |J_+|(1 - 4\lambda^\varepsilon)\lambda\min_{j \in J_+}\|z_j\|$.

For all $t \in [T_1, T_2]$, by Lemma 28 (ii) we have $\|v^t\| < \frac{\|v^*\|}{2}$, and by Lemma 28 (vii) we have $\frac{\mathrm{d}}{\mathrm{d}t}\|v^t\|^2 > \frac{\eta_d^2\nu_d^{*2}}{2\|\gamma_{[n]}\|}\|v^t\|^2$.

Hence

$$T_2 - T_1 < \left(\ln\left(\frac{1}{\lambda}\right) + \ln\left(\frac{\|v^*\|}{\min_{j \in J_+}\|z_j\|}\right)\right)\frac{4\|\gamma_{[n]}\|}{\eta_d^2\nu_d^{*2}}$$

$$\leq \left(\ln\left(\frac{1}{\lambda}\right) + \ln\left(\frac{1}{\delta}\right)\right)\frac{4d\Delta^2}{\delta^6}$$

$$< \ln\left(\frac{1}{\lambda}\right)\frac{(4 + \varepsilon/2)d\Delta^2}{\delta^6}$$

since $\ln(1/\lambda)\varepsilon/2 \geq \frac{9\cdot3}{2}n(\ln n)/\delta^3 \geq 9 \cdot 3(\ln 2)/\delta^3 > 4\ln(1/\delta)$, and so

$$T_2 < \ln\left(\frac{1}{\lambda}\right)\frac{\varepsilon}{\|\gamma_{[n]}\|} + \ln\left(\frac{1}{\lambda}\right)\frac{(4 + \varepsilon/2)d\Delta^2}{\delta^6} \leq \ln\left(\frac{1}{\lambda}\right)\frac{(4 + \varepsilon)d\Delta^2}{\delta^6} \ .$$

By Lemma 28 (ii) we have $L(\theta^{T_2}) < \frac{1}{2}\left(1 - \frac{\eta_d}{4\eta_1}\right)^2\|v^*\|\|\gamma_{[n]}\| < \frac{\|\gamma_{[n]}\|}{2}$.

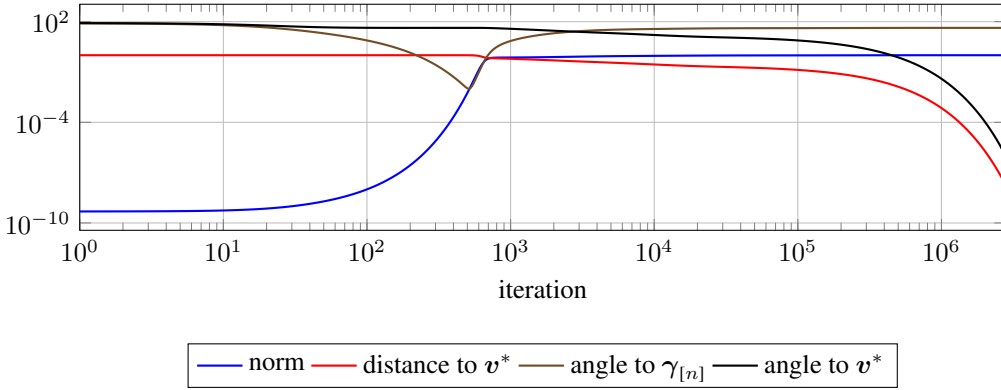

Figure 4: The evolution of several measures of the weighted sum of the hidden neurons during the training. Both axes are logarithmic, and the angles are in degrees.

For all $t \geq T_2$, by Lemma 28 (ii) we have $\|\boldsymbol{v}^t\| > \frac{\|\boldsymbol{\gamma}_{[n]}\|}{4\eta_1}$, and so

$$
\begin{aligned}
\|\nabla L(\boldsymbol{\theta}^t)\|^2 &= -\mathrm{d}L(\boldsymbol{\theta}^t)/\mathrm{d}t \\
&= \boldsymbol{g}^{t\top} \mathrm{d}\boldsymbol{v}^t/\mathrm{d}t \\
&\geq \boldsymbol{g}^{t\top} \boldsymbol{f}^t - 3\lambda^{\varepsilon/2}\|\boldsymbol{g}^t\|\|\boldsymbol{f}^t\|) \\
&> \left( (1 + \lambda^{2\varepsilon/3})\frac{\|\boldsymbol{\gamma}_{[n]}\|}{4\eta_1} - 6\lambda^{\varepsilon/2}\|\boldsymbol{v}^*\| \right)\|\boldsymbol{g}^t\|^2 \\
&> \frac{\|\boldsymbol{\gamma}_{[n]}\|}{5\eta_1}\|\boldsymbol{g}^t\|^2 \\
&\geq \frac{\eta_d}{5\eta_1}\|\boldsymbol{\gamma}_{[n]}\|\|\boldsymbol{v}^* - \boldsymbol{v}^t\|\|\boldsymbol{g}^t\| \\
&\geq \frac{2\eta_d}{5\eta_1}\|\boldsymbol{\gamma}_{[n]}\|L(\boldsymbol{\theta}^t)
\end{aligned}
$$

since $\lambda^{\varepsilon/2} \leq n^{-\frac{9\cdot3}{2}n\Delta^2/\delta^3} < \frac{\delta^2}{120\Delta^2} \leq \frac{\|\boldsymbol{\gamma}_{[n]}\|}{120\eta_1}$.

Hence for all $t \geq T_2$ we have

$$
L(\boldsymbol{\theta}^t) < \frac{\|\boldsymbol{\gamma}_{[n]}\|}{2}\exp\left(-(t - T_2)\frac{2\eta_d\|\boldsymbol{\gamma}_{[n]}\|}{5\eta_1}\right) \leq \frac{\Delta^2}{2}\exp\left(-(t - T_2)\frac{2\delta^4}{5\Delta^2}\right). \qquad \square
$$

*Example* 33. Using the single run from Example 31 again, we illustrate in Figure 4 the progression of several significant measures of the sum $\boldsymbol{v}^t$ of the hidden neurons multiplied by their last-layer weights during the training. In particular, we can see that the alignment with the vector $\boldsymbol{\gamma}_{[n]}$ reaches its maximum around iteration $500$, after which the distance and the angle to the teacher neuron $\boldsymbol{v}^*$ starts to decrease. $\qquad \square$

## F   Proofs for the implicit bias

In this section, we include an explicit subscript $\lambda$ for quantities that depend on the initialisation scale.

A key part of showing that the networks to which the training converges as time tends to infinity themselves have a limit in parameter space as the initialisation scale tends to zero is to establish the existence of that double limit for every ratio between the Euclidean norms of two hidden neurons. The following lemma does that, and provides two alternative expressions for each such double-limit ratio: the limit of the same ratio at time $T_1$ as $\lambda$ tends to zero, and the corresponding limit for the yardstick trajectories as $t$ tends to infinity.

**Lemma 34.** *For all $j, j' \in J_+$ we have*

$$\lim_{\lambda \to 0^+} \lim_{t \to \infty} \frac{\|\boldsymbol{w}_{\lambda,j}^t\|}{\|\boldsymbol{w}_{\lambda,j'}^t\|} = \lim_{\lambda \to 0^+} \frac{\|\boldsymbol{w}_{\lambda,j}^{T_{\lambda,1}}\|}{\|\boldsymbol{w}_{\lambda,j'}^{T_{\lambda,1}}\|} = \lim_{t \to \infty} \frac{\|\boldsymbol{\alpha}_j^t\|}{\|\boldsymbol{\alpha}_{j'}^t\|} \ .$$

*Proof.* Suppose $j, j' \in J_+$.

Recalling [Proposition 22](#) and letting $u_j := \operatorname{artanh} \cos \varphi_j^{T_0}$, for all $t \geq T_0$ we have

$$
\begin{aligned}
\left| \mathrm{d} \ln \frac{\|\boldsymbol{\alpha}_j^t\|}{\|\boldsymbol{\alpha}_{j'}^t\|} / \mathrm{d}t \right| &= |(\overline{\boldsymbol{\alpha}}_j^t - \overline{\boldsymbol{\alpha}}_{j'}^t)^\top \boldsymbol{\gamma}_{[n]}| \\
&= \left| \tanh(u_j + \|\boldsymbol{\gamma}_{[n]}\|(t - T_0)) - \tanh(u_{j'} + \|\boldsymbol{\gamma}_{[n]}\|(t - T_0)) \right| \|\boldsymbol{\gamma}_{[n]}\| \\
&= |\tanh(u_j - u_{j'})| \, \|\boldsymbol{\gamma}_{[n]}\| \\
&\quad \left(1 - \tanh(u_j + \|\boldsymbol{\gamma}_{[n]}\|(t - T_0)) \tanh(u_{j'} + \|\boldsymbol{\gamma}_{[n]}\|(t - T_0))\right) \\
&< |\tanh(u_j - u_{j'})| \, \|\boldsymbol{\gamma}_{[n]}\| \left(1 - \tanh^2(\|\boldsymbol{\gamma}_{[n]}\|(t - T_0))\right) \\
&< |\tanh(u_j - u_{j'})| \, \|\boldsymbol{\gamma}_{[n]}\| \left(1 - \left(1 - 2/\exp(2\|\boldsymbol{\gamma}_{[n]}\|(t - T_0))\right)^2\right) \\
&< 4|\tanh(u_j - u_{j'})| \, \|\boldsymbol{\gamma}_{[n]}\| \exp(-2\|\boldsymbol{\gamma}_{[n]}\|(t - T_0)) \ ,
\end{aligned}
$$

so $\displaystyle\lim_{t \to \infty} \frac{\|\boldsymbol{\alpha}_j^t\|}{\|\boldsymbol{\alpha}_{j'}^t\|}$ exists.

By [Lemma 23 (iv)](#), we have

$$
\begin{aligned}
&\left| \ln \frac{\|\boldsymbol{w}_{\lambda,j}^{T_{\lambda,1}}\|}{\|\boldsymbol{w}_{\lambda,j'}^{T_{\lambda,1}}\|} - \ln \frac{\|\boldsymbol{\alpha}_j^{T_{\lambda,1}}\|}{\|\boldsymbol{\alpha}_{j'}^{T_{\lambda,1}}\|} \right| \\
&\qquad = \left| \left( \ln \|\boldsymbol{w}_{\lambda,j}^{T_{\lambda,1}}/\lambda\| - \ln \|\boldsymbol{\alpha}_j^{T_{\lambda,1}}\| \right) - \left( \ln \|\boldsymbol{w}_{\lambda,j'}^{T_{\lambda,1}}/\lambda\| - \ln \|\boldsymbol{\alpha}_{j'}^{T_{\lambda,1}}\| \right) \right| \leq 2\lambda^{1-3\varepsilon} \ ,
\end{aligned}
$$

so $\displaystyle\lim_{\lambda \to 0^+} \frac{\|\boldsymbol{w}_{\lambda,j}^{T_{\lambda,1}}\|}{\|\boldsymbol{w}_{\lambda,j'}^{T_{\lambda,1}}\|} = \lim_{t \to \infty} \frac{\|\boldsymbol{\alpha}_j^t\|}{\|\boldsymbol{\alpha}_{j'}^t\|}$.

By [Lemma 28 (i), (iv)](#), and [(v)](#), for all $t \geq T_{\lambda,1}$ we have

$$\left| \mathrm{d} \ln \frac{\|\boldsymbol{w}_{\lambda,j}^t\|}{\|\boldsymbol{w}_{\lambda,j'}^t\|} / \mathrm{d}t \right| = |(\overline{\boldsymbol{w}}_{\lambda,j}^t - \overline{\boldsymbol{w}}_{\lambda,j'}^t)^\top \boldsymbol{g}_\lambda^t| < \sqrt{8}\lambda^{\varepsilon/2} \|\boldsymbol{g}_\lambda^t\| \ .$$

By [Lemma 28 (ix)](#) and [Lemma 32 (iii)](#), for all $t \geq T_{\lambda,2}$ we have

$$\|\boldsymbol{g}_\lambda^t\| \leq \eta_1 \|\boldsymbol{v}^* - \boldsymbol{v}_\lambda^t\| < \eta_1 \exp(-5\|\boldsymbol{\gamma}_{[n]}\|(t - T_{\lambda,2})/8) \ .$$

Hence $\displaystyle\lim_{t \to \infty} \frac{\|\boldsymbol{w}_{\lambda,j}^t\|}{\|\boldsymbol{w}_{\lambda,j'}^t\|}$ exists. Moreover, by [Lemma 32 (ii)](#), for all $t \geq T_{\lambda,2}$ we have

$$
\begin{aligned}
\left| \ln \frac{\|\boldsymbol{w}_{\lambda,j}^t\|}{\|\boldsymbol{w}_{\lambda,j'}^t\|} - \ln \frac{\|\boldsymbol{w}_{\lambda,j}^{T_{\lambda,1}}\|}{\|\boldsymbol{w}_{\lambda,j'}^{T_{\lambda,1}}\|} \right| &< \sqrt{8}\lambda^{\varepsilon/2} \Delta^2 \left( \ln\left(\frac{1}{\lambda}\right) \frac{(4 + \varepsilon/2)d\Delta^2}{\delta^6} + \int_0^{t - T_{\lambda,2}} \exp(-5\delta^2 t'/8) \, \mathrm{d}t' \right) \\
&< \sqrt{8}\lambda^{\varepsilon/2} \Delta^2 \left( \ln\left(\frac{1}{\lambda}\right) \frac{(4 + \varepsilon/2)d\Delta^2}{\delta^6} + \frac{8}{5\delta^2} \right) \\
&< \sqrt{8}\lambda^{\varepsilon/2} \ln\left(\frac{1}{\lambda}\right) \frac{(4 + \varepsilon)d\Delta^4}{\delta^6} \\
&< \lambda^{\varepsilon/3} \ln\left(\frac{1}{\lambda}\right)
\end{aligned}
$$

since $\lambda^{-\varepsilon/6} \geq n^{\frac{9}{2}n\Delta^2/\delta^3} \geq \left(2^{\frac{9}{4}n\Delta^2/\delta^3}\right)^2 > \left(\frac{11n\Delta^2}{\delta^3}\right)^2 > \frac{\sqrt{8}(4+\varepsilon)d\Delta^4}{\delta^6}$. Therefore

$$\left| \lim_{t\to\infty} \frac{\|\boldsymbol{w}_{\lambda,j}^t\|}{\|\boldsymbol{w}_{\lambda,j'}^t\|} - \frac{\|\boldsymbol{w}_{\lambda,j}^{T_{\lambda,1}}\|}{\|\boldsymbol{w}_{\lambda,j'}^{T_{\lambda,1}}\|} \right| < \lambda^{\varepsilon/3} \ln\left(\frac{1}{\lambda}\right),$$

so $\displaystyle\lim_{\lambda\to 0+}\lim_{t\to\infty} \frac{\|\boldsymbol{w}_{\lambda,j}^t\|}{\|\boldsymbol{w}_{\lambda,j'}^t\|} = \lim_{\lambda\to 0+} \frac{\|\boldsymbol{w}_{\lambda,j}^{T_{\lambda,1}}\|}{\|\boldsymbol{w}_{\lambda,j'}^{T_{\lambda,1}}\|}$. $\qquad\square$

We are now in a position to prove the main theorem, restated from section 6. It establishes that, as the initialisation scale $\lambda$ tends to zero, the networks with zero loss to which the gradient flow converges tend to a network in the set $\Theta_{\boldsymbol{v}^*}$ (defined in section 6) of balanced interpolators of rank 1.

**Theorem 7.** *Under Assumptions 1 and 2,* $L\left(\lim_{t\to\infty}\boldsymbol{\theta}_\lambda^t\right) = 0$ *and* $\lim_{\lambda\to 0+}\lim_{t\to\infty}\boldsymbol{\theta}_\lambda^t \in \Theta_{\boldsymbol{v}^*}$.

*Proof.* By Lemma 32 (v), we have $\lim_{t\to\infty} L(\boldsymbol{\theta}_\lambda^t) = 0$.

By Proposition 9, for all $t \in [0,\infty)$ we have

$$\int_t^\infty \left\| \frac{\mathrm{d}}{\mathrm{d}t'}\boldsymbol{\theta}_\lambda^{t'} \right\|^2 \mathrm{d}t' = -\int_t^\infty \frac{\mathrm{d}}{\mathrm{d}t'} L\left(\boldsymbol{\theta}_\lambda^{t'}\right) \mathrm{d}t' = L(\boldsymbol{\theta}_\lambda^t) .$$

Hence $\boldsymbol{\theta}_\lambda^\infty := \lim_{t\to\infty}\boldsymbol{\theta}_\lambda^t$ exists, and since the loss function is continuous, we have $L(\boldsymbol{\theta}_\lambda^\infty) = 0$.

Let us write $\boldsymbol{\theta}_\lambda^\infty = \left([a_{\lambda,1}^\infty,\ldots,a_{\lambda,m}^\infty],[\boldsymbol{w}_{\lambda,1}^\infty,\ldots,\boldsymbol{w}_{\lambda,m}^\infty]^\top\right)$, and let $\boldsymbol{v}_\lambda^\infty := \sum_{j\in J_+} a_{\lambda,j}^\infty \boldsymbol{w}_{\lambda,j}^\infty$.

Since $\Theta$ is closed, for all $j \in [m]$ we have $a_{\lambda,j}^\infty = \|\boldsymbol{w}_{\lambda,j}^\infty\|$.

Recalling $\mathrm{span}\{\boldsymbol{x}_1,\ldots,\boldsymbol{x}_n\} = \mathbb{R}^d$ and Lemma 28 (iii), we have $\boldsymbol{v}_\lambda^\infty = \boldsymbol{v}^*$.

By Lemma 28 (i), for all $j \in J_+$ we have $\overline{\boldsymbol{w}}_{\lambda,j}^{\infty\top} \boldsymbol{v}^* > 1 - 4\lambda^\varepsilon$, so

$$1 \leq \sum_{j\in J_+} \|\boldsymbol{w}_{\lambda,j}^\infty\|^2 < \frac{1}{1-4\lambda^\varepsilon} ,$$

and thus $\lim_{\lambda\to 0+} \sum_{j\in J_+} \|\boldsymbol{w}_{\lambda,j}^\infty\|^2 = 1$.

By Lemma 34, for all $j,j' \in J_+$, $\lim_{\lambda\to 0+} \frac{\|\boldsymbol{w}_{\lambda,j}^\infty\|}{\|\boldsymbol{w}_{\lambda,j'}^\infty\|}$ exists.

Hence, for all $j \in J_+$, we have that $a_j^\infty := \lim_{\lambda\to 0+}\|\boldsymbol{w}_{\lambda,j}^\infty\|$ and $\lim_{\lambda\to 0+}\overline{\boldsymbol{w}}_{\lambda,j}^\infty$ exist, and so also $\boldsymbol{w}_j^\infty := \lim_{\lambda\to 0+}\boldsymbol{w}_{\lambda,j}^\infty$ exists. Moreover, we have $a_j^\infty = \|\boldsymbol{w}_j^\infty\|$ and $\overline{\boldsymbol{w}}_j^\infty = \boldsymbol{v}^*$ for all $j \in J_+$, and we have $\sum_{j\in J_+}\|\boldsymbol{w}_j^\infty\|^2 = 1$.

By Proposition 18, Lemma 19 (ii), Lemma 21, and Assumption 1 (v), for all $j \notin J_+$, we have $\|\boldsymbol{w}_{\lambda,j}^t\| \leq \lambda\|\boldsymbol{z}_j\|$ for all $t \in [0,\infty)$, so $a_j^\infty := \lim_{\lambda\to 0+} a_{\lambda,j}^\infty = 0$ and $\boldsymbol{w}_j^\infty := \lim_{\lambda\to 0+}\boldsymbol{w}_{\lambda,j}^\infty = \boldsymbol{0}$.

Therefore $\boldsymbol{\theta}^\infty := \left([a_1^\infty,\ldots,a_m^\infty],[\boldsymbol{w}_1^\infty,\ldots,\boldsymbol{w}_m^\infty]^\top\right) \in \Theta_{\boldsymbol{v}^*}$. $\qquad\square$

*Example* 35. Continuing with the single run from Example 31, we illustrate in Figure 5 how, although the loss converges to zero exponentially fast, for a fixed positive initialisation scale the angles between the hidden neurons in the aligned bundle do not in general decrease to zero.

Figure 6 shows the course of the training from the point of view of the two measures of network complexity, namely the nuclear and square Euclidean norms: during the alignment phase they are both close to zero, they grow rapidly as the network departs from the saddle at the origin, and they converge towards 1 and 2 respectively as the loss converges to zero. $\qquad\square$

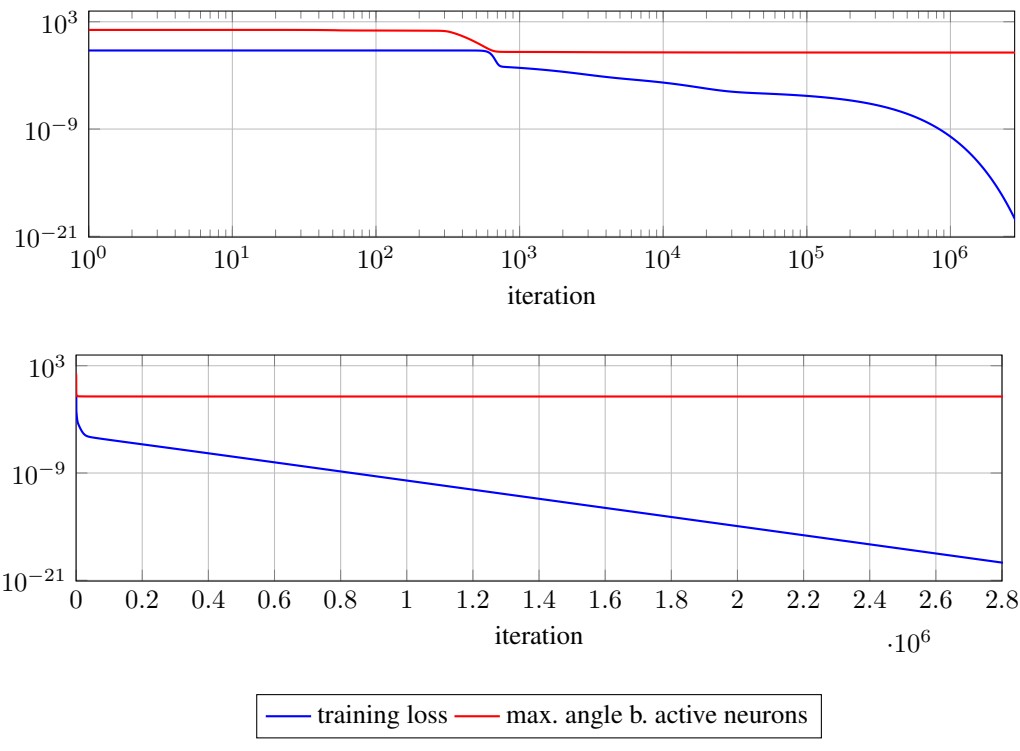

Figure 5: The evolution of the training loss and the maximum angle between active hidden neurons during the training. The two plots are of the same data, the vertical axes are logarithmic, the horizontal axis is logarithmic in the top plot and linear in the bottom plot, and the angles are in degrees.

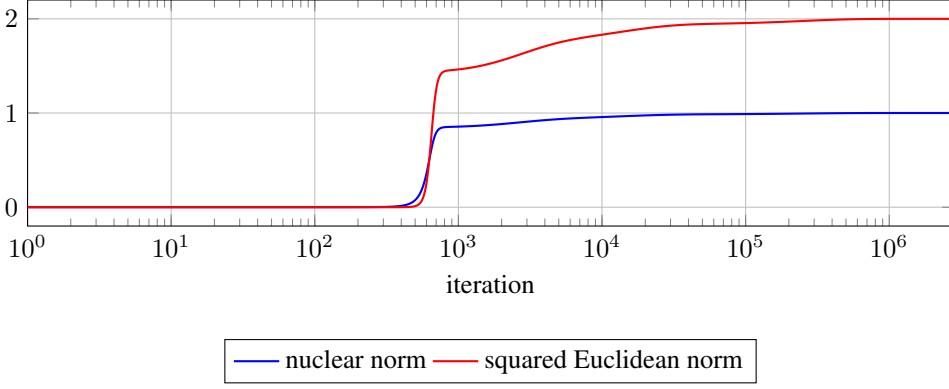

Figure 6: The evolution of the nuclear and square Euclidean norms during the training. The horizontal axis is logarithmic, and the vertical axis is linear.

# G  Proofs and examples for the interpolators

First we prove the following theorem, restated from section 7. For case $\mathcal{M} < 0$, the main part of our argument shows that, if a global minimiser of $\|\boldsymbol{\theta}\|^2$ was not a member of $\Theta_{\boldsymbol{v}^*}$, then we could obtain from each hidden neuron $\boldsymbol{w}_j$ a vector $\boldsymbol{p}_j$ such that the inner products of the inputs $\boldsymbol{x}_i$ with the vector $\sum_{j\in[m]} a_j\,\boldsymbol{p}_j$ coincide with the network outputs, the projection of each vector $\boldsymbol{p}_j$ onto the teacher neuron has length at most $\|\boldsymbol{w}_j\|$, and at least one of those inequalities is strict, leading to a contradiction. For case $\mathcal{M} > 0$, we provide counterexample interpolator networks, the Euclidean norm of whose parameters is smaller than the Euclidean norm of the networks in $\Theta_{\boldsymbol{v}^*}$.

**Theorem 8.** *Under Assumptions 1 and 3:*

  *(i) if $\mathcal{M} < 0$ then $\Theta_{\boldsymbol{v}^*}$ is the set of all global minimisers of $\|\boldsymbol{\theta}\|^2$ subject to $L(\boldsymbol{\theta}) = 0$;*

  *(ii) if $\mathcal{M} > 0$ then no point in $\Theta_{\boldsymbol{v}^*}$ is a global minimiser of $\|\boldsymbol{\theta}\|^2$ subject to $L(\boldsymbol{\theta}) = 0$.*

*Proof.* For all $\boldsymbol{\theta} = (\boldsymbol{a}, \boldsymbol{W}) \in \Theta_{\boldsymbol{v}^*}$ we have $L(\boldsymbol{\theta}) = 0$ and $\|\boldsymbol{\theta}\|^2 = \sum_{j\in[m]}(a_j^2 + \|\boldsymbol{w}_j\|^2) = 2$.

To establish the case when $\mathcal{M} < 0$, supposing $\boldsymbol{\theta} = (\boldsymbol{a}, \boldsymbol{W}) \in \mathbb{R}^m \times \mathbb{R}^{m\times d}$ is a global minimiser of $\|\boldsymbol{\theta}\|^2$ subject to $L(\boldsymbol{\theta}) = 0$, it suffices to show $\boldsymbol{\theta} \in \Theta_{\boldsymbol{v}^*}$.

By the minimality of $\|\boldsymbol{\theta}\|^2$ subject to $L(\boldsymbol{\theta}) = 0$, for all $j \in [m]$, if $a_j = 0$ then $\boldsymbol{w}_j = \boldsymbol{0}$, and also if $\forall i \in [d]\colon \sigma(\boldsymbol{w}_j^\top \boldsymbol{x}_i) = 0$ then $a_j = 0$.

For all $j \in [m]$, if $a_j = 0$ and $\boldsymbol{w}_j = \boldsymbol{0}$, then removing $a_j$ and $\boldsymbol{w}_j$ from $\boldsymbol{\theta}$ preserves the values of $\|\boldsymbol{\theta}\|^2$ and $L(\boldsymbol{\theta})$, and the truth or falsity of $\boldsymbol{\theta} \in \Theta_{\boldsymbol{v}^*}$. Hence we may assume for all $j \in [m]$ that $a_j \neq 0$ and $\exists i \in [d]\colon \sigma(\boldsymbol{w}_j^\top \boldsymbol{x}_i) \neq 0$.

For all $j \in [m]$, replacing $a_j$ by $\sqrt{\|\boldsymbol{w}_j\|/|a_j|}\, a_j$ and $\boldsymbol{w}_j$ by $\sqrt{|a_j|/\|\boldsymbol{w}_j\|}\, \boldsymbol{w}_j$ preserves $L(\boldsymbol{\theta})$, and decreases $\|\boldsymbol{\theta}\|^2$ unless $|a_j| = \|\boldsymbol{w}_j\|$. Hence $\boldsymbol{\theta} \in \Theta$.

For all $j \in [m]$, let $K_j := \{k \in [d] \mid \boldsymbol{w}_j^\top \boldsymbol{x}_k \geq 0\}$ and

$$\boldsymbol{p}_j := \sum_{k\in K_j}(\boldsymbol{w}_j^\top \boldsymbol{x}_k)\boldsymbol{\chi}_k \qquad\qquad \boldsymbol{q}_j := \sum_{k\notin K_j}-(\boldsymbol{w}_j^\top \boldsymbol{x}_k)\boldsymbol{\chi}_k \;,$$

so that $\boldsymbol{w}_j = \boldsymbol{p}_j - \boldsymbol{q}_j$. Observe also that since $\exists i \in [d]\colon \sigma(\boldsymbol{w}_j^\top \boldsymbol{x}_i) \neq 0$, we have $\boldsymbol{p}_j \neq \boldsymbol{0}$.

*Claim* 36. For all $j \in [m]$ we have $|\boldsymbol{p}_j^\top \boldsymbol{v}^*| \leq \|\boldsymbol{w}_j\|$, and if $\boldsymbol{q}_j \neq \boldsymbol{0}$ then the inequality is strict.

*Proof of claim.* Suppose $j \in [m]$. If $\boldsymbol{q}_j = \boldsymbol{0}$ then $\boldsymbol{w}_j = \boldsymbol{p}_j$. If $\boldsymbol{q}_j \neq \boldsymbol{0}$ then

$$\begin{aligned}
\|\boldsymbol{w}_j\|^2 &= \|\boldsymbol{p}_j\|^2 + \|\boldsymbol{q}_j\|^2 - 2\|\boldsymbol{p}_j\|\|\boldsymbol{q}_j\|\cos\angle(\boldsymbol{p}_j, \boldsymbol{q}_j)\\
&> \|\boldsymbol{p}_j\|^2 + \|\boldsymbol{q}_j\|^2 - 2\|\boldsymbol{p}_j\|\|\boldsymbol{q}_j\|\sin\angle(\boldsymbol{p}_j, \boldsymbol{v}^*)\\
&= \|\boldsymbol{p}_j\|^2 \cos^2\angle(\boldsymbol{p}_j, \boldsymbol{v}^*) + (\|\boldsymbol{p}_j\|\sin\angle(\boldsymbol{p}_j, \boldsymbol{v}^*) - \|\boldsymbol{q}_j\|)^2\\
&\geq \|\boldsymbol{p}_j\|^2 \cos^2\angle(\boldsymbol{p}_j, \boldsymbol{v}^*)\\
&= (\boldsymbol{p}_j^\top \boldsymbol{v}^*)^2 \;. \qquad\qquad\qquad\qquad\qquad\qquad\qquad\qquad\qquad\square
\end{aligned}$$

Now for all $i \in [d]$ we have

$$\left(\sum_{j\in[m]} a_j\,\boldsymbol{p}_j\right)^\top \boldsymbol{x}_i = \sum_{j\in[m]} a_j\,\boldsymbol{p}_j^\top \boldsymbol{x}_i = \sum_{j\in[m]} a_j\,\sigma(\boldsymbol{w}_j^\top \boldsymbol{x}_i) = \boldsymbol{v}^{*\top}\boldsymbol{x}_i \;.$$

Since $\mathrm{span}\{\boldsymbol{x}_1, \ldots, \boldsymbol{x}_d\} = \mathbb{R}^d$, we infer $\sum_{j\in[m]} a_j\,\boldsymbol{p}_j = \boldsymbol{v}^*$, so by Claim 36 we have

$$1 = \sum_{j\in[m]} a_j\,\boldsymbol{p}_j^\top \boldsymbol{v}^* \leq \sum_{j\in[m]} |a_j\,\boldsymbol{p}_j^\top \boldsymbol{v}^*| \leq \sum_{j\in[m]} \|a_j\,\boldsymbol{w}_j\| = \tfrac{1}{2}\sum_{j\in[m]} (a_j^2 + \|\boldsymbol{w}_j\|^2) = \tfrac{1}{2}\|\boldsymbol{\theta}\|^2 \leq 1 \;,$$

and if $\boldsymbol{q}_j \neq \boldsymbol{0}$ for some $j \in [m]$ then the second of the three inequalities is strict. However, all three inequalities must be equalities, so also for all $j \in [m]$ we have $\boldsymbol{q}_j = \boldsymbol{0}$. Hence $a_j\,\boldsymbol{w}_j^\top \boldsymbol{v}^* =$

$a_j\,\boldsymbol{p}_j^\top\boldsymbol{v}^* = \|a_j\,\boldsymbol{w}_j\|$, and thus $\overline{a_j\,\boldsymbol{w}_j} = \boldsymbol{v}^*$. Since $a_j < 0$ would imply $\overline{\boldsymbol{w}}_j = -\boldsymbol{v}^*$, which would contradict $\boldsymbol{q}_j = \boldsymbol{0}$, we have $a_j > 0$ and $\overline{\boldsymbol{w}}_j = \boldsymbol{v}^*$. Therefore $\boldsymbol{\theta} \in \Theta_{\boldsymbol{v}^*}$.

To establish the case when $\mathcal{M} > 0$, it suffices to exhibit $\boldsymbol{\theta} = (\boldsymbol{a}, \boldsymbol{W}) \in \mathbb{R}^m \times \mathbb{R}^{m\times d}$ such that $L(\boldsymbol{\theta}) = 0$ and $\|\boldsymbol{\theta}\|^2 < 2$.

Let $\emptyset \subsetneq K \subsetneq [d]$, $\boldsymbol{0} \neq \boldsymbol{p} \in \mathrm{cone}\{\boldsymbol{\chi}_k \mid k \in K\}$, and $\boldsymbol{0} \neq \boldsymbol{q} \in \mathrm{cone}\{\boldsymbol{\chi}_k \mid k \notin K\}$ be such that $\cos\angle(\boldsymbol{p}, \boldsymbol{q}) > \sin\angle(\boldsymbol{p}, \boldsymbol{v}^*)$. We have $\overline{\boldsymbol{p}} = \sum_{k\in K} b_k \boldsymbol{\chi}_k$ for some $b_k \geq 0$, and $\overline{\boldsymbol{q}} = \sum_{k\notin K} c_k \boldsymbol{\chi}_k$ for some $c_k \geq 0$. Since $\mathrm{span}\{\boldsymbol{\chi}_1, \ldots, \boldsymbol{\chi}_d\} = \mathbb{R}^d$, we have $\cos\angle(\boldsymbol{p}, \boldsymbol{q}) < 1$.

*Case $\angle(\boldsymbol{p}, \boldsymbol{v}^*) \leq \pi/2$.* Then $\cos\angle(\boldsymbol{p}, \boldsymbol{v}^*) > \sin\angle(\boldsymbol{p}, \boldsymbol{q})$.

Let $\xi := \min\{\min\{y_k/b_k \mid k \in K \wedge b_k \neq 0\}, \cos\angle(\boldsymbol{p}, \boldsymbol{v}^*) - \sin\angle(\boldsymbol{p}, \boldsymbol{q})\}$, $\boldsymbol{r} := \overline{\boldsymbol{p}} - \overline{\boldsymbol{q}}\,\overline{\boldsymbol{q}}^\top \overline{\boldsymbol{p}}$,

$$a_1 := 1 \qquad\qquad\qquad \boldsymbol{w}_1 := \boldsymbol{v}^* - \xi\,\overline{\boldsymbol{p}}$$
$$a_2 := \sqrt{\xi\|\boldsymbol{r}\|} \qquad\qquad \boldsymbol{w}_2 := \sqrt{\xi/\|\boldsymbol{r}\|}\,\boldsymbol{r}\,,$$

and $a_j := 0$ and $\boldsymbol{w}_j := \boldsymbol{0}$ for all $j > 2$.

From

$$\boldsymbol{r}^\top \boldsymbol{x}_i = \begin{cases} b_i & \text{if } i \in K, \\ -c_i \cos\angle(\boldsymbol{p}, \boldsymbol{q}) & \text{if } i \notin K, \end{cases}$$

it follows that $h_{\boldsymbol{\theta}}(\boldsymbol{x}_i) = y_i$ for all $i \in [d]$, i.e. $L(\boldsymbol{\theta}) = 0$.

We have

$$\begin{aligned}
\|\boldsymbol{\theta}\|^2 &= a_1^2 + \|\boldsymbol{w}_1\|^2 + a_2^2 + \|\boldsymbol{w}_2\|^2 \\
&= 2 + \xi^2 - 2\xi\,\overline{\boldsymbol{p}}^\top \boldsymbol{v}^* + 2\xi\|\boldsymbol{r}\| \\
&= 2 - \xi\big[2(\cos\angle(\boldsymbol{p}, \boldsymbol{v}^*) - \sin\angle(\boldsymbol{p}, \boldsymbol{q})) - \xi\big] \\
&\leq 2 - \xi^2\,.
\end{aligned}$$

*Case $\angle(\boldsymbol{p}, \boldsymbol{v}^*) > \pi/2$.* Then $-\cos\angle(\boldsymbol{p}, \boldsymbol{v}^*) > \sin\angle(\boldsymbol{p}, \boldsymbol{q})$.

Let $\xi := -\cos\angle(\boldsymbol{p}, \boldsymbol{v}^*) - \sin\angle(\boldsymbol{p}, \boldsymbol{q})$, $\boldsymbol{r} := \overline{\boldsymbol{p}} - \overline{\boldsymbol{q}}\,\overline{\boldsymbol{q}}^\top \overline{\boldsymbol{p}}$,

$$a_1 := 1 \qquad\qquad\qquad \boldsymbol{w}_1 := \boldsymbol{v}^* + \xi\,\overline{\boldsymbol{p}}$$
$$a_2 := -\sqrt{\xi\|\boldsymbol{r}\|} \qquad\qquad \boldsymbol{w}_2 := \sqrt{\xi/\|\boldsymbol{r}\|}\,\boldsymbol{r}\,,$$

and $a_j := 0$ and $\boldsymbol{w}_j := \boldsymbol{0}$ for all $j > 2$.

From

$$\boldsymbol{r}^\top \boldsymbol{x}_i = \begin{cases} b_i & \text{if } i \in K, \\ -c_i \cos\angle(\boldsymbol{p}, \boldsymbol{q}) & \text{if } i \notin K, \end{cases}$$

it follows that $h_{\boldsymbol{\theta}}(\boldsymbol{x}_i) = y_i$ for all $i \in [d]$, i.e. $L(\boldsymbol{\theta}) = 0$.

We have

$$\begin{aligned}
\|\boldsymbol{\theta}\|^2 &= a_1^2 + \|\boldsymbol{w}_1\|^2 + a_2^2 + \|\boldsymbol{w}_2\|^2 \\
&= 2 + \xi^2 + 2\xi\,\overline{\boldsymbol{p}}^\top \boldsymbol{v}^* + 2\xi\|\boldsymbol{r}\| \\
&= 2 - \xi\big[2(-\cos\angle(\boldsymbol{p}, \boldsymbol{v}^*) - \sin\angle(\boldsymbol{p}, \boldsymbol{q})) - \xi\big] \\
&= 2 - \xi^2\,. \qquad\qquad\qquad\qquad\qquad\qquad\qquad\quad \square
\end{aligned}$$

*Example* 37. Now we present two families of examples of a teacher neuron and training points that respectively satisfy: $\mathcal{M} < 0$ for any $d > 1$, and $\mathcal{M} > 0$ for any $d > 2$.

Let $\{\boldsymbol{e}_i\}_{i=1}^d$ denote the standard basis of $\mathbb{R}^d$.

$\mathcal{M} < 0$. Let $\xi \in (0, 1)$ and consider, for all $i \in [d]$, vectors

$$\boldsymbol{x}_i := \left(1 - \frac{d-1}{d}(1-\xi)\right)\boldsymbol{e}_i + \frac{1-\xi}{d}\sum_{k\neq i}\boldsymbol{e}_k\,.$$

Take $s := (1, \ldots, 1) \in \mathbb{R}^d$ and $v^* := \bar{s}$.

It can be checked that, for vectors $\chi_1, \ldots, \chi_d$ defined by

$$\chi_k := \frac{1}{\xi}\left(e_k - \frac{1-\xi}{d}s\right),$$

we have $[\chi_1, \ldots, \chi_d]^\top = X^{-1}$.

Notice that whenever $k \neq i$ we have $\chi_k^\top \chi_i = \frac{1}{\xi^2}\left(-2\frac{1-\xi}{d} + \left(\frac{1-\xi}{d}\right)^2 d\right) = \frac{1-\xi}{\xi^2 d}(-2 + 1 - \xi) < 0$.

Hence for all $\emptyset \subsetneq K \subsetneq [d]$, all $\mathbf{0} \neq p \in \text{cone}\{\chi_k \mid k \in K\}$, and all $\mathbf{0} \neq q \in \text{cone}\{\chi_i \mid i \notin K\}$ we have $\cos \angle(p, q) < 0$. Thus $\mathcal{M} < 0$.

It remains to verify that $\angle(v^*, x_i) < \pi/4$ for all $i$. Indeed

$$\begin{aligned}
\|x_i\|^2 &= \left(1 - \frac{d-1}{d}(1-\xi)\right)^2 + (d-1)\left(\frac{1-\xi}{d}\right)^2 \\
&= \left(\frac{1}{d} + \xi\left(1 - \frac{1}{d}\right)\right)^2 + \frac{d-1}{d^2} - \frac{d-1}{d^2}2\xi + \frac{d-1}{d^2}\xi^2 \\
&= \frac{1}{d^2} + 2\xi\frac{1}{d}\left(1 - \frac{1}{d}\right) + \left(1 - \frac{1}{d}\right)^2\xi^2 + \frac{d-1}{d^2} - \frac{d-1}{d^2}2\xi + \frac{d-1}{d^2}\xi^2 \\
&= \frac{1}{d} + \frac{(d-1)^2 + (d-1)}{d^2}\xi^2 \\
&= \frac{1}{d} + \frac{d-1}{d}\xi^2,
\end{aligned}$$

so in particular $\|s\|^2\|x_i\|^2 = 1 + (d-1)\xi^2$. Therefore

$$\begin{aligned}
\cos \angle(v^*, x_i) &= \frac{s^\top x_i}{\|s\|\|x_i\|} \\
&> \frac{\left(1 - \frac{d-1}{d}(1-\xi)\right) + (d-1)\frac{1-\xi}{d}}{1 + \frac{1}{2}(d-1)\xi^2} \\
&= \frac{1}{1 + \frac{1}{2}(d-1)\xi^2},
\end{aligned}$$

so it suffices to take $\xi \leq \sqrt{\frac{2(\sqrt{2}-1)}{d-1}}$.

$\mathcal{M} > 0$.  For $d > 2$, let $b \geq 11$, and consider the data points

$$\begin{aligned}
x_1 &:= b\,e_1 \\
x_2 &:= b\,e_1 - \sqrt{b}\,e_2 + e_3 \\
x_3 &:= b\,e_1 + \sqrt{b}\,e_2 + e_3 \\
x_i &:= b\,e_1 + e_i \quad \text{for all } 4 \leq i \leq d
\end{aligned}$$

and the teacher neuron $v^* := \frac{4}{5}e_1 + \frac{3}{5}e_3$.

For all $i$ we have

$$\cos \angle(v^*, x_i) > \frac{4b}{5\sqrt{b^2 + b + 1}} > \frac{4}{5}\frac{b}{b+1} \geq \frac{4}{5}\frac{11}{12} = \frac{11}{15} > \frac{1}{\sqrt{2}}.$$

Straightforward calculation shows that, for $[\chi_1, \ldots, \chi_d]^\top := X^{-1}$, we have

$$\begin{aligned}
\chi_2 &= -\frac{1}{2\sqrt{b}}e_2 + \frac{1}{2}e_3 \\
\chi_3 &= \phantom{-}\frac{1}{2\sqrt{b}}e_2 + \frac{1}{2}e_3
\end{aligned}$$

and hence

$$\cos \angle(\boldsymbol{\chi}_2, \boldsymbol{\chi}_3) - \sin \angle(\boldsymbol{\chi}_2, \boldsymbol{v}^*) = \frac{\frac{1}{4} - \frac{1}{4b}}{\frac{1}{4} + \frac{1}{4b}} - \sqrt{1 - \frac{9}{100(\frac{1}{4} + \frac{1}{4b})}}$$

$$= \frac{b-1}{b+1} - \sqrt{1 - \frac{9}{25}\frac{b}{b+1}}$$

$$\geq \frac{5}{6} - \sqrt{\frac{67}{100}}$$

$$> 0 \, . \qquad \qquad \square$$

*Remark* 38.    (i) For any $\boldsymbol{\theta}$ such that $L(\boldsymbol{\theta}) = 0$, we have $\|\boldsymbol{\theta}\|^2 \geq 2\, h_{\boldsymbol{\theta}}(\boldsymbol{x}_1)/\|\boldsymbol{x}_1\| = 2\cos\angle(\boldsymbol{v}^*, \boldsymbol{x}_1) > \sqrt{2}$.

(ii) For $d = 2$, since $\angle(\boldsymbol{x}_1, \boldsymbol{x}_2) < \pi/2$, we have $\angle(\boldsymbol{\chi}_1, \boldsymbol{\chi}_2) > \pi/2$, so necessarily $\mathcal{M} < 0$.

(iii) As its proof above shows, Theorem 8 remains true if we relax the correlation between the teacher neuron and the training points to $\angle(\boldsymbol{v}^*, \boldsymbol{x}_i) < \pi/2$ for all $i$.

# H    Additional information about the experiments

For both the centred and the uncentred schemes of generating the training dataset (defined in section 8), we train a one-hidden layer ReLU network by gradient descent with learning rate $0.01$, from a balanced initialisation such that $\boldsymbol{z}_j \overset{\text{i.i.d.}}{\sim} \mathcal{N}(\boldsymbol{0}, \frac{1}{dm}\boldsymbol{I}_d)$ and $s_j \overset{\text{i.i.d.}}{\sim} \mathcal{U}\{\pm 1\}$, for a range of initialisation scales $\lambda$, and for several combinations of input dimensions $d$ and network widths $m$.

The plots in Figure 7, which extends Figure 1 in the main, are obtained by varying the input dimension as $d = 4, 16, 64, 256, 1024$ while keeping the network width at $m = 200$. The plots in Figure 8 are obtained with input dimension $d = 1024$ by varying the network width as $m = 25, 50, 200$. For all twelve plots, we vary the initialisation scale as $\lambda = 4^2, 4^1, \ldots, 4^{-12}, 4^{-13}$, and we train the network until the number of iterations reaches $2 \cdot 10^7$ or the loss drops below $10^{-9}$. The plots are in line with Theorem 7, showing how the three different proxies of rank decrease as $\lambda$ decreases.

Figure 9 complements Figure 2 in the main, illustrating exponentially fast convergence of the training loss (cf. Theorem 6) and reduction of the outside distribution test loss as $\lambda$ decreases, for the uncentred scheme of generating the training dataset.

The medians plotted in Figure 7 and Figure 8, as well as the corresponding standard deviations, can be found in Tables 1–6 and Tables 7–12 respectively.

The experiments were run using Python 3.10.4 and Pytorch 1.12.1 with CUDA 11.7 on a cluster utilising Intel Xeon Platinum 8268 processors. Some experiments for dimension 1024 also used NVIDIA RTX 6000 GPUs. The time taken per iteration greatly depends on the dimension and the width. For dimension 1024 and width 200, about 300 iterations per second could be performed on the CPU. The GPU was about 20% faster in this setting. The total number of iterations performed for dimension 1024 was about 1.6 billion. Experiments for lower dimensions or smaller widths are less demanding.

Overall, these numerical results correspond to our theoretical predictions, and suggest that the training dynamics and the implicit bias we established theoretically occur in practical settings in which some of our assumptions are relaxed.

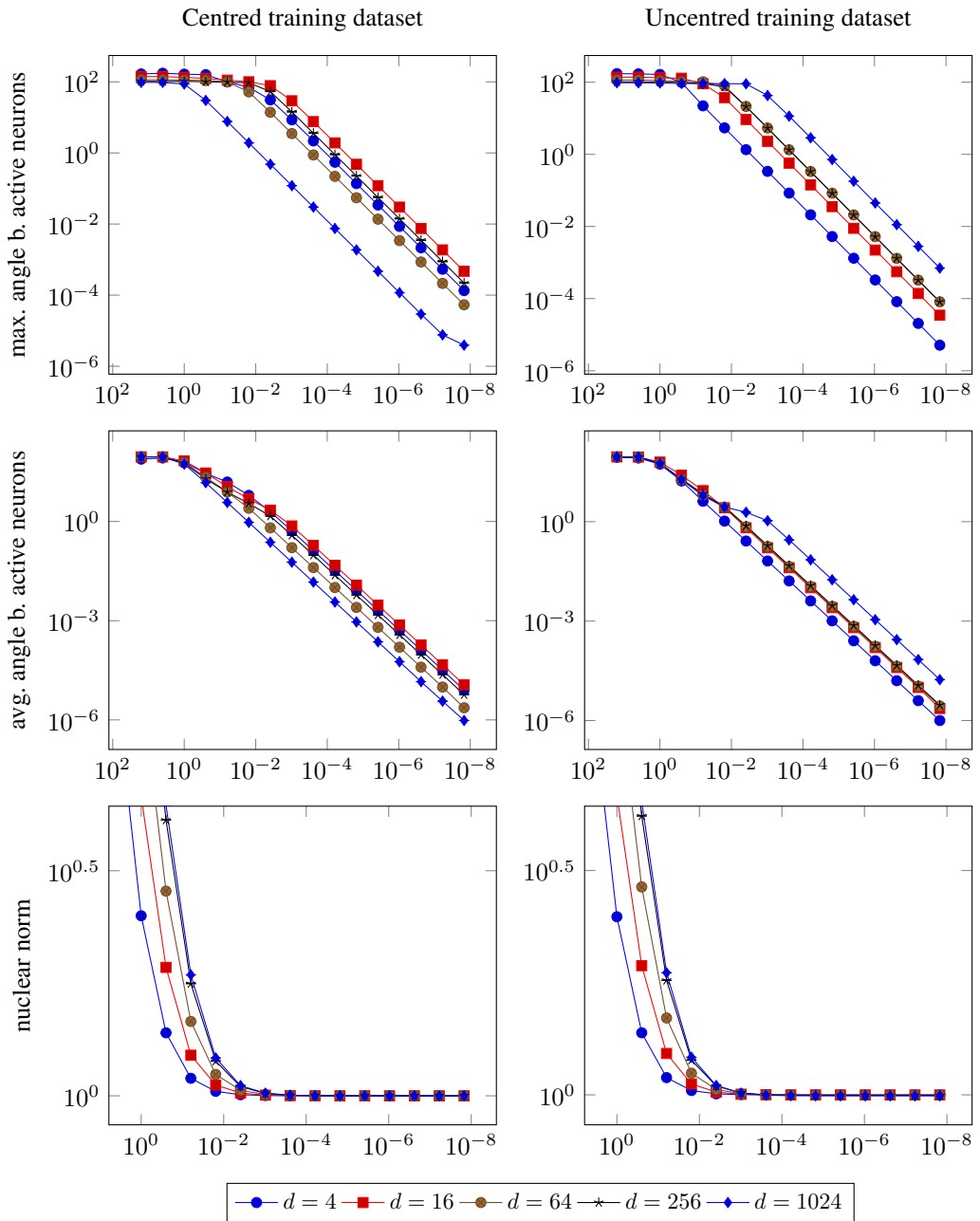

Figure 7: Dependence of the maximum angle between active hidden neurons, of the average angle between active hidden neurons, and of the nuclear norm of the hidden-layer weights on the initialisation scale $\lambda$, for the two generation schemes of the training dataset, the five different input dimensions, and network width $200$, at the end of the training. Both axes are logarithmic, and each point plotted shows the median over five trials.

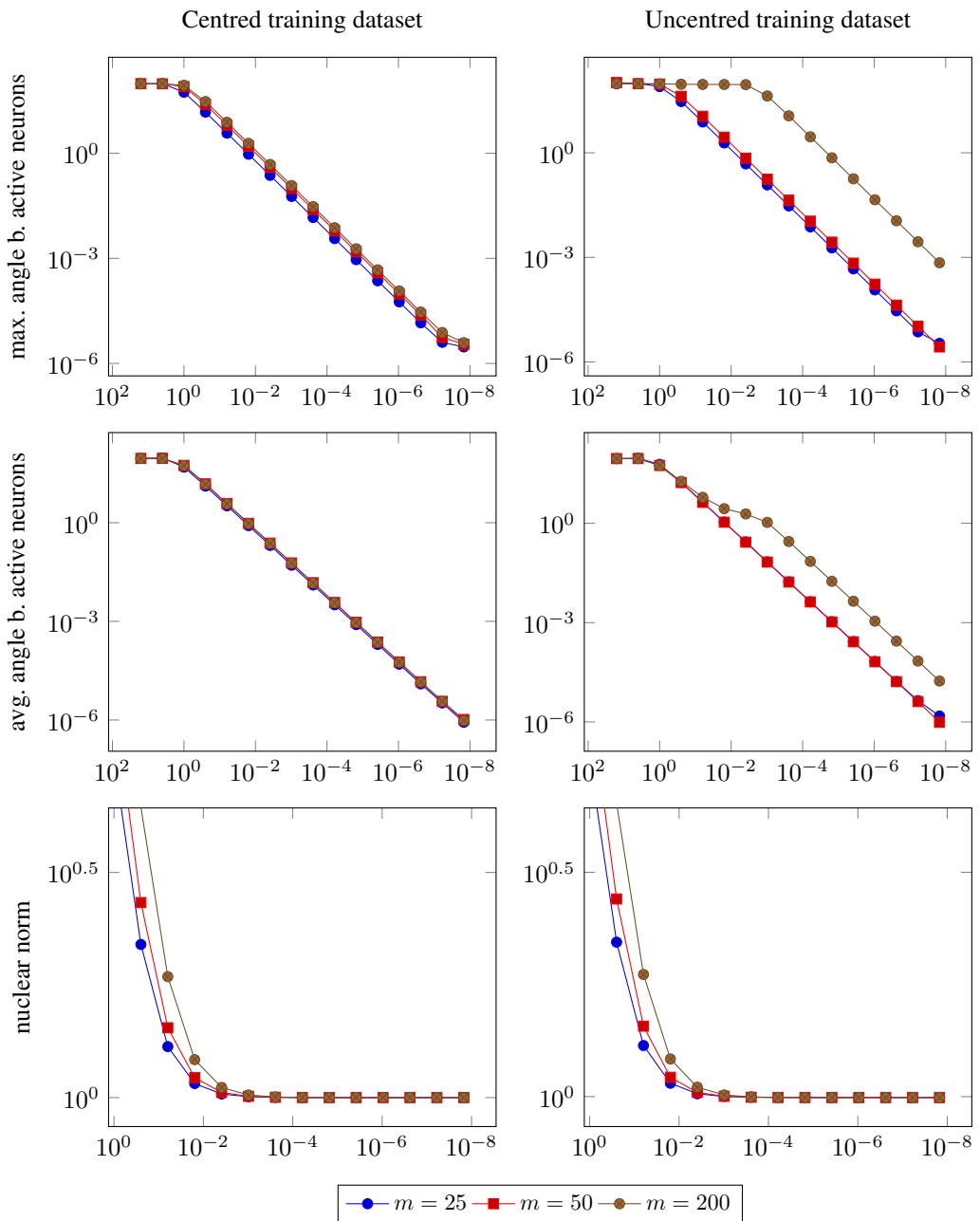

Figure 8: Dependence of the maximum angle between active hidden neurons, of the average angle between active hidden neurons, and of the nuclear norm of the hidden-layer weights on the initialisation scale $\lambda$, for the two generation schemes of the training dataset, the three different network widths, and input dimension $1024$, at the end of the training. Both axes are logarithmic, and each point plotted shows the median over five trials.

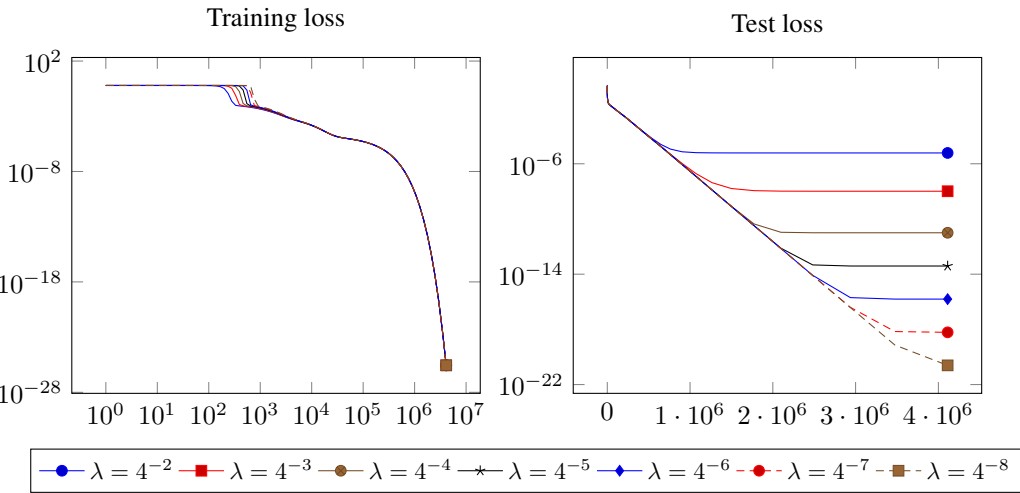

Figure 9: Evolution of the training loss, and of an outside distribution test loss, during training for an example uncentred training dataset in dimension $16$ and with $m = 25$. The horizontal axes show iterations; they are logarithmic for the training loss, and linear for the test loss. The vertical axes are logarithmic.

Table 1: The medians over five trials plotted in Figure 7 on the top left, with the standard deviations shown in parentheses, both rounded to four-digit mantissas.

| $\lambda$ | $d = 4$ | | $d = 16$ | | $d = 64$ | |
|---|---|---|---|---|---|---|
| $4^2$ | $1.665 \cdot 10^2$ | $(1.045 \cdot 10^1)$ | $1.405 \cdot 10^2$ | $(3.256 \cdot 10^0)$ | $1.136 \cdot 10^2$ | $(1.466 \cdot 10^0)$ |
| $4^1$ | $1.743 \cdot 10^2$ | $(3.577 \cdot 10^0)$ | $1.399 \cdot 10^2$ | $(3.818 \cdot 10^0)$ | $1.140 \cdot 10^2$ | $(1.252 \cdot 10^0)$ |
| $4^0$ | $1.644 \cdot 10^2$ | $(7.951 \cdot 10^0)$ | $1.336 \cdot 10^2$ | $(2.045 \cdot 10^0)$ | $1.122 \cdot 10^2$ | $(3.772 \cdot 10^0)$ |
| $4^{-1}$ | $1.592 \cdot 10^2$ | $(2.641 \cdot 10^1)$ | $1.246 \cdot 10^2$ | $(8.589 \cdot 10^0)$ | $1.089 \cdot 10^2$ | $(2.359 \cdot 10^1)$ |
| $4^{-2}$ | $9.943 \cdot 10^1$ | $(5.393 \cdot 10^1)$ | $1.114 \cdot 10^2$ | $(1.892 \cdot 10^1)$ | $9.976 \cdot 10^1$ | $(3.900 \cdot 10^1)$ |
| $4^{-3}$ | $7.037 \cdot 10^1$ | $(5.924 \cdot 10^1)$ | $1.003 \cdot 10^2$ | $(3.897 \cdot 10^1)$ | $5.205 \cdot 10^1$ | $(4.263 \cdot 10^1)$ |
| $4^{-4}$ | $3.106 \cdot 10^1$ | $(6.113 \cdot 10^1)$ | $7.797 \cdot 10^1$ | $(4.700 \cdot 10^1)$ | $1.391 \cdot 10^1$ | $(4.877 \cdot 10^1)$ |
| $4^{-5}$ | $8.538 \cdot 10^0$ | $(6.251 \cdot 10^1)$ | $2.931 \cdot 10^1$ | $(4.519 \cdot 10^1)$ | $3.500 \cdot 10^0$ | $(4.279 \cdot 10^1)$ |
| $4^{-6}$ | $2.180 \cdot 10^0$ | $(4.159 \cdot 10^1)$ | $7.605 \cdot 10^0$ | $(3.021 \cdot 10^1)$ | $8.756 \cdot 10^{-1}$ | $(2.551 \cdot 10^1)$ |
| $4^{-7}$ | $5.476 \cdot 10^{-1}$ | $(2.987 \cdot 10^1)$ | $1.915 \cdot 10^0$ | $(1.084 \cdot 10^1)$ | $2.189 \cdot 10^{-1}$ | $(6.959 \cdot 10^0)$ |
| $4^{-8}$ | $1.371 \cdot 10^{-1}$ | $(6.403 \cdot 10^0)$ | $4.793 \cdot 10^{-1}$ | $(2.760 \cdot 10^0)$ | $5.473 \cdot 10^{-2}$ | $(1.754 \cdot 10^0)$ |
| $4^{-9}$ | $3.428 \cdot 10^{-2}$ | $(1.585 \cdot 10^0)$ | $1.198 \cdot 10^{-1}$ | $(6.903 \cdot 10^{-1})$ | $1.368 \cdot 10^{-2}$ | $(4.390 \cdot 10^{-1})$ |
| $4^{-10}$ | $8.570 \cdot 10^{-3}$ | $(3.956 \cdot 10^{-1})$ | $2.996 \cdot 10^{-2}$ | $(1.728 \cdot 10^{-1})$ | $3.421 \cdot 10^{-3}$ | $(1.098 \cdot 10^{-1})$ |
| $4^{-11}$ | $2.142 \cdot 10^{-3}$ | $(9.884 \cdot 10^{-2})$ | $7.491 \cdot 10^{-3}$ | $(4.321 \cdot 10^{-2})$ | $8.552 \cdot 10^{-4}$ | $(2.744 \cdot 10^{-2})$ |
| $4^{-12}$ | $5.356 \cdot 10^{-4}$ | $(2.471 \cdot 10^{-2})$ | $1.873 \cdot 10^{-3}$ | $(1.080 \cdot 10^{-2})$ | $2.138 \cdot 10^{-4}$ | $(6.860 \cdot 10^{-3})$ |
| $4^{-13}$ | $1.339 \cdot 10^{-4}$ | $(6.176 \cdot 10^{-3})$ | $4.682 \cdot 10^{-4}$ | $(2.701 \cdot 10^{-3})$ | $5.343 \cdot 10^{-5}$ | $(1.715 \cdot 10^{-3})$ |
| $4^{-14}$ | $3.348 \cdot 10^{-5}$ | $(1.544 \cdot 10^{-3})$ | $1.170 \cdot 10^{-4}$ | $(6.751 \cdot 10^{-4})$ | $1.334 \cdot 10^{-5}$ | $(4.285 \cdot 10^{-4})$ |
| $4^{-15}$ | $8.409 \cdot 10^{-6}$ | $(3.859 \cdot 10^{-4})$ | $2.925 \cdot 10^{-5}$ | $(1.687 \cdot 10^{-4})$ | $3.415 \cdot 10^{-6}$ | $(1.068 \cdot 10^{-4})$ |

| $\lambda$ | $d = 256$ | | $d = 1024$ | |
|---|---|---|---|---|
| $4^2$ | $1.030 \cdot 10^2$ | $(7.357 \cdot 10^{-1})$ | $9.654 \cdot 10^1$ | $(2.816 \cdot 10^{-1})$ |
| $4^1$ | $1.028 \cdot 10^2$ | $(6.489 \cdot 10^{-1})$ | $9.650 \cdot 10^1$ | $(4.394 \cdot 10^{-1})$ |
| $4^0$ | $1.037 \cdot 10^2$ | $(1.092 \cdot 10^0)$ | $8.695 \cdot 10^1$ | $(5.651 \cdot 10^0)$ |
| $4^{-1}$ | $9.990 \cdot 10^1$ | $(4.216 \cdot 10^0)$ | $2.997 \cdot 10^1$ | $(1.003 \cdot 10^1)$ |
| $4^{-2}$ | $9.841 \cdot 10^1$ | $(2.837 \cdot 10^1)$ | $7.662 \cdot 10^0$ | $(2.820 \cdot 10^0)$ |
| $4^{-3}$ | $9.404 \cdot 10^1$ | $(4.252 \cdot 10^1)$ | $1.918 \cdot 10^0$ | $(7.110 \cdot 10^{-1})$ |
| $4^{-4}$ | $5.304 \cdot 10^1$ | $(4.524 \cdot 10^1)$ | $4.796 \cdot 10^{-1}$ | $(1.779 \cdot 10^{-1})$ |
| $4^{-5}$ | $1.427 \cdot 10^1$ | $(3.982 \cdot 10^1)$ | $1.199 \cdot 10^{-1}$ | $(4.447 \cdot 10^{-2})$ |
| $4^{-6}$ | $3.583 \cdot 10^0$ | $(1.824 \cdot 10^1)$ | $2.998 \cdot 10^{-2}$ | $(1.112 \cdot 10^{-2})$ |
| $4^{-7}$ | $8.963 \cdot 10^{-1}$ | $(4.782 \cdot 10^0)$ | $7.494 \cdot 10^{-3}$ | $(2.779 \cdot 10^{-3})$ |
| $4^{-8}$ | $2.241 \cdot 10^{-1}$ | $(1.199 \cdot 10^0)$ | $1.874 \cdot 10^{-3}$ | $(6.948 \cdot 10^{-4})$ |
| $4^{-9}$ | $5.602 \cdot 10^{-2}$ | $(2.998 \cdot 10^{-1})$ | $4.684 \cdot 10^{-4}$ | $(1.737 \cdot 10^{-4})$ |
| $4^{-10}$ | $1.401 \cdot 10^{-2}$ | $(7.495 \cdot 10^{-2})$ | $1.171 \cdot 10^{-4}$ | $(4.342 \cdot 10^{-5})$ |
| $4^{-11}$ | $3.501 \cdot 10^{-3}$ | $(1.874 \cdot 10^{-2})$ | $2.930 \cdot 10^{-5}$ | $(1.086 \cdot 10^{-5})$ |
| $4^{-12}$ | $8.753 \cdot 10^{-4}$ | $(4.685 \cdot 10^{-3})$ | $7.636 \cdot 10^{-6}$ | $(2.675 \cdot 10^{-6})$ |
| $4^{-13}$ | $2.188 \cdot 10^{-4}$ | $(1.171 \cdot 10^{-3})$ | $3.912 \cdot 10^{-6}$ | $(4.332 \cdot 10^{-7})$ |
| $4^{-14}$ | $5.477 \cdot 10^{-5}$ | $(2.925 \cdot 10^{-4})$ | $4.183 \cdot 10^{-6}$ | $(1.183 \cdot 10^{-6})$ |
| $4^{-15}$ | $1.371 \cdot 10^{-5}$ | $(7.244 \cdot 10^{-5})$ | $4.005 \cdot 10^{-6}$ | $(1.095 \cdot 10^{-6})$ |

Table 2: The medians over five trials plotted in Figure 7 on the top right, with the standard deviations shown in parentheses, both rounded to four-digit mantissas.

| $\lambda$ | $d = 4$ | | $d = 16$ | | $d = 64$ | |
|---|---|---|---|---|---|---|
| $4^2$ | $1.728 \cdot 10^2$ | $(8.402 \cdot 10^0)$ | $1.372 \cdot 10^2$ | $(4.366 \cdot 10^0)$ | $1.154 \cdot 10^2$ | $(1.454 \cdot 10^0)$ |
| $4^1$ | $1.704 \cdot 10^2$ | $(5.072 \cdot 10^0)$ | $1.381 \cdot 10^2$ | $(3.195 \cdot 10^0)$ | $1.152 \cdot 10^2$ | $(1.564 \cdot 10^0)$ |
| $4^0$ | $1.641 \cdot 10^2$ | $(2.687 \cdot 10^1)$ | $1.324 \cdot 10^2$ | $(6.702 \cdot 10^0)$ | $1.101 \cdot 10^2$ | $(3.607 \cdot 10^0)$ |
| $4^{-1}$ | $9.916 \cdot 10^1$ | $(6.601 \cdot 10^1)$ | $1.287 \cdot 10^2$ | $(1.970 \cdot 10^1)$ | $1.056 \cdot 10^2$ | $(8.201 \cdot 10^0)$ |
| $4^{-2}$ | $2.210 \cdot 10^1$ | $(6.712 \cdot 10^1)$ | $9.046 \cdot 10^1$ | $(2.885 \cdot 10^1)$ | $1.028 \cdot 10^2$ | $(3.421 \cdot 10^1)$ |
| $4^{-3}$ | $5.421 \cdot 10^0$ | $(6.697 \cdot 10^1)$ | $3.727 \cdot 10^1$ | $(4.346 \cdot 10^1)$ | $7.944 \cdot 10^1$ | $(4.106 \cdot 10^1)$ |
| $4^{-4}$ | $1.350 \cdot 10^0$ | $(6.609 \cdot 10^1)$ | $9.250 \cdot 10^0$ | $(5.094 \cdot 10^1)$ | $2.129 \cdot 10^1$ | $(2.172 \cdot 10^1)$ |
| $4^{-5}$ | $3.370 \cdot 10^{-1}$ | $(6.163 \cdot 10^1)$ | $2.287 \cdot 10^0$ | $(3.956 \cdot 10^1)$ | $5.368 \cdot 10^0$ | $(6.186 \cdot 10^0)$ |
| $4^{-6}$ | $8.423 \cdot 10^{-2}$ | $(6.122 \cdot 10^1)$ | $5.699 \cdot 10^{-1}$ | $(3.996 \cdot 10^1)$ | $1.340 \cdot 10^0$ | $(1.562 \cdot 10^0)$ |
| $4^{-7}$ | $2.106 \cdot 10^{-2}$ | $(6.065 \cdot 10^1)$ | $1.423 \cdot 10^{-1}$ | $(4.068 \cdot 10^1)$ | $3.348 \cdot 10^{-1}$ | $(3.912 \cdot 10^{-1})$ |
| $4^{-8}$ | $5.264 \cdot 10^{-3}$ | $(5.959 \cdot 10^1)$ | $3.557 \cdot 10^{-2}$ | $(4.091 \cdot 10^1)$ | $8.370 \cdot 10^{-2}$ | $(9.783 \cdot 10^{-2})$ |
| $4^{-9}$ | $1.316 \cdot 10^{-3}$ | $(5.841 \cdot 10^1)$ | $8.893 \cdot 10^{-3}$ | $(4.100 \cdot 10^1)$ | $2.092 \cdot 10^{-2}$ | $(2.446 \cdot 10^{-2})$ |
| $4^{-10}$ | $3.290 \cdot 10^{-4}$ | $(5.722 \cdot 10^1)$ | $2.223 \cdot 10^{-3}$ | $(1.845 \cdot 10^1)$ | $5.231 \cdot 10^{-3}$ | $(6.115 \cdot 10^{-3})$ |
| $4^{-11}$ | $8.225 \cdot 10^{-5}$ | $(5.606 \cdot 10^1)$ | $5.558 \cdot 10^{-4}$ | $(5.136 \cdot 10^0)$ | $1.308 \cdot 10^{-3}$ | $(1.529 \cdot 10^{-3})$ |
| $4^{-12}$ | $2.056 \cdot 10^{-5}$ | $(5.495 \cdot 10^1)$ | $1.389 \cdot 10^{-4}$ | $(1.309 \cdot 10^0)$ | $3.269 \cdot 10^{-4}$ | $(3.822 \cdot 10^{-4})$ |
| $4^{-13}$ | $5.123 \cdot 10^{-6}$ | $(5.389 \cdot 10^1)$ | $3.473 \cdot 10^{-5}$ | $(3.285 \cdot 10^{-1})$ | $8.173 \cdot 10^{-5}$ | $(9.554 \cdot 10^{-5})$ |
| $4^{-14}$ | $1.708 \cdot 10^{-6}$ | $(5.288 \cdot 10^1)$ | $8.707 \cdot 10^{-6}$ | $(8.221 \cdot 10^{-2})$ | $2.042 \cdot 10^{-5}$ | $(2.359 \cdot 10^{-5})$ |
| $4^{-15}$ | $1.708 \cdot 10^{-6}$ | $(5.192 \cdot 10^1)$ | $2.259 \cdot 10^{-6}$ | $(2.056 \cdot 10^{-2})$ | $5.400 \cdot 10^{-6}$ | $(5.391 \cdot 10^{-6})$ |

| $\lambda$ | $d = 256$ | | $d = 1024$ | |
|---|---|---|---|---|
| $4^2$ | $1.025 \cdot 10^2$ | $(1.676 \cdot 10^0)$ | $9.693 \cdot 10^1$ | $(1.513 \cdot 10^{-1})$ |
| $4^1$ | $1.022 \cdot 10^2$ | $(1.122 \cdot 10^0)$ | $9.623 \cdot 10^1$ | $(4.889 \cdot 10^{-1})$ |
| $4^0$ | $1.004 \cdot 10^2$ | $(2.333 \cdot 10^0)$ | $9.507 \cdot 10^1$ | $(6.628 \cdot 10^{-1})$ |
| $4^{-1}$ | $9.800 \cdot 10^1$ | $(1.214 \cdot 10^1)$ | $9.253 \cdot 10^1$ | $(6.296 \cdot 10^0)$ |
| $4^{-2}$ | $9.213 \cdot 10^1$ | $(3.434 \cdot 10^1)$ | $9.143 \cdot 10^1$ | $(2.830 \cdot 10^1)$ |
| $4^{-3}$ | $7.404 \cdot 10^1$ | $(4.140 \cdot 10^1)$ | $9.113 \cdot 10^1$ | $(4.401 \cdot 10^1)$ |
| $4^{-4}$ | $2.102 \cdot 10^1$ | $(3.818 \cdot 10^1)$ | $9.031 \cdot 10^1$ | $(4.837 \cdot 10^1)$ |
| $4^{-5}$ | $5.305 \cdot 10^0$ | $(4.104 \cdot 10^1)$ | $4.268 \cdot 10^1$ | $(4.573 \cdot 10^1)$ |
| $4^{-6}$ | $1.327 \cdot 10^0$ | $(4.139 \cdot 10^1)$ | $1.146 \cdot 10^1$ | $(3.987 \cdot 10^1)$ |
| $4^{-7}$ | $3.316 \cdot 10^{-1}$ | $(4.075 \cdot 10^1)$ | $2.875 \cdot 10^0$ | $(4.045 \cdot 10^1)$ |
| $4^{-8}$ | $8.290 \cdot 10^{-2}$ | $(4.008 \cdot 10^1)$ | $7.186 \cdot 10^{-1}$ | $(4.141 \cdot 10^1)$ |
| $4^{-9}$ | $2.072 \cdot 10^{-2}$ | $(2.250 \cdot 10^1)$ | $1.796 \cdot 10^{-1}$ | $(4.170 \cdot 10^1)$ |
| $4^{-10}$ | $5.181 \cdot 10^{-3}$ | $(6.373 \cdot 10^0)$ | $4.491 \cdot 10^{-2}$ | $(4.178 \cdot 10^1)$ |
| $4^{-11}$ | $1.295 \cdot 10^{-3}$ | $(1.614 \cdot 10^0)$ | $1.123 \cdot 10^{-2}$ | $(4.180 \cdot 10^1)$ |
| $4^{-12}$ | $3.238 \cdot 10^{-4}$ | $(4.042 \cdot 10^{-1})$ | $2.807 \cdot 10^{-3}$ | $(4.172 \cdot 10^1)$ |
| $4^{-13}$ | $8.099 \cdot 10^{-5}$ | $(1.011 \cdot 10^{-1})$ | $7.016 \cdot 10^{-4}$ | $(2.239 \cdot 10^1)$ |
| $4^{-14}$ | $2.026 \cdot 10^{-5}$ | $(2.527 \cdot 10^{-2})$ | $1.754 \cdot 10^{-4}$ | $(6.256 \cdot 10^0)$ |
| $4^{-15}$ | $5.976 \cdot 10^{-6}$ | $(6.317 \cdot 10^{-3})$ | $4.385 \cdot 10^{-5}$ | $(1.581 \cdot 10^0)$ |

Table 3: The medians over five trials plotted in Figure 7 on the middle left, with the standard deviations shown in parentheses, both rounded to four-digit mantissas.

| $\lambda$ | $d = 4$ | | $d = 16$ | | $d = 64$ | |
|---|---|---|---|---|---|---|
| $4^2$ | $7.683 \cdot 10^1$ | $(9.817 \cdot 10^0)$ | $9.016 \cdot 10^1$ | $(5.402 \cdot 10^{-1})$ | $9.001 \cdot 10^1$ | $(1.067 \cdot 10^{-1})$ |
| $4^1$ | $8.311 \cdot 10^1$ | $(3.064 \cdot 10^0)$ | $8.978 \cdot 10^1$ | $(3.437 \cdot 10^{-1})$ | $8.954 \cdot 10^1$ | $(1.429 \cdot 10^{-1})$ |
| $4^0$ | $6.196 \cdot 10^1$ | $(8.026 \cdot 10^0)$ | $6.779 \cdot 10^1$ | $(3.166 \cdot 10^0)$ | $6.012 \cdot 10^1$ | $(1.436 \cdot 10^0)$ |
| $4^{-1}$ | $2.831 \cdot 10^1$ | $(8.054 \cdot 10^0)$ | $2.917 \cdot 10^1$ | $(2.550 \cdot 10^0)$ | $2.170 \cdot 10^1$ | $(2.631 \cdot 10^0)$ |
| $4^{-2}$ | $1.555 \cdot 10^1$ | $(5.922 \cdot 10^0)$ | $1.125 \cdot 10^1$ | $(1.727 \cdot 10^0)$ | $7.769 \cdot 10^0$ | $(2.285 \cdot 10^0)$ |
| $4^{-3}$ | $6.157 \cdot 10^0$ | $(4.770 \cdot 10^0)$ | $4.808 \cdot 10^0$ | $(1.412 \cdot 10^0)$ | $2.509 \cdot 10^0$ | $(1.719 \cdot 10^0)$ |
| $4^{-4}$ | $1.966 \cdot 10^0$ | $(3.649 \cdot 10^0)$ | $2.203 \cdot 10^0$ | $(1.188 \cdot 10^0)$ | $6.430 \cdot 10^{-1}$ | $(1.313 \cdot 10^0)$ |
| $4^{-5}$ | $5.171 \cdot 10^{-1}$ | $(3.181 \cdot 10^0)$ | $7.332 \cdot 10^{-1}$ | $(1.057 \cdot 10^0)$ | $1.612 \cdot 10^{-1}$ | $(1.013 \cdot 10^0)$ |
| $4^{-6}$ | $1.308 \cdot 10^{-1}$ | $(2.053 \cdot 10^0)$ | $1.886 \cdot 10^{-1}$ | $(6.984 \cdot 10^{-1})$ | $4.031 \cdot 10^{-2}$ | $(5.756 \cdot 10^{-1})$ |
| $4^{-7}$ | $3.278 \cdot 10^{-2}$ | $(1.152 \cdot 10^0)$ | $4.741 \cdot 10^{-2}$ | $(2.505 \cdot 10^{-1})$ | $1.008 \cdot 10^{-2}$ | $(1.566 \cdot 10^{-1})$ |
| $4^{-8}$ | $8.199 \cdot 10^{-3}$ | $(2.638 \cdot 10^{-1})$ | $1.186 \cdot 10^{-2}$ | $(6.378 \cdot 10^{-2})$ | $2.520 \cdot 10^{-3}$ | $(3.948 \cdot 10^{-2})$ |
| $4^{-9}$ | $2.050 \cdot 10^{-3}$ | $(6.571 \cdot 10^{-2})$ | $2.967 \cdot 10^{-3}$ | $(1.595 \cdot 10^{-2})$ | $6.299 \cdot 10^{-4}$ | $(9.880 \cdot 10^{-3})$ |
| $4^{-10}$ | $5.126 \cdot 10^{-4}$ | $(1.641 \cdot 10^{-2})$ | $7.417 \cdot 10^{-4}$ | $(3.992 \cdot 10^{-3})$ | $1.575 \cdot 10^{-4}$ | $(2.470 \cdot 10^{-3})$ |
| $4^{-11}$ | $1.282 \cdot 10^{-4}$ | $(4.101 \cdot 10^{-3})$ | $1.854 \cdot 10^{-4}$ | $(9.984 \cdot 10^{-4})$ | $3.938 \cdot 10^{-5}$ | $(6.175 \cdot 10^{-4})$ |
| $4^{-12}$ | $3.204 \cdot 10^{-5}$ | $(1.025 \cdot 10^{-3})$ | $4.635 \cdot 10^{-5}$ | $(2.496 \cdot 10^{-4})$ | $9.849 \cdot 10^{-6}$ | $(1.544 \cdot 10^{-4})$ |
| $4^{-13}$ | $7.973 \cdot 10^{-6}$ | $(2.562 \cdot 10^{-4})$ | $1.161 \cdot 10^{-5}$ | $(6.245 \cdot 10^{-5})$ | $2.332 \cdot 10^{-6}$ | $(3.861 \cdot 10^{-5})$ |
| $4^{-14}$ | $2.029 \cdot 10^{-6}$ | $(6.408 \cdot 10^{-5})$ | $3.056 \cdot 10^{-6}$ | $(1.562 \cdot 10^{-5})$ | $9.651 \cdot 10^{-7}$ | $(9.611 \cdot 10^{-6})$ |
| $4^{-15}$ | $5.905 \cdot 10^{-7}$ | $(1.602 \cdot 10^{-5})$ | $9.897 \cdot 10^{-7}$ | $(3.929 \cdot 10^{-6})$ | $5.896 \cdot 10^{-7}$ | $(2.404 \cdot 10^{-6})$ |

| $\lambda$ | $d = 256$ | | $d = 1024$ | |
|---|---|---|---|---|
| $4^2$ | $9.011 \cdot 10^1$ | $(3.716 \cdot 10^{-2})$ | $9.013 \cdot 10^1$ | $(1.563 \cdot 10^{-2})$ |
| $4^1$ | $8.965 \cdot 10^1$ | $(7.506 \cdot 10^{-2})$ | $8.960 \cdot 10^1$ | $(9.857 \cdot 10^{-2})$ |
| $4^0$ | $5.760 \cdot 10^1$ | $(1.833 \cdot 10^0)$ | $5.427 \cdot 10^1$ | $(1.127 \cdot 10^0)$ |
| $4^{-1}$ | $1.952 \cdot 10^1$ | $(2.337 \cdot 10^0)$ | $1.494 \cdot 10^1$ | $(4.246 \cdot 10^{-1})$ |
| $4^{-2}$ | $7.718 \cdot 10^0$ | $(2.352 \cdot 10^0)$ | $3.758 \cdot 10^0$ | $(1.110 \cdot 10^{-1})$ |
| $4^{-3}$ | $3.520 \cdot 10^0$ | $(2.141 \cdot 10^0)$ | $9.400 \cdot 10^{-1}$ | $(2.784 \cdot 10^{-2})$ |
| $4^{-4}$ | $1.450 \cdot 10^0$ | $(1.730 \cdot 10^0)$ | $2.350 \cdot 10^{-1}$ | $(6.962 \cdot 10^{-3})$ |
| $4^{-5}$ | $3.816 \cdot 10^{-1}$ | $(1.407 \cdot 10^0)$ | $5.875 \cdot 10^{-2}$ | $(1.740 \cdot 10^{-3})$ |
| $4^{-6}$ | $9.573 \cdot 10^{-2}$ | $(4.887 \cdot 10^{-1})$ | $1.469 \cdot 10^{-2}$ | $(4.351 \cdot 10^{-4})$ |
| $4^{-7}$ | $2.394 \cdot 10^{-2}$ | $(1.254 \cdot 10^{-1})$ | $3.672 \cdot 10^{-3}$ | $(1.088 \cdot 10^{-4})$ |
| $4^{-8}$ | $5.985 \cdot 10^{-3}$ | $(3.140 \cdot 10^{-2})$ | $9.180 \cdot 10^{-4}$ | $(2.719 \cdot 10^{-5})$ |
| $4^{-9}$ | $1.496 \cdot 10^{-3}$ | $(7.851 \cdot 10^{-3})$ | $2.295 \cdot 10^{-4}$ | $(6.797 \cdot 10^{-6})$ |
| $4^{-10}$ | $3.741 \cdot 10^{-4}$ | $(1.963 \cdot 10^{-3})$ | $5.739 \cdot 10^{-5}$ | $(1.696 \cdot 10^{-6})$ |
| $4^{-11}$ | $9.352 \cdot 10^{-5}$ | $(4.907 \cdot 10^{-4})$ | $1.434 \cdot 10^{-5}$ | $(4.200 \cdot 10^{-7})$ |
| $4^{-12}$ | $2.343 \cdot 10^{-5}$ | $(1.227 \cdot 10^{-4})$ | $3.686 \cdot 10^{-6}$ | $(1.467 \cdot 10^{-7})$ |
| $4^{-13}$ | $5.816 \cdot 10^{-6}$ | $(3.058 \cdot 10^{-5})$ | $9.589 \cdot 10^{-7}$ | $(9.762 \cdot 10^{-8})$ |
| $4^{-14}$ | $1.857 \cdot 10^{-6}$ | $(7.695 \cdot 10^{-6})$ | $9.701 \cdot 10^{-7}$ | $(2.770 \cdot 10^{-7})$ |
| $4^{-15}$ | $9.740 \cdot 10^{-7}$ | $(1.898 \cdot 10^{-6})$ | $7.552 \cdot 10^{-7}$ | $(2.631 \cdot 10^{-7})$ |

Table 4: The medians over five trials plotted in Figure 7 on the middle right, with the standard deviations shown in parentheses, both rounded to four-digit mantissas.

| $\lambda$ | $d = 4$ | | $d = 16$ | | $d = 64$ | |
|---|---|---|---|---|---|---|
| $4^2$ | $8.693 \cdot 10^1$ | $(6.307 \cdot 10^0)$ | $9.007 \cdot 10^1$ | $(1.912 \cdot 10^{-1})$ | $9.020 \cdot 10^1$ | $(1.051 \cdot 10^{-1})$ |
| $4^1$ | $8.332 \cdot 10^1$ | $(3.578 \cdot 10^0)$ | $8.919 \cdot 10^1$ | $(3.286 \cdot 10^{-1})$ | $8.965 \cdot 10^1$ | $(1.669 \cdot 10^{-1})$ |
| $4^0$ | $5.427 \cdot 10^1$ | $(1.395 \cdot 10^1)$ | $6.270 \cdot 10^1$ | $(2.967 \cdot 10^0)$ | $5.755 \cdot 10^1$ | $(2.259 \cdot 10^0)$ |
| $4^{-1}$ | $1.690 \cdot 10^1$ | $(1.879 \cdot 10^1)$ | $2.528 \cdot 10^1$ | $(3.054 \cdot 10^0)$ | $1.834 \cdot 10^1$ | $(2.899 \cdot 10^0)$ |
| $4^{-2}$ | $4.138 \cdot 10^0$ | $(1.715 \cdot 10^1)$ | $8.632 \cdot 10^0$ | $(1.621 \cdot 10^0)$ | $6.326 \cdot 10^0$ | $(2.343 \cdot 10^0)$ |
| $4^{-3}$ | $1.041 \cdot 10^0$ | $(1.471 \cdot 10^1)$ | $2.621 \cdot 10^0$ | $(1.206 \cdot 10^0)$ | $2.697 \cdot 10^0$ | $(1.608 \cdot 10^0)$ |
| $4^{-4}$ | $2.606 \cdot 10^{-1}$ | $(1.387 \cdot 10^1)$ | $6.547 \cdot 10^{-1}$ | $(1.258 \cdot 10^0)$ | $7.072 \cdot 10^{-1}$ | $(6.715 \cdot 10^{-1})$ |
| $4^{-5}$ | $6.517 \cdot 10^{-2}$ | $(1.357 \cdot 10^1)$ | $1.632 \cdot 10^{-1}$ | $(8.705 \cdot 10^{-1})$ | $1.778 \cdot 10^{-1}$ | $(1.836 \cdot 10^{-1})$ |
| $4^{-6}$ | $1.630 \cdot 10^{-2}$ | $(1.337 \cdot 10^1)$ | $4.075 \cdot 10^{-2}$ | $(8.647 \cdot 10^{-1})$ | $4.439 \cdot 10^{-2}$ | $(4.626 \cdot 10^{-2})$ |
| $4^{-7}$ | $4.074 \cdot 10^{-3}$ | $(1.314 \cdot 10^1)$ | $1.018 \cdot 10^{-2}$ | $(8.829 \cdot 10^{-1})$ | $1.110 \cdot 10^{-2}$ | $(1.158 \cdot 10^{-2})$ |
| $4^{-8}$ | $1.018 \cdot 10^{-3}$ | $(1.291 \cdot 10^1)$ | $2.546 \cdot 10^{-3}$ | $(8.890 \cdot 10^{-1})$ | $2.774 \cdot 10^{-3}$ | $(2.897 \cdot 10^{-3})$ |
| $4^{-9}$ | $2.546 \cdot 10^{-4}$ | $(1.191 \cdot 10^1)$ | $6.364 \cdot 10^{-4}$ | $(8.913 \cdot 10^{-1})$ | $6.935 \cdot 10^{-4}$ | $(7.243 \cdot 10^{-4})$ |
| $4^{-10}$ | $6.366 \cdot 10^{-5}$ | $(1.093 \cdot 10^1)$ | $1.591 \cdot 10^{-4}$ | $(4.011 \cdot 10^{-1})$ | $1.734 \cdot 10^{-4}$ | $(1.811 \cdot 10^{-4})$ |
| $4^{-11}$ | $1.592 \cdot 10^{-5}$ | $(9.224 \cdot 10^0)$ | $3.978 \cdot 10^{-5}$ | $(1.116 \cdot 10^{-1})$ | $4.334 \cdot 10^{-5}$ | $(4.527 \cdot 10^{-5})$ |
| $4^{-12}$ | $3.972 \cdot 10^{-6}$ | $(8.167 \cdot 10^0)$ | $9.970 \cdot 10^{-6}$ | $(2.845 \cdot 10^{-2})$ | $1.083 \cdot 10^{-5}$ | $(1.132 \cdot 10^{-5})$ |
| $4^{-13}$ | $1.002 \cdot 10^{-6}$ | $(7.210 \cdot 10^0)$ | $2.308 \cdot 10^{-6}$ | $(7.141 \cdot 10^{-3})$ | $2.737 \cdot 10^{-6}$ | $(2.805 \cdot 10^{-6})$ |
| $4^{-14}$ | $3.440 \cdot 10^{-7}$ | $(6.689 \cdot 10^0)$ | $7.877 \cdot 10^{-7}$ | $(1.787 \cdot 10^{-3})$ | $9.234 \cdot 10^{-7}$ | $(7.185 \cdot 10^{-7})$ |
| $4^{-15}$ | $3.018 \cdot 10^{-7}$ | $(5.947 \cdot 10^0)$ | $4.524 \cdot 10^{-7}$ | $(4.468 \cdot 10^{-4})$ | $6.012 \cdot 10^{-7}$ | $(1.589 \cdot 10^{-7})$ |

| $\lambda$ | $d = 256$ | | $d = 1024$ | |
|---|---|---|---|---|
| $4^2$ | $9.008 \cdot 10^1$ | $(5.085 \cdot 10^{-2})$ | $9.012 \cdot 10^1$ | $(2.708 \cdot 10^{-2})$ |
| $4^1$ | $8.965 \cdot 10^1$ | $(9.344 \cdot 10^{-2})$ | $8.973 \cdot 10^1$ | $(8.384 \cdot 10^{-2})$ |
| $4^0$ | $5.698 \cdot 10^1$ | $(1.686 \cdot 10^0)$ | $5.645 \cdot 10^1$ | $(1.837 \cdot 10^0)$ |
| $4^{-1}$ | $1.935 \cdot 10^1$ | $(3.007 \cdot 10^0)$ | $1.855 \cdot 10^1$ | $(2.099 \cdot 10^0)$ |
| $4^{-2}$ | $6.903 \cdot 10^0$ | $(2.854 \cdot 10^0)$ | $6.111 \cdot 10^0$ | $(2.381 \cdot 10^0)$ |
| $4^{-3}$ | $2.820 \cdot 10^0$ | $(1.859 \cdot 10^0)$ | $2.766 \cdot 10^0$ | $(2.318 \cdot 10^0)$ |
| $4^{-4}$ | $7.530 \cdot 10^{-1}$ | $(1.521 \cdot 10^0)$ | $1.922 \cdot 10^0$ | $(2.149 \cdot 10^0)$ |
| $4^{-5}$ | $1.893 \cdot 10^{-1}$ | $(1.465 \cdot 10^0)$ | $1.066 \cdot 10^0$ | $(1.672 \cdot 10^0)$ |
| $4^{-6}$ | $4.733 \cdot 10^{-2}$ | $(1.442 \cdot 10^0)$ | $2.823 \cdot 10^{-1}$ | $(1.218 \cdot 10^0)$ |
| $4^{-7}$ | $1.183 \cdot 10^{-2}$ | $(1.117 \cdot 10^0)$ | $7.077 \cdot 10^{-2}$ | $(9.110 \cdot 10^{-1})$ |
| $4^{-8}$ | $2.958 \cdot 10^{-3}$ | $(8.501 \cdot 10^{-1})$ | $1.769 \cdot 10^{-2}$ | $(8.302 \cdot 10^{-1})$ |
| $4^{-9}$ | $7.394 \cdot 10^{-4}$ | $(4.485 \cdot 10^{-1})$ | $4.422 \cdot 10^{-3}$ | $(8.104 \cdot 10^{-1})$ |
| $4^{-10}$ | $1.849 \cdot 10^{-4}$ | $(1.261 \cdot 10^{-1})$ | $1.106 \cdot 10^{-3}$ | $(8.055 \cdot 10^{-1})$ |
| $4^{-11}$ | $4.622 \cdot 10^{-5}$ | $(3.192 \cdot 10^{-2})$ | $2.764 \cdot 10^{-4}$ | $(8.043 \cdot 10^{-1})$ |
| $4^{-12}$ | $1.156 \cdot 10^{-5}$ | $(7.995 \cdot 10^{-3})$ | $6.910 \cdot 10^{-5}$ | $(8.025 \cdot 10^{-1})$ |
| $4^{-13}$ | $2.938 \cdot 10^{-6}$ | $(1.999 \cdot 10^{-3})$ | $1.718 \cdot 10^{-5}$ | $(4.307 \cdot 10^{-1})$ |
| $4^{-14}$ | $1.209 \cdot 10^{-6}$ | $(4.999 \cdot 10^{-4})$ | $4.466 \cdot 10^{-6}$ | $(1.203 \cdot 10^{-1})$ |
| $4^{-15}$ | $8.651 \cdot 10^{-7}$ | $(1.250 \cdot 10^{-4})$ | $1.339 \cdot 10^{-6}$ | $(3.040 \cdot 10^{-2})$ |

Table 5: The medians over five trials plotted in Figure 7 on the bottom left, with the standard deviations shown in parentheses, both rounded to four-digit mantissas.

| $\lambda$ | $d = 4$ | | $d = 16$ | | $d = 64$ | |
|---|---|---|---|---|---|---|
| $4^2$ | $3.288 \cdot 10^1$ | $(1.412 \cdot 10^0)$ | $6.314 \cdot 10^1$ | $(5.680 \cdot 10^{-1})$ | $1.216 \cdot 10^2$ | $(1.296 \cdot 10^0)$ |
| $4^1$ | $7.962 \cdot 10^0$ | $(2.012 \cdot 10^{-1})$ | $1.588 \cdot 10^1$ | $(1.277 \cdot 10^{-1})$ | $3.056 \cdot 10^1$ | $(3.182 \cdot 10^{-1})$ |
| $4^0$ | $2.512 \cdot 10^0$ | $(7.491 \cdot 10^{-2})$ | $4.628 \cdot 10^0$ | $(4.922 \cdot 10^{-2})$ | $8.342 \cdot 10^0$ | $(7.887 \cdot 10^{-2})$ |
| $4^{-1}$ | $1.380 \cdot 10^0$ | $(2.005 \cdot 10^{-2})$ | $1.929 \cdot 10^0$ | $(1.723 \cdot 10^{-2})$ | $2.849 \cdot 10^0$ | $(2.055 \cdot 10^{-2})$ |
| $4^{-2}$ | $1.094 \cdot 10^0$ | $(7.294 \cdot 10^{-3})$ | $1.231 \cdot 10^0$ | $(7.694 \cdot 10^{-3})$ | $1.463 \cdot 10^0$ | $(5.370 \cdot 10^{-3})$ |
| $4^{-3}$ | $1.023 \cdot 10^0$ | $(2.200 \cdot 10^{-3})$ | $1.057 \cdot 10^0$ | $(1.075 \cdot 10^{-3})$ | $1.116 \cdot 10^0$ | $(1.341 \cdot 10^{-3})$ |
| $4^{-4}$ | $1.006 \cdot 10^0$ | $(9.145 \cdot 10^{-4})$ | $1.014 \cdot 10^0$ | $(5.712 \cdot 10^{-4})$ | $1.029 \cdot 10^0$ | $(3.568 \cdot 10^{-4})$ |
| $4^{-5}$ | $1.001 \cdot 10^0$ | $(6.534 \cdot 10^{-4})$ | $1.003 \cdot 10^0$ | $(5.127 \cdot 10^{-4})$ | $1.007 \cdot 10^0$ | $(1.379 \cdot 10^{-4})$ |
| $4^{-6}$ | $1.000 \cdot 10^0$ | $(6.009 \cdot 10^{-4})$ | $1.001 \cdot 10^0$ | $(5.074 \cdot 10^{-4})$ | $1.002 \cdot 10^0$ | $(9.865 \cdot 10^{-5})$ |
| $4^{-7}$ | $1.000 \cdot 10^0$ | $(5.887 \cdot 10^{-4})$ | $1.000 \cdot 10^0$ | $(5.070 \cdot 10^{-4})$ | $1.000 \cdot 10^0$ | $(9.271 \cdot 10^{-5})$ |
| $4^{-8}$ | $1.000 \cdot 10^0$ | $(5.857 \cdot 10^{-4})$ | $1.000 \cdot 10^0$ | $(5.070 \cdot 10^{-4})$ | $9.999 \cdot 10^{-1}$ | $(9.154 \cdot 10^{-5})$ |
| $4^{-9}$ | $1.000 \cdot 10^0$ | $(5.850 \cdot 10^{-4})$ | $9.999 \cdot 10^{-1}$ | $(5.070 \cdot 10^{-4})$ | $9.998 \cdot 10^{-1}$ | $(9.127 \cdot 10^{-5})$ |
| $4^{-10}$ | $1.000 \cdot 10^0$ | $(5.848 \cdot 10^{-4})$ | $9.999 \cdot 10^{-1}$ | $(5.070 \cdot 10^{-4})$ | $9.998 \cdot 10^{-1}$ | $(9.121 \cdot 10^{-5})$ |
| $4^{-11}$ | $1.000 \cdot 10^0$ | $(5.848 \cdot 10^{-4})$ | $9.999 \cdot 10^{-1}$ | $(5.070 \cdot 10^{-4})$ | $9.998 \cdot 10^{-1}$ | $(9.119 \cdot 10^{-5})$ |
| $4^{-12}$ | $1.000 \cdot 10^0$ | $(5.848 \cdot 10^{-4})$ | $9.999 \cdot 10^{-1}$ | $(5.070 \cdot 10^{-4})$ | $9.998 \cdot 10^{-1}$ | $(9.118 \cdot 10^{-5})$ |
| $4^{-13}$ | $1.000 \cdot 10^0$ | $(5.848 \cdot 10^{-4})$ | $9.999 \cdot 10^{-1}$ | $(5.070 \cdot 10^{-4})$ | $9.998 \cdot 10^{-1}$ | $(9.118 \cdot 10^{-5})$ |
| $4^{-14}$ | $1.000 \cdot 10^0$ | $(5.848 \cdot 10^{-4})$ | $9.999 \cdot 10^{-1}$ | $(5.070 \cdot 10^{-4})$ | $9.998 \cdot 10^{-1}$ | $(9.118 \cdot 10^{-5})$ |
| $4^{-15}$ | $1.000 \cdot 10^0$ | $(5.848 \cdot 10^{-4})$ | $9.999 \cdot 10^{-1}$ | $(5.070 \cdot 10^{-4})$ | $9.998 \cdot 10^{-1}$ | $(9.118 \cdot 10^{-5})$ |

| $\lambda$ | $d = 256$ | | $d = 1024$ | |
|---|---|---|---|---|
| $4^2$ | $2.007 \cdot 10^2$ | $(7.876 \cdot 10^{-1})$ | $2.205 \cdot 10^2$ | $(2.389 \cdot 10^{-1})$ |
| $4^1$ | $5.038 \cdot 10^1$ | $(2.024 \cdot 10^{-1})$ | $5.534 \cdot 10^1$ | $(5.103 \cdot 10^{-2})$ |
| $4^0$ | $1.334 \cdot 10^1$ | $(5.002 \cdot 10^{-2})$ | $1.461 \cdot 10^1$ | $(1.094 \cdot 10^{-2})$ |
| $4^{-1}$ | $4.098 \cdot 10^0$ | $(1.253 \cdot 10^{-2})$ | $4.421 \cdot 10^0$ | $(4.102 \cdot 10^{-3})$ |
| $4^{-2}$ | $1.776 \cdot 10^0$ | $(3.138 \cdot 10^{-3})$ | $1.856 \cdot 10^0$ | $(1.132 \cdot 10^{-3})$ |
| $4^{-3}$ | $1.194 \cdot 10^0$ | $(7.825 \cdot 10^{-4})$ | $1.214 \cdot 10^0$ | $(2.891 \cdot 10^{-4})$ |
| $4^{-4}$ | $1.048 \cdot 10^0$ | $(1.923 \cdot 10^{-4})$ | $1.054 \cdot 10^0$ | $(7.164 \cdot 10^{-5})$ |
| $4^{-5}$ | $1.012 \cdot 10^0$ | $(4.655 \cdot 10^{-5})$ | $1.013 \cdot 10^0$ | $(1.663 \cdot 10^{-5})$ |
| $4^{-6}$ | $1.003 \cdot 10^0$ | $(1.634 \cdot 10^{-5})$ | $1.003 \cdot 10^0$ | $(2.939 \cdot 10^{-6})$ |
| $4^{-7}$ | $1.001 \cdot 10^0$ | $(1.497 \cdot 10^{-5})$ | $1.001 \cdot 10^0$ | $(1.031 \cdot 10^{-6})$ |
| $4^{-8}$ | $1.000 \cdot 10^0$ | $(1.550 \cdot 10^{-5})$ | $1.000 \cdot 10^0$ | $(1.699 \cdot 10^{-6})$ |
| $4^{-9}$ | $1.000 \cdot 10^0$ | $(1.568 \cdot 10^{-5})$ | $1.000 \cdot 10^0$ | $(1.891 \cdot 10^{-6})$ |
| $4^{-10}$ | $9.999 \cdot 10^{-1}$ | $(1.573 \cdot 10^{-5})$ | $1.000 \cdot 10^0$ | $(1.940 \cdot 10^{-6})$ |
| $4^{-11}$ | $9.999 \cdot 10^{-1}$ | $(1.574 \cdot 10^{-5})$ | $1.000 \cdot 10^0$ | $(1.952 \cdot 10^{-6})$ |
| $4^{-12}$ | $9.999 \cdot 10^{-1}$ | $(1.574 \cdot 10^{-5})$ | $1.000 \cdot 10^0$ | $(1.955 \cdot 10^{-6})$ |
| $4^{-13}$ | $9.999 \cdot 10^{-1}$ | $(1.574 \cdot 10^{-5})$ | $1.000 \cdot 10^0$ | $(1.956 \cdot 10^{-6})$ |
| $4^{-14}$ | $9.999 \cdot 10^{-1}$ | $(1.574 \cdot 10^{-5})$ | $1.000 \cdot 10^0$ | $(1.956 \cdot 10^{-6})$ |
| $4^{-15}$ | $9.999 \cdot 10^{-1}$ | $(1.574 \cdot 10^{-5})$ | $1.000 \cdot 10^0$ | $(1.956 \cdot 10^{-6})$ |

Table 6: The medians over five trials plotted in Figure 7 on the bottom right, with the standard deviations shown in parentheses, both rounded to four-digit mantissas.

| $\lambda$ | $d = 4$ | | $d = 16$ | | $d = 64$ | |
|---|---|---|---|---|---|---|
| $4^2$ | $3.215 \cdot 10^1$ | $(1.355 \cdot 10^0)$ | $6.310 \cdot 10^1$ | $(5.531 \cdot 10^{-1})$ | $1.228 \cdot 10^2$ | $(7.578 \cdot 10^{-1})$ |
| $4^1$ | $7.710 \cdot 10^0$ | $(2.875 \cdot 10^{-1})$ | $1.585 \cdot 10^1$ | $(1.381 \cdot 10^{-1})$ | $3.081 \cdot 10^1$ | $(1.894 \cdot 10^{-1})$ |
| $4^0$ | $2.495 \cdot 10^0$ | $(6.191 \cdot 10^{-2})$ | $4.659 \cdot 10^0$ | $(5.141 \cdot 10^{-2})$ | $8.493 \cdot 10^0$ | $(4.228 \cdot 10^{-2})$ |
| $4^{-1}$ | $1.375 \cdot 10^0$ | $(2.960 \cdot 10^{-2})$ | $1.941 \cdot 10^0$ | $(2.509 \cdot 10^{-2})$ | $2.907 \cdot 10^0$ | $(3.309 \cdot 10^{-2})$ |
| $4^{-2}$ | $1.093 \cdot 10^0$ | $(4.173 \cdot 10^{-3})$ | $1.237 \cdot 10^0$ | $(8.799 \cdot 10^{-3})$ | $1.484 \cdot 10^0$ | $(5.647 \cdot 10^{-3})$ |
| $4^{-3}$ | $1.023 \cdot 10^0$ | $(1.006 \cdot 10^{-3})$ | $1.058 \cdot 10^0$ | $(1.431 \cdot 10^{-3})$ | $1.118 \cdot 10^0$ | $(4.962 \cdot 10^{-3})$ |
| $4^{-4}$ | $1.005 \cdot 10^0$ | $(5.226 \cdot 10^{-4})$ | $1.014 \cdot 10^0$ | $(8.382 \cdot 10^{-4})$ | $1.028 \cdot 10^0$ | $(5.080 \cdot 10^{-3})$ |
| $4^{-5}$ | $1.001 \cdot 10^0$ | $(4.275 \cdot 10^{-4})$ | $1.003 \cdot 10^0$ | $(7.240 \cdot 10^{-4})$ | $1.006 \cdot 10^0$ | $(5.213 \cdot 10^{-3})$ |
| $4^{-6}$ | $1.000 \cdot 10^0$ | $(4.136 \cdot 10^{-4})$ | $1.001 \cdot 10^0$ | $(7.028 \cdot 10^{-4})$ | $1.001 \cdot 10^0$ | $(5.204 \cdot 10^{-3})$ |
| $4^{-7}$ | $9.999 \cdot 10^{-1}$ | $(4.106 \cdot 10^{-4})$ | $9.999 \cdot 10^{-1}$ | $(6.978 \cdot 10^{-4})$ | $9.991 \cdot 10^{-1}$ | $(5.201 \cdot 10^{-3})$ |
| $4^{-8}$ | $9.999 \cdot 10^{-1}$ | $(4.099 \cdot 10^{-4})$ | $9.998 \cdot 10^{-1}$ | $(6.965 \cdot 10^{-4})$ | $9.988 \cdot 10^{-1}$ | $(5.201 \cdot 10^{-3})$ |
| $4^{-9}$ | $9.999 \cdot 10^{-1}$ | $(4.097 \cdot 10^{-4})$ | $9.997 \cdot 10^{-1}$ | $(6.962 \cdot 10^{-4})$ | $9.987 \cdot 10^{-1}$ | $(5.201 \cdot 10^{-3})$ |
| $4^{-10}$ | $9.999 \cdot 10^{-1}$ | $(4.097 \cdot 10^{-4})$ | $9.997 \cdot 10^{-1}$ | $(6.961 \cdot 10^{-4})$ | $9.987 \cdot 10^{-1}$ | $(5.201 \cdot 10^{-3})$ |
| $4^{-11}$ | $9.999 \cdot 10^{-1}$ | $(4.097 \cdot 10^{-4})$ | $9.997 \cdot 10^{-1}$ | $(6.961 \cdot 10^{-4})$ | $9.987 \cdot 10^{-1}$ | $(5.201 \cdot 10^{-3})$ |
| $4^{-12}$ | $9.999 \cdot 10^{-1}$ | $(4.096 \cdot 10^{-4})$ | $9.997 \cdot 10^{-1}$ | $(6.961 \cdot 10^{-4})$ | $9.987 \cdot 10^{-1}$ | $(5.201 \cdot 10^{-3})$ |
| $4^{-13}$ | $9.999 \cdot 10^{-1}$ | $(4.096 \cdot 10^{-4})$ | $9.997 \cdot 10^{-1}$ | $(6.961 \cdot 10^{-4})$ | $9.987 \cdot 10^{-1}$ | $(5.201 \cdot 10^{-3})$ |
| $4^{-14}$ | $9.999 \cdot 10^{-1}$ | $(4.096 \cdot 10^{-4})$ | $9.997 \cdot 10^{-1}$ | $(6.961 \cdot 10^{-4})$ | $9.987 \cdot 10^{-1}$ | $(5.201 \cdot 10^{-3})$ |
| $4^{-15}$ | $9.999 \cdot 10^{-1}$ | $(4.096 \cdot 10^{-4})$ | $9.997 \cdot 10^{-1}$ | $(6.961 \cdot 10^{-4})$ | $9.987 \cdot 10^{-1}$ | $(5.201 \cdot 10^{-3})$ |

| $\lambda$ | $d = 256$ | | $d = 1024$ | |
|---|---|---|---|---|
| $4^2$ | $1.999 \cdot 10^2$ | $(9.674 \cdot 10^{-1})$ | $2.203 \cdot 10^2$ | $(2.081 \cdot 10^{-1})$ |
| $4^1$ | $5.019 \cdot 10^1$ | $(2.479 \cdot 10^{-1})$ | $5.530 \cdot 10^1$ | $(5.434 \cdot 10^{-2})$ |
| $4^0$ | $1.340 \cdot 10^1$ | $(7.287 \cdot 10^{-2})$ | $1.469 \cdot 10^1$ | $(2.460 \cdot 10^{-2})$ |
| $4^{-1}$ | $4.185 \cdot 10^0$ | $(3.805 \cdot 10^{-2})$ | $4.507 \cdot 10^0$ | $(2.189 \cdot 10^{-2})$ |
| $4^{-2}$ | $1.800 \cdot 10^0$ | $(1.273 \cdot 10^{-2})$ | $1.872 \cdot 10^0$ | $(5.661 \cdot 10^{-3})$ |
| $4^{-3}$ | $1.195 \cdot 10^0$ | $(2.320 \cdot 10^{-3})$ | $1.213 \cdot 10^0$ | $(5.792 \cdot 10^{-4})$ |
| $4^{-4}$ | $1.047 \cdot 10^0$ | $(8.068 \cdot 10^{-4})$ | $1.049 \cdot 10^0$ | $(7.904 \cdot 10^{-4})$ |
| $4^{-5}$ | $1.010 \cdot 10^0$ | $(7.644 \cdot 10^{-4})$ | $1.009 \cdot 10^0$ | $(9.884 \cdot 10^{-4})$ |
| $4^{-6}$ | $1.000 \cdot 10^0$ | $(7.698 \cdot 10^{-4})$ | $9.984 \cdot 10^{-1}$ | $(1.029 \cdot 10^{-3})$ |
| $4^{-7}$ | $9.981 \cdot 10^{-1}$ | $(7.690 \cdot 10^{-4})$ | $9.959 \cdot 10^{-1}$ | $(1.036 \cdot 10^{-3})$ |
| $4^{-8}$ | $9.975 \cdot 10^{-1}$ | $(7.696 \cdot 10^{-4})$ | $9.952 \cdot 10^{-1}$ | $(1.037 \cdot 10^{-3})$ |
| $4^{-9}$ | $9.974 \cdot 10^{-1}$ | $(7.696 \cdot 10^{-4})$ | $9.951 \cdot 10^{-1}$ | $(1.037 \cdot 10^{-3})$ |
| $4^{-10}$ | $9.973 \cdot 10^{-1}$ | $(7.696 \cdot 10^{-4})$ | $9.950 \cdot 10^{-1}$ | $(1.037 \cdot 10^{-3})$ |
| $4^{-11}$ | $9.973 \cdot 10^{-1}$ | $(7.696 \cdot 10^{-4})$ | $9.950 \cdot 10^{-1}$ | $(1.037 \cdot 10^{-3})$ |
| $4^{-12}$ | $9.973 \cdot 10^{-1}$ | $(7.696 \cdot 10^{-4})$ | $9.950 \cdot 10^{-1}$ | $(1.037 \cdot 10^{-3})$ |
| $4^{-13}$ | $9.973 \cdot 10^{-1}$ | $(7.696 \cdot 10^{-4})$ | $9.950 \cdot 10^{-1}$ | $(1.037 \cdot 10^{-3})$ |
| $4^{-14}$ | $9.973 \cdot 10^{-1}$ | $(7.696 \cdot 10^{-4})$ | $9.950 \cdot 10^{-1}$ | $(1.037 \cdot 10^{-3})$ |
| $4^{-15}$ | $9.973 \cdot 10^{-1}$ | $(7.696 \cdot 10^{-4})$ | $9.950 \cdot 10^{-1}$ | $(1.037 \cdot 10^{-3})$ |

Table 7: The medians over five trials plotted in Figure 8 on the top left, with the standard deviations shown in parentheses, both rounded to four-digit mantissas.

| $\lambda$ | $m = 25$ | | $m = 50$ | | $m = 200$ | |
|---|---|---|---|---|---|---|
| $4^2$ | $9.575 \cdot 10^1$ | $(1.097 \cdot 10^0)$ | $9.718 \cdot 10^1$ | $(1.086 \cdot 10^0)$ | $9.654 \cdot 10^1$ | $(2.816 \cdot 10^{-1})$ |
| $4^1$ | $9.713 \cdot 10^1$ | $(1.851 \cdot 10^0)$ | $9.604 \cdot 10^1$ | $(7.676 \cdot 10^{-1})$ | $9.650 \cdot 10^1$ | $(4.394 \cdot 10^{-1})$ |
| $4^0$ | $5.524 \cdot 10^1$ | $(1.869 \cdot 10^1)$ | $7.969 \cdot 10^1$ | $(1.272 \cdot 10^1)$ | $8.695 \cdot 10^1$ | $(5.651 \cdot 10^0)$ |
| $4^{-1}$ | $1.491 \cdot 10^1$ | $(8.641 \cdot 10^0)$ | $2.501 \cdot 10^1$ | $(1.831 \cdot 10^1)$ | $2.997 \cdot 10^1$ | $(1.003 \cdot 10^1)$ |
| $4^{-2}$ | $3.746 \cdot 10^0$ | $(2.267 \cdot 10^0)$ | $6.351 \cdot 10^0$ | $(5.363 \cdot 10^0)$ | $7.662 \cdot 10^0$ | $(2.820 \cdot 10^0)$ |
| $4^{-3}$ | $9.367 \cdot 10^{-1}$ | $(5.693 \cdot 10^{-1})$ | $1.590 \cdot 10^0$ | $(1.361 \cdot 10^0)$ | $1.918 \cdot 10^0$ | $(7.110 \cdot 10^{-1})$ |
| $4^{-4}$ | $2.342 \cdot 10^{-1}$ | $(1.424 \cdot 10^{-1})$ | $3.974 \cdot 10^{-1}$ | $(3.405 \cdot 10^{-1})$ | $4.796 \cdot 10^{-1}$ | $(1.779 \cdot 10^{-1})$ |
| $4^{-5}$ | $5.854 \cdot 10^{-2}$ | $(3.559 \cdot 10^{-2})$ | $9.935 \cdot 10^{-2}$ | $(8.513 \cdot 10^{-2})$ | $1.199 \cdot 10^{-1}$ | $(4.447 \cdot 10^{-2})$ |
| $4^{-6}$ | $1.464 \cdot 10^{-2}$ | $(8.898 \cdot 10^{-3})$ | $2.484 \cdot 10^{-2}$ | $(2.128 \cdot 10^{-2})$ | $2.998 \cdot 10^{-2}$ | $(1.112 \cdot 10^{-2})$ |
| $4^{-7}$ | $3.659 \cdot 10^{-3}$ | $(2.225 \cdot 10^{-3})$ | $6.210 \cdot 10^{-3}$ | $(5.321 \cdot 10^{-3})$ | $7.494 \cdot 10^{-3}$ | $(2.779 \cdot 10^{-3})$ |
| $4^{-8}$ | $9.147 \cdot 10^{-4}$ | $(5.561 \cdot 10^{-4})$ | $1.552 \cdot 10^{-3}$ | $(1.330 \cdot 10^{-3})$ | $1.874 \cdot 10^{-3}$ | $(6.948 \cdot 10^{-4})$ |
| $4^{-9}$ | $2.287 \cdot 10^{-4}$ | $(1.390 \cdot 10^{-4})$ | $3.881 \cdot 10^{-4}$ | $(3.325 \cdot 10^{-4})$ | $4.684 \cdot 10^{-4}$ | $(1.737 \cdot 10^{-4})$ |
| $4^{-10}$ | $5.721 \cdot 10^{-5}$ | $(3.476 \cdot 10^{-5})$ | $9.702 \cdot 10^{-5}$ | $(8.313 \cdot 10^{-5})$ | $1.171 \cdot 10^{-4}$ | $(4.342 \cdot 10^{-5})$ |
| $4^{-11}$ | $1.439 \cdot 10^{-5}$ | $(8.669 \cdot 10^{-6})$ | $2.427 \cdot 10^{-5}$ | $(2.074 \cdot 10^{-5})$ | $2.930 \cdot 10^{-5}$ | $(1.086 \cdot 10^{-5})$ |
| $4^{-12}$ | $4.005 \cdot 10^{-6}$ | $(2.205 \cdot 10^{-6})$ | $5.533 \cdot 10^{-6}$ | $(5.167 \cdot 10^{-6})$ | $7.636 \cdot 10^{-6}$ | $(2.675 \cdot 10^{-6})$ |
| $4^{-13}$ | $2.958 \cdot 10^{-6}$ | $(9.800 \cdot 10^{-7})$ | $3.520 \cdot 10^{-6}$ | $(8.499 \cdot 10^{-7})$ | $3.912 \cdot 10^{-6}$ | $(4.332 \cdot 10^{-7})$ |
| $4^{-14}$ | $1.708 \cdot 10^{-6}$ | $(1.249 \cdot 10^{-6})$ | $2.700 \cdot 10^{-6}$ | $(8.285 \cdot 10^{-7})$ | $4.183 \cdot 10^{-6}$ | $(1.183 \cdot 10^{-6})$ |
| $4^{-15}$ | $2.700 \cdot 10^{-6}$ | $(9.763 \cdot 10^{-7})$ | $2.700 \cdot 10^{-6}$ | $(1.093 \cdot 10^{-6})$ | $4.005 \cdot 10^{-6}$ | $(1.095 \cdot 10^{-6})$ |

Table 8: The medians over five trials plotted in Figure 8 on the top right, with the standard deviations shown in parentheses, both rounded to four-digit mantissas.

| $\lambda$ | $m = 25$ | | $m = 50$ | | $m = 200$ | |
|---|---|---|---|---|---|---|
| $4^2$ | $9.529 \cdot 10^1$ | $(3.501 \cdot 10^0)$ | $1.036 \cdot 10^2$ | $(9.435 \cdot 10^0)$ | $9.693 \cdot 10^1$ | $(1.513 \cdot 10^{-1})$ |
| $4^1$ | $9.730 \cdot 10^1$ | $(1.043 \cdot 10^0)$ | $9.607 \cdot 10^1$ | $(9.753 \cdot 10^{-1})$ | $9.623 \cdot 10^1$ | $(4.889 \cdot 10^{-1})$ |
| $4^0$ | $7.921 \cdot 10^1$ | $(1.510 \cdot 10^1)$ | $9.347 \cdot 10^1$ | $(7.892 \cdot 10^0)$ | $9.507 \cdot 10^1$ | $(6.628 \cdot 10^{-1})$ |
| $4^{-1}$ | $2.949 \cdot 10^1$ | $(3.891 \cdot 10^1)$ | $4.184 \cdot 10^1$ | $(3.133 \cdot 10^1)$ | $9.253 \cdot 10^1$ | $(6.296 \cdot 10^0)$ |
| $4^{-2}$ | $7.645 \cdot 10^0$ | $(3.979 \cdot 10^1)$ | $1.114 \cdot 10^1$ | $(2.911 \cdot 10^1)$ | $9.143 \cdot 10^1$ | $(2.830 \cdot 10^1)$ |
| $4^{-3}$ | $1.917 \cdot 10^0$ | $(4.028 \cdot 10^1)$ | $2.798 \cdot 10^0$ | $(1.139 \cdot 10^1)$ | $9.113 \cdot 10^1$ | $(4.401 \cdot 10^1)$ |
| $4^{-4}$ | $4.796 \cdot 10^{-1}$ | $(4.148 \cdot 10^1)$ | $6.999 \cdot 10^{-1}$ | $(2.957 \cdot 10^0)$ | $9.031 \cdot 10^1$ | $(4.837 \cdot 10^1)$ |
| $4^{-5}$ | $1.199 \cdot 10^{-1}$ | $(4.186 \cdot 10^1)$ | $1.750 \cdot 10^{-1}$ | $(7.414 \cdot 10^{-1})$ | $4.268 \cdot 10^1$ | $(4.573 \cdot 10^1)$ |
| $4^{-6}$ | $2.998 \cdot 10^{-2}$ | $(4.052 \cdot 10^1)$ | $4.375 \cdot 10^{-2}$ | $(1.854 \cdot 10^{-1})$ | $1.146 \cdot 10^1$ | $(3.987 \cdot 10^1)$ |
| $4^{-7}$ | $7.494 \cdot 10^{-3}$ | $(1.466 \cdot 10^1)$ | $1.094 \cdot 10^{-2}$ | $(4.636 \cdot 10^{-2})$ | $2.875 \cdot 10^0$ | $(4.045 \cdot 10^1)$ |
| $4^{-8}$ | $1.874 \cdot 10^{-3}$ | $(3.832 \cdot 10^0)$ | $2.734 \cdot 10^{-3}$ | $(1.159 \cdot 10^{-2})$ | $7.186 \cdot 10^{-1}$ | $(4.141 \cdot 10^1)$ |
| $4^{-9}$ | $4.684 \cdot 10^{-4}$ | $(9.530 \cdot 10^{-1})$ | $6.835 \cdot 10^{-4}$ | $(2.898 \cdot 10^{-3})$ | $1.796 \cdot 10^{-1}$ | $(4.170 \cdot 10^1)$ |
| $4^{-10}$ | $1.171 \cdot 10^{-4}$ | $(2.383 \cdot 10^{-1})$ | $1.709 \cdot 10^{-4}$ | $(7.244 \cdot 10^{-4})$ | $4.491 \cdot 10^{-2}$ | $(4.178 \cdot 10^1)$ |
| $4^{-11}$ | $2.935 \cdot 10^{-5}$ | $(5.957 \cdot 10^{-2})$ | $4.270 \cdot 10^{-5}$ | $(1.811 \cdot 10^{-4})$ | $1.123 \cdot 10^{-2}$ | $(4.180 \cdot 10^1)$ |
| $4^{-12}$ | $7.295 \cdot 10^{-6}$ | $(1.489 \cdot 10^{-2})$ | $1.063 \cdot 10^{-5}$ | $(4.519 \cdot 10^{-5})$ | $2.807 \cdot 10^{-3}$ | $(4.172 \cdot 10^1)$ |
| $4^{-13}$ | $3.415 \cdot 10^{-6}$ | $(3.722 \cdot 10^{-3})$ | $2.700 \cdot 10^{-6}$ | $(1.115 \cdot 10^{-5})$ | $7.016 \cdot 10^{-4}$ | $(2.239 \cdot 10^1)$ |
| $4^{-14}$ | $3.622 \cdot 10^{-6}$ | $(9.298 \cdot 10^{-4})$ | $2.958 \cdot 10^{-6}$ | $(2.532 \cdot 10^{-6})$ | $1.754 \cdot 10^{-4}$ | $(6.256 \cdot 10^0)$ |
| $4^{-15}$ | $2.832 \cdot 10^{-6}$ | $(2.317 \cdot 10^{-4})$ | $2.561 \cdot 10^{-6}$ | $(9.373 \cdot 10^{-7})$ | $4.385 \cdot 10^{-5}$ | $(1.581 \cdot 10^0)$ |

Table 9: The medians over five trials plotted in Figure 8 on the middle left, with the standard deviations shown in parentheses, both rounded to four-digit mantissas.

| $\lambda$ | $m = 25$ | | $m = 50$ | | $m = 200$ | |
|---|---|---|---|---|---|---|
| $4^2$ | $9.013 \cdot 10^1$ | $(2.193 \cdot 10^{-1})$ | $9.019 \cdot 10^1$ | $(5.887 \cdot 10^{-2})$ | $9.013 \cdot 10^1$ | $(1.563 \cdot 10^{-2})$ |
| $4^1$ | $9.292 \cdot 10^1$ | $(5.592 \cdot 10^{-1})$ | $9.081 \cdot 10^1$ | $(6.437 \cdot 10^{-2})$ | $8.960 \cdot 10^1$ | $(9.857 \cdot 10^{-2})$ |
| $4^0$ | $4.898 \cdot 10^1$ | $(3.903 \cdot 10^0)$ | $5.533 \cdot 10^1$ | $(2.030 \cdot 10^0)$ | $5.427 \cdot 10^1$ | $(1.127 \cdot 10^0)$ |
| $4^{-1}$ | $1.301 \cdot 10^1$ | $(1.671 \cdot 10^0)$ | $1.521 \cdot 10^1$ | $(1.672 \cdot 10^0)$ | $1.494 \cdot 10^1$ | $(4.246 \cdot 10^{-1})$ |
| $4^{-2}$ | $3.265 \cdot 10^0$ | $(4.356 \cdot 10^{-1})$ | $3.826 \cdot 10^0$ | $(4.739 \cdot 10^{-1})$ | $3.758 \cdot 10^0$ | $(1.110 \cdot 10^{-1})$ |
| $4^{-3}$ | $8.164 \cdot 10^{-1}$ | $(1.093 \cdot 10^{-1})$ | $9.569 \cdot 10^{-1}$ | $(1.199 \cdot 10^{-1})$ | $9.400 \cdot 10^{-1}$ | $(2.784 \cdot 10^{-2})$ |
| $4^{-4}$ | $2.041 \cdot 10^{-1}$ | $(2.734 \cdot 10^{-2})$ | $2.392 \cdot 10^{-1}$ | $(3.000 \cdot 10^{-2})$ | $2.350 \cdot 10^{-1}$ | $(6.962 \cdot 10^{-3})$ |
| $4^{-5}$ | $5.103 \cdot 10^{-2}$ | $(6.836 \cdot 10^{-3})$ | $5.981 \cdot 10^{-2}$ | $(7.501 \cdot 10^{-3})$ | $5.875 \cdot 10^{-2}$ | $(1.740 \cdot 10^{-3})$ |
| $4^{-6}$ | $1.276 \cdot 10^{-2}$ | $(1.709 \cdot 10^{-3})$ | $1.495 \cdot 10^{-2}$ | $(1.875 \cdot 10^{-3})$ | $1.469 \cdot 10^{-2}$ | $(4.351 \cdot 10^{-4})$ |
| $4^{-7}$ | $3.189 \cdot 10^{-3}$ | $(4.272 \cdot 10^{-4})$ | $3.738 \cdot 10^{-3}$ | $(4.688 \cdot 10^{-4})$ | $3.672 \cdot 10^{-3}$ | $(1.088 \cdot 10^{-4})$ |
| $4^{-8}$ | $7.974 \cdot 10^{-4}$ | $(1.068 \cdot 10^{-4})$ | $9.345 \cdot 10^{-4}$ | $(1.172 \cdot 10^{-4})$ | $9.180 \cdot 10^{-4}$ | $(2.719 \cdot 10^{-5})$ |
| $4^{-9}$ | $1.994 \cdot 10^{-4}$ | $(2.675 \cdot 10^{-5})$ | $2.336 \cdot 10^{-4}$ | $(2.929 \cdot 10^{-5})$ | $2.295 \cdot 10^{-4}$ | $(6.797 \cdot 10^{-6})$ |
| $4^{-10}$ | $4.998 \cdot 10^{-5}$ | $(6.686 \cdot 10^{-6})$ | $5.842 \cdot 10^{-5}$ | $(7.310 \cdot 10^{-6})$ | $5.739 \cdot 10^{-5}$ | $(1.696 \cdot 10^{-6})$ |
| $4^{-11}$ | $1.258 \cdot 10^{-5}$ | $(1.667 \cdot 10^{-6})$ | $1.462 \cdot 10^{-5}$ | $(1.808 \cdot 10^{-6})$ | $1.434 \cdot 10^{-5}$ | $(4.200 \cdot 10^{-7})$ |
| $4^{-12}$ | $3.306 \cdot 10^{-6}$ | $(5.100 \cdot 10^{-7})$ | $3.694 \cdot 10^{-6}$ | $(4.799 \cdot 10^{-7})$ | $3.686 \cdot 10^{-6}$ | $(1.467 \cdot 10^{-7})$ |
| $4^{-13}$ | $8.538 \cdot 10^{-7}$ | $(1.928 \cdot 10^{-7})$ | $1.034 \cdot 10^{-6}$ | $(3.022 \cdot 10^{-7})$ | $9.589 \cdot 10^{-7}$ | $(9.762 \cdot 10^{-8})$ |
| $4^{-14}$ | $3.211 \cdot 10^{-7}$ | $(6.454 \cdot 10^{-7})$ | $4.019 \cdot 10^{-7}$ | $(3.681 \cdot 10^{-7})$ | $9.701 \cdot 10^{-7}$ | $(2.770 \cdot 10^{-7})$ |
| $4^{-15}$ | $8.748 \cdot 10^{-7}$ | $(4.134 \cdot 10^{-7})$ | $7.127 \cdot 10^{-7}$ | $(2.035 \cdot 10^{-7})$ | $7.552 \cdot 10^{-7}$ | $(2.631 \cdot 10^{-7})$ |

Table 10: The medians over five trials plotted in Figure 8 on the middle right, with the standard deviations shown in parentheses, both rounded to four-digit mantissas.

| $\lambda$ | $m = 25$ | | $m = 50$ | | $m = 200$ | |
|---|---|---|---|---|---|---|
| $4^2$ | $9.000 \cdot 10^1$ | $(2.752 \cdot 10^{-1})$ | $9.023 \cdot 10^1$ | $(2.088 \cdot 10^{-1})$ | $9.012 \cdot 10^1$ | $(2.708 \cdot 10^{-2})$ |
| $4^1$ | $9.220 \cdot 10^1$ | $(4.438 \cdot 10^{-1})$ | $9.081 \cdot 10^1$ | $(6.729 \cdot 10^{-2})$ | $8.973 \cdot 10^1$ | $(8.384 \cdot 10^{-2})$ |
| $4^0$ | $6.040 \cdot 10^1$ | $(5.291 \cdot 10^0)$ | $5.576 \cdot 10^1$ | $(2.495 \cdot 10^0)$ | $5.645 \cdot 10^1$ | $(1.837 \cdot 10^0)$ |
| $4^{-1}$ | $1.729 \cdot 10^1$ | $(9.637 \cdot 10^0)$ | $1.701 \cdot 10^1$ | $(3.629 \cdot 10^0)$ | $1.855 \cdot 10^1$ | $(2.099 \cdot 10^0)$ |
| $4^{-2}$ | $4.364 \cdot 10^0$ | $(6.059 \cdot 10^0)$ | $4.306 \cdot 10^0$ | $(3.047 \cdot 10^0)$ | $6.111 \cdot 10^0$ | $(2.381 \cdot 10^0)$ |
| $4^{-3}$ | $1.092 \cdot 10^0$ | $(4.761 \cdot 10^0)$ | $1.078 \cdot 10^0$ | $(1.174 \cdot 10^0)$ | $2.766 \cdot 10^0$ | $(2.318 \cdot 10^0)$ |
| $4^{-4}$ | $2.730 \cdot 10^{-1}$ | $(4.860 \cdot 10^0)$ | $2.694 \cdot 10^{-1}$ | $(3.045 \cdot 10^{-1})$ | $1.922 \cdot 10^0$ | $(2.149 \cdot 10^0)$ |
| $4^{-5}$ | $6.825 \cdot 10^{-2}$ | $(4.918 \cdot 10^0)$ | $6.735 \cdot 10^{-2}$ | $(7.634 \cdot 10^{-2})$ | $1.066 \cdot 10^0$ | $(1.672 \cdot 10^0)$ |
| $4^{-6}$ | $1.706 \cdot 10^{-2}$ | $(4.765 \cdot 10^0)$ | $1.684 \cdot 10^{-2}$ | $(1.909 \cdot 10^{-2})$ | $2.823 \cdot 10^{-1}$ | $(1.218 \cdot 10^0)$ |
| $4^{-7}$ | $4.266 \cdot 10^{-3}$ | $(1.724 \cdot 10^0)$ | $4.209 \cdot 10^{-3}$ | $(4.774 \cdot 10^{-3})$ | $7.077 \cdot 10^{-2}$ | $(9.110 \cdot 10^{-1})$ |
| $4^{-8}$ | $1.067 \cdot 10^{-3}$ | $(4.507 \cdot 10^{-1})$ | $1.052 \cdot 10^{-3}$ | $(1.194 \cdot 10^{-3})$ | $1.769 \cdot 10^{-2}$ | $(8.302 \cdot 10^{-1})$ |
| $4^{-9}$ | $2.666 \cdot 10^{-4}$ | $(1.121 \cdot 10^{-1})$ | $2.631 \cdot 10^{-4}$ | $(2.984 \cdot 10^{-4})$ | $4.422 \cdot 10^{-3}$ | $(8.104 \cdot 10^{-1})$ |
| $4^{-10}$ | $6.671 \cdot 10^{-5}$ | $(2.803 \cdot 10^{-2})$ | $6.579 \cdot 10^{-5}$ | $(7.462 \cdot 10^{-5})$ | $1.106 \cdot 10^{-3}$ | $(8.055 \cdot 10^{-1})$ |
| $4^{-11}$ | $1.665 \cdot 10^{-5}$ | $(7.007 \cdot 10^{-3})$ | $1.645 \cdot 10^{-5}$ | $(1.865 \cdot 10^{-5})$ | $2.764 \cdot 10^{-4}$ | $(8.043 \cdot 10^{-1})$ |
| $4^{-12}$ | $4.401 \cdot 10^{-6}$ | $(1.752 \cdot 10^{-3})$ | $4.109 \cdot 10^{-6}$ | $(4.718 \cdot 10^{-6})$ | $6.910 \cdot 10^{-5}$ | $(8.025 \cdot 10^{-1})$ |
| $4^{-13}$ | $1.507 \cdot 10^{-6}$ | $(4.378 \cdot 10^{-4})$ | $9.764 \cdot 10^{-7}$ | $(1.226 \cdot 10^{-6})$ | $1.718 \cdot 10^{-5}$ | $(4.307 \cdot 10^{-1})$ |
| $4^{-14}$ | $1.071 \cdot 10^{-6}$ | $(1.092 \cdot 10^{-4})$ | $7.966 \cdot 10^{-7}$ | $(6.012 \cdot 10^{-7})$ | $4.466 \cdot 10^{-6}$ | $(1.203 \cdot 10^{-1})$ |
| $4^{-15}$ | $7.101 \cdot 10^{-7}$ | $(2.720 \cdot 10^{-5})$ | $6.444 \cdot 10^{-7}$ | $(4.099 \cdot 10^{-7})$ | $1.339 \cdot 10^{-6}$ | $(3.040 \cdot 10^{-2})$ |

Table 11: The medians over five trials plotted in Figure 8 on the bottom left, with the standard deviations shown in parentheses, both rounded to four-digit mantissas.

| $\lambda$ | $m = 25$ | | $m = 50$ | | $m = 200$ | |
|---|---|---|---|---|---|---|
| $4^2$ | $7.823 \cdot 10^1$ | $(2.780 \cdot 10^{-1})$ | $1.118 \cdot 10^2$ | $(4.888 \cdot 10^{-1})$ | $2.205 \cdot 10^2$ | $(2.389 \cdot 10^{-1})$ |
| $4^1$ | $1.974 \cdot 10^1$ | $(8.401 \cdot 10^{-2})$ | $2.803 \cdot 10^1$ | $(1.204 \cdot 10^{-1})$ | $5.534 \cdot 10^1$ | $(5.103 \cdot 10^{-2})$ |
| $4^0$ | $5.695 \cdot 10^0$ | $(1.629 \cdot 10^{-2})$ | $7.771 \cdot 10^0$ | $(2.592 \cdot 10^{-2})$ | $1.461 \cdot 10^1$ | $(1.094 \cdot 10^{-2})$ |
| $4^{-1}$ | $2.189 \cdot 10^0$ | $(4.216 \cdot 10^{-3})$ | $2.712 \cdot 10^0$ | $(6.911 \cdot 10^{-3})$ | $4.421 \cdot 10^0$ | $(4.102 \cdot 10^{-3})$ |
| $4^{-2}$ | $1.298 \cdot 10^0$ | $(1.043 \cdot 10^{-3})$ | $1.429 \cdot 10^0$ | $(1.780 \cdot 10^{-3})$ | $1.856 \cdot 10^0$ | $(1.132 \cdot 10^{-3})$ |
| $4^{-3}$ | $1.075 \cdot 10^0$ | $(2.595 \cdot 10^{-4})$ | $1.107 \cdot 10^0$ | $(4.487 \cdot 10^{-4})$ | $1.214 \cdot 10^0$ | $(2.891 \cdot 10^{-4})$ |
| $4^{-4}$ | $1.019 \cdot 10^0$ | $(6.386 \cdot 10^{-5})$ | $1.027 \cdot 10^0$ | $(1.122 \cdot 10^{-4})$ | $1.054 \cdot 10^0$ | $(7.164 \cdot 10^{-5})$ |
| $4^{-5}$ | $1.005 \cdot 10^0$ | $(1.513 \cdot 10^{-5})$ | $1.007 \cdot 10^0$ | $(2.796 \cdot 10^{-5})$ | $1.013 \cdot 10^0$ | $(1.663 \cdot 10^{-5})$ |
| $4^{-6}$ | $1.001 \cdot 10^0$ | $(3.763 \cdot 10^{-6})$ | $1.002 \cdot 10^0$ | $(7.153 \cdot 10^{-6})$ | $1.003 \cdot 10^0$ | $(2.939 \cdot 10^{-6})$ |
| $4^{-7}$ | $1.000 \cdot 10^0$ | $(2.674 \cdot 10^{-6})$ | $1.000 \cdot 10^0$ | $(2.684 \cdot 10^{-6})$ | $1.001 \cdot 10^0$ | $(1.031 \cdot 10^{-6})$ |
| $4^{-8}$ | $1.000 \cdot 10^0$ | $(2.889 \cdot 10^{-6})$ | $1.000 \cdot 10^0$ | $(2.209 \cdot 10^{-6})$ | $1.000 \cdot 10^0$ | $(1.699 \cdot 10^{-6})$ |
| $4^{-9}$ | $1.000 \cdot 10^0$ | $(2.972 \cdot 10^{-6})$ | $1.000 \cdot 10^0$ | $(2.199 \cdot 10^{-6})$ | $1.000 \cdot 10^0$ | $(1.891 \cdot 10^{-6})$ |
| $4^{-10}$ | $1.000 \cdot 10^0$ | $(2.994 \cdot 10^{-6})$ | $1.000 \cdot 10^0$ | $(2.205 \cdot 10^{-6})$ | $1.000 \cdot 10^0$ | $(1.940 \cdot 10^{-6})$ |
| $4^{-11}$ | $1.000 \cdot 10^0$ | $(3.000 \cdot 10^{-6})$ | $1.000 \cdot 10^0$ | $(2.207 \cdot 10^{-6})$ | $1.000 \cdot 10^0$ | $(1.952 \cdot 10^{-6})$ |
| $4^{-12}$ | $1.000 \cdot 10^0$ | $(3.001 \cdot 10^{-6})$ | $1.000 \cdot 10^0$ | $(2.207 \cdot 10^{-6})$ | $1.000 \cdot 10^0$ | $(1.955 \cdot 10^{-6})$ |
| $4^{-13}$ | $1.000 \cdot 10^0$ | $(3.002 \cdot 10^{-6})$ | $1.000 \cdot 10^0$ | $(2.207 \cdot 10^{-6})$ | $1.000 \cdot 10^0$ | $(1.956 \cdot 10^{-6})$ |
| $4^{-14}$ | $1.000 \cdot 10^0$ | $(3.002 \cdot 10^{-6})$ | $1.000 \cdot 10^0$ | $(2.207 \cdot 10^{-6})$ | $1.000 \cdot 10^0$ | $(1.956 \cdot 10^{-6})$ |
| $4^{-15}$ | $1.000 \cdot 10^0$ | $(3.002 \cdot 10^{-6})$ | $1.000 \cdot 10^0$ | $(2.207 \cdot 10^{-6})$ | $1.000 \cdot 10^0$ | $(1.956 \cdot 10^{-6})$ |

Table 12: The medians over five trials plotted in Figure 8 on the bottom right, with the standard deviations shown in parentheses, both rounded to four-digit mantissas.

| $\lambda$ | $m = 25$ | | $m = 50$ | | $m = 200$ | |
|---|---|---|---|---|---|---|
| $4^2$ | $7.790 \cdot 10^1$ | $(4.055 \cdot 10^{-1})$ | $1.114 \cdot 10^2$ | $(1.844 \cdot 10^{-1})$ | $2.203 \cdot 10^2$ | $(2.081 \cdot 10^{-1})$ |
| $4^1$ | $1.960 \cdot 10^1$ | $(9.365 \cdot 10^{-2})$ | $2.799 \cdot 10^1$ | $(5.122 \cdot 10^{-2})$ | $5.530 \cdot 10^1$ | $(5.434 \cdot 10^{-2})$ |
| $4^0$ | $5.687 \cdot 10^0$ | $(2.398 \cdot 10^{-2})$ | $7.818 \cdot 10^0$ | $(2.092 \cdot 10^{-2})$ | $1.469 \cdot 10^1$ | $(2.460 \cdot 10^{-2})$ |
| $4^{-1}$ | $2.209 \cdot 10^0$ | $(6.457 \cdot 10^{-3})$ | $2.758 \cdot 10^0$ | $(1.127 \cdot 10^{-2})$ | $4.507 \cdot 10^0$ | $(2.189 \cdot 10^{-2})$ |
| $4^{-2}$ | $1.300 \cdot 10^0$ | $(3.430 \cdot 10^{-3})$ | $1.436 \cdot 10^0$ | $(2.735 \cdot 10^{-3})$ | $1.872 \cdot 10^0$ | $(5.661 \cdot 10^{-3})$ |
| $4^{-3}$ | $1.071 \cdot 10^0$ | $(9.842 \cdot 10^{-4})$ | $1.105 \cdot 10^0$ | $(6.528 \cdot 10^{-4})$ | $1.213 \cdot 10^0$ | $(5.792 \cdot 10^{-4})$ |
| $4^{-4}$ | $1.015 \cdot 10^0$ | $(6.093 \cdot 10^{-4})$ | $1.022 \cdot 10^0$ | $(6.194 \cdot 10^{-4})$ | $1.049 \cdot 10^0$ | $(7.904 \cdot 10^{-4})$ |
| $4^{-5}$ | $1.001 \cdot 10^0$ | $(5.900 \cdot 10^{-4})$ | $1.002 \cdot 10^0$ | $(6.740 \cdot 10^{-4})$ | $1.009 \cdot 10^0$ | $(9.884 \cdot 10^{-4})$ |
| $4^{-6}$ | $9.972 \cdot 10^{-1}$ | $(5.898 \cdot 10^{-4})$ | $9.967 \cdot 10^{-1}$ | $(6.937 \cdot 10^{-4})$ | $9.984 \cdot 10^{-1}$ | $(1.029 \cdot 10^{-3})$ |
| $4^{-7}$ | $9.963 \cdot 10^{-1}$ | $(5.864 \cdot 10^{-4})$ | $9.954 \cdot 10^{-1}$ | $(6.985 \cdot 10^{-4})$ | $9.959 \cdot 10^{-1}$ | $(1.036 \cdot 10^{-3})$ |
| $4^{-8}$ | $9.961 \cdot 10^{-1}$ | $(5.857 \cdot 10^{-4})$ | $9.951 \cdot 10^{-1}$ | $(6.994 \cdot 10^{-4})$ | $9.952 \cdot 10^{-1}$ | $(1.037 \cdot 10^{-3})$ |
| $4^{-9}$ | $9.961 \cdot 10^{-1}$ | $(5.856 \cdot 10^{-4})$ | $9.950 \cdot 10^{-1}$ | $(6.996 \cdot 10^{-4})$ | $9.951 \cdot 10^{-1}$ | $(1.037 \cdot 10^{-3})$ |
| $4^{-10}$ | $9.961 \cdot 10^{-1}$ | $(5.855 \cdot 10^{-4})$ | $9.950 \cdot 10^{-1}$ | $(6.996 \cdot 10^{-4})$ | $9.950 \cdot 10^{-1}$ | $(1.037 \cdot 10^{-3})$ |
| $4^{-11}$ | $9.960 \cdot 10^{-1}$ | $(5.855 \cdot 10^{-4})$ | $9.950 \cdot 10^{-1}$ | $(6.996 \cdot 10^{-4})$ | $9.950 \cdot 10^{-1}$ | $(1.037 \cdot 10^{-3})$ |
| $4^{-12}$ | $9.960 \cdot 10^{-1}$ | $(5.855 \cdot 10^{-4})$ | $9.950 \cdot 10^{-1}$ | $(6.996 \cdot 10^{-4})$ | $9.950 \cdot 10^{-1}$ | $(1.037 \cdot 10^{-3})$ |
| $4^{-13}$ | $9.960 \cdot 10^{-1}$ | $(5.855 \cdot 10^{-4})$ | $9.950 \cdot 10^{-1}$ | $(6.996 \cdot 10^{-4})$ | $9.950 \cdot 10^{-1}$ | $(1.037 \cdot 10^{-3})$ |
| $4^{-14}$ | $9.960 \cdot 10^{-1}$ | $(5.855 \cdot 10^{-4})$ | $9.950 \cdot 10^{-1}$ | $(6.996 \cdot 10^{-4})$ | $9.950 \cdot 10^{-1}$ | $(1.037 \cdot 10^{-3})$ |
| $4^{-15}$ | $9.960 \cdot 10^{-1}$ | $(5.855 \cdot 10^{-4})$ | $9.950 \cdot 10^{-1}$ | $(6.996 \cdot 10^{-4})$ | $9.950 \cdot 10^{-1}$ | $(1.037 \cdot 10^{-3})$ |

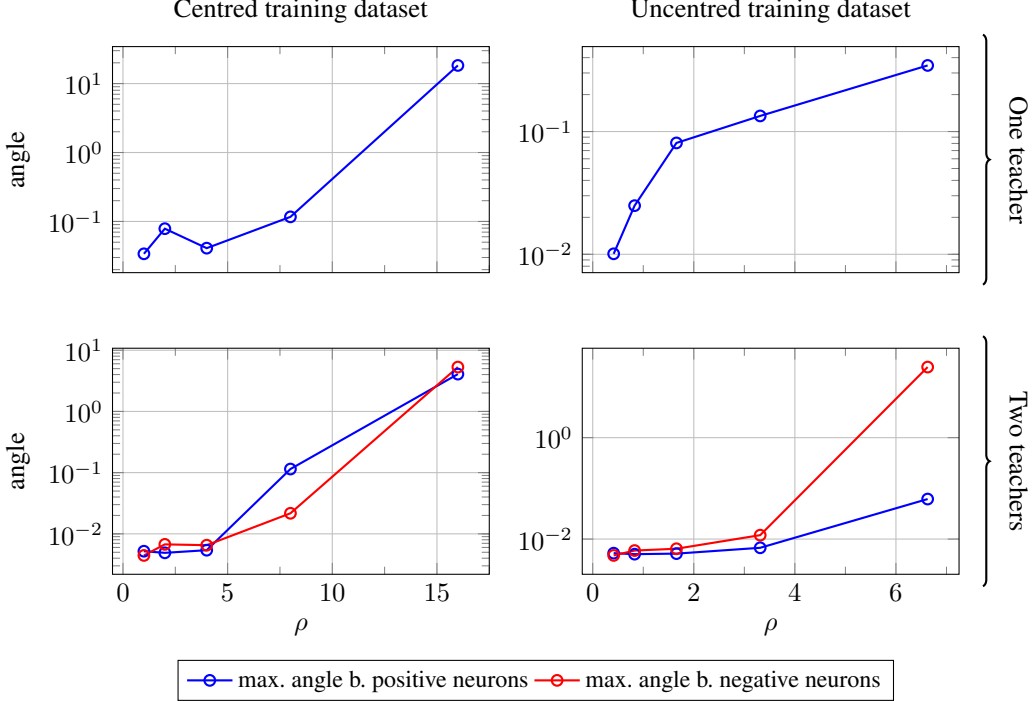

Figure 10: The maximum angle between hidden neurons that start with a positive (blue) and negative (red) inner product with the first teacher neuron. The first teacher neuron has norm 1 and the second teacher neuron has norm 3. The vertical axes are logarithmic and the angles are in degrees. The horizontal axes show different multipliers $\rho$ for the variance of the distribution of the data points (cf. section 8). The input dimension is $d = 16$ and, for each teacher neuron, we sample $d$ data points from the distribution specified in the main. Each point in the plot shows the median over 15 trials of the angle in degrees at the end of training. The training runs for $2 \cdot 10^7$ iterations or until the loss reaches $10^{-9}$. The width of the network is $m = 25$, and the initialisation scale is $\lambda = 4^{-7}$.

## I  Further experiments

Here we report on experiments in which we explore the effects of adding a second teacher neuron whose direction is opposite to that of the first, and of increasing the scale $\rho$ of the noise used to generate the synthetic datasets (cf. section 8) so that quickly most of the data points exceed the $\pi/4$ angle with their corresponding teacher neuron.

In Figure 10, the growing maximum angles between neurons at the end of the training indicate that we no longer have a single (or one per teacher neuron) aligned bundle of neurons forming and sticking together for the rest of the training.

In the bottom two plots of Figure 10 and in Figure 11, for small scales $\rho$ (where the smallest values are such that the angles between the data points and the corresponding teacher neuron concentrate around $\pi/4$), the phenomena we identified theoretically still seem to hold, where the training passes near a second saddle point as we outlined in section 9.

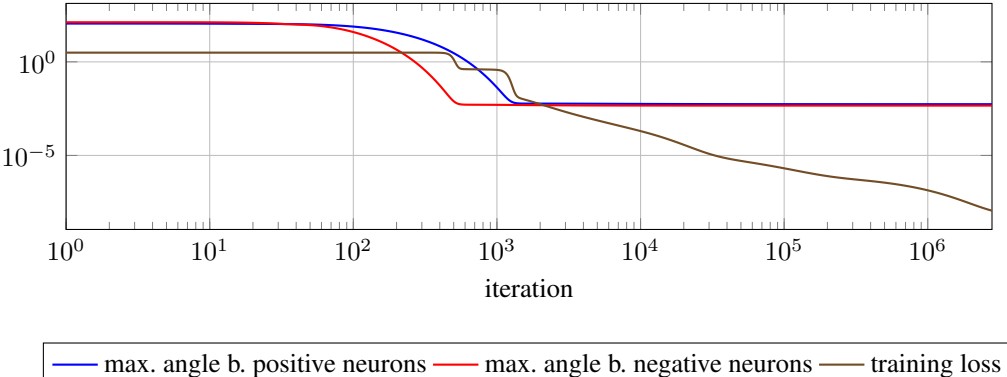

Figure 11: The evolution of the training loss and the maximum angle between positive and negative hidden neurons during the training in dimension 16. The vertical axes are logarithmic and the angles are in degrees. This is one example of a run contributing to Figure 10. Specifically, in this run the training dataset is uncentered and $\rho = \sqrt{2} - 1$. The two fast drops in loss (after passing of the first and then the second saddle point) coincide with the times at which the respective group of hidden neurons aligns.