# OpenReview forum: "Learning a Neuron by a Shallow ReLU Network: Dynamics and Implicit Bias for Correlated Inputs"
_NeurIPS.cc/2023/Conference — NeurIPS 2023 poster_

### Official Review · Reviewer_JZbY · 2023-06-19

**Soundness:** 3 good
**Presentation:** 3 good
**Contribution:** 3 good
**Rating:** 7
**Confidence:** 3

**Summary:**

The authors analyze the dynamics and implicit bias of gradient flow with the square loss when learning a single ReLU neuron using a one-hidden-layer ReLU network. They assume that the training data are correlated with the teacher neuron (the angles are smaller than $\pi/4$), and that gradient flow starts from a small and balanced initialization. They give a non-asymptotic convergence analysis. In the limit where the initialization scale tends to zero, the resulting network has rank $1$. Namely, all non-zero neurons point in the direction of the teacher neuron. On the other hand, the resulting network might not have minimal Euclidean norm. Thus, there is an implicit bias for rank minimization but not for norm minimization.

**Strengths:**

Understanding convergence and implicit bias in overparameterized networks is an important question, that has attracted much interest in recent years. The paper gives a detailed analysis of the trajectory and implicit bias. The analysis is under strong assumptions (single-neuron teacher, correlated training data, small and balanced initialization, etc.), but these assumptions are present also in existing results, and the analysis of gradient flow is challenging even under such assumptions. Finally, the paper is well-written.

**Weaknesses:**

In Assumption 1:
- Item (iv): why is it a measure-zero event?
- Items (iv) and (v): I think that the assumptions should specify properties of the training data and the training algorithm. Then, the properties of the trajectory should be shown using these assumptions. Can you specify items (iv) and (v) as assumptions on the data+algorithm?

Other than that, I don’t have major concerns. An obvious limitation is the strong assumptions, and specifically the assumption on the correlated training data, but as I already mentioned, I think that it is reasonable here.


**Questions:**

See the “weaknesses” section.

**Limitations:**

The authors discussed the limitations.

---

> ### Author Rebuttal · Authors · 2023-08-08
>
> > In Assumption 1:
> > - Item (iv): why is it a measure-zero event?
> > - Items (iv) and (v): I think that the assumptions should specify properties of the training data and the training algorithm. Then, the properties of the trajectory should be shown using these assumptions. Can you specify items (iv) and (v) as assumptions on the data+algorithm?
>
> Regarding item (iv), a single yardstick trajectory at any time $t$ follows a direction $\mathbf{\gamma}\_S$ for some $S \subseteq [n]$.  The set $S$ changes at most $n$ times, namely at the crossing of $\bigcup\_{i \in [n]} H\_i$, where $H\_i$ is the set of vectors orthogonal to the training point $\mathbf{x}\_i$.  Consider a set of targets $H\_i \cap H\_k$, of dimension $d-2$.  It can be reached by a straight-line trajectory from a convex polyhedron $P$ of dimension at most $d-1$.  Thus, the previous change of direction occurs at a vector from $P \cap \bigcup_{i \in [n]} H\_i$, which is a polyhedron of dimension at most $d-2$ as well.  Thus, in order for a vector to reach $\bigcup_{i < k} H\_i \cap H\_k$ by a yardstick trajectory, it must belong to a finite union of affine subspaces of dimension $d-1$.  Therefore, if each $\mathbf{z}\_j$ at the initialisation is sampled from any absolutely continuous distribution on $\mathbb{R}^d$, the probability that Assumption 1 (iv) is violated is $0$.
>
> Regarding item (v), we would say it is an assumption about the gradient flow, and it has featured in the literature, e.g. in Eberle et al. “Existence, uniqueness, and convergence rates for gradient flows in the training of artificial neural networks with ReLU activation” ERA 31(5): 2519-2554, 2023, where on page 2535 it is paraphrased as “the set of all degenerate neurons of the GF solution at time $t \in [0, \infty)$ is non-decreasing in the time variable” (in fact, our Assumption 1 (v) is slightly weaker than that).  The trajectories of the gradient flow that this assumption excludes cannot arise as limits of paths of gradient descent with any fixed value for the derivative of $\sigma$ (the non-linear ReLU function) at zero: if that value is $0$, then any deactivated neuron cannot activate subsequently even if there are data points exactly on its ReLU boundary; and if that value is greater than $0$, then in our setting the deactivation of any neuron would necessarily involve removing all data points from its ReLU boundary.  Thus arguably these impractical trajectories might a priori exist only due to a quirk of the theoretical setup of the gradient flow.
>
> We are happy to expand on these points in the main paper, and we shall provide in the appendix a more detailed proof of the measure zero for item (iv).

---

> > ### Comment · Reviewer_JZbY · 2023-08-12
> >
> > Thanks for the response

---

### Official Review · Reviewer_ytEj · 2023-06-28

**Soundness:** 3 good
**Presentation:** 2 fair
**Contribution:** 3 good
**Rating:** 6
**Confidence:** 4

**Summary:**

The paper studies the problem of learning a single ReLU using a 2-layer ReLU network using gradient flow on both layers. The main assumption is that the data is correlated with the target neuron, while other milder assumptions are also used (e.g. specific initialization and spectral assumption on the data matrix). The main result is a two-phase convergence to a global minimum. Several experiments are also given.


**Strengths:**

- The main convergence result is novel AFAIK, and shows an interesting convergence dynamic, where in the first phase the learned weights either align with the target neuron or deactivate, and in the second phase converge to a global minimum.
- The connection between the global minima of the problem and the minimal norm solution in Section 7 is interesting and brings forward the question of whether minimizing the empirical loss results in a minimal norm solution, which was studied in previous works too (e.g. Vardi et al 2022).
- The experimental part shows empirically the behavior of the angles between the learned weights and the target neuron.


**Weaknesses:**

- The assumption that the data is correlated with the target neuron is pretty strong. The motivation for taking an angle of at most \pi/4 between each data point and the target neuron is also not clear. What changes if the angle is larger\smaller? I think the authors should elaborate more on this assumptions, and what breaks if it is not assumed. To compare, other papers about studying a single neuron usually consider a data distribution spread in all directions (e.g. Frei et al. 2020, Yehudai & Shamir 2020).
- The presentation of the main result is not clear. I think there should be a single Theorem stating the convergence result, with an explicit convergence rate. Currently, there is no single result just a lemma for each phase, and it is difficult to parse the main out-take of the paper. In such a paper, I think the convergence rate of the entire procedure is crucial to fully understand the quality of the result.
- The paper is very technical, and in my opinion, doesn’t provide enough intuition to understand the quantities that are used. For example, the definition of \delta in line 238. Can’t \delta be zero or at least exponentially small (if the angle between two data points is very close to \pi/2).
- The result in Section 7 is interesting but not quite clear. What does the quantity M represents? What can we say about the dataset itself so that either option (i) or (ii) of Theorem 8 is applied?
- I think that claiming that the implicit bias for the problem studied here is to minimize the norm is a bit misleading. As I understand it, all the learned weights either align with the target neuron or deactivate, this means that the solution converges to a specific form of rank-1 matrix (where each row is either v^* or 0).


**Questions:**

- What changes if we assume that the angle between the data samples and the target neurons is \alpha, where \alpha > \pi/4?
- What is the total convergence rate of gradient flow for learning a single neuron?
- Is it possible to extend the result to gradient flow? or possibly SGD?
- Is there an explicit condition on the dataset that can be given so that the set of interpolators also minimizes the norm of the predictor


**Limitations:**

The authors do address adequately to the limitations of the paper, although I think it is important to elaborate more on the main assumption about the dataset.

---

> ### Author Rebuttal · Authors · 2023-08-08
>
> > [...] where in the first phase the learned weights either align with the target neuron or deactivate, [...]
>
> Please see the first item in our response to reviewer PDpC.
>
> > The assumption that the data is correlated with the target neuron is pretty strong. The motivation for taking an angle of at most $\pi/4$ between each data point and the target neuron is also not clear. What changes if the angle is larger\smaller? I think the authors should elaborate more on this assumptions, and what breaks if it is not assumed. [...]
>
> Please see our response to all the reviewers jointly.
>
> > The presentation of the main result is not clear. I think there should be a single Theorem stating the convergence result, with an explicit convergence rate. Currently, there is no single result just a lemma for each phase, and it is difficult to parse the main out-take of the paper. [...]
>
> > What is the total convergence rate of gradient flow for learning a single neuron?
>
> From Lemma 6, in the final stage of the training, the mean square empirical loss converges to zero at an exponential rate $L(\mathbf{\theta}^{T\_2+t}) = \exp(-\Omega(t\delta^4/\Delta^2))$, which is $\exp(-\Omega(t))$ for fixed $\delta$ and $\Delta$.
>
> The time $T\_2$ to reach the final stage by Assumption 2 satisfies $T_2 = O((\ln m+n^2) d \Delta^4/(\delta^8\ln\delta))$, which is $O((\ln m+n^2)d)$ for fixed $\delta$ and $\Delta$.  This reflects the lengthy escape from the saddle at the origin due to the small initialisation scale.  (After submission, we succeeded in reducing the $n^2$ term here to $n\ln n$.)
>
> We are happy to make these bounds clearer in the paper.
>
> > The paper is very technical, and in my opinion, doesn’t provide enough intuition to understand the quantities that are used. For example, the definition of $\delta$ in line 238. Can’t $\delta$ be zero or at least exponentially small (if the angle between two data points is very close to $\pi/2$).
>
> In our results, the dataset and the unscaled initialisation are given rather than sampled from particular distributions.  This generality comes at the price of having the explicit parameters $\delta$ and $\Delta$, which otherwise typically would be replaced by lower and upper bounds (respectively) on measurements of the dataset and the unscaled initialisation that hold with high probabilities.
>
> That $\delta$ is positive follows from Assumption 1, however indeed it may be arbitrarily small since in our worst-case approach the dataset is given by an adversary.
>
> We remark that e.g. Boursier et al. regarded quantities like our $\delta$ and $\Delta$ as constants (see their Appendix B.2).
>
> > The result in Section 7 is interesting but not quite clear. What does the quantity $\mathcal{M}$ represents? What can we say about the dataset itself so that either option (i) or (ii) of Theorem 8 is applied?
>
> > Is there an explicit condition on the dataset that can be given so that the set of interpolators also minimizes the norm of the predictor
>
> At present we do not have an alternative characterisation of the dichotomy shown in Theorem 8.
>
> The following is a sufficient condition for option (i) to apply, i.e. for the set of rank-1 interpolators to also minimise the norm: the inverse of the Gram matrix of the dataset (in our setting this Gram matrix is positive) is a Z-matrix, i.e. the inner product of any two distinct rows of the inverse of the dataset matrix $\mathbf X$ is non-positive.
>
> As we point out in Remark 37 (ii) in Appendix H, option (i) of Theorem 8 also always holds in dimension $d=2$.
>
> That option (ii) may occur is related to (in fact, it implies) the known fact that, for $d>2$, it is not the case that the inverse of every non-singular symmetric non-negative $d \times d$ matrix is a Z-matrix (see e.g. Markham “Nonnegative matrices whose inverses are M-matrices" Proc. Am. Math. Soc. 36 (2), 326-330, 1972).  However, additional work was involved to construct our family of examples in Example 36 (case $\mathcal M>0$) in Appendix H.
>
> > I think that claiming that the implicit bias for the problem studied here is to minimize the norm is a bit misleading. As I understand it, all the learned weights either align with the target neuron or deactivate, this means that the solution converges to a specific form of rank-1 matrix (where each row is either $\mathbf{v}^*$ or 0).
>
> In the definition of $\Theta\_{\mathbf{v}^*}$ in Section 6, the normalised non-zero hidden neurons equal $\mathbf{v}^*$. Thus Theorem 7 asserts that, as the initialisation scale decreases to zero, the training converges to a rank-1 matrix in which each row is either a positive scalar multiple of $\mathbf{v}^*$ or $0$. However, the scalars can be quite different, i.e. the hidden neurons in the rank-1 network might have quite different norms (that depend on the initialisation).
>
> We agree that the norm is not necessarily minimised; indeed Theorem 8 (ii) captures the scenario where it is not.
>
> > Is it possible to extend the result to gradient flow? or possibly SGD?
>
> Thank you for this suggestion. We believe an extension to gradient descent would be possible, by proving an upper bound on the learning rate such that essentially the same behaviour occurs as with gradient flow (as in e.g. Cheridito et al. “A proof of convergence for gradient descent in the training of artificial neural networks for constant target functions” J. Complex. 72, 101646, 2022). At the moment we prefer to leave this to future work, so as not to distract readers from our main goal: analysis of the training dynamics and characterisation of the implicit bias of learning a non-trivial regression task using a non-linear network in a theoretically challenging setting.
>
> An intriguing related question is to what extent the phenomena described in this and other works that assume a small initialisation are consistent with training by gradient descent (possibly stochastic) that uses adaptive learning rates to speed up departures from near saddle points.

---

> > ### Comment · Reviewer_ytEj · 2023-08-18
> >
> > I thank the authors for the response. I still think there are some issues with the presentation of the paper, most notably I think it would make it much clearer if there is a single theorem statement which provides the total convergence time, rather than several lemmas. I will keep my score.

---

> > > ### Author Response · Authors · 2023-08-21
> > >
> > > We did not say so explicitly, but yes, we shall be happy to put the total convergence time in a single theorem as you have suggested --- thank you for this idea. More generally, we shall improve the presentation based on the suggestions in all four reviews.

---

### Official Review · Reviewer_PDpC · 2023-07-06

**Soundness:** 3 good
**Presentation:** 3 good
**Contribution:** 3 good
**Rating:** 6
**Confidence:** 3

**Summary:**

This paper studies how a two-layer ReLU network can fit a single neuron. The authors consider the case where all
training points are correlated with the teacher neuron and show that gradient flow from small initialization can
converge to a zero-loss solution.
They divide the training into two stages. In the first stage, the neurons are small and will align with the teacher
neuron or deactivate, depending on the sign of the output weight, and in the second stage, the aligned neurons will grow
and fit the target function.
In addition, they show that as the initialization scale goes to $0$, gradient flow converges to a rank-$1$ solution.

**Strengths:**

* Overall, the presentation is clear, and the main text is relatively easy to follow (see the weakness part of the
  review for some minor issues.)
* The use of the yardsticks $\omega_j$ is interesting.
* The geometric argument for the second phase seems novel and may be of independent interest.
* Theorem 8, which shows the set of balanced rank-1 interpolating networks and the set of minimum-norm interpolating
  networks can be the same or disjoint, depending on a certain quantity, is surprising.

**Weaknesses:**

* The presentation is overall clear, but the notations are cumbersome.
  * For example, I don't think using $w$ and $\omega$ simultaneously is a good idea, especially when they co-occur a lot
    and represent two closely related objects.
  * The detailed definitions of $\delta$ and $S_l$ can be moved into the appendix, and define them informally in the
    main text and maybe briefly explain how small can $\delta$ be and the intuition behind $S_l$.
* I personally don't like the $\exp(-n)$ initialization scale, though I will accept it as it has been used in previous
  works. I think it is somewhat cheating because, with it, you can make sure the norm of the neurons is sufficiently small
  so that you can ignore them for any polynomially long time.
* The setting is quite restricted and unrealistic as it requires the angle of all inputs and the teacher vector to be
  smaller than $\pi / 4$.

**Questions:**

* Is it possible to adapt your strategy to the Gaussian inputs case? This seems to be a more natural generalization to
  the orthogonal input setting.
* Could you intuitively explain the meaning of the conditions in the definition of $S_l$?

**Limitations:**

This is a theoretical work and I cannot see any potential negative societal impacts.

---

> ### Author Rebuttal · Authors · 2023-08-08
>
> > In the first stage, the neurons are small and will align with the teacher neuron or deactivate, [...]
>
> Just to clarify that the alignment in the first phase is with the vector $\mathbf{\gamma}\_{[n]} = \frac{1}{n} \sum_{i = 1}^n y\_i \mathbf{x}\_i$, whose direction is in general different from that of the teacher neuron.  (E.g. the teacher neuron might even be outside of the cone spanned by the data points.)  This is important because it means that, after the first phase, the aligned neurons may need to change direction significantly to fit the target function.  In Example 32 at the end of Appendix F, we show an example run in which the alignment with $\mathbf{\gamma}\_{[n]}$ happens relatively early while the norm is relatively small, whereas alignment with the teacher neuron $\mathbf{v}^*$ happens only towards the final convergence after the growth of the norm.
>
> > The presentation is overall clear, but the notations are cumbersome. [...]
>
> We are happy to implement these suggestions.  The intention behind the $\mathbf{w}$ and $\mathbf{\omega}$ notations was that their similar looks would remind the reader that they represent related objects, however thank you for pointing out that it may be confusing.
>
> > I personally don't like the $\exp(-n)$ initialization scale, though I will accept it as it has been used in previous works. [...]
>
> The exponential dependence of the initialisation scale on the dataset cardinality $n$ is indeed impractical when $n$ is not small.  However, we think it is worthwhile to analyse in detail even such small initialisation scales and to determine a bound under which the analysis holds, partly to provide a solid basis from which to investigate the border to another regime such as mean field or lazy.  It is encouraging that our numerical experiments paint a picture which broadly follows the patterns identified in our theoretical results even for initialisation scales that are significantly larger than our theoretical bound.  Relaxing that bound, and in particular seeking to reach polynomial dependence on $n$, is therefore an interesting direction for future work.  We expect one challenge will be that the first phase of the training will produce bundles of neurons that are not as tightly aligned as we can guarantee in this work.
>
> > The setting is quite restricted and unrealistic as it requires the angle of all inputs and the teacher vector to be smaller than $\pi / 4$.
>
> Please see our response to all the reviewers jointly.
>
> > Is it possible to adapt your strategy to the Gaussian inputs case? This seems to be a more natural generalization to the orthogonal input setting.
>
> We expect that various parts of our submission could be adapted to the Gaussian inputs case, either for a population loss or for an empirical loss with a sufficiently large number of samples.
>
> Xu and Du COLT 2023 considered learning a ReLU neuron by a ReLU network with Gaussian inputs, where only the hidden layer is trained and every last-layer weight is fixed to $1$.  Having a population loss over the Gaussian inputs makes some aspects of the training dynamics simpler than in our setting.  In particular, already the first phase aligns the neurons to the teacher.
>
> We think it is worthwhile to consider finite datasets, especially in the context of seeking to determine the implicit bias of gradient-based algorithms for regression tasks, where a central question is what interpolants the training converges to in predictor space.  The results in this submission establish that, for any dimension $d > 1$ and in our correlated setting, $d$ linearly independent inputs (the algebraic minimum) suffice to learn the teacher neuron; and that the implicit bias is such that, as the initialisation scale tends to zero, exactly the teacher neuron is converged to and moreover with a network of rank $1$.  (In the orthogonal setting, the number of inputs is at most $d$.)  We suggest one of the main contributions of this work is a full account of the second phase of the training, which due to working with an empirical loss over a finite dataset (rather than a population loss over a spherically symmetric data distribution) may start with the bundle of neurons aligned to a direction which is far from that of the teacher.
>
> > Could you intuitively explain the meaning of the conditions in the definition of $\mathcal{S}\_\ell$?
>
> The entire set $\mathcal S$ is open, bounded, and connected.  It is an invariant for the training dynamics in predictor space even when small noise is added to the derivative: namely, $\mathcal{S}$ contains the trajectory of the bundle $\mathbf{v}$ of neurons from the end of the alignment phase onwards, travelling from near the origin to near the teacher $\mathbf{v}^*$, see e.g.  Example 30 (and Figure 3) in Appendix F.
>
> The $\Omega\_k$, $\Phi\_\ell$, and $\Psi$ conditions control the dynamics: the ratios $\nu\_k / \nu\_k^*$ grow towards $1$ and then overshoot $1$, all sequentially one coordinate at a time: each component $\mathcal{S}\_\ell$ basically contains the segment of the trajectory during which the $\ell$th coordinate of the bundle grows towards its target, and the “handover” to $\mathcal{S}\_{\ell + 1}$ (for $\ell < d$) happens exactly at the point of the overshoot.  The coordinates, $\nu\_k$ and $\nu\_k^*$, are with respect to the basis consisting of the eigenvectors of $\frac{1}{n} \mathbf{X} \mathbf{X}^\top$.
>
> The non-linearity of the $\Psi\_{k, k’}^\uparrow$ constraint makes the boundary of the set $\mathcal{S}$ repelling for the approximate training dynamics when approached from the inside.
>
> Finally, $\Xi$ defines an ellipsoid of all vectors that have an acute angle with the derivative of the training dynamics in predictor space.  Moreover, $\Xi$ adds a “padding” inside the boundary determined by the small quantity $\lambda^{\varepsilon / 3}$ to account for the approximate derivatives.
>
> If you think it would be helpful, we shall add explanations along those lines to the paper.

---

> > ### Comment · Reviewer_PDpC · 2023-08-16
> >
> > Thank you for the clarifications. I will keep my score.

---

### Official Review · Reviewer_6KPF · 2023-07-12

**Soundness:** 3 good
**Presentation:** 4 excellent
**Contribution:** 3 good
**Rating:** 6
**Confidence:** 3

**Summary:**

Convergence and implicit bias of non-linear networks is an important open question in deep learning theory.  The paper studies these questions in the case of regression with ReLU networks with a single teacher neuron. It proves that at a vanishing initialization scale the student neurons align with the teacher (or gets deactivated). It also shows an interesting counter example such that the implicit bias as initialization tends to zero need not be a minimum norm interpolator.

**Strengths:**

a) Going beyond the orthogonal data, the paper proposes an interesting setting which helps analyse the case of correlated inputs.

b) The geometric technique to study the convergence after the alignment phase is novel.

c) The scenario proposed where the implicit bias as $\lambda \to 0$ is a rank minimizing one instead of a minimum norm interpolator is a very interesting contribution.

**Weaknesses:**

a) The setting is simplified: there is only a single teacher neuron and all the labels are only positive. It directly that neurons (at least yardstick neurons) with negative last layer decreases in norm. The assumption that the inputs are correlated further ensures that they are deactivated. This makes the analysis easier.

b) The technical analysis follows the same strategy as Boursier et. al. Some aspects are easier as there is only one saddle to escape.

**Questions:**

a) It is mentioned in the contributions there exists a case where that the minimum norm interpolator is rank 2 instead of rank 1. Can the authors comment how the non-linearity plays a role here? Can the authors a simple example providing a more detailed illustration?

**Limitations:**

a) The experiments are conducted with synthetic data and only in the case of single teacher neuron. It would interesting some empirical evidence in the case of more than one teacher if the phenomenon characterized in the paper holds in more generality.

---

> ### Author Rebuttal · Authors · 2023-08-08
>
> > b) The technical analysis follows the same strategy as Boursier et. al. Some aspects are easier as there is only one saddle to escape.
>
> We think that the presence of the negative labels and the consequent second saddle in Boursier et al. did not introduce major difficulties in that work, i.e. that the main technical achievements in Boursier et al. lie elsewhere in that work.  Indeed, the proofs of the technical results that take care of the negative labels in Boursier et al. (Lemma 8 and Lemma 11) proceed mostly along the same lines as the proofs of the corresponding results about the positive labels (Lemma 7 and Lemma 10, respectively).
>
> We remarked in the second paragraph of Appendix A that our results should be straightforward to extend to a setting with an orthogonally separable dataset labelled by two teacher neurons, which would allow both positive and negative labels, and result in a second saddle.  We chose not to develop that case in the paper because it would be a relatively shallow addition from a technical point of view.  Please see the bottom plot in the attached PDF for an experimental illustration.
>
> We suggest that, in technical aspects (proofs), our correlated setting is considerably more difficult to analyse compared to the orthogonal setting of Boursier et al.  In particular, our handling of the second phase during which the active neurons simultaneously grow and turn involved a novel geometric technique.
>
> > a) It is mentioned in the contributions there exists a case where that the minimum norm interpolator is rank 2 instead of rank 1. Can the authors comment how the non-linearity plays a role here? Can the authors a simple example providing a more detailed illustration?
>
> A family of examples is provided in Example 36 (case $\mathcal{M} > 0$) in Appendix H, for any dimension $d > 2$.  The core of the construction are three data points, e.g. $(16, 0, 0)$, $(16, -4, 1)$, $(16, 4, 1)$, and a teacher neuron, e.g. $(0.8, 0, 0.6)$, whose norm is $1$.  In this instance, the labels are therefore $12.8$, $13.4$, $13.4$, respectively.  However the same labels are produced by a sum of two ReLU neurons $(0.8, 0.05, 0.4)$ and $(0, -0.1, 0)$, whose sum of norms is approximately $0.896 + 0.1 = 0.996 < 1$.  In order for the second neuron to contribute to the label of just one data point out of three, non-linearity is essential.  Without it the sum of the two neurons would label the third data point by $13$ rather than $13.4$.  It is straightforward to obtain from this example a balanced one-hidden layer ReLU network with two neurons which thereby has smaller norm than any interpolator of rank $1$.
>
> If you think it would be helpful, we are happy to include a short example along these lines in the main paper?
>
> > a) The experiments are conducted with synthetic data and only in the case of single teacher neuron. It would interesting some empirical evidence in the case of more than one teacher if the phenomenon characterized in the paper holds in more generality.
>
> Learning a multi-neuron ReLU teacher network is known to be challenging to analyse.  Already in a uni-variate setting with biases, it is easy to come up with example datasets of only a few data points for which during the training several bundles of neurons emerge that simultaneously change their norms and directions, and apparently interact significantly with each other.  Such an example was provided by Boursier et al. in their Figure 4 in Appendix A.1.  We believe that our work provides a substantial and important step towards understanding such phenomena in arbitrary dimension, where we focused on alignment of the neurons and their simultaneous growth and turning, leaving for future work significant interactions between multiple bundles of neurons.
>
> Please see the middle two and the bottom plot in the attached PDF for some results from additional experiments with two teacher neurons.  For small scales of the noise used to generate the synthetic datasets (where the smallest scales are such that the angles between the data points and the corresponding teacher concentrate around $\pi / 4$), the phenomena we identified theoretically still seem to hold, where the training passes near a second saddle point as we outlined in the second paragraph of Appendix A.

---

### Author Rebuttal · Authors · 2023-08-08

We thank all the reviewers for their positive and encouraging reviews, and for comments and questions, which will help us improve the submission.

We attach a PDF with plots from a few additional experiments, and refer to it in our responses to some of the reviewers.

In what follows we elaborate on the $\pi / 4$ condition on the angles between the data points and the teacher neuron, which reviewers PDpC and ytEj commented on and asked about, and which may be of interest also to the other reviewers.

* Theoretically, whilst the $\pi/4$ condition is strong, it enables us to focus on datasets that are not orthogonal, thus taking on the main challenge posed by Boursier et al. NeurIPS 2022. Notice, e.g., that orthogonal datasets in $\mathbb R^d$ may have at most $d$ data points, whereas in our work the number of data points is unbounded.

* A key place where we rely on the $\pi/4$ condition is to show that, during the second phase of the training, all neurons that form the aligned bundle are active on all (and thus the same set) of data points (see Proposition 26 in Appendix F).  The $\pi / 4$ condition also implies that the angle between any two data points is less than $\pi / 2$, which streamlines the first phase and ensures that it produces only one bundle of aligned neurons.  We remark that, in dimension $2$ (which covers the uni-variate setting with biases), the $\pi / 4$ restriction with the teacher neuron can in fact be replaced by this weaker $\pi / 2$ restriction between data points.

* Please also see the top four figures in the attached PDF for some results of additional experiments that explore the effects of increasing the scale of the noise used to generate the synthetic datasets so that quickly most of the data points exceed the $\pi / 4$ angle with the corresponding teacher neuron.  The growing maximum angles between neurons at the end of the training indicate that we no longer have a single (or one per teacher neuron) aligned bundle of neurons forming and sticking together for the rest of the training.

We believe this submission paves the way for future work to generalise both the strictly orthogonal datasets of Boursier et al. and the correlated datasets with the $\pi / 4$ bound considered here.

We are happy to add remarks such as above to the paper.  We shall make clear all the places in the proofs where we use the $\pi / 4$ condition, and we shall report on the additional experiments in the appendix.

---

### Decision · Program_Chairs · 2023-09-21

**Decision:**

Accept (poster)

**Comment:**

The authors consider the dynamics of two-layer relu networks trained by gradient flow on inputs such that the labels are given by a single neuron which is correlated with all training points.  The reviewers were unanimous in voicing support for acceptance, albeit most of the reviewers were somewhat unenthusiastic in their support.  The reviewers valued the importance of understanding the implicit bias of gradient flow and the difficulty in trajectory analyses even in the simplified setting considered by the authors; one reviewer was impressed by some of the arguments used.  The primary concern brought forward by reviewers was the clarity of the writing and organization of the paper, with secondary concerns concerning the strength of the assumptions.  On the whole, I think the contributions are significant and I recommend acceptance.

I would recommend that the authors revise the main section in accordance with reviewer ytEj's suggestions.  I would also confirm that the camera ready and supplemental version have the same citation list, I noticed some odd differences in these.